# Glioblastoma remodelling of human neural circuits decreases survival

Saritha Krishna[1], Abrar Choudhury[1], Michael B. Keough[2], Kyounghee Seo[1], Lijun Ni[2], Sofia Kakaizada[1], Anthony Lee[1], Alexander Aabedi[1], Galina Popova[1], Benjamin Lipkin[3], Caroline Cao[1], Cesar Nava Gonzales[1], Rasika Sudharshan[1], Andrew Egladyous[1], Nyle Almeida[1], Yalan Zhang[1], Annette M. Molinaro[1], Humsa S. Venkatesh[2], Andy G. S. Daniel[1], Kiarash Shamardani[2], Jeanette Hyer[1], Edward F. Chang[1,4], Anne Findlay[5], Joanna J. Phillips[1,6], Srikantan Nagarajan[5], David R. Raleigh[1,6,7], David Brang[3], Michelle Monje[2,8] & Shawn L. Hervey-Jumper[1,4 ✉]

Gliomas synaptically integrate into neural circuits[1,2]. Previous research has demonstrated bidirectional interactions between neurons and glioma cells, with neuronal activity driving glioma growth[1–4] and gliomas increasing neuronal excitability[2,5–8]. Here we sought to determine how glioma-induced neuronal changes influence neural circuits underlying cognition and whether these interactions influence patient survival. Using intracranial brain recordings during lexical retrieval language tasks in awake humans together with site-specific tumour tissue biopsies and cell biology experiments, we find that gliomas remodel functional neural circuitry such that task-relevant neural responses activate tumour-infiltrated cortex well beyond the cortical regions that are normally recruited in the healthy brain. Site-directed biopsies from regions within the tumour that exhibit high functional connectivity between the tumour and the rest of the brain are enriched for a glioblastoma subpopulation that exhibits a distinct synaptogenic and neuronotrophic phenotype. Tumour cells from functionally connected regions secrete the synaptogenic factor thrombospondin-1, which contributes to the differential neuron–glioma interactions observed in functionally connected tumour regions compared with tumour regions with less functional connectivity. Pharmacological inhibition of thrombospondin-1 using the FDA-approved drug gabapentin decreases glioblastoma proliferation. The degree of functional connectivity between glioblastoma and the normal brain negatively affects both patient survival and performance in language tasks. These data demonstrate that high-grade gliomas functionally remodel neural circuits in the human brain, which both promotes tumour progression and impairs cognition.

Malignant brain tumours such as glioblastomas exist within the context of complex neural circuitry. Neuronal activity promotes glioma growth through both paracrine signalling (neuroligin-3 and brain-derived neurotrophic factor (BDNF)) and AMPAR (α-amino-3-hydroxy-5-methyl-4-isoxazole propionic acid receptor)-mediated excitatory electrochemical synapses[1–4].

Likewise, glioblastomas influence neurons, inducing neuronal hyperexcitability through the secretion of non-synaptic glutamate and synaptogenic factors[5,6] and reducing inhibitory interneurons[7]. Beyond preclinical models, we previously demonstrated in awake, resting patients that glioblastoma-infiltrated cortex exhibits increased neuronal excitability[2]. The mechanisms by which glioblastomas maintain the ability to engage with neuronal circuitry and alter cortical function remain incompletely understood[9]. Deciphering the processes by which gliomas remodel neural circuits may uncover therapeutic vulnerabilities for these lethal brain cancers. To address these gaps in knowledge, we performed intraoperative electrophysiology while patients engaged in language tasks: we analysed local field potentials in glioblastoma-infiltrated cortex during speech initiation, determined the decodability of neural responses and revealed biological drivers of synaptic enrichment in glioblastoma cells (Extended Data Fig. 1 and Supplementary Tables 1 and 2).

## Glioblastomas remodel neural circuits

Glioblastomas and other high-grade gliomas interact with neural elements, resulting in cellular- and network-level changes[10–13]. While neurons within glioblastoma-infiltrated brain are hyperexcitable at

[1]Department of Neurological Surgery, University of California, San Francisco, San Francisco, CA, USA. [2]Department of Neurology, Stanford University, Stanford, CA, USA. [3]Department of Psychology, University of Michigan, Ann Arbor, MI, USA. [4]Weill Institute for Neurosciences, University of California San Francisco, San Francisco, CA, USA. [5]Department of Radiology and Biomedical Imaging, University of California, San Francisco, San Francisco, CA, USA. [6]Department of Pathology, University of California, San Francisco, San Francisco, CA, USA. [7]Department of Radiation Oncology, University of California, San Francisco, San Francisco, USA. [8]Howard Hughes Medical Institute, Stanford, CA, USA. ✉e-mail: Shawn.Hervey-Jumper@ucsf.edu

rest, the extent of task-specific neuronal hyperexcitability and the ability to extract neural features from glioma-infiltrated cortex remain unclear. To examine cognitive task-specific neuronal activity from glioblastoma-infiltrated cortex, we selected a cohort of adult patients with cortically projecting tumours in the lateral prefrontal cortex (LPFC; Extended Data Fig. 2a). Electrocorticography (ECoG) electrodes were placed over tumour-infiltrated and normal-appearing cortex. ECoG signals filtered between 70–110 Hz were used for analysis of high-gamma band range power (HGp), which is strongly related to local neuronal population spikes[14,15] and is increased by cortical hyperexcitability[16]. Spectral data demonstrated clear separation of frequencies across tumour and non-tumour electrodes (Fig. 1a and Extended Data Fig. 3a).

ECoG was recorded from the dominant hemisphere LPFC during auditory and visual picture naming as an illustrative example of a well-defined cognitive neuronal circuit[17]. While patients were awake and speaking, HGp was recorded for single-electrode (Extended Data Fig. 3b) and group-level analysis. HGp data from control and non-tumour conditions demonstrate the expected neural time course of speech motor planning within the LPFC (Extended Data Figs. 2b and 3c,d), consistent with previously established models of speech initiation demonstrated in non-human primates and humans[18,19]. We next performed the same analysis focused only on electrode arrays recording from tumour-infiltrated cortex. Countering the theory that glioblastoma–synaptic integration may result in physiologically disorganized neural responses, we found task-relevant neural activity within the entire region of the tumour-infiltrated cortex, including cortical regions that are not typically implicated in speech production (Fig. 1b)—a notable finding that indicates tumour-induced functional remodelling of language circuitry. Similarly, we found that, across WHO grade 2–4 glioma subtypes, task-specific neuronal responses for speech initiation are maintained within the LPFC (Extended Data Fig. 2c,d). These findings demonstrate that neuronal activity within tumour-affected cortex is physiologically organized, including neuronal activity elicited by speech tasks in regions that are outside of regions that are typically involved in speech production.

In light of this finding of preserved task-evoked neural responses from tumour-infiltrated cortex, we next examined whether the magnitude of neural responses may differ in tumour-affected cortical language areas. We therefore pair-matched tumour-infiltrated and normal-appearing cortex (Extended Data Fig. 3d,e), demonstrating increased HGp during speech production in glioblastoma-infiltrated cortex, consistent with hyperexcitability (Fig. 1c,d).

Neural computations for speech vary by condition. Vocalization of infrequently used (low frequency) words, for example, requires a more intricate coordination of articulatory elements than that of commonly used (high frequency) words[20,21]. We therefore determined the decodability of neuronal signals from normal-appearing and glioblastoma-infiltrated cortex using a logistic regression classifier to distinguish between low-frequency and high-frequency word trial conditions (Fig. 1e). We implemented identical training and leave-one-participant-out cross-validation paradigms for both conditions. Normal-appearing cortex produced above-chance decoding between low- and high-frequency word trials. By contrast, glioblastoma-infiltrated cortex did not decode word trials above chance. These data further demonstrate that glioblastoma infiltration into the human cortex maintains task-specific neuronal responses, including neuronal hyperexcitability, yet tumour-affected cortex loses the ability to decode complex word conditions.

## Synaptogenic tumour cells promote connectivity

Having demonstrated that gliomas remodel neuronal circuits, we next examined whether specific molecularly defined glioma cellular subpopulations influence functional integration of the tumour into neural circuitry. Glioblastoma cells are heterogeneous[22–24] and previous findings indicate that oligodendrocyte-precursor-cell-like subpopulations are enriched for synaptic gene expression[2], whereas astrocyte-like subpopulations secrete synaptogenic factors[8,25]. Thus, functionally connected regions may vary within tumours and differences in functional connectivity between tumour regions may be due at least in part to varying subpopulations of glioma cells. With the goal of sampling functionally connected regions within gliomas, we measured neuronal oscillations within glioma-infiltrated brain using magnetoencephalography (MEG) and sampled primary patient glioblastoma tissues with varying functional connectivity during surgical tumour resection[26–28]. The connectivity of an individual voxel was derived by the mean imaginary coherence between the index voxel and the rest of the brain[29–31]. Intratumoural functional connectivity correlated with neuronal activity within tumour-infiltrated cortex and high functional connectivity (HFC) voxels were identified both within tumour regions that were contrast-enhancing or T2/FLAIR hyperintense on magnetic resonance imaging (MRI; Extended Data Fig. 4).

To investigate the differences between functionally connected, HFC and non-functionally connected low functional connectivity (LFC) tumour regions, we performed bulk and single-cell RNA sequencing (RNA-seq) analyses. Bulk RNA-seq transcriptomic analysis revealed upregulation in HFC tumour regions of genes that are involved in the assembly of neural circuits, including axon pathfinding genes (*NTNG1*, also known as netrin G1), synapse-associated genes (for example, *SYNPO*, also known as synaptopodin) and synaptogenic factors including a sevenfold upregulation of thrombospondin-1 (*THBS1*, encoding TSP-1). *THBS1*, which encodes a known synaptogenic factor that is secreted in the healthy brain by astrocytes[32], was particularly interesting in the context of the observed remodelling of functional language circuitry described above (Extended Data Fig. 5a–c and Supplementary Table 3).

To further assess cellular subpopulation contribution to *THBS1* expression, we performed single-cell sequencing analysis of biopsy samples from HFC and LFC tumour regions (Supplementary Table 4). Malignant tumour cells were inferred on the basis of the expression programs and detection of tumour-specific genetic alterations, including copy-number variants (Extended Data Fig. 6a–e). We found that 2.44% of all tumour cells expressed *THBS1*, and that HFC tumour cells expressed higher levels of *THBS1* compared with LFC tumour cells (Fig. 2a and Extended Data Fig. 6g). Within LFC-region samples, *THBS1* expression primarily derives from a non-tumour astrocyte population (Extended Data Fig. 6e–g). This suggests that, within low-connectivity intratumoural regions, astrocytes chiefly express *THBS1*, whereas, within HFC regions, high-grade glioma cells express *THBS1* in addition to astrocytes and myeloid cells, which may promote the observed neural circuit remodelling (Extended Data Fig. 6h–j). Notably, myeloid cells, which include bone-marrow-derived macrophages, microglia, dendritic cells and neutrophils, chiefly comprise the glioblastoma tumour immune microenvironment (Extended Data Fig. 6d,e), and the microglial cell surface molecules CD36 and CD47 can function as TSP-1 receptors[33,34]. Although the role of TSP-1 in the tumour immune microenvironment is not yet clear, myeloid cell expression of TSP-1 suggests that multiple cell types in the tumour microenvironment of HFC regions may contribute to altered synaptic connectivity. Elevated expression of *THBS1* within HFC regions was confirmed by protein-level analysis using HFC and LFC patient-derived glioblastoma biopsy tissues. Concordant with transcriptomic profiles, immunohistochemistry analysis demonstrated increased TSP-1 expression within HFC tissues (Extended Data Fig. 7a). Immunofluorescence and confocal microscopy analysis confirmed that malignant tumour cells express TSP-1 in HFC tissue (Fig. 2b). The fact that a subpopulation of malignant tumour cells in HFC regions produce TSP-1 suggests a differential potential of tumour cells in the HFC regions to promote synaptogenesis and thereby connectivity, consistent with the cancer biology principal that

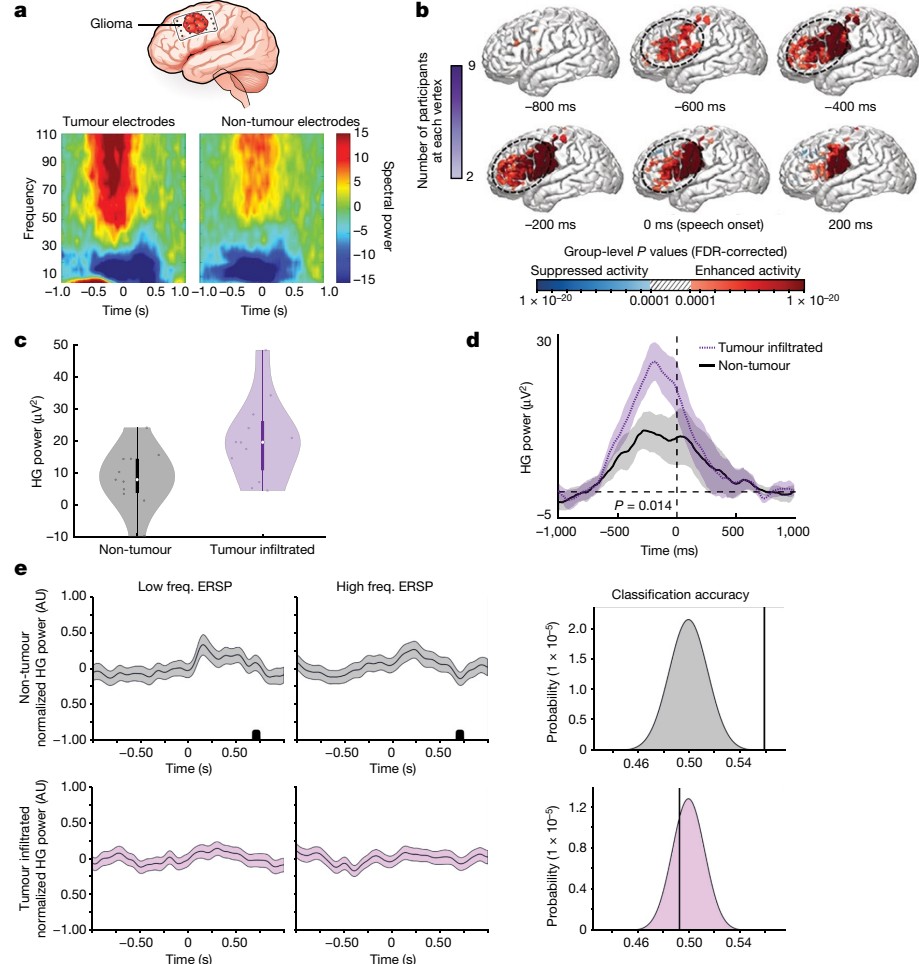

**Fig. 1 | High-grade gliomas remodel long-range functional neural circuits.**
**a**, In participants with dominant hemisphere glioblastomas, we applied
subdural ECoG over the posterior lateral frontal cortex during an audiovisual
speech initiation task to assess circuit dynamics. Spectral data show the
expected pattern of HGp increasing above 50 Hz in addition to clear separation
of frequencies across tumour and non-tumour electrodes. **b**, The posterior
lateral frontal cortex (outlined area) time series of HGp within tumour-
infiltrated cortex between −600 ms and speech onset (0 ms). **c**, High-gamma
(HG) recordings from averaged electrodes within each patient, while averaging
the effect across the sampled region of cortex for an individual showing greater
HG power within electrodes overlying tumour-infiltrated cortex (n = 14
patients, $F_{1,21}$ = 25.562, P = 0.00005). Data are median (centre dot), first to third
quartiles (bars) and the minimum and maximum points (whiskers). **d**, Electrodes
were compared between non-tumour and tumour-infiltrated regions; the
false-discovery rate (FDR)-corrected HGp demonstrates task-relevant

hyperexcitability (P = 0.016). Data are mean ± s.e.m. **e**, Event-related spectral
perturbations (ERSPs) during a naming task for low-frequency words (low
freq., left column) and high-frequency words (high freq., middle column) in
normal-appearing non-tumour regions (top row) and glioma-infiltrated
(bottom row) cortex. Signals from high-frequency word trials were able to
be decoded above chance in normal-appearing cortex (mean accuracy =
0.56, P = 0.000089) but not in glioma-infiltrated cortex (mean classifier
accuracy = 0.49, P = 0.72) using a regularized logistic regression classifier with
leave-one-participant-out cross-validation (right column). Data are mean
± 95% confidence interval. AU, arbitrary units. For **b**–**d**, statistical analysis was
performed using two-sided linear mixed-effects models (**b**–**d**), and corrections
for multiple comparisons were performed using FDR adjustment (**b** and **d**). For
**e**, P values were determined using two-tailed Student's t-tests with Bonferroni
multiple-comparison correction for the number of timepoints (left, ERSP) and
one-sided Z-tests (right, classification accuracy).

cellular subpopulations assume distinct roles within the heterogenous
cancer ecosystem, which may be defined at least in part by functional
connectivity measures.

Hypothesizing that this subpopulation of HFC glioma cells may pro-
mote synaptogenesis and consequent remodelling of connectivity as
observed in glioma-associated language networks above, we next exam-
ined whether HFC-associated glioma cells promote structural synapse
formation, similar to normal astrocytes[35–37] and certain astrocyte-like
glioblastoma cells[8,25]. We first analysed primary patient glioblastoma
biopsies from HFC and LFC regions using immunohistochemistry
and confocal microscopy. We found increased presynaptic neuronal
puncta (synapsin-1; Fig. 2c) together with increased postsynaptic
puncta density and cluster size on neurons (PSD95⁺neurofilament⁺),
and synapsin–PSD95 puncta colocalization (Fig. 2d and Extended Data

Fig. 7b) within HFC regions compared with LFC regions. Together, these
data indicate increased synapse stability and/or synapse formation in
high-connectivity regions of glioblastoma, supporting a role for TSP-1
in glioma-associated neural-circuit remodelling.

Primary patient-derived glioma cultures from HFC and LFC tumour
regions were generated to perform further mechanistic experiments.
We co-cultured high-grade glioma cells from HFC and LFC tumour
regions with mouse hippocampal neurons to test the effects of
TSP-1^high- and TSP-1^low-expressing primary patient-derived glioma
cells on synaptic connectivity of neurons (Extended Data Fig. 7c,d).
We then quantified the size of postsynaptic puncta (marked by the
postsynaptic marker homer-1) and the number of colocalized pre-
and postsynaptic puncta in HFC and LFC co-cultures with neurons.
This demonstrated an increased number of colocalization points and

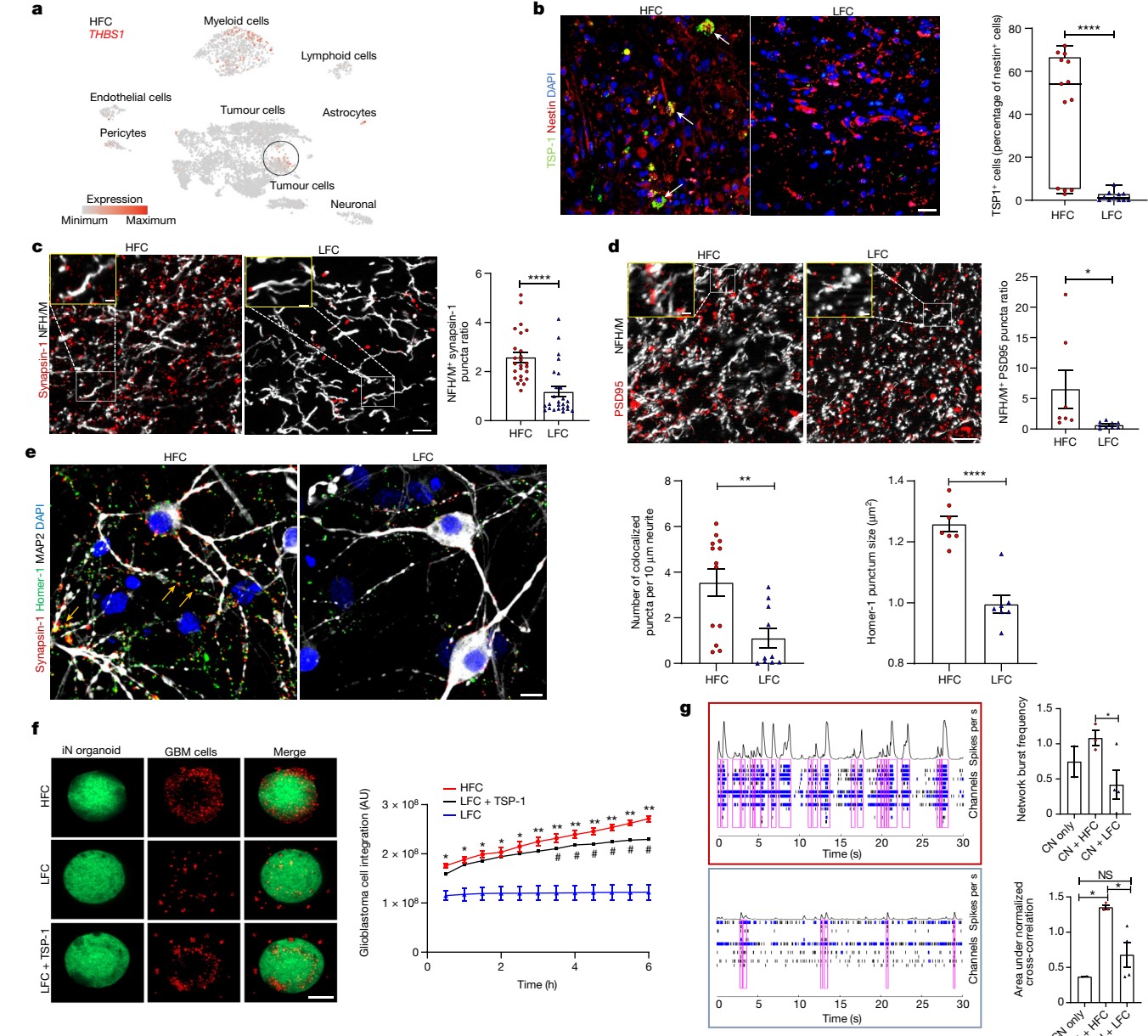

**Fig. 2 | Tumour-infiltrated circuits exhibit areas of synaptic remodelling characterized by glioma cells expressing synaptogenic factors. a**, Single-cell RNA-seq feature plot analysis of *THBS1* in HFC (*n* = 6,666 cells, 3 participants) tissues; within HFC samples, *THBS1* is primarily in glioblastoma cells (circled). **b**, TSP-1 immunofluorescence analysis of nestin-positive tumour cells in HFC and LFC tissues. *n* = 13 (HFC) and *n* = 11 (LFC) sections, 3 per group. *P* = 0.000073. Scale bar, 50 μm. The box plot shows the median (centre line), interquartile range (box limits) and minimum and maximum values (whiskers). **c**, The synapsin-1 puncta count in HFC and LFC glioblastoma tissue samples. *n* = 25 regions, 4 per group. *P* = 0.000014. Red, synapsin-1 (presynaptic puncta); white, neurofilament heavy and medium (neurons). Scale bar, 10 μm. Inset: magnified view of synapsin-1 puncta on neurons. Scale bar, 3 μm. **d**, PSD95 puncta count. *n* = 7 (HFC) and *n* = 9 (LFC) sections, 3 per group. *P* = 0.04. Red, PSD95 (postsynaptic puncta); white, neurofilament heavy and medium chains (NFH/M) (neurons). Scale bar, 10 μm. Inset: magnified view of PSD95 puncta on neurons. Scale bar, 3 μm. **e**, Representative confocal images showing synaptic punctum colocalization (yellow arrows). Red, synapsin-1; green, homer-1 (postsynaptic puncta); white, MAP2 (neurons); blue, 4′,6-diamidino-2-phenylindole dihydrochloride (DAPI). Scale bar, 10 μm. Quantification of the number of

colocalized pre- and postsynaptic puncta (*n* = 13 (HFC) and *n* = 10 (LFC) regions, 2 per group; *P* = 0.005) and homer-1 puncta size in neuron–glioma co-culture (*P* = 0.000024). **f**, TSP-1 rescue of induced neuron (iN) organoids in co-culture with HFC and LFC cells for 6 h. Scale bar, 300 μm. Quantification of glioblastoma (GBM) cell integration measured on the basis of the fluorescence intensity of RFP-positive glioblastoma cells in the organoids. Significant differences between HFC and LFC groups (asterisks) and LFC and LFC + TSP-1 (hash) are indicated. *n* = 2 (HFC and LFC groups) and *n* = 1 (LFC + TSP-1 group). Scale bar, 300 μm. **g**, Representative MEA raster plots showing individual spikes (tick mark), bursts (cluster of spikes in blue) and synchronized network bursts (pink) after 48 h co-culture of cortical neurons (CN) with HFC and LFC cells (outlined in red and blue, respectively). Quantification of network burst frequency (Hz) (*n* = 2 (CN only), *n* = 3 (CN + HFC) and *n* = 4 (CN + LFC); *P* = 0.05) and network synchrony (area under normalized cross-correlation; *n* = 2 (CN only), *n* = 3 (CN + HFC) and *n* = 4 (CN + LFC); *P* = 0.0129 (CN versus CN + HFC); *P* = 0.0308 (CN + HFC versus CN + LFC)). Data are mean ± s.e.m. (**b**–**g**). *P* values were determined using two-tailed Student's *t*-tests (**b**–**f**) and one-way analysis of variance (ANOVA) with Tukey's post hoc test (**g**). *\*P* < 0.05, *\*\*P* < 0.01, *\*\*\*\*P* < 0.0001; NS, not significant.

postsynaptic homer-1[+] punctum size in glioma and neuronal processes in HFC–neuron co-cultures compared with LFC–neuron co-cultures (Fig. 2e), additionally indicating a role for HFC glioma cells in synaptogenesis.

To further investigate the functional distinctions between malignant subpopulations isolated from HFC and LFC regions, we tested neuron–glioma interactions in a neuronal organoid model. We co-cultured HFC and LFC glioma cells with GFP-labelled human neuron organoids generated from an induced pluripotent stem (iPS) cell line integrated with a doxycycline-inducible human *NGN2* transgene to drive neuronal differentiation[38]. Quantification of postsynaptic homer-1 in induced-neuron organoids revealed a relative increase in postsynaptic puncta density when co-cultured with HFC glioma cells compared with LFC glioma cells (Extended Data Fig. 7e). Live-cell imaging of neuronal organoids co-cultured with HFC and LFC glioma cells revealed that HFC glioma cultures exhibit prominent neuronal tropism and integrate extensively in the organoids, whereas LFC glioma cells displayed minimal integration with neuron organoids (Fig. 2f and Supplementary Videos 1 and 2). Notably, exogenous administration of TSP-1 to induced-neuron–LFC co-culture reversed this phenotype and promoted robust LFC glioma integration into the neuronal organoid (Fig. 2f and Supplementary Video 3), further implicating TSP-1 in neuron–glioma interactions. The electrophysical properties of TSP-1[high]-expressing cells in co-culture with neurons were analysed using multi-electrode array (MEA) electrophysiology. After co-culture for 48 h, the total number of network bursts (a measure of neuronal activity) from cortical neuron co-culture with TSP-1[high]-expressing HFC cells was increased relative to cortical neurons alone or under LFC co-culture conditions. Neurons in co-culture with HFC glioma cells also demonstrated increased network synchrony as measured by the area under normalized cross-correlation (the area under interelectrode cross-correlation normalized to the autocorrelations; Fig. 2g and Extended Data Fig. 7f).

Gliomas exhibit intratumoural heterogeneity with subpopulations of cancer cells assuming particular roles[23,24]. The human data presented above demonstrate localizational heterogeneity of functional integration in glioblastoma with normal brain circuity and suggest that, within intratumoural regions of HFC, a tumour subpopulation with synaptogenic properties exists. We next examined the structural synapses in TSP-1[high]-expressing HFC glioma cell-infiltrated mouse brain. RFP-labelled HFC or LFC glioma cells were stereotactically xenografted into the CA1 region of the mouse hippocampus[2] (Fig. 3a). After a period of engraftment and growth, immuno-electron microscopy analysis identified neuron-to-neuron and neuron-to-glioma synapses[2] (Fig. 3b). The total number of synapses (neuron-to-neuron and neuron-to-glioma combined) was significantly higher in HFC glioma xenografts than in LFC glioma xenografts (Fig. 3b and Extended Data Fig. 7g), further demonstrating a greater synaptogenic potential of glioma cells isolated from HFC patient tumour regions.

## HFC promotes tumour progression

Neurons promote glioma cell proliferation[1–4] and we hypothesized that HFC cells may represent a cellular subpopulation within glioblastomas that are differentially regulated by neuronal factors. We found that primary patient biopsies from HFC and LFC regions demonstrated increased Ki-67 proliferative marker staining within HFC regions (Fig. 3c). To test whether HFC cells differentially proliferate in response to neuronal factors compared with LFC primary patient cultures, patient-derived HFC and LFC cells cultured alone or in co-culture with mouse hippocampal neurons were treated with 5-ethynyl-2′-deoxyuridine (EdU) overnight. HFC glioma cells exhibit a fivefold increase in proliferation when cultured with neurons. By contrast, the LFC glioma in vitro cell proliferation index (determined as the fraction of DAPI cells co-expressing EdU) is similar with and without hippocampal neurons in vitro (Fig. 3d and Extended Data Fig. 8). These results indicate that the ability of HFC cells to proliferate is contingent on the presence of neuronally secreted factors and that, in the absence of neuronal signals, they tend to acquire a dormant tumour phenotype.

Given the neuronal tropism exhibited by HFC glioma cells together with the concept that neural network integration requires invasion of brain parenchyma to reach and colocalize with neuronal elements, we next tested the effects of neuronal conditioned medium on invasion of HFC and LFC glioma cells using a spheroid invasion assay. LFC glioma cells demonstrated no differences in spheroid volume in the presence or absence of neuronal conditioned medium; however, HFC glioma cells exhibited an increased spheroid invasion area in response to neuronal conditioned medium. In addition to increased invasion area, HFC glioma cells extended long processes representing tumour microtubes in response to neuronal conditioned medium (Extended Data Fig. 9a). Tumour microtubes connect glioma cells in a gap-junction-coupled network[1,39–41] through which neuronal-activity-induced currents are amplified[2]. Scanning electron microscopy (SEM) was performed on TSP-1[high]-expressing HFC and LFC cells in the presence or absence of neuronal conditioned medium, demonstrating robust cytoplasmic extensions connecting HFC cells (Fig. 3e). We also quantified the change in mean spheroid volume. We found that neuronal conditioned medium increased both invasion and microtube length in HFC but not LFC cultures (Fig. 3f and Extended Data Fig. 9a). Concordantly, the invasive marker MET was increased within HFC samples compared with LFC samples (Extended Data Fig. 9b,c). Primary patient-derived HFC cells were then transduced with a short hairpin RNA (shRNA) control or shRNA against *THBS1* to knockdown TSP-1. Cell viability was confirmed using a live/dead assay with robust knockdown of our target protein (Extended Data Fig. 10a,b). Knockdown of *THBS1* in HFC cells decreased the number of tumour microtubes relative to the control conditions (Fig. 3g), consistent with the known role for TSP-1 in tumour microtube formation[39].

Glioblastoma cell invasion bears negative prognostic value. We therefore performed survival studies of mice that were orthotopically xenografted with patient-derived HFC or LFC glioma cells. Mice bearing HFC tumours exhibited greater tumour burden and shorter survival compared with LFC-tumour-xenografted mice (Fig. 3h and Extended Data Fig. 9d). Taken together, these results suggest that functionally connected intratumoural regions are enriched for a tumour cell population that is differentially responsive to neuronal signals and exhibits a proliferative, invasive and integrative phenotype in the neuronal microenvironment that negatively influences survival in a preclinical model.

## Glioma connectivity shortens patient survival

We next investigated the effects of tumour-intrinsic functional connectivity on patient survival and cognition. First, we tested the hypothesis that gliomas exhibiting increased functional connectivity may be more aggressive, given the robust influence of neuronal activity on tumour progression[2–4]. We performed a human survival analysis of patients with newly diagnosed glioblastoma. After controlling for known correlates of survival (age, tumour volume, completion of chemotherapy and radiation, and extent of tumour resection)[42], neural oscillations and functional connectivity were measured within tumour-infiltrated brain using MEG (Supplementary Tables 2 and 5). A Kaplan–Meier survival analysis illustrates an overall survival of 71 weeks for patients with functional connectivity compared with an overall survival of 123 weeks for participants without HFC voxels, illustrating a striking inverse relationship between survival and functional connectivity of the tumour (mean follow-up time, 50.5 months) (Extended Data Fig. 11a). To identify clinically relevant survival risk groups, we next used recursive partitioning survival analysis using the partDSA algorithm[42–44]. Within this analysis we controlled for important prognostic variables such as MGMT promoter methylation status. Overall survival risk was based on

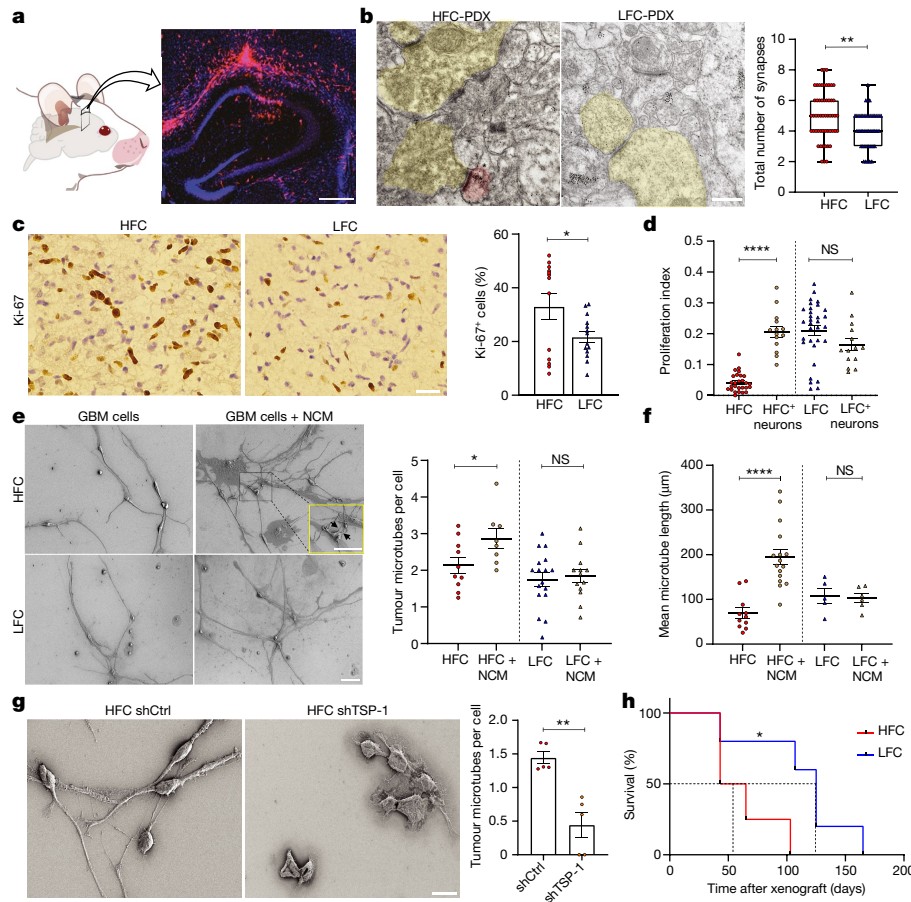

**Fig. 3 | High-grade gliomas exhibit bidirectional interactions with HFC brain regions. a**, Representative micrograph showing RFP-labelled glioblastoma xenografted into the mouse hippocampus. Scale bar, 500 μm. **b**, Immuno-electron microscopy analysis of HFC or LFC cell xenografts. The asterisk denotes immuno-gold particle labelling of RFP. Postsynaptic density in RFP⁺ tumour cells (pseudocoloured red), synaptic cleft and clustered synaptic vesicles in apposing presynaptic neuron (pseudocoloured yellow) identify both neuron–glioma synapses in HFC-PDX (left) and neuron–neuron synapses in LFC-PDX (right). Quantification of the total number (neuron–neuron combined with neuron–glioma) of synapses per field of view in HFC/LFC xenografts. *n* = 4 mice per group. *P* = 0.0019. Scale bar, 1,000 nm. Data are median (centre line), with first and third quartiles (box limits) and the minimum and maximum points (whiskers). **c**, Representative immunohistochemistry images in glioblastoma tissues demonstrate increased Ki-67 protein expression in HFC samples. *n* = 13 (HFC) and *n* = 14 (LFC) regions, 4 per group. *P* = 0.04. Scale bar, 50 μm. **d**, Glioblastoma cells from HFC tissues show a marked increase in the proliferative index when

co-cultured with mouse hippocampal neurons. *n* = 27 (HFC), *n* = 14 (HFC + neurons), *n* = 32 (LFC) and *n* = 14 regions (LFC + neurons), 3 per group. **e**, SEM images of HFC and LFC cells cultured in the presence or absence of neuronal conditioned medium (NCM) shows tumour microtubes (TMTs) that connect neighbouring cells through cytoplasmic extensions. Quantification of TMTs per cell. *n* = 10 (HFC), *n* = 8 (HFC + NCM), *n* = 17 (LFC) and *n* = 13 (LFC + NCM) regions, 2 per group. *P* = 0.0455. Scale bars, 20 μm (full fields) and 10 μm (magnified view). **f**, Quantification of the mean microtube length per spheroid. *n* = 11 (HFC), *n* = 16 (HFC + NCM), *n* = 5 (LFC) and *n* = 6 (LFC + NCM) spheroids, 1 per group. *P* = 0.000011. **g**, Representative SEM images showing TMTs and quantification of TMTs per cell from HFC shCtrl and HFC shTSP-1 conditions. *n* = 5 regions, 2 per group. *P* = 0.0012. Scale bar, 20 μm. **h**, Kaplan–Meier survival curves of mice bearing HFC or LFC xenografts. *n* = 4 (HFC) and *n* = 5 (LFC). *P* = 0.03. Data are mean ± s.e.m. (**b**–**h**). *P* values were determined using two-tailed Student's *t*-tests (**b**–**g**) and two-tailed log-rank analysis (**h**). *P < 0.05, **P < 0.01, ****P < 0.0001; NS, not significant.

the interactive effects of all known prognostic variables (for example, age at diagnosis, sex, tumour location, chemotherapy, radiotherapy, the presence of functional connectivity within the tumour, pre- and post-operative tumour volume, and the extent of resection). The first division was based on known risk factors such as age and extent of tumour resection. Within this hierarchical model of partitioning, the degree of connectivity was identified as the next most important variable, which divided risk groups 2 and 3. Risk group 1 (black) had the worst outcomes and is the combination of patients older than 72 years or any age with less than 97% extent of tumour resection (subtotal resection). Risk group 3 (grey) had the best survival, and these are patients are younger than 62 years with over 97% extent of tumour resection and absence of functional connectivity in the tumour. Intermediate risk group 2 (red) revealed an interesting interaction between age and HFC. This group had two subsets: patients with over 97% resection of

tumour and age younger than 72 years with intratumoural connectivity; and those between 62 and 72 years without functional integration (Fig. 4a,b). These results demonstrate the notable prognostic value of connectivity on survival. We next examined whether TSP-1, a secreted synaptogenic protein[32,35], can be identified in the patient serum and whether circulating TSP-1 is correlated with functional connectivity. Circulating TSP-1 levels in the patient serum exhibited a notable positive correlation with intratumoural functional connectivity (Fig. 4c).

We hypothesized that, beyond survival, intratumoural functional connectivity may also influence cognition. We therefore performed visual picture and auditory naming testing in our cohort of patients with dominant hemisphere glioblastoma, given their correlation with aphasia in clinical populations[45,46]. Linear regression of the number of HFC voxels within tumours with language task performance demonstrated an inverse relationship between language cognitive performance and

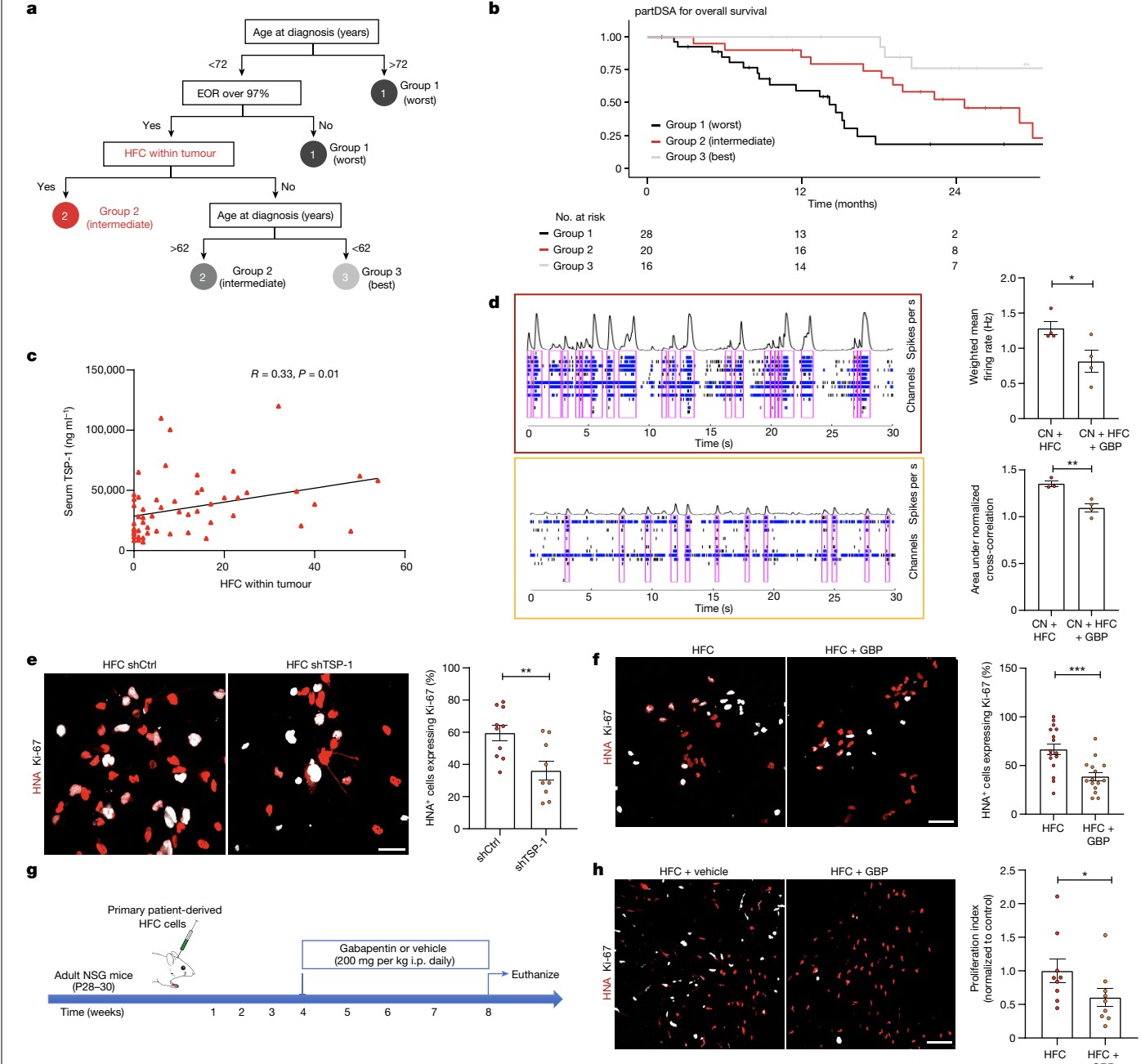

**Fig. 4 | Intratumoural connectivity in patients with high-grade glioma is correlated with survival and TSP-1. a,b,** Schematic (**a**) and partDSA model (**b**) of overall survival in patients, incorporating the effects of glioblastoma intrinsic functional connectivity, therapeutic and clinical factors by recursive partitioning results into three risk groups. Risk group 1 (black) patients have the shortest survival, including a combination of (1) patients older than 72 and (2) patients younger than 72 with an extent of tumour resection (EOR) of less than 97%. Risk group 3 (grey) patients have the best survival, including patients who are younger than 62 with an extent of tumour resection of greater than 97% and no intratumoural connectivity. Intermediate risk group 2 (red) comprises a combination of patients with greater than 97% extent of resection and (1) an age of younger than 72 with tumour intrinsic connectivity or (2) patients between 62 and 72 years without connectivity. **c,** Linear regression statistics illustrate that serum TSP-1 is correlated with the extent of intratumoural functional connectivity. $n = 56$. $P = 0.01$. **d,** Representative MEA raster plots showing neuronal spikes (black tick marks), bursts (cluster of spikes in blue) and synchronized network bursts (pink) of neuron–HFC co-cultures (outlined in red) and 24–48 h exposure of neuron–HFC co-culture to (50 μM) GBP (outlined in orange). Quantification of the weighted mean firing rate (Hz) and network synchrony (area under normalized cross-correlation) from HFC and HFC + GBP glioma–neuron co-culture (weighted mean firing rate: $n = 4$ well, 2 per group; $P = 0.04$; area under normalized cross-correlation: $n = 3$ (HFC) and $n = 4$ (HFC + GBP); $P = 0.007$). **e,** Representative confocal images from neuron–HFC glioma co-culture showing a decrease in HFC cell proliferation after *THBS1* knockdown using shRNA ($n = 10$ (HFC shCtrl) and $n = 9$ (HFC shTSP-1); $P = 0.0068$). Red, HNA (human nuclei); white, Ki-67. Scale bar, 30 μm. **f,** Representative confocal images from neuron–HFC glioma co-culture showing a decrease in HFC cell proliferation after gabapentin (32 μM) treatment for TSP-1 inhibition. $n = 16$ (HFC) and $n = 15$ (HFC + GBP), 2 per group. $P = 0.0007$. Red, HNA (human nuclei); white, Ki-67. Scale bar, 30 μm. **g,** Schematic for gabapentin treatment of HFC xenografted mice. i.p., intraperitoneal. **h,** Representative confocal images, and quantification demonstrating a decrease in the proliferation index (Ki-67⁺HNA⁺/HNA⁺) after gabapentin treatment in mice bearing HFC xenografts. $n = 9$ mice per group. $P = 0.046$. Red, HNA (human nuclei); white, Ki-67. Scale bar, 70 μm. Data are mean ± s.e.m. (**d**–**f** and **h**). *P* values were determined using two-sided linear regression analysis (**c**), and two-tailed (**d**–**f**) and one-tailed (**h**) Student's *t*-tests. *$P < 0.05$, **$P < 0.01$, ***$P < 0.001$; NS, not significant.

tumour functional connectivity (Extended Data Fig. 11b,c). Together, these findings suggest that functional integration of glioblastoma into neural circuits negatively influences cognition and survival.

## TSP-1 as a therapeutic target

Given the premise that TSP-1 serves as a regulator of neuronal activity-driven glioma growth, we sought to target TSP-1 therapeutically using gabapentin (GBP), which blocks the thrombospondin receptor α2δ-1 (ref. 47). In neuron–glioma co-cultures, individual spikes, bursts (cluster of spikes) and synchronized network bursts were reduced after 24–48 h exposure to GBP (Fig. 4d). Primary patient-derived HFC cells were transduced with shRNA-control or shRNA against *THBS1* or treated with GBP. Pharmacological TSP-1 inhibition using GBP did not influence the proliferation of HFC cells grown alone in culture, verifying that there were no tumour cell-intrinsic effects of GBP (Extended Data Fig. 12). By contrast, genetic or pharmacological targeting of TSP-1 resulted in a marked decrease in proliferation of HFC glioma cells co-cultured with neurons (Fig. 4e,f). GBP administration to mice bearing HFC patient-derived xenografts (PDX) resulted in a marked decrease in glioma proliferation (Ki-67⁺HNA⁺/HNA⁺) in gabapentin-treated mice bearing HFC xenografts relative to vehicle-treated controls (Fig. 4g,h).

## Discussion

Integration of high-grade glioma into neural networks is manifested by bidirectional interactions whereby neuronal activity increases glioma growth[1–4,48] and gliomas increase neuronal excitability[5–8]. To understand whether glioma–neuronal interactions influence neural circuit dynamics, we used short-range electrocorticography analysis of tumour-infiltrated cortex in humans to demonstrate language-task-specific activation as well as functional remodelling of language circuits. We further demonstrated that distinct intratumoural regions maintain functional connectivity through a subpopulation of TSP-1-expressing malignant cells (HFC glioma cells). This molecularly distinct glioma subpopulation is differentially responsive to neuronal signals, exhibiting a synaptogenic, proliferative, invasive and integrative profile. Previous research has demonstrated that neuronal activity promotes glioma proliferation through paracrine and synaptic signalling[1–4], and we have now shown that patients with glioblastoma exhibiting functional connectivity between the tumour and the rest of the brain experience a shorter overall survival compared with patients without HFC. Pharmacological inhibition of TSP-1 decreases glioblastoma cell proliferation and network synchrony within the tumour microenvironment, highlighting a potential therapeutic strategy to be assessed in future clinical studies.

The neuronal microenvironment has emerged as a crucial regulator of glioma growth. Both paracrine signalling and connectivity remodelling may contribute to network-level changes in patients, affecting both cognition and survival. In patients, the role of neural network dynamics on survival and cognition remains poorly understood and how glioma–network interactions influence cognition remains unanswered. In fact, some studies using a heterogenous population of patients with both IDH wild type (WT) and mutant WHO grade III and IV gliomas have suggested that functional connectivity improves overall survival[49–51]; however, such previous research has been confounded by functional connectivity methods that are heavily influenced by the presence of tumour vascularity, limited spatial resolution and a heterogenous patient cohort. Nonetheless, the evidence in this study that glioblastomas remodel functional circuits and that functional connectivity negatively influences survival does not address direction of causality. It remains possible that glioma originating in functionally connected cortical regions are more strongly connected and may therefore exhibit greater network distribution, thereby encouraging

distinct glioblastoma subpopulations with the ability to migrate[52]. A better understanding of the cross-talk between neurons and gliomas as well as how functional integration affects clinical outcomes may open the door to a range of pharmacological and neuromodulation therapeutic strategies focused on improving cognitive outcomes and survival.

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

## Methods

### Patients, samples, consent and ethics approval

Each participant in the study was recruited from a prospective registry of adults aged 18–85 with newly diagnosed frontal, temporal and parietal IDH-WT high-grade gliomas with detailed language assessments and baseline MEG recordings. Inclusionary criteria included the following: native English speaking, aged 18–85 years, and no previous history of psychiatric illness, neurological illness, or drug or alcohol abuse. All human electrocorticography data were obtained during lexical retrieval language tasks from 14 adult awake patients undergoing intraoperative brain mapping for surgical resection. Tumours from eight patients were used for RNA-seq experiments. Site-directed tumour biopsies from 19 patients were used for immunofluorescence/immunohistochemistry analysis and 24 patients were used for immunocytochemistry and cell-based functional assays. Tumours from eight patients were used for mouse xenograft experiments. All of the participants provided written informed consent to participate in this study, which was approved by the University of California, San Francisco (UCSF) institutional review board (IRB) for human research (UCSF CC-171027, CHR 17-23215) and performed in accordance with the Declaration of Helsinki.

We began by studying short-range circuit dynamics in a subset of 14 patients with dominant hemisphere glioblastoma infiltrating speech production areas of the inferior frontal lobe using ECoG in the intraoperative setting (Extended Data Fig. 1a). We then focused molecular studies on patients with surgically treated IDH-WT glioblastoma, performed extraoperative language assessments and imaginary coherence as a long-range measure of functional connectivity using MEG (Extended Data Fig. 1a,b). This enabled us to import functional connectivity data into the operating room in which we performed site-specific tissue biopsies of human glioma from regions with differing measures of functional connectivity for in vivo and in vitro cell biology experiments including primary patient cultures ($n = 19$ patients) and multimodal tissue profiling, including microscopy, sequencing, proteomics and patient-derived tumour xenografting (Extended Data Fig. 1c). This layered approach—combining clinical variables, cognition assessments, human and animal models of network dynamics, in addition to cell biology—served as a platform to study the clinical implications of glioma–neuron interactions (Extended Data Fig. 1d and Supplementary Tables 1 and 2).

### Human ECoG and data analyses

The hemisphere of language dominance was determined using baseline magnetic source imaging. In brief, the participants sat in a 275-channel whole-head CTF Omega 2000 system (CTF Systems) sampling at 1,200 Hz while they performed an auditory-verb generation task. The resulting time series were then reconstructed in source space with an adaptive spatial filter after registration with high-resolution MRI. Finally, changes in beta-band activity during verb generation were compared across hemispheres to generate an overall laterality index. All of the participants were left-dominant and underwent electrophysiological recording of the left hemisphere. We implemented an intraoperative testing paradigm that was previously established[9]. Noise in the operating room was minimized through rigorous enforcement of the following: (1) all personnel were requested to cease verbal communication; (2) telephones and alarms were muted; and (3) surgical suction and all other non-essential machinery were temporarily shut down. A 15 inch laptop computer (60 Hz refresh rate) running a custom MATLAB script integrated with PsychToolbox 3 (http://psychtoolbox.org/) was placed 30 cm away from each participant. The script initiated a picture-naming task that consisted of a single block of 48 unique stimuli, each depicting a common object or animal through coloured line drawings. Each stimulus was presented at the point of central fixation and occupied 75% of the display. After presentation of each stimulus, the participants were required to vocalize a single word that best described the item.

Intraoperative photographs with and without subdural electrodes present were used to localize each electrode contact combined with stereotactic techniques[9,53]. Images were registered using landmarks from gyral anatomy and vascular arrangement to preoperative T1- and T2-weighted MRI scans. Tumour boundaries were localized on MRI scans and electrodes within 10 mm of necrotic tumour core tissue were identified as 'tumour' contacts. Electrodes overlying the hypointense core of the tumour extending from the contrast enhancing rim to the edge of FLAIR were considered to be tumour electrodes, and electrodes completely outside of any T1 post gadolinium or FLAIR signal were considered to be non-tumour or normal appearing by a trained co-author blinded to the electrophysiologic data[2]. Glioma-infiltrated regions were defined on the basis of two criteria previously established in the literature[9], including mass-like region of T2-weighted FLAIR sequences signal. Imaging was confirmed by gross inspection of the cortex confirming dilation and/or an abnormal vascular pattern. Previous research has shown that regions of non-enhancing disease consist of infiltrating tumour cells intermixed with neurons and normal glial cells[2,54]. These labels were reviewed by the study principal investigator (S.L.H.-J.) and compared to labels derived during intraoperative stereotactic neuronavigation to reach a consensus (Brainlab).

Each participant received a training session 2 days before participation to ensure familiarity with the task. ECoG signals were acquired during a period after stopping the administration of anaesthetics (minimum drug wash out period of 20 min) and the patient was judged to be alert and awake after an extensive post-emergence wakefulness assessment to ensure adequate arousal[55]. Intraoperative tasks consisted of naming pictorial representations of common objects and animals (picture naming) and naming common objects and animals through auditory descriptions (auditory naming)[56]. Post-operative videos were reanalysed to ensure that all data were collected and correct responses only were included for analysis. Audio was sampled at 44.1 kHz from a dual-channel microphone placed 5 cm from the participant and electrophysiological signals were amplified (g.tec). Recordings were acquired at 4,800 Hz and downsampled to 1,200 Hz during the initial stages of processing. During offline analyses, audio and electrophysiological recordings were manually aligned, resampled and segmented into epochs (speech-locked). These epochs set time = 0 ms as speech onset and included ±2,000 ms for a total of 4,000 ms of signal per trial. Trials were discarded if (1) an incorrect response was given (including fillers and interjections) or (2) there was a greater than 2 s delay between stimulus presentation and response so as to maintain consistent trial dynamics and ensure that the neural signal indeed reflected the experimental manipulations. Channels with excessive noise artifacts were visually identified and removed if their kurtosis exceeded 5.0. After the rejection of artifactual channels, data were referenced to a common average, high-pass filtered at 0.1 Hz to remove slow-drift artifacts, and bandpass filtered between 70–110 Hz using a 300-Order FIR filter to focus the analyses on the high-gamma band range, which is strongly related to local mean population spiking rates. To extract the ERSPs, electrophysiological signals were first downsampled to 600 Hz, then high-pass filtered at 0.1 Hz to remove DC-offset and low-frequency drift, notch-filtered at 60 Hz and its harmonics to remove line noise, and bandpass-filtered between 70 and 170 Hz (that is, the high-gamma range) using a Hamming windowed sinc FIR filter. These signals were finally smoothed using a 100 ms Gaussian kernel, downsampled to 100 Hz and $z$-scored across each trial. Electrodes were subsequently rereferenced to the common average for each participant to facilitate group comparisons, and regions of interest were defined according to the Automated Anatomical Labelling Atlas (https://www.gin.cnrs.fr/en/tools/aal/). The location of grid implantation was solely directed by clinical indications. The accuracy of the final registration for each participant was independently confirmed using gyral and sulcal anatomy to triangulate the location of each electrode registered to the template surface and was then compared to intraoperative photographs of the

actual cortex with the overlying grid(s)[57]. The HGp was then calculated using the square of the Hilbert transform on the filtered data. The HGp was then averaged across the resting-state time series, yielding a single measure of neural responsivity for each electrode contact. The HGp was then averaged across patients during the task response period, yielding a single measure of neuronal responsivity for each channel. The HGp levels were then compared between tumour and normal appearing channels. Linear mixed-effects modelling was used to perform statistical comparisons with repeated measures using the nlme package in R (v.3.1-161; https://cran.r-project.org/web/packages/nlme/citation.html). The signal's origin (that is, normal-appearing/glioma-infiltrated cortex) was modelled as a fixed effect and the participants were modelled as random effects. For continuous variables without repeated measures, $t$-tests were used. A threshold of $P < 0.05$ was used to denote statistical significance and corrections for multiple comparisons were made using the Bonferroni method.

To decode between low-frequency words (for example, rooster) and high-frequency words (for example, car), signals from normal-appearing and glioma-infiltrated electrodes were extracted from the anterior temporal lobe after participant-level registration to a common MNI atlas. Responses were time-locked to speech onset and the signal envelope was extracted using a Hilbert transform after applying a bandpass filter in the high-gamma range (70–170 Hz). Subsequently, an l2-regularized logistic regression classifier was trained (cost of 1) to distinguish neural responses during vocalization of low-frequency words (for example, rooster) from high-frequency words (for example, car). Model performance was determined by taking the accuracy on a held-out participant and averaging it across all folds (that is, leave-one-participant-out cross-validation) and statistical significance was determined by testing this accuracy against a binomial distribution. This process was conducted separately for normal-appearing and glioma-infiltrated cortex using an identical preprocessing, training and testing paradigm.

## MEG recordings and data analysis
MEG recordings were performed according to an established protocol[30,31]. In brief, the study participants had continuous resting state MEG recorded with a 275-channel whole-head CTF Omega 2000 system (CTF Systems) using a sampling rate of 1,200 Hz. During resting-state recordings, the participants were awake with their eyes closed. Surface landmarks were co-registered to structural magnetic resonance images to generate the head shape. Within the alpha frequency band, an artifact-free 1 min epoch was selected for further analysis if the patient's head movement did not exceed 0.5 cm. This artifact-free, 1 min epoch was then analysed using the NUTMEG software suite (v.4; UCSF Biomagnetic Imaging Laboratory) to reconstruct whole-brain oscillatory activity from MEG sensors so as to construct functional connectivity (imaginary coherence (IC)) metrics[54,58,59]. Spatially normalized structural magnetic resonance images were used to overlay a volume-of-interest projection (grid size = 8 mm; approximately 3,000 voxels per participant) such that each voxel contained the entire time series of activity for that location derived by all the MEG sensor recordings. The time series within each voxel was then bandpass-filtered for the alpha band (8–12 Hz) and reconstructed in source space using a minimum-variance adaptive spatial filtering technique[54,60]. The alpha frequency band was selected because it was the most consistently identified peak in the power spectra from this sampling window in our patient series. Functional connectivity estimates were calculated using IC, a technique known to reduce overestimation biases in MEG data generated from common references, cross-talk and volume conduction[26,28].

Resting-state MEG was also used to measure intratumoural gamma activity. A spatial beamformer was applied to extract neural signals at the voxel level from manually defined regions of interest corresponding to FLAIR signal abnormality (that is, within the infiltrative margin of the tumour)[61]. These source-space signals were then downsampled to 300 Hz, notch filtered at 60 Hz to remove line noise and rereferenced to the common average. Spectral activity from 1 to 50 Hz was estimated at each voxel using Thomson's multitaper method (pmtm in MATLAB R2021b) with 29 Slepian tapers. Next, gamma power from 30 to 50 Hz was computed at each voxel after subtracting the aperiodic component from each spectrum by fitting a Lorentzian function in semi-log space[62]. A point estimate of intratumoural gamma activity was subsequently computed by averaging the activity across all voxels for each participant and regressed against the corresponding number of manually counted intratumoural HFC nodes.

## Functional connectivity map
The functional connectivity of an individual voxel was derived by the mean IC between the index voxel and the rest of the brain, referenced to its contralesional pair[30]. It is possible that there are regions within gliomas with varying amounts of functional connectivity. Moreover, there are individual patients with more or less functional connectivity. We have addressed these differences in our experimental model. Intratumoural differences in functional connectivity were addressed by the following: in comparison to contralesional voxels, we used a two-tailed $t$-test to test the null hypothesis that the $Z$-transformed connectivity IC between the index voxel and non-tumour voxel is equal to the mean of the $Z$-transformed connectivity between all contralateral voxels and the same set of voxels. The resultant functional connectivity values were separated into tertiles: upper tertile (HFC) and lower tertile (LFC). Functional connectivity maps were created by projecting connectivity data onto each individual patient's preoperative structural magnetic resonance images and imported into the operating room neuronavigation console. Stereotactic site-directed biopsies from HFC (upper tertile) and LFC (lower tertile) intratumoural regions were taken and $x, y, z$ coordinates determined using Brainlab neuro-navigation. Thus, only the extremes of intratumoural connectivity (high and low connectivity, HFC and LFC, respectively) were analysed for these experiments. Rather than raw values, each functional connectivity measure represents a $Z$-transformed value and it therefore remains likely that the HFC distinction for one patient does not perfectly coincide with the HFC distinction in another patient's tumour (intertumoural heterogeneity).

## Measurement of tumour volume and calculation of volumetric extent of resection
Pre-operative and post-operative tumour volumes were quantified using BrainLab Smartbrush (v.2.6; Brainlab). Pre-operative MRI scans were obtained within 24 h before resection, and post-operative scans were all obtained within 72 h after resection. Total contrast-enhancing tumour volumes were measured at both pre-operative and post-operative timepoints. The total contrast-enhancing tumour volume was measured on T1-weighted post-contrast images, and the non-enhancing tumour volume was measured on T2 or FLAIR sequences. Manual segmentation was performed with region-of-interest analysis 'painting' inclusion regions based on fluid-attenuated inversion-recovery (FLAIR) sequences from pre- and post-operative MRI scans to quantify tumour volume. The extent of resection was calculated as follows: (pre-operative tumour volume − post-operative tumour volume)/pre-operative tumour volume × 100%. Manual segmentations were performed for which the tumour volumetric measurements were verified for accuracy after an initial training period. Volumetric measurements were performed blinded to patients' clinical outcomes. All of the patients in the cohort had available preoperative and postoperative MRI scans for analysis. To ensure that post-operative FLAIR signal was not surgically induced oedema or ischaemia, FLAIR pre- and post-operative MRIs were carefully compared alongside DWI sequences before including each region in the volume segmentation[42]. HFC voxels with T1 post gadolinium contrast enhancing tumour were considered to be HFC-positive for survival analysis.

## Language assessments

One to two days before tumour resection, patients underwent baseline language evaluation, which consisted of naming pictorial representations of common objects and animals (picture naming) and naming common objects and animals through auditory descriptions (auditory naming). Visual picture naming and auditory stimulus naming testing were used given their known significance and clinical correlation with outcomes in clinical patient population[63,64]. The correct answers for these tasks (delivered on a laptop with a 15 inch monitor (60 Hz refresh rate) positioned two feet away from the seated patient in a quiet clinical setting) were matched on word frequency (that is, commonality within the English language) using SUBTLEX$_{WF}$ scores provided by the Elixcon project and content category. Task stimuli were randomized and presented using PsychToolbox. The task order was randomly selected by the psychometrist for each participant. Slides were manually advanced by the psychometrist either immediately after the participant provided a response or after 6 s if no response was given. The tasks were scored on a scale from 0 to 4 by a trained clinical research coordinator who was initially blinded to all clinical data (including imaging studies). No participants had uncorrectable visual or hearing loss. Details of the administration and scoring of auditory and picture naming language tasks can be found in previous studies[27,55,65].

## Isolation and culture of primary patient-derived glioblastoma cells

Tumour tissues with high (HFC) and low (LFC) functional connectivity sampled during surgery based on preoperative MEG were processed for quality control by a certified neuropathologist and were subsequently used to generate primary patient-derived cultures. Patient-matched samples were acquired from site-directed HFC and LFC intratumoural regions from the same patient. Intratumoural HFC and LFC tissues were dissociated both mechanically and enzymatically and then passed through a 40 μm filter to remove debris. The filtered cell suspension was then treated with ACK lysis buffer (Invitrogen) to remove red blood cells and subsequently cultured as free-floating neurospheres in a defined, serum-free medium designated tumour sphere culture medium, consisting of Dulbecco's modified Eagle's medium (DMEM-F12; Invitrogen), B27 (Invitrogen), N2 (Invitrogen), human-EGF (20 ng ml$^{-1}$; Peprotech), human-FGF (20 ng ml$^{-1}$; Peprotech). Normocin (InvivoGen) was also added to the cell culture medium in combination with penicillin–streptomycin (Invitrogen) to prevent mycoplasma, bacterial and fungal contaminations. Cell cultures were routinely tested for mycoplasma (PCR Mycoplasma Test Kit I/C, PromoCell) and no positive results were obtained (Extended Data Fig. 7d).

## Bulk RNA-seq and analysis

RNA was isolated from HFC ($n = 3$) and LFC ($n = 4$) tumour samples using the RNeasy Plus Universal Mini Kit (QIAGEN) and RNA quality was confirmed using the Advanced Analytical Fragment Analyzer. RNA-seq libraries were generated using the TruSeq Stranded RNA Library Prep Kit v2 (RS-122- 2001, Illumina) and 100 bp paired-end reads were sequenced on the Illumina HiSeq 2500 system to at least 26 million reads per sample at the Functional Genomics Core Facility at UCSF. Quality control of FASTQ files was performed using FASTQC (http://www.bioinformatics.babraham.ac.uk/projects/fastqc/). Reads were trimmed with Trimmomatic (v.0.32)[66] to remove leading and trailing bases with quality scores of less than 20 as well as any bases that did not have an average quality score of 20 within a sliding window of 4 bases. Any reads shorter than 72 bases after trimming were removed. Reads were subsequently mapped to the human reference genome GRCh38 (https://www.ncbi.nlm.nih.gov/assembly/GCF_000001405.39/)[67] using HISAT2[68] (v.2.1.0) with the default parameters. For differential expression analysis, we extracted exon-level count data from the mapped HISAT2 output using featureCounts[69]. Differentially expression analysis was performed using DESeq2[70] using the apeglm parameter[71] to accurately calculate log-transformed fold changes and setting a false-discovery rate of 0.05. Differentially expressed genes were identified as those with log-transformed fold changes of greater than 1 and an adjusted P value of less than 0.05. Unsupervised gene expression principal component analysis and volcano plots of IDH-WT glioblastoma (Extended Data Fig. 5b,c) revealed 144 differentially expressed genes between HFC and LFC tumour regions, including 40 genes involved in nervous system development (Supplementary Table 3).

## Single-cell sequencing

**Single-cell suspension generation.** Fresh tumour samples were acquired from the operating room and transported to the laboratory space in PBS and on ice. Tumour tissue was minced with #10 scalpels (Integra LifeSciences) and then digested in papain (Worthington Biochemical, LK003178) for 45 min at 37 °C. Digested tumour tissue was then incubated in red blood cell lysis buffer (eBioscience, 00-4300-54) for 10 min at room temperature. Finally, the samples were sequentially filtered through 70 μm and 40 μm filters to generate a single-cell suspension.

**Single-cell sequencing and analysis.** Single-cell suspensions of three patient-matched HFC and LFC tumour tissues were generated as described above and processed for single-cell RNA-seq using the Chromium Next GEM Single Cell 3′ GEM, Library & Gel Bead Kit v3.1 on the 10x Chromium controller (10x Genomics) using the manufacturer's recommended default protocol and settings, at a target cell recovery of 5,000 cells per sample. Although single-cell sequencing does not capture all cell types within the central nervous system microenvironment, the sequencing pipeline used in this study has been demonstrated to identify neurons and was therefore chosen for use in physiologically annotated fresh glioblastoma samples, compared with single-nucleus RNA-seq, which is commonly applied for frozen archived tissues[72,73]. One hundred base pair paired-end reads were sequenced on the Illumina NovaSeq 6000 system at the Center for Advanced Technology at the University of California San Francisco, and the resulting FASTQ files were processed using the CellRanger analysis suite (v.3.0.2; https://github.com/10XGenomics/cellranger) for alignment to the hg38 reference genome, identification of empty droplets, and determination of the count threshold for further analysis. A cell quality filter of greater than 500 features but fewer than 10,000 features per cell, and less than 20% of read counts attributed to mitochondrial genes, was used. Single-cell UMI count data were preprocessed in Seurat (v.3.0.1)[74,75] using the sctransform workflow[76], with scaling based on the regression of UMI count and the percentage of reads attributed to mitochondrial genes per cell. Dimensionality reduction was performed using principal component analysis and then principal component loadings were corrected for batch effects using Harmony[77]. Uniform manifold approximation and projection was performed on the reduced data with a minimum distance metric of 0.4 and Louvain clustering was performed using a resolution of 0.2. Marker selection was performed in Seurat using a minimum difference in the fraction of detection of 0.5 and a minimum log-transformed fold change of 0.5. We assessed the single-cell transcriptome from 6,666 HFC-region cells and 7,065 LFC-region cells (Supplementary Table 4).

## Immunohistochemistry and immunofluorescence analysis

After rehydration, 5.0 μm paraffin-embedded sections were processed for antigen retrieval followed by blocking and primary antibody incubation overnight at 4 °C. The following primary antibodies were used: rabbit anti-synapsin 1 (1:1,000, EMD Millipore), mouse anti-PSD95 (1:100, UC Davis), mouse anti-nestin (1:500, Abcam), mouse anti-neurofilament (M+H; 1:1,000, Novus Biologicals), mouse anti-TSP-1 (1:20, Invitrogen), rabbit anti-TSP-1 (1:50, Abcam), rabbit anti-MET (1:100, Abcam) and rabbit anti-Ki-67 (1:100, Abcam). We used species-specific secondary antibodies: Alexa 488 goat anti-chicken IgG, Alexa 488 goat anti-rabbit

IgG, Alexa 568 goat anti-rabbit IgG, Alexa 568 goat anti-mouse IgG, Alexa 647 goat anti-rabbit IgG, all used at 1:500 (Invitrogen). After DAPI nuclear counter staining (Vector Laboratories, 1:1,000), coverslips were mounted with Fluoromount-G mounting medium (SouthernBiotech) for immunofluorescence analysis. The number of synapsin-1 and PSD95 puncta was quantified using spots (with automatic intensity maximum spot detection thresholds and a spot diameter of 1.0 μm) detection function of Imaris. The ratio of pre- and postsynaptic puncta was calculated by dividing the total number of synapsin-1 or PSD95 puncta on neurofilament-positive neurons to the total number of cells stained with DAPI in 135 μm × 135 μm field areas for quantification. Alternatively, the sections were incubated in DAB horseradish peroxidase (Vector Laboratories) for chemical colorimetric detection after incubation in ImmPress anti-rabbit IgG (Novus Biologicals) and counterstained with Harris haematoxylin for immunohistochemistry analysis.

### Glioma–mouse hippocampal neuron co-culture
Glioma cells were plated on poly-D-lysine and laminin-coated coverslips (Neuvitro) at a density of 10,000 cells per well in 24-well plates. Approximately 24 h later, 40,000 embryonic mouse hippocampal neurons (Gibco) were seeded on top of the glioma cells and maintained with serum-free Neurobasal medium supplemented with B27, gentamicin and GlutaMAX (Gibco). After 2 weeks of co-culture, cells were fixed with 4% paraformaldehyde (PFA) for 30 min at 4 °C and incubated in blocking solution (5% normal donkey and goat serum, 0.25% Triton X-100 in PBS) at room temperature for 1 h. Next, they were treated with primary antibodies diluted in the blocking solutions overnight at 4 °C. The following antibodies were used: rabbit anti-homer-1 (1:250, Pierce), mouse anti-synapsin-1 (1:200) and chicken anti-MAP2 (1:500, Abcam). The coverslips were then rinsed three times in PBS and incubated in secondary antibody solution (Alexa 488 goat anti-chicken IgG; Alexa 568 goat anti-mouse IgG, and Alexa 647 goat anti-rabbit IgG, all used at 1:500 (Invitrogen) in antibody diluent solution for 1 h at room temperature. The coverslips were rinsed three times in PBS and then mounted with VECTA antifade mounting medium with DAPI (Vector Laboratories).

**Confocal imaging and quantification of synapsin-1 and homer-1 staining and colocalization analysis.** Images were captured at 1,024 × 1,024 resolution using a ×10 objective on the Nikon C2 confocal microscope. The confocal microscope settings for the homer-1 Alexa488 and synapsin-1 Alexa647 channels were held constant across all of the samples that were used for the experiment. Collected images were then imported into Imaris software (Imaris v.9.2.1, Bitplane) and the threshold value for each channel was manually adjusted and the colocalized voxels of the synapsin-1 puncta with the homer-1 marker was detected by creating a colocalization channel using the built-in the colocalization module of the Imaris software. Furthermore, the colocalization events were quantified by running the built-in spot detection algorithm of Imaris in conjunction with the colocalization channel. Next, dendrites labelled by MAP2 was visualized in TRITC channel and reconstructed using the Filament tool of Imaris software; the number of colocalized puncta representing synapses were counted and presented as the number of synapsin-1- and homer-1-positive puncta per 10 μm of dendrite length. Areas of homer-1 immunolabelled synaptic puncta were reconstructed using Imaris software Surface tool on maximal-intensity projections. Surfaces were built using a surface area detail level of 0.1 μm, thresholding by absolute intensity and taking all voxel >1.0 into account. The area sizes of individual anti-homer1-immunostained puncta were analysed and the mean values were calculated.

**Induced neuron organoid and glioma co-culture.** Induced neuron organoids were generated from a WTC11 iPS cell clone integrated by human NGN2 transgene induction as described previously[38,78]. In brief, iNeuron organoids were generated by the transgenic human iPS cell WTC11 line by NGN2 induction through addition of 2 μg ml⁻¹ doxycycline

in the 1:1 mixture of Neurobasal and BrainPhys neural medium containing 1% B-27 supplement, 0.5% GlutaMAX, 0.2 μM compound E, 10 ng ml⁻¹ BDNF and 10 ng ml⁻¹ NT-3 for 10 days to induce neuronal differentiation. Next, neuron maturation was triggered by feeding the organoids with approximately 8-month-old organoid conditioned medium derived from astrocytes. Astrocytes were differentiated from the human iPS cell WTC11 line and cultured in a medium consisting of DMEM/F12 containing GlutaMAX, sodium bicarbonate, sodium pyruvate, N-2 supplement, B-27 supplement (Gibco), 2 μg ml⁻¹ heparin, 10 ng ml⁻¹ EGF and 10 ng ml⁻¹ FGF2. Neuron organoids were characterized as postmitotic and stained for MAP2 and βIII-tubulin to validate neuronal induction efficiency. After 14 days of neuronal differentiation, HFC/LFC glioma cells labelled with RFP were added to the neuron organoid culture at a ratio of 1:3. Before iNeuron induction, the transgenic human iPS cell line WTC11 was transduced with GFP lentivirus. A Zeiss Cell Observer spinning-disc confocal microscope (Carl Zeiss) fitted with a temperature- and carbon-dioxide-controlled chamber was used to record live interactions of glioma cells with neuron organoids. Organoids were imaged every 10 min for a 6 h period, starting at the time of co-culture initiation, using a 10× objective with 0.4 NA. To assess the effect of exogenous TSP-1 on the functional integration between glioma cells and neurons, human recombinant TSP-1 (R&D Systems) was applied at a dose of 5 μg ml⁻¹ to the LFC-neuron organoid co-culture. Live-cell image analyses were performed using ImageJ. In brief, a region of interest was drawn around each GFP-positive neuron organoid and the fluorescence intensity (integrated density) of the RFP-positive glioblastoma cells was measured in the outlined regions of interest for each of the indicated timepoints. At the end of two weeks, organoids from HFC and LFC co-cultures were embedded in OCT and sectioned at 10 μm thickness for homer-1 immunofluorescence staining. Determination of homer-1 expression was performed by analysing homer-1 puncta density of neuron-organoid-HFC and LFC co-cultures.

### MEA recordings
**Preparation of MEA plates.** We prepared 24-well CytoView multi-electrode plates (Axion Biosystems) before the addition of cells by coating with poly-D-lysine (Thermo Fisher Scientific), laminin (Fisher Scientific) and fibronectin (Corning). In brief, 1 day before the establishment of cultures, a solution of 0.1 mg ml⁻¹ of poly-D-lysine was added to the MEA plates at a volume of 100 μl per well and incubated at room temperature for 2 h. After 2 h, poly-D-lysine was aspirated and the plates were washed three times with sterile water, and allowed to air dry in a biosafety cabinet and stored at 4 °C. The next day, the plates were coated with 100 μl of 5 μg ml⁻¹ laminin and 1 μg ml⁻¹ fibronectin and incubated for 2 h at 37 °C before cell seeding.

**Preparation of cortical cultures.** Primary cortical cultures were established from E18 CD1 mice (Charles River Laboratories). Timed-pregnant CD1 dams were killed by $CO_2$ euthanasia in accordance with UCSF Institutional Animal Care and Use Committee (IACUC). Dissection of complete cortex from E18 embryos was performed in ice-cold HBSS (Gibco) under a dissecting microscope (Zeiss). Dissected cortices were minced to 1 mm² pieces and enzymatically digested in 5 ml of 0.25% trypsin reconstituted from 2.5% trypsin (Corning) in calcium- and magnesium-free Hank's Balanced Salt Solution (Worthington Biochemical Corporation) for 30 min at 37 °C. Then, 0.5 ml of 10 mg ml⁻¹ of DNase (Sigma-Aldrich) was added in the last 5 min of dissociation. Mechanical dissociation was then carried about by trituration using fire-polished glass Pasteur pipettes until tissue was homogeneously suspended with no visible sections/aggregates and subsequently filtered through a 40 μm cell strainer (Thermo Fisher Scientific). Cells were collected by centrifugation at 500*g* for 5 min and the resulting cell pellet was resuspended in fresh complete BrainPhys culture medium (1× BrainPhys culture medium (StemCell Technologies) supplemented with B27 (Invitrogen), N2 (Invitrogen) and penicillin–streptomycin

antibiotics (Invitrogen). Then, 10 μl of cell suspension was mixed 1:1 with Trypan Blue, and the viable cell concentration was quantified using a haemocytometer. Further dilution was performed to bring viable cell concentration to 100,000 cells per 10 μl. Droplets of 10 μl were then added directly over the electrode field of each pretreated MEA well and stored in the cell culture incubator for 1 h to allow cell adhesion. The wells were then carefully flooded with 500 μl complete BrainPhys medium and the cultures were maintained and allowed to mature in a tissue culture incubator with semi-weekly half-volume medium changes.

**Recordings of spontaneous neuronal activity and analysis.** Spontaneous extracellular neuronal recordings were carried out using the Maestro Edge system with an integrated heating system and temperature controller (Axion Biosystems) in combination with the Axion 24-well CytoView MEA plates (each well housing a 4 × 4 16-channel electrode array that are 350 μm away from each other) and Axion Integrated Studio (AxIS) Navigator (v.3.5.2; Axion Biosystems). In brief, to record spontaneous neuronal activity, the Neural Real-Time module was used. The neuronal firing events/action potentials (herein referred to as the spike) was defined by applying an adaptive threshold crossing method, that sets the threshold for spike detection for each channel/ electrode to 5 s.d. of the noise level[79]; activity exceeding this threshold was counted as a spike. Unless otherwise stated, all analysis considers only active channels, defined as channels exhibiting ≥5 spikes per min. Raw data files were obtained by sampling the channels simultaneously with a gain of 1,000× and a sampling frequency of 12.5 kHz per channel using a band-pass filter (200–3,000 Hz). To detect single-electrode bursting activity, an interspike interval threshold was used, setting the minimum number of spikes at 5 and the maximum interspike interval at 100 ms. The network bursting activity (simultaneous bursts at multiple MEA electrodes) was analysed by Neural Metric Tool (v.1.2.3; Axion Biosystems). For this purpose, the Adaptive algorithm was selected using the following settings: minimum number of spikes = 50 and minimum electrodes = 35%. Quantification of network synchrony was computed through AxIS software by calculating the area under the normalized cross-correlogram (AUNCC) as described previously[80–83]. AUNCC represents the area under interelectrode cross-correlation normalized to the autocorrelations, with higher values indicating greater synchronicity of the network. For additional neural data analysis, including mean firing rate of each electrode (the ratio of the total number of spikes per second and the total duration of recording (1,800 s)) and weighted mean firing rate (defined as the spike rate per well multiplied by the number of active electrodes in the associated well), raw data files were processed offline using the Statistics Compiler function in AxIS. Statistics Compiler output files were processed in Microsoft Excel (Microsoft) and with custom Python scripts to organize and extract individual parameter data for each well of each MEA plate and for data normalization. Raster plots illustrating spike histogram and network bursts were generated using Neural Metric Tool (Axion Biosystems).

**Glioma–neuron co-culture and gabapentin treatment.** Spontaneous neuronal activity from cortical cultures grown on MEA plates was recorded in 30 min sessions on days in vitro 1 (DIV1), DIV7 and DIV15. Bright-field images were captured at each of the above timepoints to assess the neuronal cell density and electrode coverage. Primary cortical neurons showed a constant maturation trend from DIV7 to DIV15, and the co-culture experiments were initiated when neurons showed a synchronous activity pattern network at DIV15. Baseline data were therefore recorded on DIV15 immediately before addition of glioma cells in the presence or absence of gabapentin. For glioma cell co-culture, a single-cell suspension from cultured neurospheres of primary patient-derived HFC and LFC were prepared and diluted to a viable cell concentration of 20,000 cells per 5 μl. Droplets of 5 μl were then plated on top of differentiating neurons in the MEA plate.

After plating, glioma cells were allowed to adhere for approximately 1 h after which HFC cultures were exposed for next 24–48 h to a working concentration of 50 μM gabapentin[84,85] diluted in complete BrainPhys medium or an equivalent amount of vehicle (sterile water) as a control. Each condition was run on two wells (experimental replicates). Neurons from two different embryos were used as biological triplicates (n = 2). Presented data from MEA recordings reflects well-wide averages from active electrodes, with the number of wells per condition represented by n values.

## Mice and housing conditions

All in vivo experiments were conducted in accordance with the protocols approved by the UCSF Institutional Animal Care and Use Committee (IACUC) and performed in accordance with institutional guidelines. Animals were maintained under pathogen-free conditions, in temperature- and humidity-controlled housing, with free access to food and water, under a 12 h–12 h light–dark cycle. For brain tumour xenograft experiments, the IACUC does not set a limit on maximal tumour volume but rather on indications of morbidity. These limits were not exceeded in any of the experiments as mice were euthanized if they exhibited signs of neurological morbidity or lost 15% or more of their body weight.

## Orthotopic xenografting for neuronal circuit integration and mouse survival experiments

For all xenograft studies, NSG mice (NOD-SCID-IL2R gamma chain-deficient, The Jackson Laboratory) were used. Male and female mice were used equally. For immuno-electron microscopy experiments, a single-cell suspension from cultured neurospheres of HFC and LFC (n = 2 each) labelled with red fluorescent protein (RFP) were prepared in sterile DMEM immediately before the xenograft procedure. Mice (n = 8; 2 biological replicates per patient line) at postnatal day 28–30 were anaesthetized with 1–4% isoflurane and placed into a stereotactic apparatus. The cranium was exposed through a midline incision under aseptic conditions. Approximately 50,000 cells in 2 μl sterile PBS were stereotactically implanted into the CA1 region of the hippocampus through a 31-gauge burr hole, using a digital pump at infusion rate of 0.4 μl min$^{-1}$ and 31-gauge Hamilton syringe. Stereotactic coordinates used were as follows: 1.5 mm lateral to midline, 1.8 mm posterior to bregma, −1.4 mm deep to cranial surface. At the completion of infusion, the syringe needle was allowed to remain in place for a minimum of 2 min, then manually withdrawn at a rate of 0.875 mm min$^{-1}$ to minimize backflow of the injected cell suspension. For survival studies, morbidity criteria used were either: reduction of weight by 15% initial weight, or clinical signs such as hunched posture, lethargy or persistent decumbency. Kaplan–Meier survival analysis using log-rank testing was performed to determine statistical significance.

## Quantification of tumour cell burden

Cell quantification was performed by a blinded investigator at 10–20× magnification using the Zeiss LSM800 scanning confocal microscope and Zen 2011 imaging software (Carl Zeiss). The area for quantification was selected for a 1-in-6 series of 40 μm coronal sections (240 μm apart from one another). Immunohistochemistry was performed on brain sections from HFC and LFC xenografts to stain for human nuclear antigen (HNA)-positive tumour cells. The tumour burden was evaluated using blinded rank-order analysis as previously reported[86]. For each mouse, the section with the maximal amount of tumour burden was selected as defined by the number of HNA-positive cells. From this section, a tiled ×10 image of the centre of the section (3 × 3 tiles stitched together) was created, therefore generating a single image for each mouse that represents the maximal amount of tumour burden. Next, the images generated from each mouse were compared and ranked in order from the image with the least to maximum number of cells. Subsequently, scores were assigned and ranged from 1 to 24 with 1

and 24 representing images with the lowest and highest number of HNA-positive cells, respectively. After scoring, the experimenter was unblinded and the scores from each image were assigned to each of the respective HFC or LFC experimental conditions. Statistical differences between HFC and LFC scores were evaluated using two-tailed unpaired Mann–Whitney tests.

## Sample preparation and image acquisition for electron microscopy

Twelve weeks after xenografting, mice were euthanized by transcardial perfusion with Karnovsky's fixative: 2% glutaraldehyde (EMS, 16000) and 4% PFA (EMS, 15700) in 0.1 M sodium cacodylate (EMS, 12300), pH 7.4. Transmission electron microscopy (TEM) was performed in the tumour mass within the CA1 region of the hippocampus for all xenograft analysis. The samples were then post-fixed in 1% osmium tetroxide (EMS, 19100) for 1 h at 4 °C, washed three times with ultrafiltered water, then en bloc stained overnight at 4 °C. The samples were dehydrated in graded ethanol (50%, 75% and 95%) for 15 min each at 4 °C; the samples were then allowed to equilibrate to room temperature and were rinsed in 100% ethanol twice, followed by acetonitrile for 15 min. The samples were infiltrated with EMbed-812 resin (EMS, 14120) mixed 1:1 with acetonitrile for 2 h followed by 2:1 EMbed-812:acetonitrile overnight. The samples were then placed into EMbed-812 for 2 h, then placed into TAAB capsules filled with fresh resin, which were then placed into a 65 °C oven overnight. Sections were taken between 40 nm and 60 nm on a Leica Ultracut S (Leica) and mounted on 100-mesh Ni grids (EMS FCF100-Ni). For immunohistochemistry, microetching was done with 10% periodic acid and eluting of osmium with 10% sodium metaperiodate for 15 min at room temperature on parafilm. Grids were rinsed with water three times, followed by 0.5 M glycine quench, and then incubated in blocking solution (0.5% BSA, 0.5% ovalbumin in PBST) at room temperature for 20 min. Primary goat anti-RFP (1: 300, ABIN6254205) was diluted in the same blocking solution and incubated overnight at 4 °C. The next day, grids were rinsed in PBS three times, and incubated in secondary antibodies (1:10 10 nm gold-conjugated IgG, TED Pella, 15796) for 1 h at room temperature and rinsed with PBST followed by water. For each staining set, samples that did not contain any RFP-expressing cells were stained simultaneously to control for any non-specific binding. Grids were contrast stained for 30 s in 3.5% uranyl acetate in 50% acetone followed by staining in 0.2% lead citrate for 90 s. The samples were imaged using a JEOL JEM-1400 TEM at 120 kV and images were collected using a Gatan Orius digital camera.

## Electron microscopy data analysis

Sections from the xenografted hippocampi of mice were imaged as above using TEM imaging. Here 101 sections of HFC xenografts across 4 mice and 104 sections of LFC xenografts across 3 mice were analysed. Electron microscopy images were taken at ×6,000 with a field of view of 15.75 µm². Glioma cells were counted and analysed after unequivocal identification of a cluster of immunogold particle labelling with 15 or more particles. The total number of synapses, including neuron-to-neuron and neuron-to-glioma synapses (identified by: (1) the presence of synaptic vesicle clusters; (2) visually apparent synaptic cleft; and (3) identification of clear postsynaptic density in the glioma cell) were counted.

## EdU assay

The EdU-incorporation assay was performed using an EdU assay kit (Invitrogen) according to the manufacturer's instructions. Patient-derived HFC/LFC glioma cells were seeded on poly-D-lysine- and laminin-coated coverslips at 10,000 cells per well of a 24-well plate. After 24 h of seeding, embryonic mouse hippocampal neurons were added to the glioma cells at 40,000 cells per well for the glioma-neuron co-culture group. After 72 h, glioma cells alone or in coculture with neurons were treated with 20 µM EdU overnight at 37 °C. Subsequently, the cells were fixed with 4% PFA and stained using the Click-iT EdU kit protocol.

The proliferation index was then determined by quantifying the fraction of EdU-labelled cells/DAPI-labelled cells using confocal microscopy at ×10 magnification.

## 3D spheroid invasion assay

Glioma cell invasion was evaluated by performing an invasion assay using the Cultrex 3D Spheroid Cell Invasion Assay Kit (Trevigen) according to the manufacturer's protocol. In brief, 3,000 cells were resuspended in 50 µl of spheroid formation matrix solution (prepared in culture medium) in a round-bottom 96-well plate. Spheroids were allowed to form for 72 h and images were taken at ×10 magnification before addition of invasion matrix (0 h). Working on ice, 50 µl of invasion matrix was then added to each well and the plate was incubated at 37 °C. After 1 h of gel formation, for the glioma cell + mouse conditioned medium (mCM) group, 100 µl of mouse hippocampal neuron supernatant was added to the wells to assess the effect of neuronal secreted factors on the invasiveness of the glioma cells. After 24 h of incubation at 37 °C, invasions were observed under a microscope and images were taken at ×10. Microtube lengths as well as the area of each spheroid measured at 0 h (pre-invasion) and 24 h (post-invasion) were analysed using ImageJ and the difference was used to calculate the total area of cell invasion.

## SEM analysis

SEM was performed as described previously[39] to investigate the tumour microtubes in greater detail. Primary patient-derived HFC/LFC glioma cells were seeded on poly-D-lysine- and laminin-coated coverslips (Neuvitro) at 10,000 cells per well of a 24-well plate. For the glioma cell + mCM and glioma cell + mCM + GBP groups, mouse neuronal conditioned medium was added to the wells in the absence or presence of gabapentin (32 µM), respectively, to assess the effect of neuronal secreted factors and gabapentin modulation on tumour microtube formation of glioma cells[47,87,88]. After culture for 1 week, the samples were fixed with 4% PFA for 30 min at 4 °C, washed with PBS and ultrafiltered water. After serial stepwise ethanol dehydration, coverslips were mounted on SEM stubs (TED Pella) using conductive adhesive tape (12 mm OD PELCO Tabs, TED Pella). The samples were then sputter-coated with a 2 nm layer of gold palladium. Tumour microtubes were observed under field emission SEM (Sigma 500; Carl Zeiss Microscopy) and micrographs were recorded at an accelerating voltage of 1.0 kV.

## THBS1 shRNA knockdown

Primary patient-derived HFC glioma cells were seeded in 6-well plates at a density of $2 \times 10^5$ cells per well. After overnight incubation, cells were infected with lentiviral particles expressing a shRNA targeting THBS1 or a control scramble construct expressing green fluorescent protein (GFP) according to the manufacturer's protocol using polybrene transfection reagent and subsequently kept in a 5% $CO_2$ incubator at 37 °C for 24 h. After 24 h, the medium was replaced with fresh complete culture medium. Cells were checked 72 h after transduction for GFP expression to evaluate the efficiency of the transduction. The lentiviral shRNA constructs targeting THBS1 (5′-AGACATCTTCCAAGCATATAA-3′) and control scrambled shRNA (5′-CCTAAGGTTAAGTCGCCCTCG-3′) were designed and constructed by VectorBuilder.

## Cell viability assay

The cell viability of primary patient-derived HFC cells after THBS1 knockdown was evaluated using the Live/Dead Viability/Cytotoxicity kit (Molecular Probes). One-week after transduction, HFC cells expressing scramble shRNA or shRNA targeting THBS1 were seeded on poly-D-lysine- and laminin-coated coverslips (Neuvitro) at 10,000 cells per well of a 24-well plate. After 1 week of culture, a 500 µl cell-staining solution at a final concentration of 2 µM calcein AM and 4 µM ethidium bromide (EthD-1) in DPBS was added to each well, and the plates were

incubated for 45 min at room temperature in dark. As an indicator of cell viability, the calcein-AM is metabolically converted by intracellular esterase activity resulting in the green, fluorescent product, calcein. EthD-1 is excluded from live cells but is readily taken up by dead cells and stains the DNA emitting red fluorescence. Live and dead cells were imaged from four random fields per well and were visualized under a fluorescence microscope. The percentage of live cells was calculated as the number of live cells (in green) divided by the number of total cells (green + red) per image field.

## In vitro cell proliferation assessment after *THBS1* shRNA knockdown

After 1 week of transduction, primary patient-derived high connectivity glioma cells expressing scramble shRNA or shRNA against *THBS1* were seeded on poly-D-lysine- and laminin-coated coverslips (Neuvitro) at 10,000 cells per well of a 24-well plate. Approximately 24 h later, 40,000 embryonic mouse hippocampal neurons (Gibco) were seeded on top of the glioma cells and maintained with serum-free Neurobasal medium supplemented with B27, gentamicin and GlutaMAX (Gibco). After 1 week of co-culture, cells were fixed with 4% PFA for 30 min at 4 °C and incubated in blocking solution (5% normal donkey and goat serum, 0.25% Triton X-100 in PBS) at room temperature for 1 h. Next, cells were treated with primary antibodies diluted in the blocking solutions overnight at 4 °C. The following antibodies were used: rabbit anti-Ki-67 (1:500, Abcam) and human nuclear antigen (HNA; mouse anti-human nuclei, 235-1; 1:100, Millipore). The coverslips were then rinsed three times in PBS and incubated in secondary antibody solution (Alexa 488 goat anti-rabbit IgG, and Alexa 647 goat anti-mouse IgG) all used at 1:500 (Invitrogen) in antibody diluent solution for 1 h at room temperature. The coverslips were rinsed three times in PBS and then mounted with VECTA antifade mounting medium with DAPI (Vector Laboratories). To calculate the proliferation index, the total number of HNA-positive cells co-labelled with Ki-67 was divided by the total number of human nuclei-labelled cells per image field visualized using confocal microscopy at ×40 magnification.

## In vitro cell proliferation assessment after pharmacological inhibition of TSP-1 by gabapentin

The effect of TSP-1 inhibition by gabapentin on glioma cell proliferation was evaluated by Ki-67 immunofluorescence staining, as described above. In brief, primary patient-derived high connectivity glioma cells were seeded on poly-D-lysine- and laminin-coated coverslips (Neuvitro) at 10,000 cells per well of a 24-well plate. Approximately 24 h later, 40,000 embryonic mouse hippocampal neurons (Gibco) were seeded on top of the glioma cells and maintained with serum-free Neurobasal medium supplemented with B27, gentamicin and GlutaMAX (Gibco). The next day, cultures were treated either with vehicle (sterile water) or 32 µM gabapentin followed by daily half-medium switches of fresh 32 µM gabapentin until 1 week of coculture[88]. Subsequently, cells were fixed and immunostained for Ki-67 and HNA labelling for proliferation assessment as described above.

## Orthotopic xenografting for proliferation assessment and pharmacological inhibition of TSP-1 by gabapentin

A single-cell suspension from cultured neurospheres of primary patient-derived HFC and LFC were prepared in sterile HBSS immediately before the xenograft procedure by dissociation with TrypLE (Thermo Fisher Scientific). Mice (4–6 biological replicates per patient line) at postnatal day 28–30 were anaesthetized with 1–4% isoflurane and placed into a stereotactic apparatus. The cranium was exposed through midline incision under aseptic conditions. Approximately 150,000 cells in 3 µl sterile HBSS were stereotactically implanted into the premotor cortex (M2) through a 26-gauge burr hole, using a digital pump at infusion rate of 1.0 µl min⁻¹. Stereotactic coordinates used were as follows: 1.0 mm lateral to midline, 1.0 mm anterior to bregma, −1.0 mm deep to cortical surface. Four weeks after xenograft, HFC/LFC-bearing mice were treated with systemic administration of gabapentin (200 mg kg⁻¹; Sigma-Aldrich; formulated in saline) through intraperitoneal injection for 28 consecutive days. Controls were treated with an identical volume of the relevant vehicle. At 8 weeks after xenograft, mice were euthanized and coronal brain sections at 40 µm were obtained for immunohistochemistry.

## Confocal imaging and quantification of tumour burden and cell proliferation

Cell quantification was performed by a blinded investigator at ×10–20 magnification using the Zeiss LSM800 scanning confocal microscope and Zen imaging software (Carl Zeiss). The area for quantification was selected for a 1-in-6 series of 40 µm coronal sections (240 µm apart from one another). Within 3 sections of maximal tumour burden, all HNA-positive (mouse anti-human nuclei, 235-1; 1:100, Millipore) cells were quantified to determine the tumour burden within the areas quantified. HNA-positive tumour cells were then assessed for double-labelling with Ki-67. To calculate the proliferation index (the percentage of proliferating tumour cells for each animal), the total number of HNA-positive cells co-labelled with Ki-67 across all areas quantified was divided by the total number of human nuclei-positive cells counted across all areas quantified. Differences in proliferation indices were calculated using unpaired, one-tailed Student's *t*-tests.

## ELISA

Peripheral blood samples from newly diagnosed patients with glioblastoma ($n = 56$) were collected and allowed to clot for 30 min at room temperature before centrifugation for 15 min at 1,000$g$. The serum was stored at −80 °C until analysis. The TSP-1 level was determined using the Quantikine immunosorbent assay kits according to the manufacturer's instructions (R&D Systems). To confirm the functional protein-level knockdown of *THBS1*, the TSP-1 level was also measured in cell culture supernatants collected from scramble and *THBS1*-shRNA transduced HFC cells (collected one-week after infection) using the Picokine ELISA kit (Boster Biological Technology).

## Statistics and reproducibility

Statistical tests were conducted using Prism (GraphPad) software unless otherwise indicated. The significance test of different groups was determined using Student's *t*-tests and one-way ANOVA with Tukey's post hoc tests. $P \leq 0.05$ was considered to be statistically significant. Two-tailed unpaired Mann–Whitney tests were used to analyse the tumour cell burden of HFC and LFC hippocampal xenografts using the same significance denotation as above. Two-tailed log-rank analyses were used to analyse the statistical significance of Kaplan–Meier survival curves for human patients. All of the micrographs shown are representative of three independently conducted experiments, with similar results obtained. Statistical analyses for RNA-seq data are described above in the respective sections. The tumour volumes of patients with glioblastoma were calculated by manual segmentation with region-of-interest analysis 'painting' inclusion regions on the basis of FLAIR sequences. Volumetric measurements were made blinded to patients' clinical outcomes. Manual segmentations were performed by co-authors A.E., A.A. and A.L. with tumour volumetrics verified for accuracy after an initial training period. Student's *t*-tests and $\chi^2$ tests were used to compare continuous and categorical variables between patient cohorts, respectively. Patient overall survival (OS) was defined as the time from the date of first surgery or original biopsy (if it occurred before surgery) until death or the last contact date. Alive patients were censored at the time of loss to follow-up or last follow-up date. Median follow-up was estimated using the reverse Kaplan–Meier method.

To identify clinically relevant survival risk groups in a multivariate setting, we used recursive partitioning analyses for survival data using the partDSA algorithm (v.0.9.14)[42–44]. Survival trees use recursive

partitioning to divide patients into different risk groups. The Brier score was the chosen loss function for splitting and pruning. Such methods are nonparametric and, therefore, do not require the proportional hazards assumption. All known prognostic variables were included in the trees, including age at diagnosis, sex, MGMT promoter methylation status, tumour location, chemotherapy, radiotherapy, the presence of functional connectivity within the tumour, pre- and post-operative tumour volume and the extent of resection. MGMT methylation status was included in both univariate Cox proportional-hazard modelling in addition to multivariate recursive partitioning survival analysis. The tree that minimized the fivefold cross-validated error as well as the most parsimonious tree within one standard error of the overall minimum error were selected for review. Leaves of the resulting trees defined the final risk groups from which the corresponding Kaplan–Meier curves were generated. Median OS times and hazard ratios were generated and compared between risk groups using the Kaplan–Meier method and the Cox proportional hazards model, respectively. The proportional hazards assumption was verified.

## Reporting summary

Further information on research design is available in the Nature Portfolio Reporting Summary linked to this article.

## Data availability

Bulk and single-cell RNA-seq data of primary patient-derived samples reported in this manuscript have been deposited at the NCBI Gene Expression Omnibus under the accession code GSE223065. The publicly available GRCh38 (hg38; https://www.ncbi.nlm.nih.gov/assembly/GCF_000001405.39/) was used in this study. Source data are provided with this paper.

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

**Acknowledgements** This study was supported by NIH grants K08NS110919 and P50CA097257; Robert Wood Johnson Foundation grant 74259; the UCSF LoGlio Collective and Resonance Philanthropies; and U19 CA264339, Tom Paquin Brain Cancer Research Fund to S.L.H.-J.; the Sullivan Brain Cancer Fund to S. Krishna; NIH grants F30CA246808 and T32GM007618 to A.C.; NIH grant K99CA252001 to H.S.V.; NIH grant R01NS100440 to S.N.; NIH grant R00DC013828 to D.B.; NIH grants R01NS092597, DP1NS111132, P50CA165962, R01CA258384, R01CA263500 and U19CA264504, a Robert J. Kleberg Jr and Helen C. Kleberg Foundation grant and funding from Cancer Research UK to M.M.; American Brain Tumour Association grant MSSF1900021 and Lucien Rubinstein Award to N.A.; and the UCSF Physician Scientist Scholar Program, the UCSF Wolfe Meningioma Program Project, NIH grant K08CA212279 to D.R.R. We thank J. Phillips, A. Shai and the staff of the UCSF Brain Tumour Center Biorepository and Pathology Core; the staff within the Biological Imaging Development CoLab (BIDC) at UCSF for confocal microscopy and Imaris software support; T. Ozawa and R. Santos of the UCSF Brain Tumour Center Preclinical Therapeutics Core; N. Jumper for proofreading; and K. Probst for illustrations.

**Author contributions** S. Krishna and S.L.H.-J. designed, conducted and analysed experiments. M.M. contributed to study design, data analysis and manuscript editing. A.C. and D.R. contributed to bulk and single cell transcriptomic analyses. M.B.K. and K. Shamardani contributed to in vivo PDX gabapentin experiment. K. Seo contributed to in vitro organoid experiments. L.N. contributed to electron microscopy data acquisition and analyses. S. Kakaizada and N.A. contributed to language and cognitive assessments. Y.Z., A.M.M., A.L. and A.A. contributed to human survival analyses. G.P. and J.H. contributed to the MEA experiment. J.J.P., C.C., C.G. and R.S. contributed to quantitative imaging analysis. A.E., A.F. and S.N. contributed to MEG data acquisition and quantification. H.S.V. contributed to orthotopic xenografting. E.F.C., B.L., A.G.S.D. and D.B. contributed to intraoperative electrocorticography analysis. S. Krishna and S.L.H.-J. wrote the manuscript. S.L.H.-J. conceived the project and supervised all aspects of the work.

**Competing interests** The authors declare no competing interests.

**Additional information**
**Correspondence and requests for materials** should be addressed to Shawn L. Hervey-Jumper.

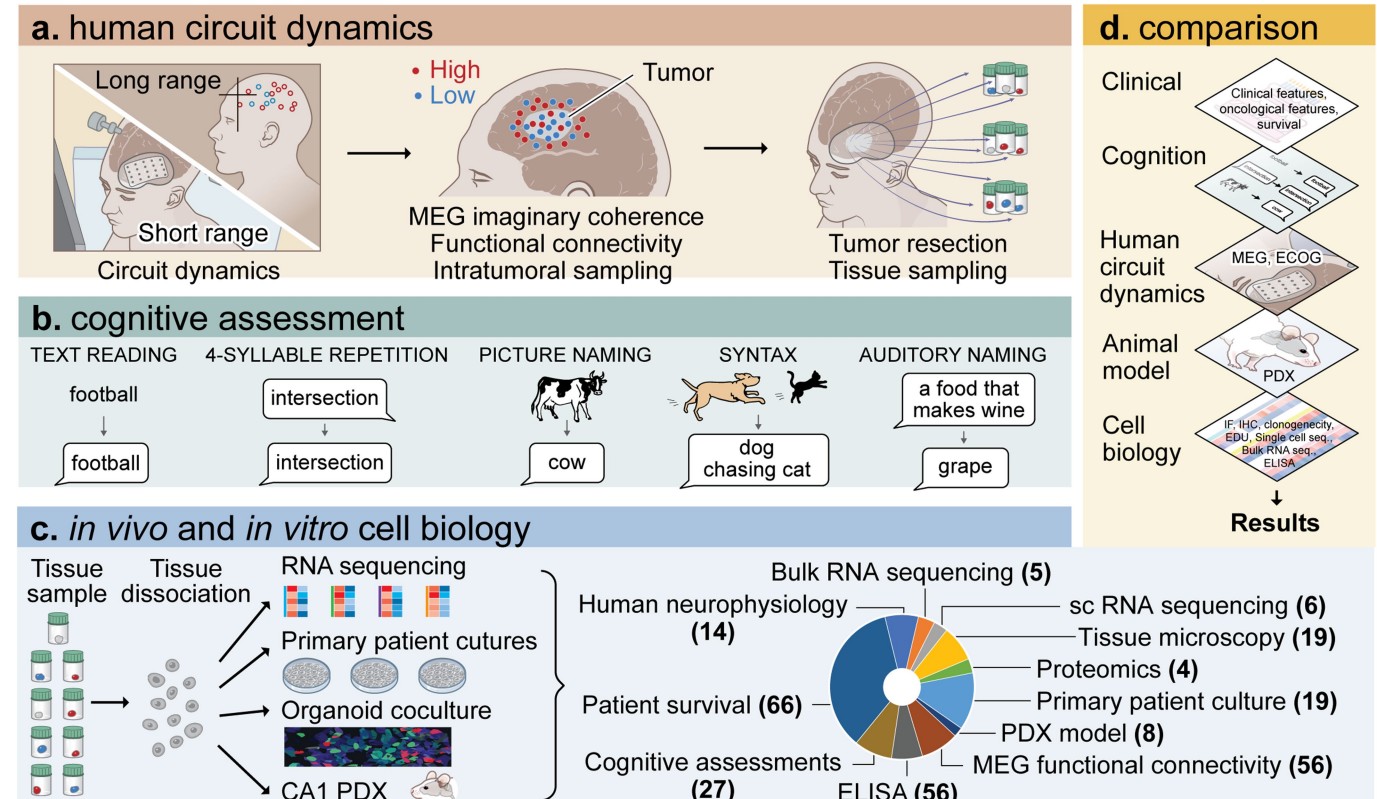

**Extended Data Fig. 1 | Experimental workflow. a**, Schematic of study workflow. In human participants with dominant hemisphere gliomas, we applied subcortical high density electrocorticography during audiovisual speech initiation to assess tumour intrinsic neuronal circuit dynamics. Focusing on glioblastoma, we then assessed long-range functional connectivity using magnetoencephalography (MEG) imaginary coherence. **b**, Extra-operative language assessments were performed for correlation with biological assays.

**c**, Long-range measure of tumour intrinsic functional connectivity identified regions of high and low connectivity for site specific biopsies which were used for *in vivo* and *in vitro* cell biology experiments. **d**, Multiple layered approach including clinical variables, cognition assessments, human and animal model network dynamics, and cell biology experiments serves as a platform for glioma influence on neural circuit dynamics.

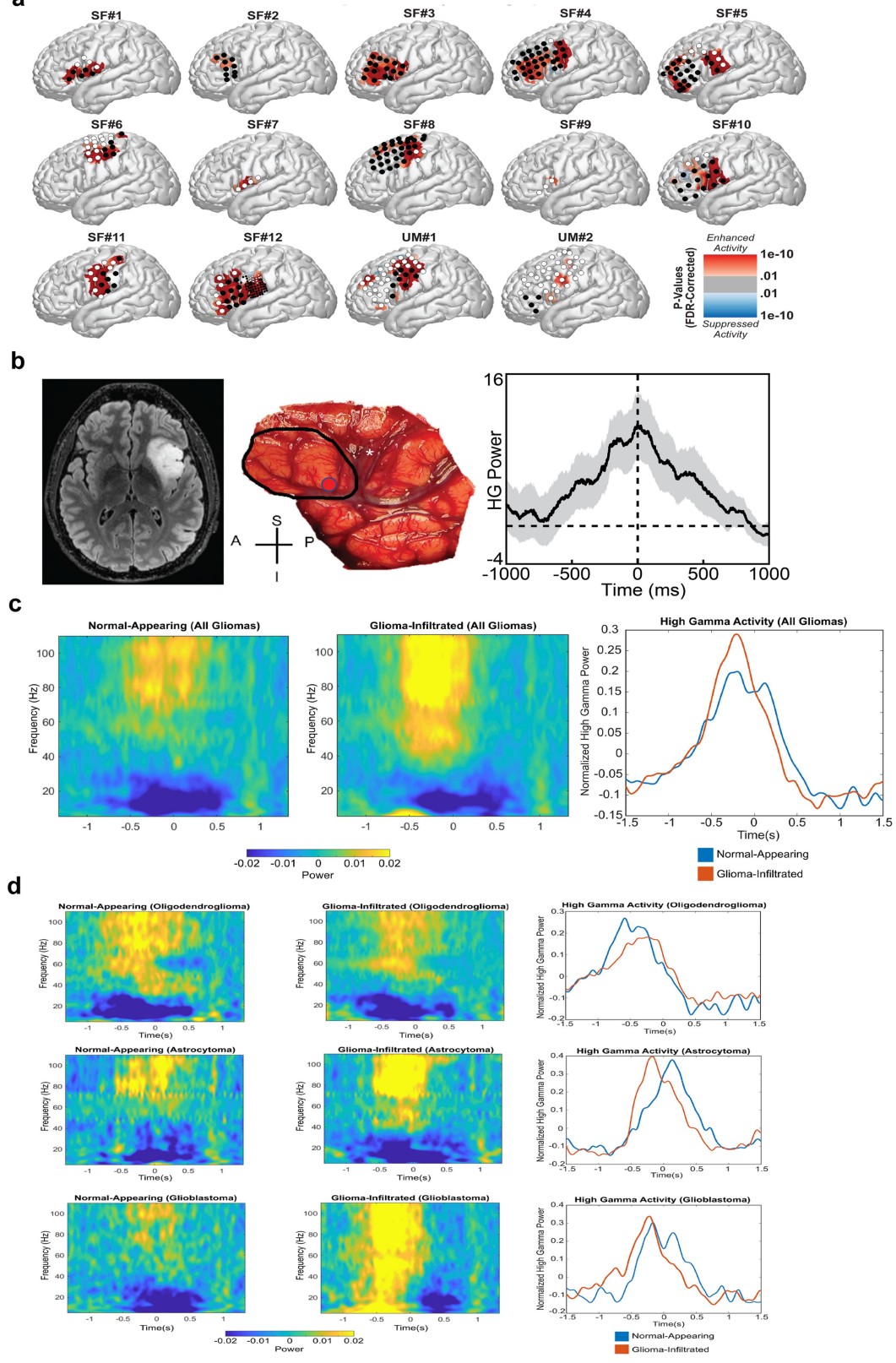

**Extended Data Fig. 2** | See next page for caption.

**Extended Data Fig. 2 | Electrode location and spectral data across cortically infiltrating diffuse glioma. a**, Electrodes overlying normal appearing and glioma-infiltrated regions a cohort of 14 adult patients with cortically projecting glioma in the lateral prefrontal cortex. Electrodes over non-tumour regions are shown in white and those over tumour-infiltrative regions in black. **b**, Positive control conditions included speech initiation responses within non-infiltrated cortex of the left lateral prefrontal cortex (LPFC) for non-cortically projecting glioblastoma. (Left) Axial FLAIR MRI demonstrates tumour location within insular cortex. Hemisphere of language dominance on the left was performed according to study protocol. (Middle) Black outline illustrates LPFC with ECoG recordings obtained from electrode A24, denoted by the red dot. White star represents frontal lobe motor cortex. (Right) Identical to non-tumour comparisons for cortically projecting gliomas, speech responses demonstrate elevate high gamma power (HGp) prior to speech onset consistent with speech motor planning. Dark line and shaded region represent mean and 95% confidence interval, respectively. **c**, Selectivity of maintained tumour intrinsic task-specific cortical responses is identified across diffuse glioma subtypes (adapted from Aabedi et al. 2021). Spectral data show clear separation of frequencies across tumour (glioma-infiltrated) and non-tumour (normal-appearing) electrodes. Group level analysis of participants (n = 12) demonstrates speech initiation responses across WHO 2–4 diffuse glioma. **d**, Glioma subtype-specific speech initiation spectral responses for electrodes above normal-appearing and glioma-infiltrated cortex showing conserved phenotype with task-specific hyperexcitability observed only in participants with glioblastoma. Subtypes: grade 2 and 3 oligodendroglioma (n = 4), grade 2 and 3 astrocytoma (n = 4), and glioblastoma (n = 4). *P* value determined by two-tailed Student's t-test and corrections for multiple comparisons were made using the FDR method (a).

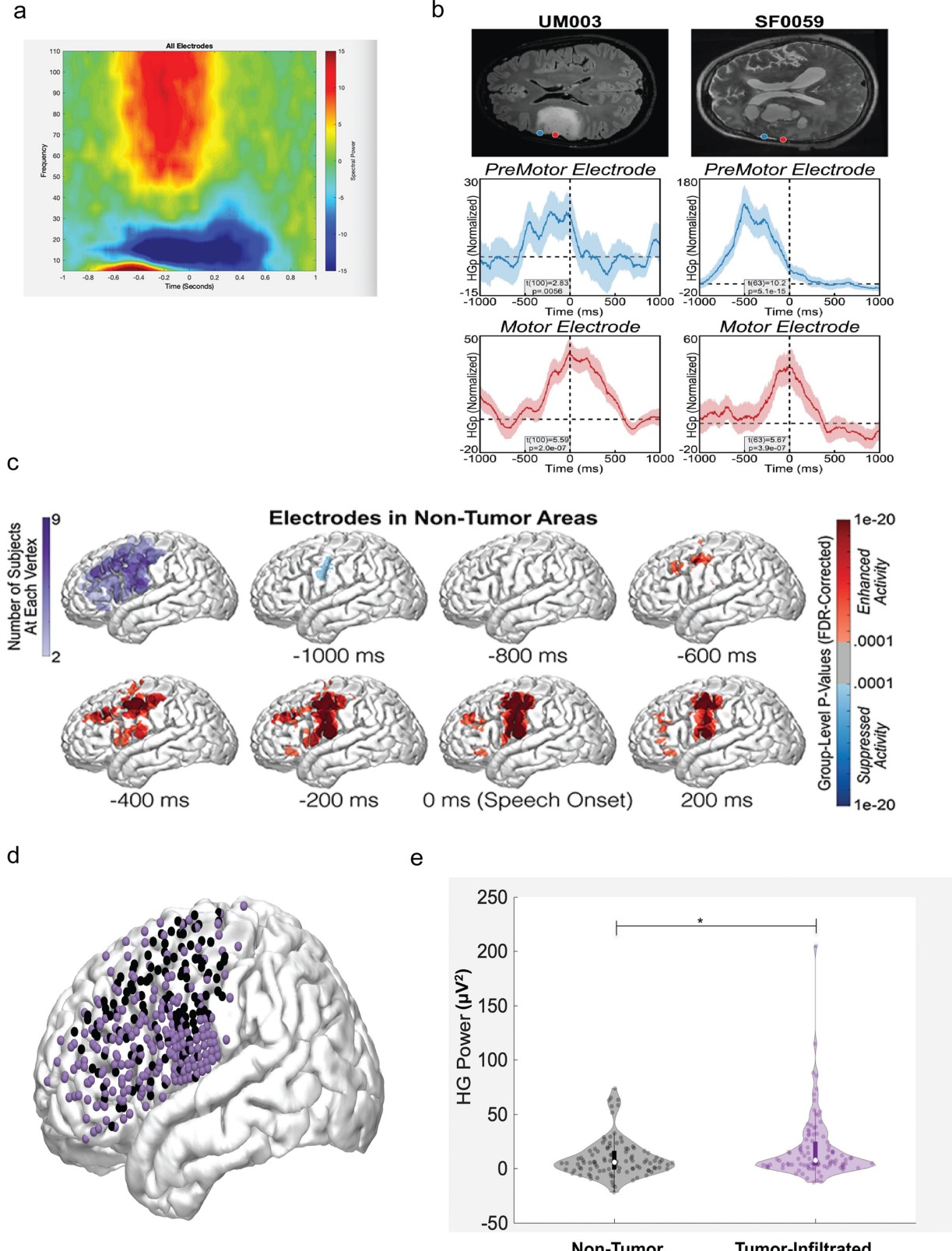

**Extended Data Fig. 3 | Speech initiation neural activity in lateral prefrontal cortex (LPFC). a**, Spectral data of time vs frequency from 100–150 Hz for all tumour and non-tumour electrodes show the expected pattern of HgP increasing above 50 Hz. **b**, High gamma power (HGp) recording from single electrodes overlying tumour-infiltrated regions of brain. Dark line and shaded region represent mean and 95% confidence interval, respectively. **c**, Reconstructed time series of HGp from non-tumour electrodes demonstrating expected spatial and temporal pattern of neural activity within lateral prefrontal cortex.

**d**, **e**, Electrodes matched to anatomical areas across tumour and non-tumour demonstrate hyperexcitability with glioma-infiltrated cortex. Plots are centred on median, with dark bars indicating the first and third quartiles, and whiskers the minimum and maximum points (n = 101 per group, *P* = 0.0163). *P* value determined by two-sided linear mixed-effects model with corrections for multiple comparisons made using the FDR method (b) and two-tailed Student's t-test (e). *P < 0.05.

**a**

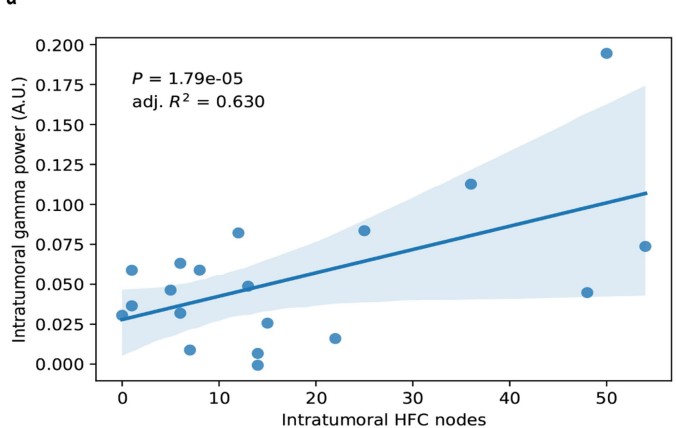

**b**

| RAW COUNTS | | | Fraction of Total (%) | | |
|---|---|---|---|---|---|
| | CE | FLAIR | | CE count | FLAIR count |
| HFC | 26 | 19 | HFC | 57.78 | 42.22 |
| LFC | 26 | 32 | LFC | 44.83 | 55.17 |

**c**

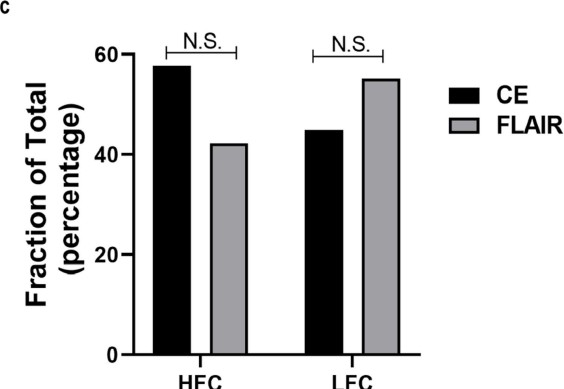

**Extended Data Fig. 4 | Gamma power and tumour-intrinsic connectivity imaging correlations. a**, Linear regression statistics illustrating that gamma power (a measure of neuronal activity) correlates with number of intratumoural high functional connectivity voxels in glioblastoma (n = 18 patients; $P$ = 0.00002). Shaded area represents the 95% confidence interval predicted by the linear regression model. **b**, Sampling of functionally connected intratumoural regions using MEG was performed exclusively in participants with dominant hemisphere glioblastoma at the point of initial diagnosis. Site-directed tissue biopsies from HFC and LFC regions were taken as determined by MRI. Table illustrates site-specific sampling of each annotated specimen as it relates to contrast enhancing (CE) region and FLAIR tumour. Site-specific samples were acquired without regard for whether they originated from enhancing or FLAIR regions. **c**, While samples were not acquired based on whether they originated from contrast enhancing or FLAIR regions, the stereotactic coordinates of each sample were acquired. While 57.78% of HFC samples originated from contrast enhancing regions, this did not reach statistical significance. P = 0.1923 two-sided chi square, P = 0.235 two-sided Fisher's exact test. $P$ value determined by two-sided linear regression analysis (a). NS, not significant.

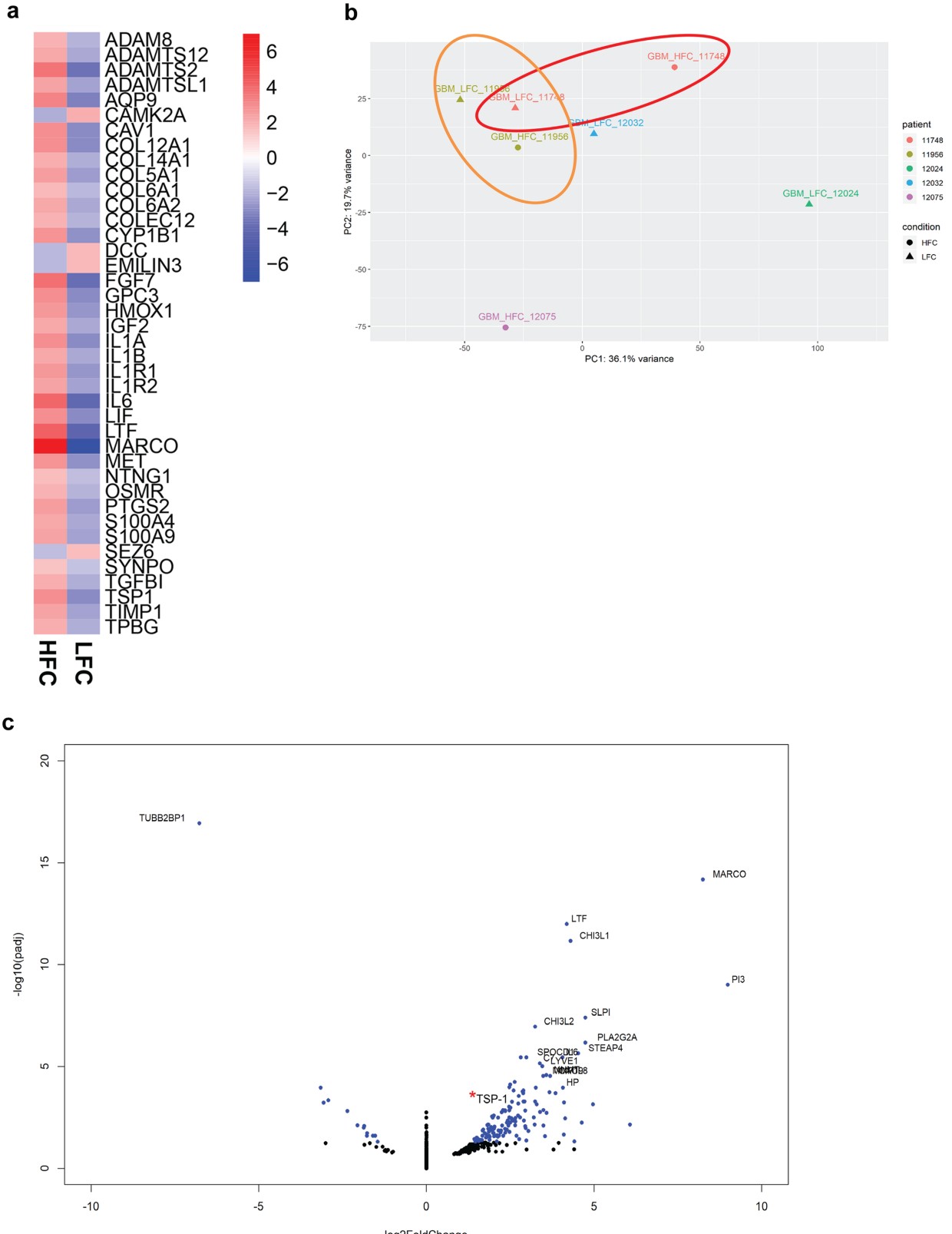

**Extended Data Fig. 5 | Neurogenic gene expression in glioblastoma. a**, Bulk RNA transcriptomic profile of HFC tissues showed a neurogenic signature including elevated (7-fold) expression of thrombospondin-1 (*TSP-1*) (n = 3-4 per group). **b**, Unsupervised principal component (PCA) analysis of bulk RNA sequencing data obtained from glioblastoma primary patient HFC (n = 3) and LFC (n = 4) samples. **c**, Volcano plots of IDH-WT glioblastoma samples revealed 144 differentially expressed genes between HFC and LFC tumour regions. The blue dots represent all differentially expressed genes, where differential expression is defined by the parameters: adjusted p-value < 0.05 and absolute log2fold change > 1. *P* value calculated using two-sided Wald test and adjusted for multiple comparisons with the Benjamini-Hochberg method.

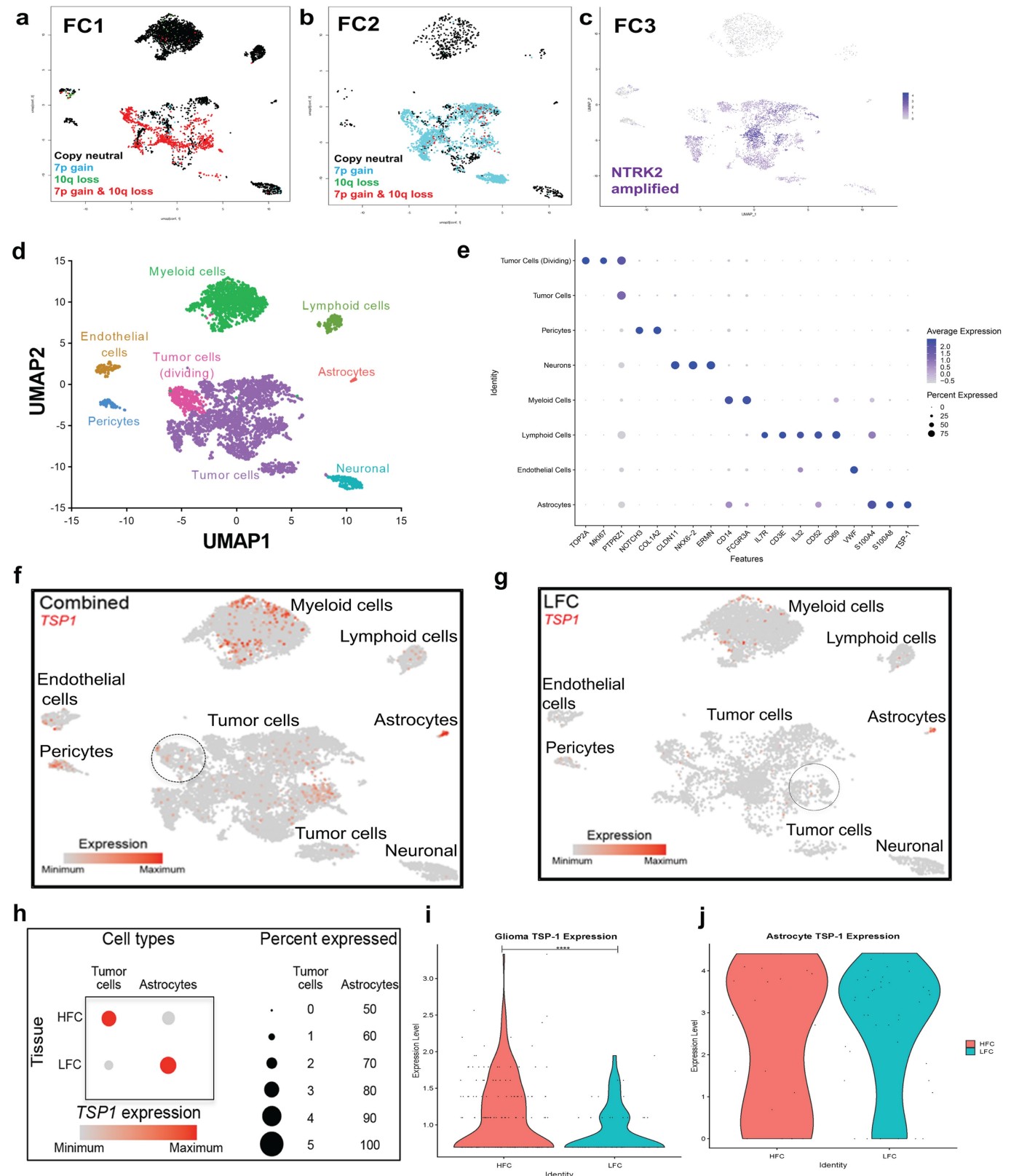

**Extended Data Fig. 6** | See next page for caption.

**Extended Data Fig. 6 | TSP-1 expression in single-cell primary patient-derived glioblastoma. a–c**, Tumour cell validation using copy number variant assessment on three matched pairs of HFC and LFC samples from FC1 (SF#1), FC2 (SF#2), and FC3 (SF#3) glioblastoma patients. Trisomy 7 and monosomy 10 co-occur in most cells in FC1. Trisomy 7 is an early event, while monosomy 10 is a late event in FC2. FC3 patient sample contains no copy number variation but has high level amplification of *NTRK2* gene. **d**, Single-cell RNA transcriptomic profile UMAP confirms distinct cell populations including non-tumour astrocytes and neurons. **e**, Gene enrichment profile used to identify each of the UMAP cell populations. **f, g**, Feature plot for *TSP-1* in combined (HFC + LFC) and LFC (n = 7,065 cells, 3 participants) population; within LFC samples, *TSP-1* expression is primarily from non-tumour astrocytes (suggesting that within low connectivity intratumoural regions, normal astrocytes secrete *TSP-1* to generate connectivity mirroring normal physiology). **h**, Dot plots showing *TSP-1* expression (grey to red scale) and percentage (number of cells expressing the gene) of *TSP-1*-positive cells in tumour cells and non-tumour astrocyte populations in HFC and LFC samples (n = 3 per group). Out of the total HFC tumour cells (n = 5325, 3 patients), 157 cells are *TSP-1* positive accounting for a percentage of 2.95, while only 1.59% (51 cells out of a total of 3212 LFC tumour cells [n = 3 patients]) express *TSP-1*. However, in the non-tumour astrocyte population, the number of *TSP-1*-positive cells are higher in LFC (n = 34 out of a total of 41 astrocytes, accounting for 82.9%) compared to HFC (n = 15 out of a total of 20 astrocytes, accounting for 75%) samples. **i**, Violin plots illustrating significantly increased TSP-1 expression within HFC region glioblastoma cells relative to LFC regions ($P = 1.4*10^{-7}$). **j**, Compared to HFC, slight trend of increased TSP-1 expression within non-tumour astrocytes of LFC population. However, this trend did not reach statistical significance, likely due to the small number of non-tumour astrocytes captured ($P = 0.45$). *P* values determined by two-tailed Student's t-test.

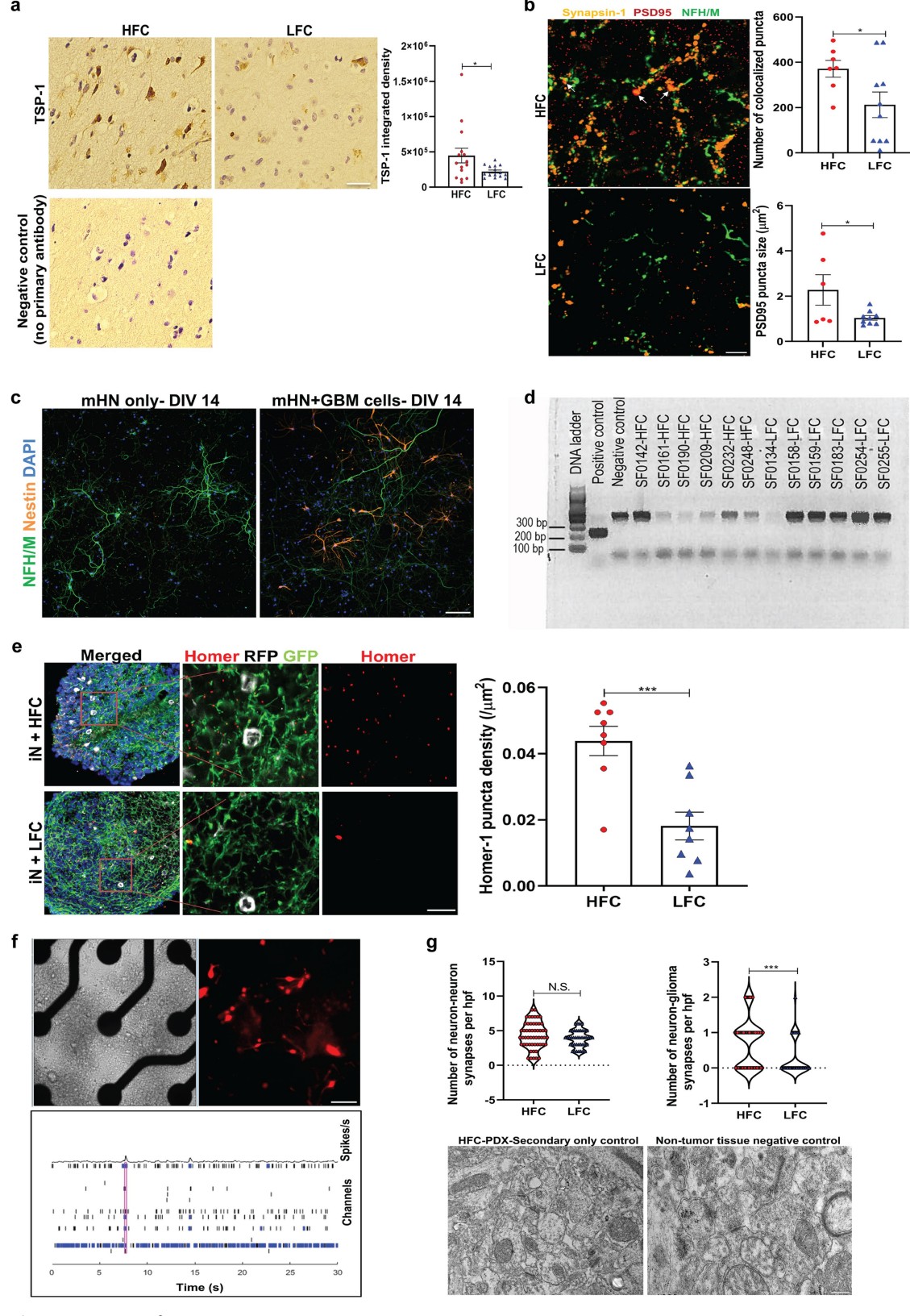

**Extended Data Fig. 7** | See next page for caption.

**Extended Data Fig. 7 | TSP-1 expression, synaptic puncta colocalization in primary patient-derived glioblastoma tissues, neuron organoid-glioblastoma co-culture model and structural synapse formation in patient-derived xenograft models. a**, TSP-1 (immunohistochemistry) expression in arbitrary units (A.U.) in HFC and LFC tissues (n = 16 regions; 3 per group; $P = 0.04$). Scale bar, 50 μm. **b**, Representative confocal images of primary patient-derived HFC and LFC tissues showing regions of synaptic puncta colocalization (white arrows). Orange, synapsin-1 (presynaptic puncta); red, PSD95 (postsynaptic puncta); green, neurofilament (neurons). Scale bar, 15 μm. Quantification of the number of colocalized pre- and postsynaptic puncta (HFC: n = 7; LFC: n = 10 regions, 2 per group; $P = 0.05$). Quantification of postsynaptic PSD95 puncta size (HFC: n = 6; LFC: n = 9, 2 per group; $P = 0.0441$). Scale bar, 15 μm. **c**, Primary patient-derived cultures and mouse hippocampal neuron controls. Neurofilament (heavy and medium chains) and nestin antibodies used as specific markers to label mouse hippocampal neurons and glioblastoma cells, respectively in glioma-neuron co-culture. Left panel: mouse hippocampal neurons alone in culture for 14 days only express neurofilament (green) and not nestin (orange). Right panel: Nestin (orange) expression in GBM cells co-cultured with neurofilament (green) labelled mouse hippocampal neurons for 14 days. Scale bar, 100 μm. **d**, Cell cultures tested for mycoplasma using a commercially available kit (PCR Mycoplasma Test Kit I/C, PromoCell, Heidelberg, Germany) shows absence of a positive band at ~270 bp. Tested primary patient-derived lines shows internal control DNA at ~479 bp indicated a successfully performed PCR. **e**, Neuron organoids (GFP labelled) were generated from an iPSC cell line integrated with doxycycline inducible human NGN2 transgene and co-cultured with RFP labelled HFC and LFC cells (pseudo-coloured white) for two weeks. Quantification of postsynaptic Homer-1 puncta density (calculated by dividing the number of puncta measured with the area of the image field) in 2-week induced neuron (iN) organoid sections (n = 8 per group; $P = 0.0009$). Scale bar, 10 μm. **f**, Multi-electrode array of glioma-neuron co-culture and control conditions. Magnified view of multi-electrode array (MEA), showing RFP-labelled glioblastoma cells in co-culture with neurons (top row). Scale bar, 100 μm. Representative raster plot showing individual spikes/extracellular action potentials (tick mark), bursts (cluster of spikes in blue) and synchronized network bursts (pink) of mouse cortical neuron (DIV 18) only condition (bottom row). The cumulative trace above the raster plots depicts the population spike time histogram indicating the synchronized activity between the different electrodes (network burst). **g**, Structural synapses in primary patient-derived glioblastoma xenografts. Quantification of neuron-to-neuron ($P = 0.1381$) and neuron-glioma synapses ($P = 0.0005$) per high power field (hpf) in HFC and LFC xenografts (top row). Specificity negative controls for immuno-gold labelling (bottom row). (Left) HFC xenograft with secondary antibody only (no primary antibody) control and (right) non-glioma bearing negative control tissue demonstrating few randomly distributed immunogold particles across the tissue specimen. Scale bar, 1000 nm. Data presented as mean ± s.e.m. $P$ values determined by two-tailed Student's t-test. *$P < 0.05$; **$P < 0.01$; ***$P < 0.001$; NS, not significant.

**a**

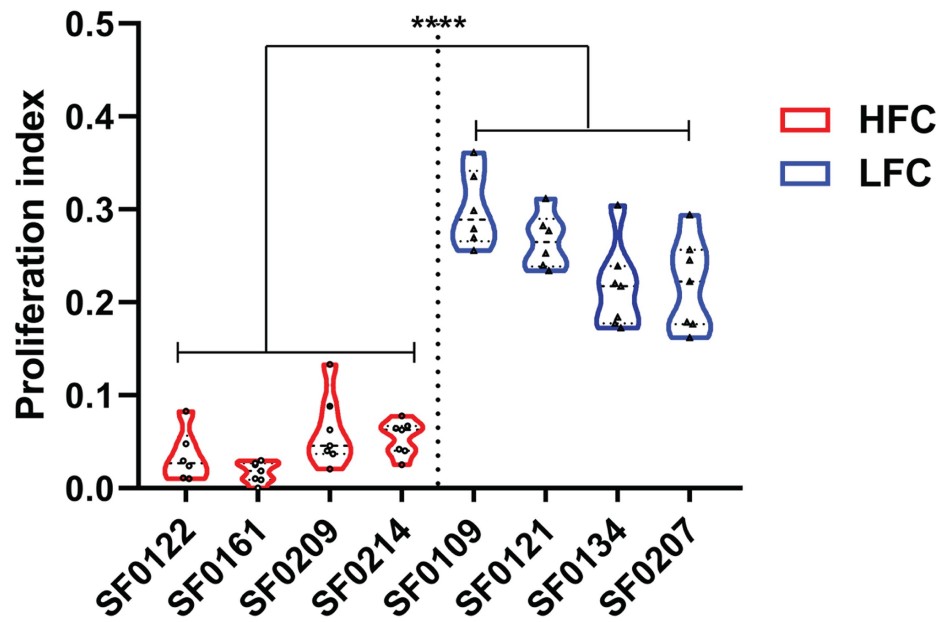

**b**

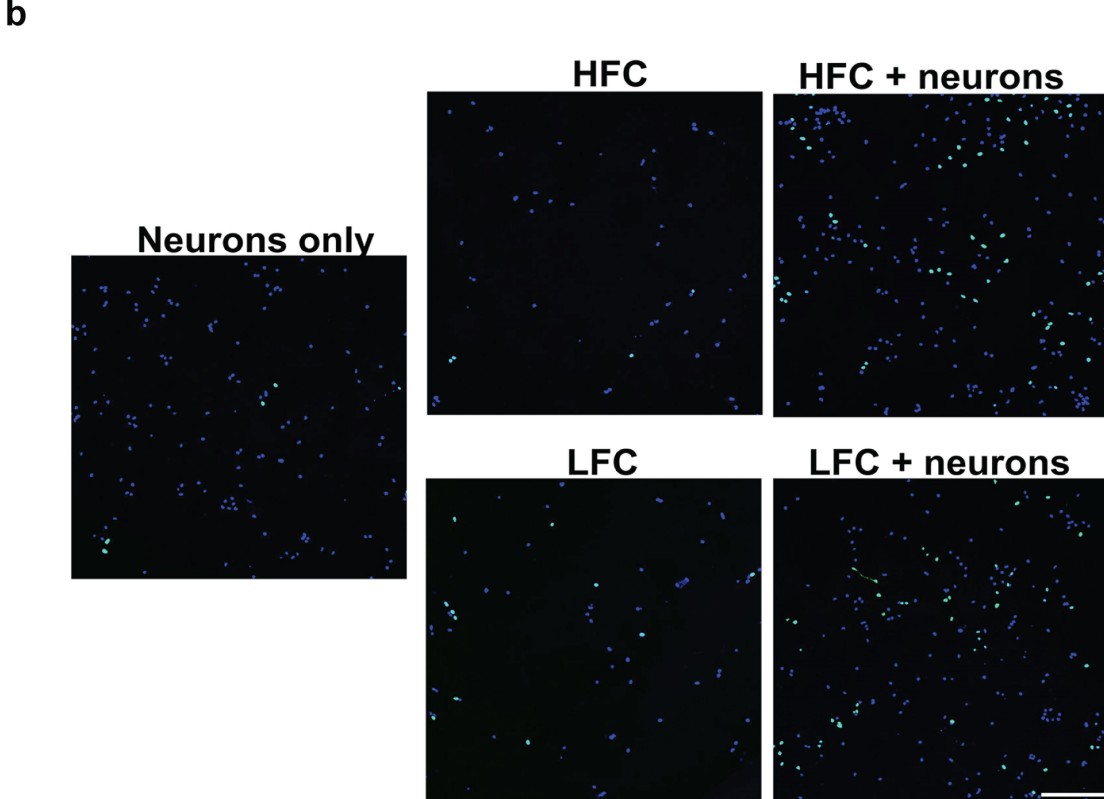

**Extended Data Fig. 8 | Proliferation of primary patient-derived glioblastoma cell monoculture and neuron co-culture conditions.** Primary patient glioblastoma cells from HFC regions illustrate marked increase in proliferation when co-cultured with mouse hippocampal neurons. **a**, Quantification of proliferation indices of HFC (n = 4) and LFC (n = 4) glioma cells alone in culture (in the absence of neurons) from individual patient lines, determined by quantifying the fraction of EdU labelled cells/DAPI labelled cells; P = 0.00005). **b**, Representative confocal images illustrating proliferating HFC and LFC glioma cells (EdU+, green) in the absence or presence of mouse hippocampal neurons (72h co-culture). Scale bar, 100 μm. Data presented as mean ± s.e.m. P value determined by two-tailed Student's t-test. ****P < 0.0001.

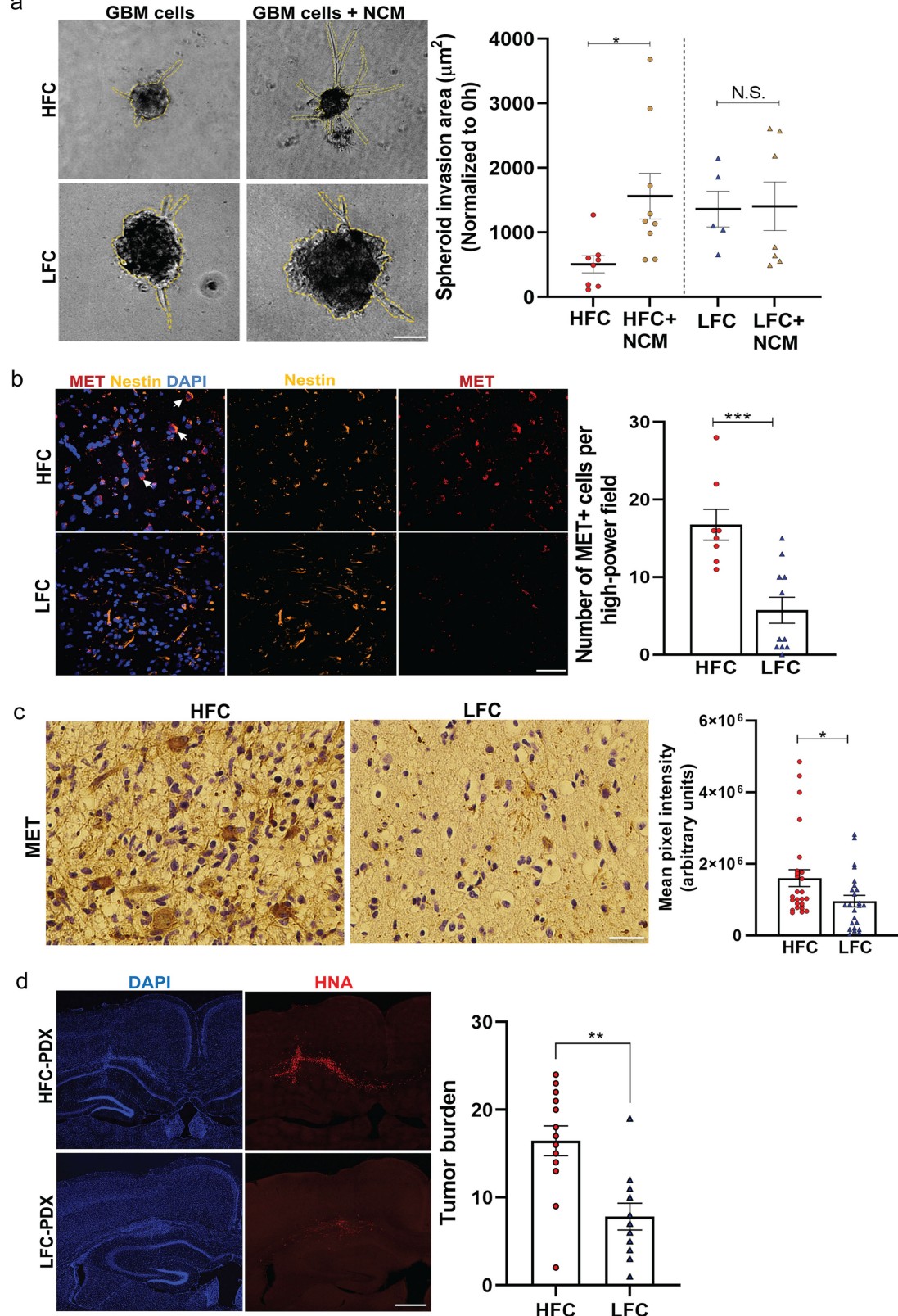

**Extended Data Fig. 9** | See next page for caption.

**Extended Data Fig. 9 | Activity-dependent invasion of TSP-1 positive HFC cells.**
**a**, 3D spheroid invasion assay showing representative micrographs imaged 24 h after addition of invasion matrix. Analysis includes quantification of mean spheroid invasion area normalized for each sample to the initial (0 h) spheroid area (HFC: n = 11; HFC + NCM: n = 16; LFC: n = 5; LFC + NCM: n = 6 spheroids, 1 per group; $P$ = 0.02). Scale bar, 200 µm. Data presented as mean ± s.e.m. $P$ values determined by two-tailed Student's t-test. **b**, Representative confocal images of primary patient-derived HFC and LFC tissues showing MET-positive glioma cells (white arrows). Red, MET; orange, Nestin (HFC/LFC-GBM cells); blue, DAPI. Scale bar, 30 µm. Quantification of MET-positive glioma cells per high-power field (HFC: n = 8; LFC: n = 11, 3 per group; $P$ = 0.0005). **c**, Representative immunohistochemistry images of MET staining in HFC and LFC tissues demonstrate increased tissue level protein expression (HFC: n = 26; LFC: n = 24 regions, 4 per group; $P$ = 0.0329). Scale bar, 50 µm. Data presented as mean ± s.e.m. $P$ values determined by two-tailed Student's t-test. **d**, Representative confocal images showing the diffuse infiltrative pattern of HFC cells in the hippocampus in comparison to the LFC cells. Quantification of tumour burden of HFC and LFC hippocampal xenografts using rank order analysis (HFC: n = 13; LFC: n = 11 regions; $P$ = 0.002). Scale bar, 100 µm. Data presented as mean ± s.e.m. $P$ value determined by two-tailed Mann-Whitney test. *$P$ < 0.05; **$P$ < 0.01; ***$P$ < 0.001; NS, not significant.

**a**

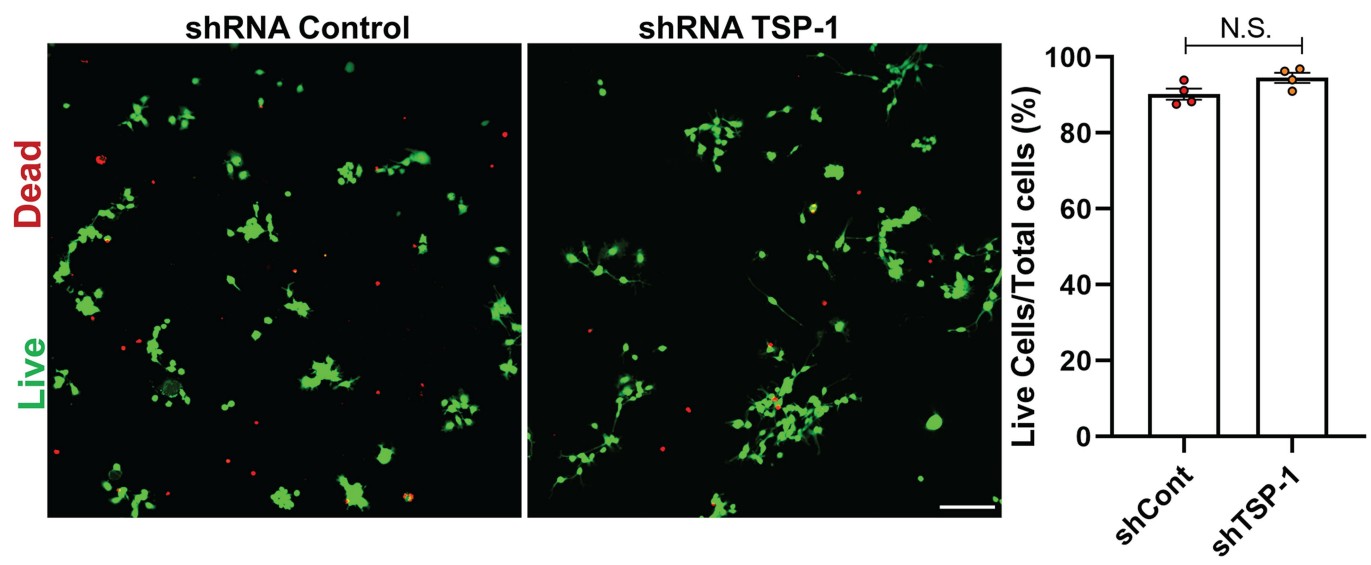

**b**

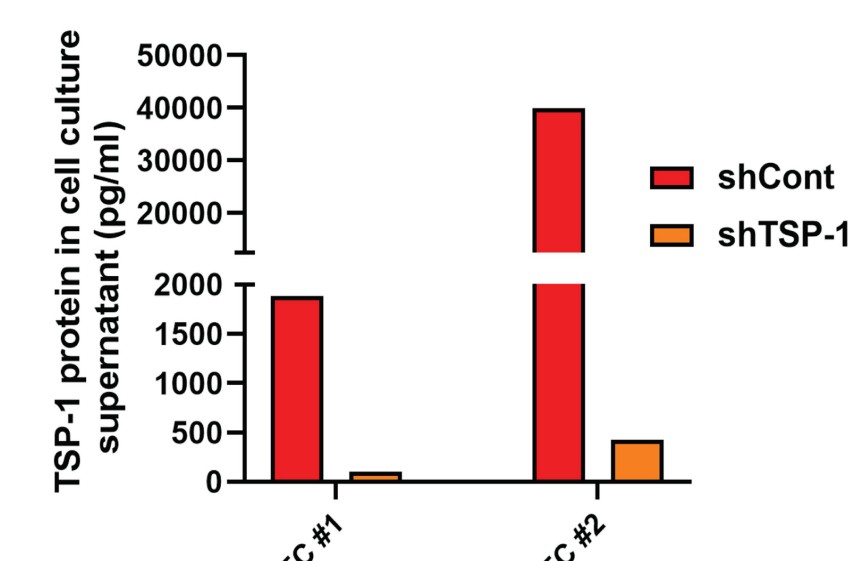

**Extended Data Fig. 10 | Cell viability and TSP-1 knockdown validation.** **a**, Cell viability determined by live/dead cell assay. Representative images illustrating no significant cell death 2 weeks post-transduction of HFC cells with control (shCont) or TSP-1 shRNAs. Live (green) and dead (red) cells were imaged from four random fields per well and were visualized under a fluorescence microscope. The percentage of live cells were calculated as the number of live cells (in green) divided by the total number of cells (green + red) per image field (n = 4 regions per group; $P$ = 0.0720). Scale bar, 100 μm. **b**, ELISA experiments performed on cell culture supernatants demonstrating strong reduction of TSP-1 expression after knockdown of TSP-1 in two different primary patient-derived HFC cell lines compared to the control scramble condition (n = 2 per group). Data presented as mean ± s.e.m. $P$ values determined by two-tailed Student's t-test. N.S., not significant.

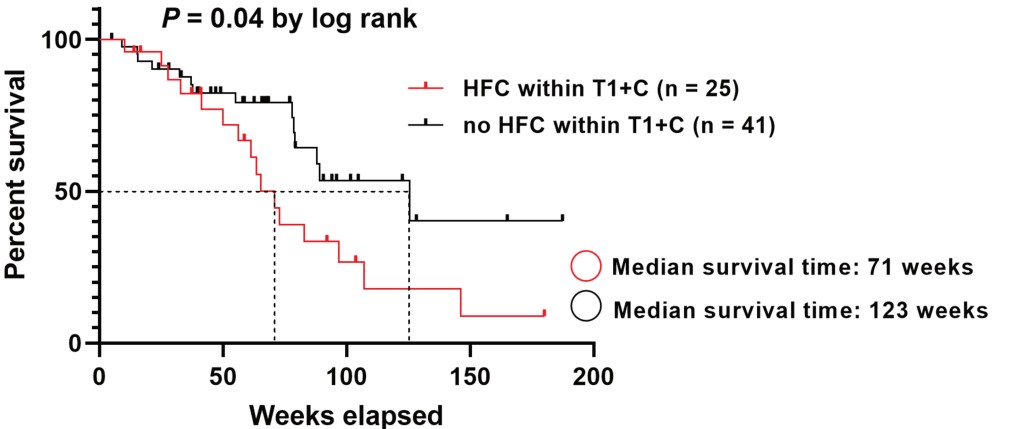

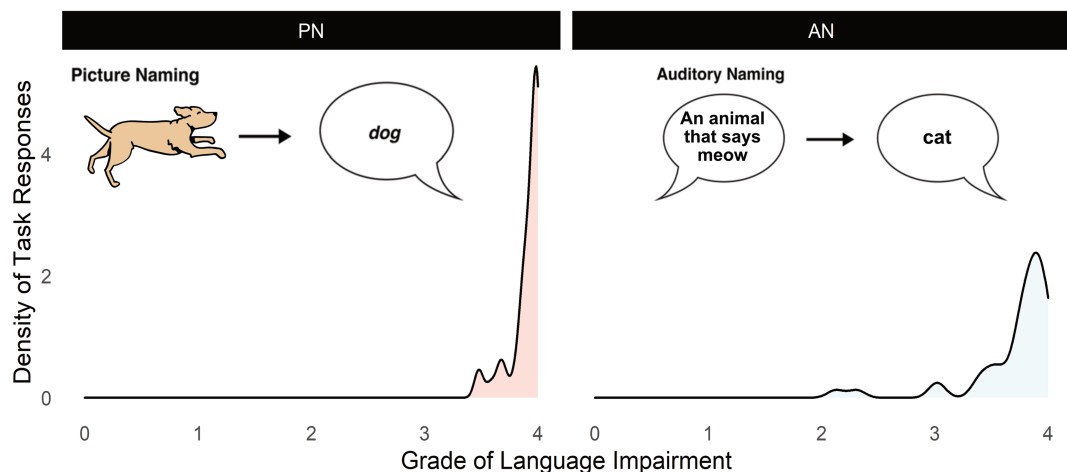

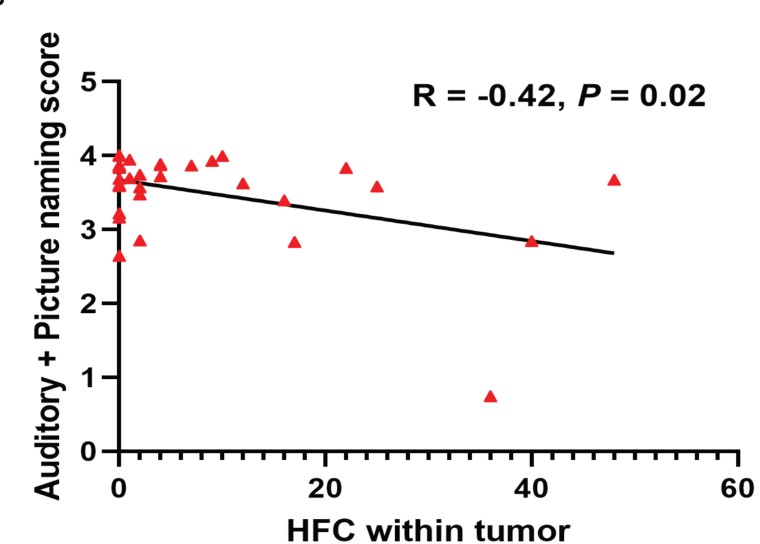

**Extended Data Fig. 11 | Patient survival and language task performance.** **a**, Kaplan-Meier human survival analysis illustrates 71-week overall survival for patients with HFC voxels as determined by contrast-enhanced T1-weighted images as compared to 123-weeks for participants without HFC voxels within their glioblastoma (mean follow-up months 50.5, range 4.9–155.9 months). **b**, Picture and auditory naming language task performance across the study population. **c**, Linear regression statistics illustrating a negative correlation between the number of intratumoural high functional connectivity voxels and baseline auditory and picture naming scores (n = 31, P = 0.0181). P value determined by two-tailed linear regression analysis.

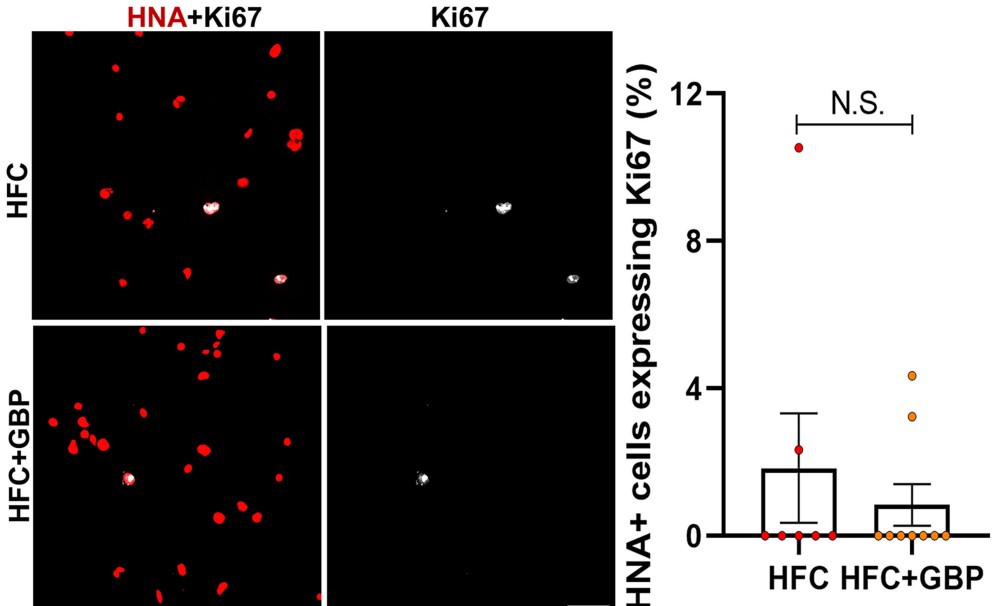

**Extended Data Fig. 12 | Anti-proliferative effects of TSP-1 inhibition in glioblastoma is limited to activity-dependent mechanisms.** Representative confocal images from HFC glioma monoculture showing no significant change in proliferation (as measured by the number of human nuclear antigen (HNA)-positive cells co-labelled with Ki67 divided by the total number of HNA-positive tumour cells counted across all areas quantified) upon pharmacological TSP-1 inhibition using (32 µM) gabapentin (HFC: n = 7; LFC: n = 9 regions, 2 per group; $P = 0.50$). Red, HNA (human nuclei); white, Ki67. Scale bar, 30 µm. Data presented as mean ± s.e.m. $P$ values determined by two-tailed Student's t-test. NS, not significant.

# Reporting Summary

## Statistics

For all statistical analyses, confirm that the following items are present in the figure legend, table legend, main text, or Methods section.

| n/a | Confirmed | |
|---|---|---|
| ☐ | ☒ | The exact sample size (*n*) for each experimental group/condition, given as a discrete number and unit of measurement |
| ☐ | ☒ | A statement on whether measurements were taken from distinct samples or whether the same sample was measured repeatedly |
| ☐ | ☒ | The statistical test(s) used AND whether they are one- or two-sided *Only common tests should be described solely by name; describe more complex techniques in the Methods section.* |
| ☐ | ☒ | A description of all covariates tested |
| ☐ | ☒ | A description of any assumptions or corrections, such as tests of normality and adjustment for multiple comparisons |
| ☐ | ☒ | A full description of the statistical parameters including central tendency (e.g. means) or other basic estimates (e.g. regression coefficient) AND variation (e.g. standard deviation) or associated estimates of uncertainty (e.g. confidence intervals) |
| ☐ | ☒ | For null hypothesis testing, the test statistic (e.g. *F*, *t*, *r*) with confidence intervals, effect sizes, degrees of freedom and *P* value noted *Give P values as exact values whenever suitable.* |
| ☒ | ☐ | For Bayesian analysis, information on the choice of priors and Markov chain Monte Carlo settings |
| ☒ | ☐ | For hierarchical and complex designs, identification of the appropriate level for tests and full reporting of outcomes |
| ☒ | ☐ | Estimates of effect sizes (e.g. Cohen's *d*, Pearson's *r*), indicating how they were calculated |

*Our web collection on statistics for biologists contains articles on many of the points above.*

## Software and code

Policy information about availability of computer code

| Data collection | Magnetoencephalography (MEG) recordings were analyzed using NUTMEG software suite version 4. For Bulk RNA-seq analysis, reads were trimmed with Trimmomatic and were mapped to the human reference genome GRCh38 using HISAT2 (v 2.1.0) and exon level count data were extracted from the mapped HISAT2 output using featureCounts. Differential gene expression analysis and log2 fold change calculation was performed using DESeq2. Confocal images were acquired using Nikon C2 confocal microscope and analyzed using Imaris Software (Imaris 9.2.1, Bitplane). Quantification of tumor cell burden was performed using a Zeiss LSM800 scanning confocal microscope and Zen 2011 imaging software (Carl Zeiss Inc.). Pre-operative and post-operative tumor volumes were quantified using BrainLab Smartbrush software (v 2.6) (Brainlab, Munich, Germany). Linear mixed effects modeling was used to perform statistical comparisons with repeated measures via the nlme package in R version 3.1-161 (https://cran.r-project.org/web/packages/nlme/citation.html). Multielectrode array (MEA) was collected using Axion integarted studio (AxIS) version 3.5.2 software. |
|---|---|
| Data analysis | Statistical analyses were conducted using Prism v8.0 (GraphPad software). Seurat R v3.0.1 was used for QC, analysis, and exploration of single-cell RNA-seq data. Confocal image analyses was done using Imaris 9.2.1 and Fiji ImageJ v2.0. Axion integarted studio (AxIS) version 3.5.2 software, Neural Metric Tool v1.2.3 software (Axion Biosystems) and Statistics Compiler function in AxIS was used for MEA analysis. Recursive partitioning analyses for survival data was performed using partDSA algorithm version 0.9.14. |

For manuscripts utilizing custom algorithms or software that are central to the research but not yet described in published literature, software must be made available to editors and reviewers. We strongly encourage code deposition in a community repository (e.g. GitHub). See the Nature Portfolio guidelines for submitting code & software for further information.

# Data

Policy information about <u>availability of data</u>

All manuscripts must include a <u>data availability statement</u>. This statement should provide the following information, where applicable:

- Accession codes, unique identifiers, or web links for publicly available datasets
- A description of any restrictions on data availability
- For clinical datasets or third party data, please ensure that the statement adheres to our <u>policy</u>

Bulk and single-cell RNA sequencing data of primary patient-derived samples reported in this manuscript are deposited in the NCBI Gene Expression Omnibus under the accession GSE223065. The publicly available GRCh38 (hg38, https://www.ncbi.nlm.nih.gov/assembly/GCF_000001405.39/) was used in this study. This manuscript contains no custom code or mathematical algorithms. All unique materials such as patient-derived cell cultures are available and can be obtained by contacting the corresponding author and with a standard MTA with University of California, San Francisco. The remaining data are available within the Article, and from Supplementary Information.

# Human research participants

Policy information about <u>studies involving human research participants and Sex and Gender in Research.</u>

| | |
|---|---|
| Reporting on sex and gender | The human subject data used in this work are all from adult males/females diagnosed with high-grade glioblastoma. Our findings apply to both sexes. A detailed information of patient's sex and age used in this study is provided in Extended Tables 2 and 5. Sex of patients was determined based on self reporting. We included sex as one of the prognostic variables besides other factors, such as age at diagnosis, tumor location, chemotherapy and radiotherapy to identify clinically relevant survival risk groups in a multivariate setting. |
| Population characteristics | The human subject data used in this work were all from adult males/females diagnosed with high-grade glioblastoma. Patient sample used for each experiment and clinical information of the patients used for human survival analysis is available in Extended Data Tables 1, 2 and 5. |
| Recruitment | All study participants were individuals seeking care for presumed diffuse glioma at University of California San Francisco. Each participant in this study was recruited from a prospective registry of adults aged 18–85 with newly diagnosed frontal, temporal, and parietal high-grade glioma. Inclusion criteria was patients with suspected brain tumor on magnetic resonance imaging (MRI). Exclusion criteria was any vulnerable population: children (age < 18). All human electrocorticography data was obtained during lexical retrieval language tasks from adult awake patients undergoing intraoperative brain mapping for surgical resection. Subjects with tumors projecting to the cortical surface as determined by absence of FLAIR or T1 post gadolinium enhancement were selected for analysis. All human magnetoencephalography recordings were obtained during resting state from adult patients aged 18–85 with newly diagnosed frontal, temporal, and parietal gliomas. Patients were recruited by a brain tumor center clinical research coordinator who was not involved in clinical patient care in order to limit the potential for enrollment bias. |
| Ethics oversight | This study complied with all relevant ethical regulations and was approved by the University of California, San Francisco (UCSF) institutional review board for human research (UCSF CHR 17-23215). |

Note that full information on the approval of the study protocol must also be provided in the manuscript.

# Field-specific reporting

Please select the one below that is the best fit for your research. If you are not sure, read the appropriate sections before making your selection.

☒ Life sciences　　　☐ Behavioural & social sciences　　　☐ Ecological, evolutionary & environmental sciences

For a reference copy of the document with all sections, see nature.com/documents/nr-reporting-summary-flat.pdf

# Life sciences study design

All studies must disclose on these points even when the disclosure is negative.

| | |
|---|---|
| Sample size | Patients presented for resection of glioma who gave consent for tumor sampling for research were included in the study, which was approved by the institutional review board (17-23215). No sample size calculation was performed, and all samples from patients meeting inclusion and exclusion criteria were included. Sample selection criteria are detailed in the Methods section. |
| Data exclusions | No data were excluded from the analyses. |
| Replication | All experiments were performed at least in triplicates and measurements were reproducible with biological replicates performed on separate cohort of patient samples/animals/cells. |
| Randomization | Primary Patient-derived tumor tissue biopsies and cells cultured from the tumor tissues were allocated based on imaging annotation for high |

| | |
|---|---|
| Randomization | functional connectivity (HFC) or low functional connectivity (LFC) experimental groups based on the mean imaginary coherence (IC) between the index MEG voxel, and the rest of the brain, referenced to its contralesional pair. All animals xenografted with individual cell lines used for immuno-electron microscopy and survival experiments were analyzed in the same way- no randomization was necessary. For pharmacological study, mice xenografted with HFC cells were randomized and intraperitoneally treated with gabapentin or corresponding vehicle. |
| Blinding | Investigators were blinded to the study groups being analyzed. |

# Reporting for specific materials, systems and methods

We require information from authors about some types of materials, experimental systems and methods used in many studies. Here, indicate whether each material, system or method listed is relevant to your study. If you are not sure if a list item applies to your research, read the appropriate section before selecting a response.

## Materials & experimental systems

| n/a | Involved in the study |
|---|---|
| ☐ | ☒ Antibodies |
| ☐ | ☒ Eukaryotic cell lines |
| ☒ | ☐ Palaeontology and archaeology |
| ☐ | ☒ Animals and other organisms |
| ☐ | ☒ Clinical data |
| ☒ | ☐ Dual use research of concern |

## Methods

| n/a | Involved in the study |
|---|---|
| ☒ | ☐ ChIP-seq |
| ☒ | ☐ Flow cytometry |
| ☐ | ☒ MRI-based neuroimaging |

## Antibodies

| | |
|---|---|
| Antibodies used | Primary antibodies used in immunohistochemistry: chicken anti-neurofilament-H (1:1000;  #NFH Aves Labs; Lot # NFH877982), chicken anti-neurofilament-M (1:1000;  #NFM Aves Labs; NFM4907982), mouse anti-neurofilament antibody (1:1000; #NB300-134 Novus Biologicals; Lot # 021 521), rabbit anti-homer-1 (1:500; #PA5-21487 Pierce; Lot#TK2679033A) mouse anti-nestin (1:500; #Ab22035 Abcam; Lot#GR3276045-7), rabbit anti-synapsin 1 (1:1000; #AB1543 EMD Millipore; Lot#3051501), mouse anti-PSD95 clone  K28/43 (1:100; #75-028 Neuromab; Lot# 455.7JD.22G), mouse anti-TSP-1 (1:20; #MA5-13398 Invitrogen),  rabbit anti-TSP-1 (1:50, #Ab85762 Abcam; GR3279809-1) rabbit anti-MET (1:100; #Ab51067 Abcam; Lot#GR261314-23), rabbit anti-Ki67 (1:100; #Ab15580 Abcam; Lot#GR3293864-1), mouse anti-human nuclei clone 235-1 (HNA, 1:100, Millipore), chicken anti-MAP2 (1:500; #Ab15452 Abcam).<br><br>For secondary antibodies: Alexa 488 goat anti-chicken IgG (#A11039; Lot#1937504), Alexa 488 goat anti-rabbit IgG (#A11034; Lot#1971418), Alexa 568 goat anti-rabbit IgG (#A11036; Lot#1924788), Alexa 568 goat anti-mouse IgG (#A11004; Lot#1906485), Alexa 647 goat anti-rabbit IgG (#A21245; Lot#2051068) all used at 1:500 (Invitrogen).<br><br>Primary antibody used in immuno-electron microscopy: goat anti-RFP (1: 300; #ABIN6254205 Antibodies-online Inc.; Lot#0040180316) and secondary antibody (1:10; #15796 TED Pella;  Lot#008330). |
| Validation | All the antibodies used in the study were bought from commercial vendors and were validated by the manufacturers, and/or in other studies:<br>Neurofilament (mouse, Novus Biological, NB300-134): Sikora J et al. X-linked Christianson syndrome: heterozygous female Slc9a6 knockout mice develop mosaic neuropathological changes and related behavioral abnormalities. Dis Model Mech. 2015. Validated in IHC by provider.<br><br>Homer (Pierce, PA5-21487): Gresa-Arribas N et al. Human neurexin-3α antibodies associate with encephalitis and alter synapse development. Neurology. 2016 Jun 14;86(24):2235-42. Validated in ICC/IF by provider.<br><br>Synapsin-1 (EMD Millipore, AB1543): Lin et al. Identification of diverse astrocyte populations and their malignant analogs. Nat Neurosci. 2017 Mar;20(3):396-405. Validated in ICC/IF by provider.<br><br>PSD95 (Neuromab, 75-028): Lin et al. Identification of diverse astrocyte populations and their malignant analogs. Nat Neurosci. 2017 Mar;20(3):396-405. Validated in ICC/IF by provider.<br><br>TSP-1 (Invitrogen, MA5-13398): Delaunay K et al. Meteorin Is a Novel Therapeutic Target for Wet Age-Related Macular Degeneration. J Clin Med. 2021 Jul 2;10(13):2973. Validated in IHC and IF by provider.<br><br>TSP-1 (Abcam, Ab85762): Zhang Y et al. Role of Elevated Thrombospondin-1 in Kainic Acid-Induced Status Epilepticus. Neurosci Bull 36:263-276 (2020). Validated in IHC-P, and ICC/IF by provider.<br><br>MET (Abcam, Ab51067): Hu H et al. Mutational Landscape of Secondary Glioblastoma Guides MET-Targeted Trial in Brain Tumor. Cell 175:1665-1678.e18 (2018). Validated in IHC-P by provider.<br><br>MAP2 (Abcam, Ab15452): Zabolocki M   et al. BrainPhys neuronal medium optimized for imaging and optogenetics in vitro. Nat Commun 11:5550 (2020). Validated in ICC by provider. |

RFP (Antibodies-online Inc., ABIN6254205): Semerci F et al. Lunatic fringe-mediated Notch signaling regulates adult hippocampal neural stem cell maintenance. Elife. 2017 Jul 12;6: e24660. Validated in IF and IHC-P by provider.

The chicken anti-neurofilament (M+H), mouse anti-nestin, rabbit anti-Ki67, and mouse anti-human nuclei used for immunohisto-chemical and immunocytochemistry analysis were all according to PMID: 31534222.

# Eukaryotic cell lines

Policy information about cell lines and Sex and Gender in Research

| | |
|---|---|
| Cell line source(s) | The eukaryotic cell lines SF#29, 31, 32, 33, 34, 35, 36, 44, 45,49, 51, 53, 54, 56, 57, 59, 60, 62 and 63 derived from primary patient-derived high-grade gliomas were generated in the Shawn Hervey-Jumper lab from site-directed biopsies (HFC/LFC), and referenced in the Supplementary Table 1 of the manuscript. |
| Authentication | Short Tandem Repeat (STR) fingerprinting is performed every 3 months on all cell cultures to ensure authenticity. |
| Mycoplasma contamination | All cell cultures are routinely tested for mycoplasma contamination and all cultures were tested negative. |
| Commonly misidentified lines (See ICLAC register) | No commonly misidentified lines were used. |

# Animals and other research organisms

Policy information about studies involving animals; ARRIVE guidelines recommended for reporting animal research, and Sex and Gender in Research

| | |
|---|---|
| Laboratory animals | NOD-SCID-IL2R gamma chain-deficient (NSG) and female athymic mice were used between 4-12 weeks of age. Mouse housing conditions have been described in the manuscript. |
| Wild animals | No wild animals were used. |
| Reporting on sex | Both male and female NOD-SCID-IL2R gamma chain-deficient (NSG) and female athymic mice were used between 4-12 weeks of age. |
| Field-collected samples | No field-collected samples were used. |
| Ethics oversight | Study was approved by UCSF Institutional Animal Care and Use Committee (AN192389-01G). |

Note that full information on the approval of the study protocol must also be provided in the manuscript.

# Clinical data

Policy information about clinical studies
All manuscripts should comply with the ICMJE guidelines for publication of clinical research and a completed CONSORT checklist must be included with all submissions.

| | |
|---|---|
| Clinical trial registration | *Provide the trial registration number from ClinicalTrials.gov or an equivalent agency.* |
| Study protocol | Each participant in the study was recruited from a prospective registry of adults aged 18–85 with newly diagnosed frontal, temporal, and parietal IDH-wild type high-grade gliomas with detailed language assessments and baseline MEG recordings. Inclusionary criteria included the following: native English-speaking, between the ages of 18- 85, and no prior history of psychiatric illness, neurologic illness or drug or alcohol abuse. All human electrocorticography data were obtained during lexical retrieval language tasks from 14 adult awake patients undergoing intraoperative brain mapping for surgical resection. Tumors from 8 patients were used for RNA-sequencing experiments. Site-directed tumor biopsies from 19 patients were used for immunofluorescence/immunohistochemistry analysis and 24 patients were used for immunocytochemistry and cell-based functional assays. Tumors from 8 patients were used for mouse xenograft experiments. All participants provided written informed consent to participate in this study, which was approved by the University of California, San Francisco (UCSF) institutional review board (IRB) for human research (UCSF CC-171027, CHR 17-23215) and performed in accordance with the Declaration of Helsinki. |
| Data collection | Language assessments including picture naming, text reading, auditory naming, syntax, and 4 syllable repetition are collected by a rained clinical research coordinator at the time of initial diagnosis as well as during the intra operative setting during standard of care clinical protocol. |
| Outcomes | Primary outcomes include auditory and picture naming task performance, scored according to Wilson et al (Wilson, S. M., Eriksson, D. K., Schneck, S. M., & Lucanie, J. M. (2018). A quick aphasia battery for efficient, reliable, and multidimensional assessment of language function. PLoS ONE, 13(2), e0192773. http://doi.org/10.1371/journal.pone.0192773) |

# Magnetic resonance imaging

## Experimental design

| | |
|---|---|
| Design type | Resting state |

| Design specifications | An artifact-free, 1-minute epoch was analyzed using the NUTMEG software suite (UCSF Biomagnetic Imaging Laboratory) to reconstruct whole-brain oscillatory activity from MEG sensors so as to construct functional connectivity (imaginary coherence) metrics. |
|---|---|
| Behavioral performance measures | MRI/MEG was aquired as a means of measuring imaginary coherence functional connectivity. There were no behavioral outcomes assessed in the MRI portion of this study. |

## Acquisition

| Imaging type(s) | Structural MRI. Imaging Acquisition and MEG Data Analysis High-resolution MRI was performed using a 3-T unit to provide anatomical detail. |
|---|---|
| Field strength | 3T |
| Sequence & imaging parameters | The protocol included the following sequences: 1) a T1-weighted, 3D, spoiled gradient–recalled acquisition in steady-state sequence, with TR 6–9 msec, TE 2–3 msec, and flip angle 12°–15°; and 2) a T2-weighted, 3D, fast spin echo sequence, with TR 2000–3800 msec and TE 87–159 msec. Both sequences had a slice thickness between 1 and 1.5 mm, an acquisition matrix from 256 × 256 to 288 × 288, contained between 114 and 428 slices. Patients were lying awake with their eyes closed in a magnetically shielded room while a 275-channel whole-head CTF Omega 2000 system (CTF Systems, Inc.) captured their continuous resting-state MEG using a sampling rate of 1200 Hz. The locations of the MEG coils were triangulated at the beginning and end of the recording run and later coregistered to a structural MR image to generate the head shape. |
| Area of acquisition | Whole brain |
| Diffusion MRI | ☐ Used   ☒ Not used |

## Preprocessing

| Preprocessing software | Data were analyzed using the NUTMEG software suite. |
|---|---|
| Normalization | Normalization was visualized using SPM and checked by eye for proper normalization. |
| Normalization template | MNI |
| Noise and artifact removal | The time series within each voxel was then bandpass filtered for the alpha band (1–20 Hz) and reconstructed in source space using a minimum-variance adaptive spatial filtering technique. |
| Volume censoring | 60 second noise free periods were chosen for analysis. |

## Statistical modeling & inference

| Model type and settings | Functional connectivity estimates were calculated using IC, a technique known to reduce overestimation biases in MEG data generated from common references, cross-talk, and volume conduction. |
|---|---|
| Effect(s) tested | The alpha band was selected because it was the most consistently identified peak in the power spectra from this sampling window in our patient series. The imaginary coherence technique was used to estimate functional connectivity because it has been previously been shown to reduce overestimation biases in MEG data. |
| Specify type of analysis: | ☒ Whole brain   ☐ ROI-based   ☐ Both |
| Statistic type for inference (See Eklund et al. 2016) | A univariate regression model was generated to analyze each of the independent variables to determine significance. A significance level of $p < .05$ was used. Statistical testing was conducted |
| Correction | FDR |

## Models & analysis

| n/a | Involved in the study |
|---|---|
| ☐ | ☒ Functional and/or effective connectivity |
| ☒ | ☐ Graph analysis |
| ☐ | ☒ Multivariate modeling or predictive analysis |

| Functional and/or effective connectivity | Imaginary Coherence |
|---|---|
| Multivariate modeling and predictive analysis | To identify clinically relevant survival risk groups in a multivariate setting, we employed recursive partitioning analyses for survival data via the partDSA algorithm. Survival trees use recursive partitioning to divide patients into different risk groups. The Brier score was the chosen loss function for splitting and pruning. Such methods are non-parametric and, therefore, do not require the proportional hazards assumption. All known prognostic variables were included in the trees, including age at diagnosis, sex, tumor location, chemotherapy, radiotherapy, the presence of functional connectivity within the tumor, pre- and post- |

operative tumor volume, and EOR. The tree that minimized the five-fold cross-validated error as well as the most parsimonious tree within one standard error of the overall minimum error were selected for review. Leaves of the resulting trees defined the final risk groups from which the corresponding Kaplan-Meier curves were generated. Median OS times and hazard ratios were generated and compared between risk groups using the Kaplan-Meier method and Cox proportional hazards model, respectively. The proportional hazards assumption was verified.

