## [Peer Review File · Nature]

Manuscript Title: Glioblastoma remodeling of human neural circuits decreases survival

Reviewer Comments & Author Rebuttals

Reviewer Reports on the Initial Version:

Referees' comments:

Referee #1 (Remarks to the Author):

Krishna et. al. present a fascinating manuscript in which they propose and document bidirectional neuronal/electrical interactions between human brains and their resident gliomas. The ideas are novel, would radically change our concepts about glial brain tumors, and are potentially clinically actionable.

I have a few suggestions to improve the quality of the manuscript.

1) While I appreciate the complexity of the topic, the manuscript in its current format is replete with jargon, and long sentences that are difficult to swallow in one mental bite, even if one is reasonably familiar with the subject area. I would encourage the authors to write the manuscript for a wider audience.

2) In the manuscript, the authors are predominantly looking at IDH-mut gliomas. I commend them for primarily focusing on one, molecularly homogenous entity. Do they have any control data from other types of tumors? IDH-wt gliomas would be good, but perhaps even better would be some metastatic (non neuronal) tumors, or some low grade glioneuronal neoplasms such as ganglioglioma etc. This would help to define if their findings are more generalized to tumors of the cortex, or if this is primarily an IDH-mut phenotype. Selection of tumors that occurred later in life (i.e., metastases) versus early in life (i.e., ganglioglioma) might also allow some appreciation of the temporal aspects of this phenomenon and if they can still occur in older humans.

3) While not critical for the current manuscript, I would point out to the authors (they may already know this) that many investigators using single cell transcriptional methods have found that certain CNS cell types, particularly neurons are 'missed' using many methods, which has been attributed to the large number of processes that neurons have. One way to make sure that a critical cell type has not been missed is to use Nuc-seq, in which only nuclei are isolated, and which is thought to give a truer representation of the cell types present.

4) Have the authors speculated on whether the phenomena they describe here, and in their prior publications could contribute the well-known phenotype of glioblastoma invasion? Are the invading cells following a gradient of electricity or TSP1? Could this be a strategy to limit glioma invasion?

5) My final comment is more high level, and could be ignored if the authors and the editors disagree with me. I really don't like the title, and I really don't like the focus on survival. I am much more excited with the idea that gliomas can remodel the brain, and that perhaps 'by thought alone', a human patient can interact with, and influence the course of their glial neoplasm. That is quite a mindful. I love the portion of the last sentence of the abstract which reads "These data demonstrate that high-grade gliomas functionally remodel neural circuits in the human brain". This is incredibly fascinating to me, and I suspect that it will also be fascinating to neuroscientists,

clinicians, and the lay public. This whole concept to me implies that it is possible, to create a structure in the brain post-natally, with which one could communicate by thought alone, and which might predict that it is possible far in the future to create a mind machine interface. Obviously that sort of wild speculation has no place in a manuscript (yet), but I hope to illustrate to the authors why the idea of the cortex and the tumor talking to each other is much more interesting than an increase in survival.

With appropriate editing and addition of controls I feel that this will be a fundamental addition to the literature, which may spawn an entirely new field of study.

Referee #2 (Remarks to the Author):

In this study, Krishna et al. perform an integrative analysis of intra-operatively and pre-operatively neurophysiological recordings from glioma patients, together with molecular and functional studies and patient survival analyses. Using electrocorticography, they find indications for a higher excitability of neurons in tumor regions, compared to non-tumor regions, and larger areas involved in speech initiation than you would normally expect. Next they used MEG studies to define functionally connected (HFC) vs less-connected (LFC) regions of the brain and the tumor area, which they used during the operation for differential tumor sampling. Studies with HFC vs LFC – derived tumor cells demonstrated enrichment of the synaptogenic molecule TSP-1 in a small population and more in HFC cells, and first hints that synapse formation could be higher in HFC regions/cells. HFC glioblastoma cells show a high interaction potential with neurons. Finally, they provide mouse and patient data indicating an association of intratumoral connectivity measures with survival.

It cannot be stressed enough how important such carefully conducted joint clinical/molecular/functional/preclinical studies are to advance our understanding of cancer. Specifically, the emerging field of cancer – neural interactions, here in the context of neurons and glioma cells, can greatly profit from it. Therefore, in general, the design and conduct of this study is an important contribution to the field. The data is mostly presented in a compelling way; however, a lot of details did not become clear to me when reading the text, those need clarification (see below), and more experimental support. Among other things there is more robust data needed to support several key conclusions; and more information about the various (largely fruitful) technologies applied to fully understand the meaning and impact of the data generated.

Major points:

1. For all studies involving patients and patient material, it would be very important to get a full picture (summarized shortly in the main part; extensively provided in the supplement) about the important features of all subjects included. This includes: glioma subtype and grade; contrast enhancement / necrosis present or not on MRI scan; CE or T2+ regions recorded/sampled; residual tumor present or not after resection (if applicable); total tumor volume (CE, T2); additional tumor therapy; IDH status; MGMT status (the 58% - 72% of methylated tumors appears extremely high for primary glioblastoma; if true, where is this bias originating from? Important question, since this is a known important predictive/prognostic biomarker); etc.
2. How are tumor-infiltrated areas defined in human participants? As gliomas are a whole-brain disease by definition, it would be great to have a (maybe already published) non-tumor cohort where the in-vivo electrophysiological measurements could be compared to. – Line 34, “normal appearing regions of brain”: Where exactly? How far away from tumor region? How determined that this region is “normal” (inspection; 5-ALA; MRI; ephys;....)

3. Couldn't it be possible that the detection of "task-relevant neural activity within the entire region of tumor-infiltrated cortex" is less a tumor-specific phenomenon, but rather an unspecific feature that occurs after all kind of brain lesions? Again, reference data / recordings in this respect? – Line 53, increased HGP- again, diffuse glioma-specific? Comparison to other brain lesions (e.g. obtained by surgery of noninfiltrative brain tumors, or during epilepsy surgery) would be very helpful to understand this important point better.
4. Fig. 1g: This panel needs much better explanation and guidance: where exactly are non-tumor areas? What are we actually seeing? Is there a quantification to substantiate the conclusions? I find it difficult to understand this panel and the conclusions.
5. Line 72/73, LFC vs HFC tumor regions: what is the characteristics of these regions (MRI? All with pathological signal? CE? Glioma subtype? – etc, see above). – Judged from 2d and 2e, the regions appear very tumor cell-low density, so most probably they are MRI negative? (Everything else would be difficult to comprehend). – This is also important for clinical translation of these findings: which parts of this multi-stage disease (with different stages of progression and brain infiltration at different anatomical sites, which is always present in the same patient in this disease) have been studied here?
6. Line 88: ONLY non-tumor astrocytes express TSP-1: the data tell something different. According to Ext. Data Fig. 5 legend, 2.95% of HFC vs 1.59% of LFC express TSP-1. Moreover, statistics is required to substantiate the claim that there is a difference between HFC vs LFC
7. Fig. 2b: Why is TSP1 also upregulated in myeloid cells ? How can the authors be sure that the upregulation is specific ? Could it be normalized to the expression of e.g. myeloid cells and re-analyzed ? Were all datasets integrated analyzed ?
8. Figure 2d: Can the authors explain why TSP1 is staining whole cells in glioma tissue ? From published data a more punctate staining would be expected. Specificity experiments are needed (negative and positive control).
9. Figure 2e: The nestin staining does not seem to specifically stain cells. Normally, this staining should also stain somata with the nucleus spared. The co-localisation does not look convincing and could be attributed to nonspecific stainings.
10. Line 110: "synaptogenesis and consequent remodeling of connectivity" – how well is this statement generally established in neuroscience? More data in this respect would be helpful.
11. Figure 2f/g: PSD-95 seem partially to form bigger cluster than expected for synapses. Quantification of cluster size needed. Colocalisation of PSD95 and synapsin is needed to be sure that indeed synapses are detected. Differences of regions could explain the different ratios between NFHM/synapsin and NFHM/PSD95 ratios but this needs to be explained.
12. Figure 2i: Colocalisation analyses together with synapsin are needed. In all synaptic analyses the cluster size needs to be determined. Are the cluster sizes different between glioma and normal synapses ? – All in all, as it is, the data does not allow to convincingly assess the question whether structural synapse formation is really promoted or not.
13. Line 124: throughout the manuscript, it is important to understand how the technologies were exactly applied to measure intra-patient heterogeneity and inter-patient heterogeneity. How is intratumoral functional connectivity per individual patient measured and quantified – is it a composite value?
14. Line 134: a 1.4 fold upregulation of CLU is not impressive in such (proteomics) screening experiments. What is the statistics? What about the other factors here? Why was CLU selected (and many others not which appear much more upregulated?)
15. Figure 3a: Why was homer intensity quantified ? What does this parameter tell us? What about Homer punctae density? Again, colocalisation analyses with e.g. synapsin needed.
16. Fig 3d, EM images: Please make clear where exactly the Immunogold particles are located. Color coding: for LFC-PDX, it appears that a synapse between two non-malignant neuronal structures is shown (pre- and postsynaptic). If yellow means pre-synaptic, only one of the two marks can be correct. Moreover, specificity of RFP is unclear. The clusters of immunogold in LFC-PDX that are clumped together are typically seen when non-specific staining occurs. Single immunogold particles are localized in the presynaptic bouton (HFC-PDX). Specificity controls are needed (negative control - not glioma bearing). What does the quantification mean (total number of synapses ?) ? Synapses per field of view ? Synaptic density needs to be determined properly

with either 3D reconstructions or at least stereological quantifications. Which role do perisynaptic contacts play? How many models have been analyzed? At least three pairs, rather six pairs are needed to make a point about HFC vs LFC. In general, it would be desirable to see more EM (and also patch clamp) experimental data for important parts of the study: to A) substantiate the existence of synapses, and B) to define the synaptic subtypes, and the mode of transmission (fast vs slow waves).

17. Fig. 3f: How do the authors explain that LFC glioma cells have a HIGHER proliferation index than HFC glioma cells as baseline, and after co-culture with neurons, both show very similar proliferation indices? Isn't that in contrast to the other findings?+

18. Fig. 3g: Provide high-res images / ideally histological sections to validate TM nature. - Again, when assessing the spheroid invasion area, what sticks out as particularly low (significantly lower than all other groups) is HFC cells without conditioned medium, while HFC+mCM, and both LFC groups are higher. The question is: why is that? Together with 3f, it appears that HFC cells without neuronal interactions are particularly "malignancy-deficient". Any hypothesis why this is the case? Any data to explain it?

19. Figure 4a: The in-vitro monoculture proliferative capacity should be determined. How many cell lines ? How many patient pairs ? Knockdown/knockout of TSP1 ? Can this be addressed pharmacologically ?

20. Line 215: tumor boundary: Needs better specification (see above). Any other factors (residual tumor mass, which could be higher in this situation and at the same time is a negative prognostic factor?). One would need to know more parameters to gain better confidence that the survival differences are (partly or mainly) due to the different MEG parameters.

21. No electrophysiology from xenograft is shown. This is needed to understand which role the fast and slow currents (Venkatesh 2019, Venkataramani 2019) play. Can also more synapses be observed functionally with electrophysiology?

22. Extended Data Fig. 10: Glioma cell marker are needed, and quantification of MET-positive glioma cell density; mean pixel intensity is not really helpful here.

23. Line 236/237: "...and that distinct intratumoral regions maintain functional connectivity through a subpopulation of TSP-1 expressing malignant cells (HFC glioma cells)." - Functional connectivity? To make this claim, optimally electrophysiological single cell data would be required, and/or ultramicroscopy/EM of TSP1 pos vs neg cells.

Minor points:

1. What is "organoid intensity ?" (Fig. 3b)

2. Figure 3, headline: "functional" not "funictonal"

3. Line 58: can this really be concluded at this point? I would suggest to tone down the language here.

4. Line 65: make clearer to the reader: first MEG – then surgery.

5. Line 70: examples of this methodology? How exactly performed? Maps?

6. Line 85: higher levels: quantification is hidden in Fig. Legend ED Fig. 5 – reference better for clarity. Please provide statistics, too.

7. Line 145: "assuming" – since there is so little known about this area, I would make it clearer that there is a big black spot regarding this point.

8. Line 226: negatively influences: appears a too strong statement, at least with the current data provided. Currently, it is more "might/could influence".

9. Line 150: "activity-dependent potassium-evoked currents in more astrocyte-like glioma cells". I do not think this is fully established.

Referee #3 (Remarks to the Author):

I commend the authors on a very large body of work that has culminated into this manuscript.

However, the work permeates a variety of fields in neuroscience and molecular biology and is likely going to be too complicated for all except a small niche of experts with knowledge of all of the many domains in which data are collected and analyzed. Given the effort that has gone into this, and some of the interesting findings, I would urge them to parcellate this into more readily digestible bodies of work. Broad commentary asides, the paper suffers from some fundamental flaws that I outline below.

A. ECoG analysis: The point of this work is to say that gliomas remodel functional circuits. Using recordings in the OR during awake craniotomies for the resections of gliomas – they make the argument that there is greater gamma activation in the electrodes overlying tumor.

1) Comparisons are made in amplitude of activation in the same region across individuals and between functional regions in the same individual. Comparison of the amplitude of activations across individuals in the same brain regions (some with a tumor in that region and some without) is flawed, as this assumes that all individuals must activate equally if recordings are performed in homologous regions.

2) When I look at the maps of electrodes over tumor and non-tumor cortex, the area that is distinctly different between these is prefrontal cortex. Estimating distinctions in activation between these two groups of electrodes is meaningless. They are comparisons across regions – and not surprisingly there is greater activation prior to the onset of articulation in prefrontal cortex relative to primary motor cortex. The “pair matching” in Extended data 2 is once again biased by spatial distinctions – these comparisons of the amplitude of activation between cortical sites in the same individual with very large numbers of electrodes and trials are only weakly significant, with a relatively modest p value, and no measure of the magnitude of the effect is provided. Comparisons within individuals are also confounded by amplitudes of activation intrinsic to these regions (e.g. ventral prefrontal cortex may activate more than dorsal). The highly variable spectro-temporal responses are unaccounted for in the analysis. As such this is not an appropriate use of ECoG data

3) It is never made clear which electrodes lie over the tumor and which do not. This is hard to derive from the group figures. In one example, there appears to be a deep seated tumor with intact cortex over it and in another the tumor is directly below the recording electrode

4) No individual spectral data are presented to illustrate the quality of these intra-operative recordings that are often contaminated by movement, RF interference and epileptiform activity, which if not recognized and used to clean the data could easily confound the derivation of the mean gamma power responses. In extended figure 2a the amplitude of activation of the two relatively homologous electrodes, both over tumor and both in “premotor” cortex varies enormously – 5 fold in the second patient (SF0059) relative to the first (UM003), illustrating the pitfalls of comparisons across regions in small groups.

5) Overall, I feel that these ECoG data and analysis, flawed as they are, are a distraction from the other points made by the paper and I wonder if whether it was in any way critical to make some of the other points in the paper.

B. Magnetoencephalography (MEG) was used to categorize cells in the outer tertiles as coming from HFC vs. LFC sites. All connectivity was estimated in the alpha band. MEG suffers from relatively poor spatial localization capacity. The impact of cortical edema, brain shift and the inability to compute inverse models in the absence of accurate individualized cortical models, which is almost always the case in gliomas due to failure of automated parcellation schema, all limit the ability of MEG. Thus the premise via which these cells are categorized is questionable. Given that ECoG was performed in all these cases, measures of functional connectivity derived from such direct recordings should be feasible and utilizing them to categorize tumor cells based on connectivity, would have been much more accurate and meaningful. It would be relatively straightforward to make such estimates using ECoG data.

C. Direction of causality: Even assuming that the MEG data are spatially accurate, a possible alternate explanation for the molecular findings may lie in the fact that functionally eloquent regions are more strongly connected – thus they have a greater number and broader distribution

of fiber pathways via which glial cells can disseminate across the brain, encouraging distinct GBM sub-populations that are more capable of migration to be seen at these sites. Thus it could well be the brain that influences what type of tumor exists in eloquent vs non eloquent sites, and not vice versa.

- 1) Thus the question becomes: what is the normal variation in the glial expression of TSP-1 in eloquent vs non eloquent regions?
- 2) The spatial disparity in the locations sampling may also impact the molecular distinctions [the finding via RNA transcriptomics and IHC that in LFC tumoral regions, only non-tumor astrocytes express TSP-1, while in HFC regions, high-grade glioma cells also express TSP-1] that are proposed as a mechanism of potential increased connectivity. It is entirely possible that TSP-1 may be a normal mechanism of enhanced connectivity in HFC regions, and amplified in their neoplastic manifestation.
- 3) The same factors may impact the greater connectivity in HFC xenografts and in organoids.
- 4) At the very least, this alternate interpretation should permeate the discussion. Optimally, experiments to disambiguate the activity derived impact of neurons in eloquent cortex in rendering HFC glial cells distinct from LFC glial cells should be derived.

D. In humans, the impressive survival differences in the KM plots fit well with the established literature for much poorer prognosis of patients with tumors in the eloquent cortex (that is essentially a surrogate for the HFC terminology), who also suffer from a lower functional performance score. As such this is more confirmatory than a discovery

- 1) It is not made clear whether the two groups received the same and roughly equivalent treatments. It would be helpful to know the PFS as well as the reason for death. Was there a difference in spread locally or more distant between HFC and non HFC groups?
- 2) Is it possible that these different subpopulations may be more resistant to chemotherapy or even radiotherapy – this may be worth adding this to the discussion as potential translational strategy.
- 3) For a small group of patients such as this, MGMT status is important to know and to account for in the analysis, as it may affect disproportionately affect survival in small sample sizes.

E. Minor points:

- 1) It does not appear that measurements of tumor volume in the mice to demonstrate differences between the xenografted HFC or LFC cells was performed – this must have been performed and it would be good to look at to explain such a different survival.
- 2) The claim that “gliomas remodel functional neural circuitry such that task-relevant neural responses activate tumor-infiltrated cortex, beyond cortical excitation normally recruited in the healthy brain” is overblown. The finding of non-traditional language sites are activated during lexical access is hardly surprising as functional reorganization secondary to gliomas is well known and is entirely expected in such cases.
- 3) The number of patients in the HFC and no HFC groups in figure 4b need to be explicit, as it’s a bit confusing and difficult to visualize each death on the plot.
- 4) This work appears to miss the opportunity to build upon prior publications (Venkatesh et al - Electrical and synaptic integration of glioma into neural circuits - Nature 2019), by not seeking to modulate the influence of glioma activity on neuronal excitability via potassium fluxes in vivo, or to directly modulate activity regulated glioma growth – natural directions given the rich datasets and the skill they have brought to bear in performance of this work.

Author Rebuttals to Initial Comments:

Referees' comments:

Referee #1 (Remarks to the Author): Referee #1:

Reviewer 1 Comment 1: Krishna et. al. present a fascinating manuscript in which they propose and document bidirectional neuronal/electrical interactions between human brains and their resident gliomas. The ideas are novel, would radically change our concepts about glial brain tumors, and are potentially clinically actionable.

Reviewer 1 Response 1: *Thank you for this important comment. We agree that these data represent a fundamental change in our understanding of molecular drivers of glioblastoma proliferation as well as radically change in the way human brain cancers are studied. We are thrilled to address each of your comments which have greatly improved this manuscript.*

I have a few suggestions to improve the quality of the manuscript.

Reviewer 1 Comment 2: While I appreciate the complexity of the topic, the manuscript in its current format is replete with jargon, and long sentences that are difficult to swallow in one mental bite, even if one is reasonably familiar with the subject area. I would encourage the authors to write the manuscript for a wider audience.

Reviewer 1 Response 2: *Thank you for this important comment. Readability for a general audience is critically important. We have addressed the readability concern with a full edit of the manuscript for clarity.*

Reviewer 1 Comment 3: In the manuscript, the authors are predominantly looking at IDH-mut gliomas. I commend them for primarily focusing on one, molecularly homogenous entity. Do they have any control data from other types of tumors? IDH-wt gliomas would be good, but perhaps even better would be some metastatic (non neuronal) tumors, or some low grade glioneuronal neoplasms such as ganglioglioma etc. This would help to define if their findings are more generalized to tumors of the cortex, or if this is primarily an IDH-mut phenotype. Selection of tumors that occurred later in life (i.e., metastases) versus early in life (i.e., ganglioglioma) might also allow some appreciation of the temporal aspects of this phenomenon and if they can still occur in older humans.

Reviewer 1 Response 3: *Thank you for bringing to light two important considerations: (1) better use of control experiments and (2) the generalizability of brain cancer remodeling of cortical circuits across differing primary and metastatic tumor subtypes. In this manuscript our goal was to focus on a specific diffuse glioma subtype. All experiments in this paper are focused on isocitrate dehydrogenase wild type glioblastoma (according to WHO 2021 classification) remodeling of neuronal circuits as well as molecular drivers of cortical neuron remodeling. We address circuit remodeling applying both short range measures of neuronal activity in a behavioral task-specific manner (electrocorticography through cortically projecting gliomas) as well as long-range measures of functional connectivity (imaginary coherence magnetoencephalography).*

We agree that proper controls and detail of experimental conditions are essential. Our experimental design was to separate tumor-infiltrated from normal-appearing electrodes as two conditions given that this exact approach has been applied and published extensively over the past decade⁷⁻¹². However, as raised by Reviewer #1, glioma remodeling of functional circuits may impact broader cortical regions of speech initiation therefore separate control conditions would be beneficial. The presence of audiovisual speech responses within the lateral prefrontal cortex (LPFC) has previously been published within the context of human epilepsy, therefore this experiment was not added to the study (only appropriate citations)¹⁷⁻²². Control conditions for this experiment are based on the absence of pre speech onset HGp prior to task administration followed by post speech onset HGp suppression with this distinct electrophysiological pattern replicated over hundreds of stimuli, trials, and electrodes (Fig. 1b-d). However, as an additional positive control demonstrating preserved speech initiation cortical responses, we have included data below for a diffuse glioma control in which cortical sampling is obtained under clinical context from LPFC for a non-cortically projecting tumor within the insular cortex. Identical to the non-tumor electrodes from cortically projecting gliomas, we discovered that group-level HGp demonstrates the expected neural time-course within LPFC, showing activation anterior to primary motor cortex between 600 milliseconds (ms) before speech onset (0 ms), and maximal activation in motor cortex at speech onset consistent with prior established models of speech initiation.

Reviewer #1, next inquired about the selectivity of maintained tumor intrinsic task-specific cortical responses across diffuse glioma subtypes. The excellent question was raised whether task neuronal activity with hyperexcitable cortical responses identified in glioblastoma, would differ by histology and between gliomas which develop early in life (ganglioglioma and DNET) vs gliomas occurring later in life (IDH-wild type glioblastoma) and compare these findings with non-glial malignancies such as brain metastasis. Based on this comment, recordings were taken from participants with cortically projecting diffuse low-grade and high-grade gliomas. Oligodendrogliomas WHO grade 2 1p/19q-codeleted, IDH mutant WHO grade 2 and 3 astrocytomas, and glioblastomas were included. Because isocitrate dehydrogenase (IDH) mutant 1p/19q-codeleted oligodendrogliomas may rely on distinct mechanisms to infiltrate the parenchyma and

modulate cortical dynamics, we performed additional analyses and directly compared these results with the 1p/19q-codeleted oligodendrogliomas. We find that across glioma subtype, tumor intrinsic task-specific neuronal responses for speech initiation are maintained within the LPFC.

Next, we separated data by glioma subtypes illustrating this conserved phenotype with task-specific hyperexcitability observed only in participants with glioblastoma.

Interestingly, brain metastasis infiltrated cortex (thin overlying cortical mantle) for lung adenocarcinoma illustrates similar preserved task-specific responses suggestive of the conservation of neuronal signals within the tumor neuronal microenvironment. This stands in stark contrast to ECoG recording from complete cortical infiltration with no remaining overlying cortex from brain metastasis setting in which no neuronal activity is identified (data not shown here).

While our initial goal was to investigate glioma-circuit remodeling across molecular subtypes of glial tumors, it quickly became evident that separation by molecular subtypes may yield markedly different mechanisms, particularly given our goal of applying circuit and cellular level drivers of cortical remodeling. The observed pattern of task-specific neuronal activity within glioma-infiltrated cortex is preserved across brain cancers. However, the temporal pattern and balance of excitatory to inhibitory inputs likely differ. During the review and resubmission period for this paper, we identified no cortically projecting glioneuronal tumors such as DNET or ganglioglioma. After inquiry with a wide network of collaborators and colleagues, we were unable to determine whether glioneuronal tumors maintain task-specific cortical responses. While task-specific cortical hyperexcitability is identified across tumor subtype, only glioblastoma demonstrated hyperexcitability in our analysis. Furthermore, the temporal pattern of behavioral responses may have implications on task accuracy (which is a focus of our future work).

In line with Reviewer #1 recommendations that this manuscript remain focused on glioblastoma (IDH-wild type), we have provided the above additional control experiments as extended figures 2b-e in the revised manuscript. The manuscript text has been revised to read as follows (updated text is in bold italics):

Glioblastomas remodel functional neural circuits

High-grade gliomas interact with normal neuronal elements, resulting in both cellular and network level changes. While high-grade gliomas influence neuronal excitability at rest, the effects of task-related activity on glioma-infiltrated neural circuit function and the impact of glioma-neuron interactions on neural circuit connectivity remain unknown. To examine cognitive task-related neuronal activity in the setting of high-grade glioma, we selected a cohort of 14 adult patients with cortically projecting glioblastoma in the lateral prefrontal cortex (LPFC) (Extended Data Fig. 2a), classically referred to as Broca's area (Extended Data Fig. 3). In the operating room, tumor boundaries were localized on magnetic resonance imaging (MRI) and electrocorticography (ECoG) electrodes placed

over the tumor-infiltrated cortical region and normal-appearing cortex. ECoG signals filtered between 70-110 Hz were used for analysis of the high-gamma band range power (HGp), which is strongly related to local neuronal population spikes and is increased by cortical hyperexcitability. **Spectral data demonstrated the expected pattern of HGp increasing above 50 Hz in addition to clear separation of frequencies across tumor and non-tumor electrodes (Fig. 1a, Extended Data Fig. 4a).**

ECoG was recorded from the dominant hemisphere LPFC during auditory and visual picture naming tasks as an illustrative example of a well-defined cognitive neuronal circuit with defined physiology. In humans, speech initiation occurs in the LPFC (Broca's speech area, Brodmann area 44). While patients were fully awake and engaged in these language tasks, HGp was recorded from single electrodes overlying tumor-infiltrated and normal-appearing regions of brain (Extended Data Fig. 4b). These recordings provide simultaneous high spatial and temporal resolution while sampling the neuronal population activity during auditory and visual initiation of speech within the LPFC.

Group-level HGp from non-tumor electrodes and control conditions demonstrates the expected neural time-course within LPFC, showing activation anterior to primary motor cortex between 600 milliseconds (ms) before speech onset (0 ms), and maximal activation in motor cortex at speech onset (Extended Data Figs. 2a-d), consistent with prior established models of speech initiation previously demonstrated in non-human primates and humans. We then performed the same time series focused only on electrode arrays recording from tumor-infiltrated cortex. Countering the theory that glioblastoma-synaptic integration may result in physiologically disorganized neural responses, we found task-relevant neural activity within the entire region of tumor-infiltrated cortex. Strikingly, this includes speech initiation-induced recruitment of not only LPFC, Broca's region, as expected, but also regions of tumor-infiltrated cortex not normally involved in speech initiation (Fig. 1b). **Similarly, we found that across WHO grade 2-4 glioma subtypes, tumor intrinsic task-specific neuronal responses for speech initiation are maintained within the LPFC (Extended Data Fig. 2c-d).** Taken together, these findings suggest that in subjects with glioblastoma affecting the dominant hemisphere LPFC, naming tasks induce physiologically organized neuronal activity within tumor-infiltrated cortex, well beyond the cortical territory normally recruited during this language task.

Reviewer 1 Comment 4: While not critical for the current manuscript, I would point out to the authors (they may already know this) that many investigators using single cell transcriptional methods have found that certain CNS cell types, particularly neurons are 'missed' using many methods, which has been attributed to the large number of processes that neurons have. One way to make sure that a critical cell type has not been missed is to use Nuc-seq, in which only nuclei are isolated, and which is thought to give a truer representation of the cell types present.

Reviewer 1 Response 4: Thank you for your comment. We agree that compared to single nucleus RNA-seq (sNuc-seq), single cell RNA-seq may not capture all cell types within the glioma microenvironment. We also understand that sNuc-seq is less susceptible to perturbations of gene expression occurring during cell isolation, such as increased expression of immediate early genes that can obscure transcriptional signatures of neuronal activity. The primary goal of our sequencing experiments was the identification of cancer cell populations which may drive neuronal activity and therefore

circuit remodeling. Therefore, we are confident that malignant glioma cells express TSP-1, and the focus of these experiments is glioblastoma-derived TSP-1 which has been demonstrated by single-cell sequencing and confirmed at the protein level. Previous single cell sequencing studies have identified neuronal cell types, albeit fewer in number when using scRNA-seq^{13,14}. We have edited the manuscript to point out that our single cell pipeline may not capture all cell types within the central nervous system microenvironment, particularly neurons.

The manuscript text has been revised to read as follows (updated text is in italics):

Single-cell sequencing and analysis

Single-cell suspensions of 3 patient-matched HFC and LFC tumor tissues were generated as described above and processed for single-cell RNA-seq using the Chromium Next GEM Single Cell 3' GEM, Library & Gel Bead Kit v3.1 on a 10x Chromium controller (10x Genomics, Pleasanton, CA) using the manufacturer recommended default protocol and settings, at a target cell recovery of 5,000 cells per sample. ***While single cell sequencing does not capture all cell types within the central nervous system microenvironment, the sequencing pipeline used in this study has been demonstrated to identify neurons and was therefore chosen for use in physiologically annotated fresh glioblastoma specimens, compared with single nucleus RNA-sequencing which is commonly applied for frozen archived tissues.*** One hundred base pair paired-end reads were sequenced on an Illumina NovaSeq 6000 at the Center for Advanced Technology at the University of California San Francisco, and the resulting FASTQ files were processed using the Cell Ranger analysis suite version 3.0.2 (<https://github.com/10XGenomics/cellranger>) for alignment to the hg38 reference genome, identification of empty droplets, and determination of the count threshold for further analysis. A cell quality filter of greater than 500 features but fewer than 10,000 features per cell, and less than 20% of read counts attributed to mitochondrial genes, was used. Single cell UMI count data were preprocessed in Seurat 3.0.1 using the SCTransform workflow, with scaling based on regression of UMI count and percentage of reads attributed to mitochondrial genes per cell. Dimensionality reduction was performed using principal component analysis and then principal component loadings were corrected for batch effects using Harmony. Uniform Manifold Approximation and Projection (UMAP) was performed on the reduced data with a minimum distance metric of 0.4 and Louvain clustering was performed using a resolution of 0.2. Marker selection was performed in Seurat using a minimum difference in the fraction of detection of 0.5 and a minimum log-fold change of 0.5. We assessed the single cell transcriptome from 6,666 HFC region cells and 7,065 LFC region cells (Extended Data Table 4).

Reviewer 1 Comment 5: Have the authors speculated on whether the phenomena they describe here, and in their prior publications could contribute the well-known phenotype of glioblastoma invasion? Are the invading cells following a gradient of electricity or TSP1? Could this be a strategy to limit glioma invasion?

Reviewer 1 Response 5: *Thank you for your comment. Reviewer 1 proposed a fascinating experiment which may deserve its own unique story. Activity regulated glioblastoma invasion is a topic of great interest in the cancer neuroscience field therefore we have not focused specifically on this set of experiments. The results*

presented in this study, specifically from the spheroid tumor microtube assay (Revised manuscript Fig. 3e, f and Extended Data Fig. 17) demonstrates that glioma cells from functionally connected regions exhibit a distinct invasive phenotype in the neuronal microenvironment. It is worth noting that HFC cells alone in culture display few tumor microtubes as they require neuronal signals in order to shift towards a proliferative and structurally connected tumor cell phenotype. We hypothesize that this change is primarily driven by TSP-1 paracrine signaling mediated by HFC glioma cells in the presence of neuronal factors. The tumor microtube phenotype in which glioma-neuron co-cultures establish tumor microtubes has been defined as “functional integration” by Venkataramani and Winkler¹⁵. Similarly, glioma-neuron co-cultures establish glioma invasion in 2D culture conditions by Venkatesh and Monje¹⁶. In our experiments using primary patient cultures of TSP-1 high-expressing astrocyte-like glioma cells originating from within functionally connected regions of brain, these cells demonstrate greater number and length of microtubes (Revised manuscript Fig. 3e, f).

Based on reviewer 1 comments, we believe that there would be great value in determining whether the observed HFC phenotype is causally related to TSP-1, and we have therefore performed additional experiments by both genetic and pharmacological targeting approaches to address the causal relationship of TSP-1 with the invasive and proliferative tumor phenotype of TSP-1 high-expressing glioblastoma cells. Primary patient-derived HFC tumor cells were either transduced with shRNA targeting TSP-1 to knockdown thrombospondin-1 or treated with the FDA approved drug gabapentin to pharmacologically inhibit TSP-1. We found that compared to control shRNA condition, HFC cells transduced with TSP-1-shRNA exhibited significantly fewer number of tumor microtubes (Revised manuscript Fig. 3g), consistent with the known role of TSP-1 in tumor microtube formation¹⁷. Interestingly, knockdown of TSP-1 also resulted in significant reduction in the number of Ki67-positive proliferating tumor cells in the neuron-HFC glioma co-culture (Revised manuscript Fig. 4e). Changes in the proliferative potential of HFC cells in the presence of the TSP-1 inhibitor, gabapentin, was further assessed in both *in vitro* neuron-glioma co-culture and *in vivo* patient-derived HFC xenograft models. We found that similar to the gene editing results, pharmacological inhibition of TSP-1 using gabapentin significantly decreased the proliferation of HFC cells both *in vitro* (Revised manuscript Fig. 4f) and *in vivo* (Revised manuscript Fig. 4g, h). We are adding the relevant new figures below for your convenience.

In vitro-TSP-1 shRNA-Tumor microtubes and Ki67 analysis

(Left) Primary patient-derived HFC cells were transduced with shRNA control or shRNA TSP-1. Representative SEM images showing tumor microtubes (TMT) and quantification of TMTs per cell from HFC shRNA-control and HFC shRNA TSP-1 conditions (HFC-shControl vs. HFC-shTSP-1: 1.44 ± 0.09 vs. 0.44 ± 0.18 , $n = 2/\text{group}$). ($P = 0.0012$). Scale bar, 20 μm . (Right) Primary patient-derived HFC cells were transduced with shRNA control or shRNA TSP-1. Representative confocal images from neuron-HFC glioma co-culture showing marked decrease in proliferation of HFC cells (as measured by the number of human nuclear antigen (HNA)-positive cells co-labelled with Ki67 divided

by the total number of HNA-positive tumor cells counted across all areas quantified) upon TSP-1 silencing using shRNA (HFC-shControl vs. HFC-shTSP-1: $59.63 \pm 4.88\%$ vs. $36.17 \pm 5.92\%$, $n = 2/\text{group}$) ($P = 0.0068$). Red, HNA (human nuclei); white, Ki67. Scale bar, $30\ \mu\text{m}$.

In vitro- Gabapentin treatment- Ki67 analysis

Representative confocal images from neuron-HFC glioma co-culture showing marked decrease in proliferation of HFC cells (as measured by the number of human nuclear antigen (HNA)-positive cells co-labelled with Ki67 divided by the total number of HNA-positive tumor cells counted across all areas quantified) upon pharmacological TSP-1 inhibition using ($32\ \mu\text{M}$) gabapentin (HFC vs. HFC + GBP: $66.67 \pm 5.82\%$ vs. $38.77 \pm 4.33\%$, $n = 2/\text{group}$) ($P = 0.0007$). Red, HNA (human nuclei); white, Ki67. Scale bar, $30\ \mu\text{m}$.

In vivo- Primary patient-derived HFC xenograft- Gabapentin treatment- Ki67 analysis

(Top row) Schematic representation of the in vivo gabapentin (GBP) treatment paradigm of HFC patient-derived xenografted (PDX) mice. (Bottom row) Representative confocal images, and quantification demonstrating marked decrease in proliferation index ($\text{Ki67}^+\text{HNA}^+/\text{HNA}^+$) of gabapentin treated mice bearing HFC xenografts (HFC + Vehicle

vs. HFC + GBP: 1.00 ± 0.17 vs. 0.76 ± 0.14 , $n = 9$ mice/group) ($P = 0.046$). Red, HNA (human nuclei); white, Ki67. Scale bar, 70 μm . Data presented as mean \pm s.e.m (c-f, h). P values determined by two-tailed Student's t -test.

Reviewer 1 Comment 6: My final comment is more high level, and could be ignored if the authors and the editors disagree with me. I really don't like the title, and I really don't like the focus on survival. I am much more excited with the idea that gliomas can remodel the brain, and that perhaps 'by thought alone', a human patient can interact with, and influence the course of their glial neoplasm. That is quite a mindful. I love the portion of the last sentence of the abstract which reads "These data demonstrate that high-grade gliomas functionally remodel neural circuits in the human brain". This is incredibly fascinating to me, and I suspect that it will also be fascinating to neuroscientists, clinicians, and the lay public. This whole concept to me implies that it is possible, to create a structure in the brain post-natally, with which one could communicate by thought alone, and which might predict that it is possible far in the future to create a mind machine interface. Obviously that sort of wild speculation has no place in a manuscript (yet), but I hope to illustrate to the authors why the idea of the cortex and the tumor talking to each other is much more interesting than an increase in survival.

Reviewer 1 Response 6: *Thank you for your insightful comment. We completely agree with Reviewer 1. Malignant gliomas classically have long been thought of as unconnected neoplastic entities, separate from the neuronal and glial brain microenvironment. These data provide compelling evidence that neural networks are not destroyed but remodeled and therapies impacting neural networks may influence outcome. We would however like to maintain the survival endpoints in the study given the importance of these findings in cancer biology. In fact, while Kaplan-Meier survival statistics remains an essential method to estimate longevity, it misses interactions between clinical and molecular variables. The interplay between factors such as glioblastoma molecular subtype (IDH, MGMT, etc.), patient (age, functional status), and treatment factors (such as extent of tumor resection, chemoradiation, etc.) has been a topic of intense interest¹⁸. Recent cancer research studies have attempted to move beyond supervised multivariate survival models. While in the initial submitted draft, we demonstrated using mouse (Revised manuscript Fig. 3h) and human (Revised manuscript Extended Data Fig. 21) Kaplan-Meier survival analysis illustrating 71-week overall survival for patients with HFC voxels as determined by contrast-enhanced T1-weighted images as compared to 123-weeks for participants without HFC voxels. In our revised draft, we have now applied an unsupervised machine learning approach to segment survival outcomes through recursive partitioning analysis (RPA) of Post-Stupp era (2005)/IDH wild-type glioblastoma patients. Variables analyzed for this experiment included those published in Molinaro et al. work¹⁸. However, a nested dataset within this cohort of 70 patients with chemoradiation treated IDH-wild-type glioblastoma had MEG measures of tumor intrinsic neuronal oscillations and thereby connectivity (35 events 20-month median follow-up). Patients were stratified in a binary manner as having any neuronal oscillations within the tumor or none.*

Using this approach, three risk groups were determined by RPA. Risk group 1 (black) had the worst outcomes and are the combination of patients older than 72 and patients younger than 72 with less than 97% extent of tumor resection. Risk group 3 (gray) have the best survival, and these are patients younger than 62 with over 97%

extent of tumor resection and without functional connectivity in the tumor. Intermediate risk group 2 (red) revealed an interesting interaction between age and HFC. This group had two subsets: patients with over 97% resection of tumor and age younger than 72 with intratumoral connectivity; and those between 62 and 72 years old without functional integration. Taken together, these data demonstrate that neuronal activity within malignant gliomas negatively impacts survival with importance demonstrated by machine learning segmentation of outcomes and quantified to the extent that the presence of neuronal activity may be the equivalent to older patient age regardless of the extent of tumor surgically removed. Given the strength of these data we would like to maintain survival data in the manuscript and title (including the addition of our new machine learning segmentation of survival outcomes as Fig. 4a, b in the revised manuscript). For revision # 1 we have maintained survival in the title however would be happy to remove survival from the title if requested by Reviewer #1 or editorial staff.

The text has been revised to read as follows (updated text is in bold italics):

Glioma functional connectivity shortens survival

We next explored the effects of high functional connectivity within gliomas on survival and cognition. First, we tested the hypothesis that gliomas exhibiting increased functional connectivity may be more aggressive, given the robust influence of neuronal activity on tumor progression. To investigate patient outcomes, we performed a human survival analysis of patients with molecularly uniform newly diagnosed IDH-WT glioblastoma. After controlling for known correlates of survival (age, tumor volume, completion of chemotherapy and radiation, and extent of tumor resection), neural oscillations and functional connectivity were measured within tumor-infiltrated brain using MEG (Extended Tables 2 and 5). Subjects were classified by the presence or absence of HFC voxels within the tumor boundary. Kaplan-Meier survival analysis illustrates 71-week overall survival for patients with HFC voxels as compared to 123-week overall survival for participants without HFC voxels, illustrating a striking inverse relationship between

survival and functional connectivity of the tumor (mean follow-up months 50.5 months) ($P = 0.04$) (Extended Data Fig. 21). **To identify clinically relevant survival risk groups for newly diagnosed glioblastoma patients treated with chemoradiation with the presence or absence of HFC voxels within the tumor, we employed recursive partitioning survival trees via the partDSA algorithm. Survival trees use recursive partitioning to divide patients into risk groups based on the interactive effects of all included prognostic variables (e.g., age at diagnosis, sex, tumor location, chemotherapy, radiotherapy, the presence of functional connectivity within the tumor, pre- and post-operative tumor volume, and extent of resection). Risk group 1 (black) had the worst outcomes and are the combination of patients older than 72 and patients younger than 72 with less than 97% extent of tumor resection. Risk group 3 (gray) have the best survival, and these are patients younger than 62 with over 97% extent of tumor resection and without functional connectivity in the tumor. Intermediate risk group 2 (red) revealed an interesting interaction between age and HFC. This group had two subsets: patients with over 97% resection of tumor and age younger than 72 with intratumoral connectivity; and those between 62 and 72 years old without functional integration (Fig. 4a, b). These results demonstrate the striking prognostic value of HFC on survival.** We next examined whether TSP-1, a secreted synaptogenic protein, can be identified in patient serum and whether circulating TSP-1 is correlated with functional connectivity as measured by magnetoencephalography imaginary coherence. Circulating TSP-1 levels in patient serum exhibited a striking positive correlation with intratumoral functional connectivity ($P = 0.01$) (Fig. 4c), identifying a possible clinical correlate for functional connectivity in glioma patients.

Reviewer 1 Comment 7: With appropriate editing and addition of controls I feel that this will be a fundamental addition to the literature, which may spawn an entirely new field of study.

Reviewer 1 Response 7: Thank you for sharing these comments which we believe has significantly improved the quality of our work.

Referee #2 (Remarks to the Author): Referee #2:

In this study, Krishna et al. perform an integrative analysis of intra-operatively and pre-operatively neurophysiological recordings from glioma patients, together with molecular and functional studies and patient survival analyses. Using electrocorticography, they find indications for a higher excitability of neurons in tumor regions, compared to non-tumor regions, and larger areas involved in speech initiation than you would normally expect. Next they used MEG studies to define functionally connected (HFC) vs less-connected (LFC) regions of the brain and the tumor area, which they used during the operation for differential tumor sampling. Studies with HFC vs LFC – derived tumor cells demonstrated enrichment of the synaptogenic molecule TSP-1 in a small population and more in HFC cells, and first hints that synapse formation could be higher in HFC regions/cells. HFC glioblastoma cells show a high interaction potential with neurons. Finally, they provide mouse and patient data indicating an association of intratumoral connectivity measures with survival.

It cannot be stressed enough how important such carefully conducted joint clinical/molecular/functional/preclinical studies are to advance our understanding of cancer. Specifically, the emerging field of cancer – neural interactions, here in the context of neurons and glioma cells, can greatly profit from it. Therefore, in general, the design and conduct of this study is an important contribution to the field. The data is mostly presented in a compelling way; however, a lot of details did not become clear to me when reading the text, those need clarification (see below), and more experimental support. Among other things there is more robust data needed to support several key conclusions; and more information about the various (largely fruitful) technologies applied to fully understand the meaning and impact of the data generated.

Major points:

Reviewer 2 Comment 1: For all studies involving patients and patient material, it would be very important to get a full picture (summarized shortly in the main part; extensively provided in the supplement) about the important features of all subjects included. This includes: glioma subtype and grade; contrast enhancement / necrosis present or not on MRI scan; CE or T2+ regions recorded/sampled; residual tumor present or not after resection (if applicable); total tumor volume (CE, T2); additional tumor therapy; IDH status; MGMT status (the 58% - 72% of methylated tumors appears extremely high for primary glioblastoma; if true, where is this bias originating from? Important question, since this is a known important predictive/prognostic biomarker); etc.

Reviewer 2 Response 1: *Thank you for this important comment and for careful review of our work. We agree completely that full description of the imaging, clinical and molecular features of all primary patient data should be included in this study. All experiments in this paper are focused on newly diagnosed isocitrate dehydrogenase wild type glioblastoma (according to WHO 2021 classification) and for the survival data, we used IDH WT glioblastoma patients that underwent the same treatment regimen. We have added a new table (Revised manuscript- Extended Data Table 2) providing a complete summary of the clinical and molecular features of the patients used for our experiments (also included below for your convenience).*

Taking this important concept one step further it is well established that the interplay between variables such as glioblastoma molecular classification (IDH, MGMT, etc), patient (age, functional status), and treatment factors (such as extent of tumor resection and chemoradiation) determines survival outcomes¹⁸. Recent studies have attempted to move beyond supervised multivariate analysis of survival outcomes. In the initial submitted draft we demonstrated using mouse (Revised manuscript Fig. 3h) and human (Revised manuscript Extended Data Fig. 21) Kaplan-Meier survival analyses illustrating 71-week overall survival for patients with HFC voxels as determined by contrast-enhanced T1-weighted images as compared to 123-weeks for participants without HFC voxels. We have now performed glioblastoma risk modeling using recursive partitioning analysis (RPA) for post-Stupp era IDH wild-type glioblastoma patients. Variables analyzed for this experiment included those published in Molinaro et al. work¹⁸. A nested dataset within this cohort of 70 patients had MEG measures of tumor intrinsic neuronal oscillations and thereby connectivity (35 events 20-month median follow-up). Patients were stratified in a binary manner as having any neuronal oscillations within the tumor or none. Using this approach, three risk groups were determined by RPA. Risk group 1 (black) had the worst outcomes and are the combination of patients older than

72 and patients younger than 72 with less than 97% extent of tumor resection. Risk group 3 (gray) have the best survival, and these are patients younger than 62 with over 97% extent of tumor resection and without functional connectivity in the tumor. Intermediate risk group 2 (red) revealed an interesting interaction between age and HFC. This group had two subsets: patients with over 97% resection of tumor and age younger than 72 with intratumoral connectivity; and those between 62 and 72 years old without functional integration. Therefore, taken together, these data demonstrate in humans that neuronal activity within malignant gliomas negatively impacts survival with importance demonstrated by machine learning segmentation of outcomes and quantified to the extent that the presence of neuronal activity may be the equivalent to older patient age regardless of the extent of tumor surgically removed. We have added this new analysis shown below as Fig. 4a, b in the revised manuscript.

New Extended Data Table 2. Patient summary-clinical and molecular features

Study #	Sex	Age (yr)	Preoperative tumor volume (ml)	Residual tumor (ml)	EOR (%)	MGMT methylation	EGFR amplification	Tumor Location	Tumor type	IDH status
SF#1	M	56.07	23.67	0.00	100.00	no	nonamp	R frontal	GBM	wt
SF#2	M	64.93	9.20	0.00	100.00	yes	amp	L temporal	GBM	wt
SF#3	M	60.96	49.68	3.95	92.06	yes	nonamp	L temporal	GBM	wt
SF#4	M	60.42	20.30	0.00	100.00	yes	amp	L frontal/insula	GBM	wt
SF#5	M	76.72	8.94	0.00	100.00	no	nonamp	L frontal	GBM	wt
SF#6	M	72	23.59	4.79	79.70	yes	nonamp	L temporal	GBM	wt
SF#7	F	64.14	78.37	0.00	100.00	yes	amp	R frontal	GBM	wt
SF#8	M	78.12	77.49	0.00	100.00	yes	amp	L frontal	GBM	wt
SF#9	F	62.2	14.13	0.00	100.00	no	amp	L temporal	GBM	wt
SF#10	F	59.02	6.38	0.00	100.00	yes	amp	R frontal	GBM	wt
SF#11	M	57.26	46.07	1.09	97.64	no	nonamp	L temporal	GBM	wt
SF#12	M	55.34	53.10	7.33	86.19	yes	nonamp	L frontal	GBM	wt
SF#13	F	66.92	3.20	0.00	100.00	yes	amp	L temporal	GBM	wt
SF#14	M	29.3	36.88	0.00	100.00	yes	amp	R insula	GBM	wt
SF#15	M	51.15	29.67	2.63	91.14	no	nonamp	L thalamus	GBM	wt
SF#16	M	48.92	31.05	0.00	100.00	no	nonamp	L frontal	GBM	wt
SF#17	F	49.3	55.55	3.49	93.72	yes	amp	L frontal	GBM	wt
SF#18	M	60.98	67.50	0.00	100.00	yes	amp	L frontal	GBM	wt
SF#19	M	80.1	45.41	0.75	98.35	yes	nonamp	L parietal	GBM	wt

SF#2 0	F	72.8 9	6.12	0.00	100.0 0	yes	nonamp	R frontal	GBM	wt
SF#2 1	F	56.3 4	81.00	0.00	100.0 0	yes	amp	L parietal	GBM	wt
SF#2 2	M	63.2 2	7.01	0.53	92.41	yes	amp	L parietal	GBM	wt
SF#2 3	F	60.0 4	15.69	0.00	100.0 0	yes	amp	L parietal	GBM	wt
SF#2 4	F	69.6 2	92.46	0.00	100.0 0	no	nonamp	R frontal	GBM	wt
SF#2 5	M	60.3 8	101.96	0.21	99.79	yes	nonamp	R frontal	GBM	wt
SF#2 6	F	52.6 1	7.32	0.00	100.0 0	yes	nonamp	L temporal	GBM	wt
SF#2 7	M	52.7 2	41.21	3.58	91.31	no	amp	R frontal	GBM	wt
SF#2 8	F	59.8 9	9.51	0.00	100.0 0	no	nonamp	L temporal	GBM	wt
SF#2 9	M	44.9 2	34.59	0.00	100.0 0	yes	amp	R temporal	GBM	wt
SF#3 0	F	65.5 3	15.77	0.48	96.96	no	unknown	Parietal	GBM	wt
SF#3 1	F	67.8 9	14.14	0.00	100.0 0	yes	nonamp	L multifocal	GBM	wt
SF#3 2	M	71.4 9	21.80	0.00	100.0 0	no	amp	L temporal	GBM	wt
SF#3 3	F	61.7 2	52.27	0.17	99.67	no	amp	R occipital	GBM	wt
SF#3 4	M	69.0 3	57.01	0.00	100.0 0	yes	nonamp	L parietal	GBM	wt
SF#3 5	F	46.9	35.99	0.00	100.0 0	yes	yes	R temporal	GBM	wt
SF#3 6	M	51.7 3	37.45	2.20	94.13	no	amp	L frontal	GBM	wt
SF#3 7	M	69.6 7	59.66	0.33	99.45	yes	nonamp	R parieto- occipital	GBM	wt
SF#3 8	M	69.0 6	74.40	2.49	96.65	yes	nonamp	R parietal	GBM	wt
SF#3 9	F	69.8 3	6.85	0.17	97.51	yes	amp		GBM	wt
SF#4 0	M	54.0 5	61.69	3.00	95.13	yes	amp	L frontal	GBM	wt
SF#4 1	F	57.3 9	9.25	0.00	100.0 0	yes	nonamp	L frontal	GBM	wt
SF#4 2	M	60.2 1	101.06	0.41	99.59	no	nonamp	L temporal	GBM	wt
SF#4 3	M	76.7 5	44.02	1.13	97.43	yes	nonamp	R temporal	GBM	wt
SF#4 4	M	43.8 1	17.72	0.00	100.0 0	no	amp	R frontal	GBM	wt

The manuscript text has been revised to read as follows (updated text is in bold italics):

Glioma functional connectivity shortens survival

We next explored the effects of high functional connectivity within gliomas on survival and cognition. First, we tested the hypothesis that gliomas exhibiting increased functional connectivity may be more aggressive, given the robust influence of neuronal activity on tumor progression. To investigate patient outcomes, we performed a human survival analysis of patients with molecularly uniform newly diagnosed IDH-WT glioblastoma. After controlling for known correlates of survival (age, tumor volume, completion of chemotherapy and radiation, and extent of tumor resection), neural oscillations and functional connectivity were measured within tumor-infiltrated brain using MEG (Extended Tables 2 and 5). Subjects were classified by the presence or absence of HFC voxels within the tumor boundary. Kaplan-Meier survival analysis illustrates 71-week overall survival for patients with HFC voxels as compared to 123-week overall survival for participants without HFC voxels, illustrating a striking inverse relationship between survival and functional connectivity of the tumor (mean follow-up months 50.5 months) ($P = 0.04$) (Extended Data Fig. 21). **To identify clinically relevant survival risk groups for newly diagnosed glioblastoma patients treated with chemoradiation with the presence or absence of HFC voxels within the tumor, we employed recursive partitioning survival trees via the partDSA algorithm. Survival trees use recursive partitioning to divide patients into risk groups based on the interactive effects of all included prognostic variables (e.g., age at diagnosis, sex, tumor location, chemotherapy, radiotherapy, the presence of functional connectivity within the tumor, pre- and post-operative tumor volume, and extent of resection). Risk group 1 (black) had the worst outcomes and are the combination of patients older than 72 and patients younger than 72 with less than 97% extent of tumor resection. Risk group 3 (gray) have the best survival, and these are patients younger than 62 with over 97% extent of tumor resection and without functional connectivity in the**

tumor. Intermediate risk group 2 (red) revealed an interesting interaction between age and HFC. This group had two subsets: patients with over 97% resection of tumor and age younger than 72 with intratumoral connectivity; and those between 62 and 72 years old without functional integration (Fig. 4a, b). These results demonstrate the striking prognostic value of HFC on survival. We next examined whether TSP-1, a secreted synaptogenic protein, can be identified in patient serum and whether circulating TSP-1 is correlated with functional connectivity as measured by magnetoencephalography imaginary coherence. Circulating TSP-1 levels in patient serum exhibited a striking positive correlation with intratumoral functional connectivity ($P = 0.01$) (Fig. 4c), identifying a possible clinical correlate for functional connectivity in glioma patients.

Reviewer 2 Comment 2: How are tumor-infiltrated areas defined in human participants? As gliomas are a whole-brain disease by definition, it would be great to have a (maybe already published) non-tumor cohort where the in-vivo electrophysiological measurements could be compared to. – Line 34, “normal appearing regions of brain”: Where exactly? How far away from tumor region? How determined that this region is “normal” (inspection; 5-ALA; MRI; ephys;...)

Reviewer 2 Response 2: *Thank you for this this incredibly interesting comment. As reviewer 2 described, high-grade gliomas are by definition a “whole brain” disease. In this study, glioma-infiltrated cortex is defined by the presence of expansile FLAIR signal on magnetic resonance imaging (MRI). Electrode localization for in-vivo ECoG recordings are detailed in the Methods under Human Electrocorticography (ECoG) and Data Analyses section (provided here for your convenience). Tumor boundaries were localized on MRI scans and electrodes overlying the hyperintense core of the tumor on T1 post gadolinium enhanced sequences, extending through the contrast enhancing rim to the edge of FLAIR were considered ‘infiltrative margin’. In contrast, electrodes completely outside of any FLAIR signal abnormality were termed ‘healthy’ in our original submission. In retrospect, we agree with Reviewer #2 that that a more succinct and biologically appropriate term for this region of brain would be ‘normal appearing’ or ‘non-tumor’. It has been well established in the literature that despite lack of grossly evident tumor seen by structural MRI, this does not preclude the presence of microscopic disease histologically within non-FLAIR regions of brain in patients with glioblastoma. In fact, we have recently quantified the extent of tumor infiltration within glioma infiltrated cortex by acquiring glioma margin specimens for tumor specific tissue characterization^{19,20}. Used generalized linear mixed models, we found that glioma specific characterization (IDH and P53) confirming scant or even abundant mitotically active malignant cells throughout the FLAIR region (particularly 1.5 cm away from the infiltrative margin)²⁰. As stated above, all glioma-infiltrated electrodes in this study were within the FLAIR region. The extent to which a quantified specific number (or percentage) of malignant cells (and therefore tumor burden) influences neuronal circuit dynamics is a fascinating question which we have not addressed in this paper. As we are sure Reviewer #2 is aware, glioma-specific markers are limited and as stated in our comments to reviewer 1, the less common IDH mutant gliomas were excluded from these experiments. Therefore, we limited regions outside of visibly abnormal FLAIR*

signal on MRI as “normal-appearing” and have redefined this terminology in our resubmitted draft.

Reviewer #2 also made the excellent comment regarding control experiments for all human electrophysiology experiments. As a reminder all human ECOG experiments are obtained under clinical context therefore we take institutional review and approval of our study protocol very seriously. The human in-vivo electrophysiology measurements performed in these experiments are identical to experiments performed in epilepsy patients using both implanted and intraoperative recordings. The presence of time specific audiovisual speech responses within the lateral prefrontal cortex (LPFC) has previously been published within the context of human epilepsy therefore this experiment was not added to the study (only appropriate citations) ¹⁷⁻²². The impressive body of work includes many but not limited to the following references⁷⁻¹²: Chang et al PNAS 2013, Fonken et al J Neurophysiol 2016, Haller et al Nat Hum Behav 2018, Chang et al Nat Neurosci 2010. However, glioma remodeling of functional circuits may impact broader cortical regions of speech initiation, therefore separate control conditions would be beneficial. Control conditions for this experiment are based on the absence of pre speech onset HGp prior to task administration followed by post speech onset HGp suppression with this distinct electrophysiological pattern replicated over hundreds of stimuli, trials, and electrodes (Fig. 1b-d). However, as an additional positive control demonstrating preserved speech initiation cortical responses, we have included data below for a diffuse glioma control in which cortical sampling is obtained under clinical context from LPFC for a non-cortically projecting tumor within the insular cortex. Identical to the non-tumor electrodes from cortically projecting gliomas, we discovered that group-level HGp demonstrates the expected neural time-course within LPFC, showing activation anterior to primary motor cortex between 600 milliseconds (ms) before speech onset (0 ms), and maximal activation in motor cortex at speech onset consistent with prior established models of speech initiation.

Because of this excellent comment, we have included new control experiments in the revised manuscript as extended figure 2b. We are confident that these two control conditions add clarity to the human ECoG behavioral experiments.

Reviewer 2 Comment 3: Couldn't it be possible that the detection of “task-relevant neural activity within the entire region of tumor-infiltrated cortex” is less a tumor-specific phenomenon, but rather an unspecific feature that occurs after all kind of brain lesions? Again, reference data / recordings in this respect? – Line 53, increased HGp- again, diffuse glioma-specific? Comparison to other brain lesions (e.g. obtained by surgery of non-infiltrative brain tumors, or during epilepsy surgery) would be very helpful to understand this important point better.

Reviewer 2 Response 3: *The extent to which diverse and differing brain cancers interface with the non-neoplastic microenvironment (i.e., non-tumor neurons and glia) ultimately influencing survival remains incompletely understood. The insightful comment that non-glioma controls would be helpful for data interpretation was raised by Reviewer #1 and is a very important concept to address. As a matter of context, the clinical condition in which one can obtain task-specific cortical electrophysiology data passively, with the spatial and temporal resolution required for these analyses are centered on diagnosis such as epilepsy, low and high-grade gliomas, as well as brain metastasis within cortical and subcortical regions of brain of presumed functional significance.*

We have addressed this comment by first expanding our analysis of task-specific responses within the LPFC and task-specific hyperexcitability which was identified in IDH-wild type glioblastoma-infiltrated cortex. We accomplished this goal by expanding analysis to WHO 2 IDH mutant oligodendroglioma and astrocytoma-infiltrated cortex. We then analyzed task speech initiation responses within LPFC from a non-glial origin tumor (lung adenocarcinoma) in order to understand if this biology is generalizable across brain cancers. We discovered that across glioma subtype (WHO 2-4), tumor intrinsic task specific neuronal responses for speech initiation are maintained within the LPFC.

Next, we separate data by glioma subtypes illustrating this conserved phenotype with task-specific cortical hyperexcitability observed only in participants with glioblastoma.

Interestingly, brain metastasis infiltrated cortex (thin overlying cortical mantle) for lung adenocarcinoma illustrates similar preserved task-specific responses suggestive of the conservation of neuronal signals within the tumor neuronal microenvironment. This stands in stark contrast to ECoG recording from complete cortical infiltration with no remaining overlying cortex from brain metastasis setting in which no HGP neuronal activity is identified (data not shown here).

While our initial goal was to investigate glioma-circuit remodeling across molecular subtypes of glial tumors, it quickly became evident that separation by molecular subtypes may yield markedly different mechanisms, particularly given our goal of applying circuit and cellular level drivers of cortical remodeling. The observed pattern of task-specific neuronal activity within glioma-infiltrated cortex is preserved across brain cancers. However, the temporal pattern and balance of excitatory to inhibitory inputs likely differ. During the review and resubmission period for this paper, we identified no cortically projecting glioneuronal tumors such as DNET or ganglioglioma. After inquiry with a wide network of collaborators and colleagues, we were unable to determine whether glioneuronal tumors maintain task-specific cortical responses. While task-specific cortical hyperexcitability is identified across tumor subtypes, only glioblastoma demonstrated hyperexcitability in our analysis. Furthermore, the temporal pattern of behavioral responses may have implications on task accuracy (which is a focus of our future work).

Lastly, the notion that within the periglioma microenvironment, that glioma-infiltrated cortex engages in network level activity is an entirely novel view of brain cancer. In fact, we have been fascinated by the concept that despite task relevant neuronal activity, patients with gliomas often present with neurological impairments which raises the important question of why. Inspired by the control experiments proposed by Reviewer #2, we have now applied neuronal decoding experiments in this dataset. Recently Leonard and others have studied how speech sequence are encoded, by using high-resolution direct cortical recordings from human lateral superior temporal cortex as subjects listened to words and non-words with varying transition probabilities between sound segments²¹. They found that neural responses encoded language-level probability of upcoming speech sounds suggesting acoustic representations with linguistic information encoded within neural substrate. We therefore applied a similar information theory, decoding framework within tumor-infiltrated and normal appearing cortical signals. In this experiment, audiovisual stimuli are separated and evenly

presented to the study participant based on level of complexity. Beyond maintained temporal representations of neuronal activity within glioma-infiltrated cortex, glioma-infiltrated cortex was unable to predict nuanced aspects of word retrieval such as word frequency (see below). Vocalization of low frequency words, for instance, requires a more intricate coordination of articulatory elements than that of high frequency words. In order to identify differences in computational properties of normal-appearing and glioblastoma-infiltrated cortex, we determined the decodability of their signals using a regularized logistic regression classifier to distinguish between low and high frequency word trial conditions using event-related responses. We implemented identical training and leave-one-subject-out cross-validation paradigms for both conditions. We have incorporated this new analysis in the revised manuscript as Fig. 1e. Despite normal appearing task-specific neuronal activity, glioma-infiltrated cortex maintains the ability to perform basic computational properties, yet loses the ability to perform complex aspects of speech and cognition.

In line with Reviewer #2 recommendations that this manuscript remain focused on glioblastoma (IDH-wild type), we have provided the above additional control experiments as extended figures 2b-e in the revised manuscript. The manuscript text has been revised to read as follows (updated text is in italics):

*The manuscript text under the heading ‘**Glioblastomas remodel functional neural circuits**’ has been revised to read as follows (updated text is in bold italics):*

High-grade gliomas interact with normal neuronal elements, resulting in both cellular and network level changes. While high-grade gliomas influence neuronal excitability at rest, the effects of task-related activity on glioma-infiltrated neural circuit function and the impact of glioma-neuron interactions on neural circuit connectivity remain unknown. To examine cognitive task-related neuronal activity in the setting of high-grade glioma, we selected a cohort of 14 adult patients with cortically projecting glioblastoma in the lateral prefrontal cortex (LPFC) (Extended Data Fig. 2a), classically referred to as Broca's area (Extended Data Fig. 3). In the operating room, tumor boundaries were localized on magnetic resonance imaging (MRI) and electrocorticography (ECoG) electrodes placed over the tumor-infiltrated cortical region and normal-appearing cortex. ECoG signals filtered between 70-110 Hz were used for analysis of the high-gamma band range power (HGp), which is strongly related to local neuronal population spikes and is increased by cortical hyperexcitability. ***Spectral data demonstrated the expected pattern of HGp increasing above 50 Hz in addition to clear separation of frequencies across tumor and non-tumor electrodes (Fig. 1a, Extended Data Fig. 4a).***

ECoG was recorded from the dominant hemisphere LPFC during auditory and visual picture naming tasks as an illustrative example of a well-defined cognitive neuronal circuit with defined physiology. In humans, speech initiation occurs in the LPFC (Broca's speech area, Brodmann area 44). While patients were fully awake and engaged in these language tasks, HGp was recorded from single electrodes overlying tumor-infiltrated and normal-appearing regions of brain (Extended Data Fig. 4b). These recordings provide simultaneous high spatial and temporal resolution while sampling the neuronal population activity during auditory and visual initiation of speech within the LPFC.

Group-level HGp from non-tumor electrodes and control conditions demonstrates the expected neural time-course within LPFC, showing activation anterior to primary motor cortex between 600 milliseconds (ms) before speech onset (0 ms), and maximal activation in motor cortex at speech onset (Extended Data Figs. 2b, 4c, d), consistent with prior established models of speech initiation previously demonstrated in non-human primates and humans. We then performed the same time series focused only on electrode arrays recording from tumor-infiltrated cortex. Countering the theory that glioblastoma-synaptic integration may result in physiologically disorganized neural responses, we found task-relevant neural activity within the entire region of tumor-infiltrated cortex. Strikingly, this includes speech initiation-induced recruitment of not only LPFC, Broca's region, as expected, but also regions of tumor-infiltrated cortex not normally involved in speech initiation (Fig. 1b). ***Similarly, we found that across WHO grade 2-4 glioma subtype, tumor intrinsic task-specific neuronal responses for speech initiation are maintained within the LPFC (Extended Data Fig. 2d, e).*** Taken together, these findings suggest that in subjects with glioblastoma affecting the dominant hemisphere LPFC, naming tasks induce physiologically organized neuronal activity within tumor-infiltrated cortex, well beyond the cortical territory normally recruited during this language task.

Task-evoked neural responses from tumor-infiltrated regions may be less than in non-tumor tissues. Thus, we wanted to understand whether the magnitude of task-related neural activity within tumor-infiltrated regions of brain oscillates similar to non-tumor regions. We therefore pair-matched each cortical electrode array (Extended Data Fig. 4d, e) which confirmed increased HGp within this expanded region of cortex infiltrated by tumor ($P = 0.016$) (Fig. 1c, d). ***Given the presence of coordinated neural responses,***

we set out to determine whether there are alterations in the high gamma activity elicited by computationally demanding tasks varied across cortical conditions. Vocalization of low frequency words, for instance, requires a more intricate coordination of articulatory elements than that of high frequency words^{22,23}. Therefore, in order to identify differences in computational properties of normal-appearing and glioblastoma-infiltrated cortex we determined the decodability of their signals using a regularized logistic regression classifier to distinguish between low and high frequency word trial conditions using event-related responses in the anterior temporal lobe (Fig. 1e). We implemented identical training and leave-one-subject-out cross-validation paradigms for both conditions. Normal-appearing cortex produced above-chance decoding between low and high frequency word trials (mean classifier accuracy = 0.56, $P < 0.001$). By contrast, glioblastoma-infiltrated cortex was not able to decode word trials above chance (mean classifier accuracy = 0.49, $P = 0.72$). These data further demonstrate that glioblastoma integration into cortical regions results in functional neural circuit remodeling and task-specific hyperexcitability, yet there is loss of computational properties within these cortical regions (Fig. 1f).

Reviewer 2 Comment 4: Fig. 1g: This panel needs much better explanation and guidance: where exactly are non-tumor areas? What are we actually seeing? Is there a quantification to substantiate the conclusions? I find it difficult to understand this panel and the conclusions.

Reviewer 2 Response 4: *Thank you for bringing this to our attention. We have modified this panel (see below) and have added additional information in the figure legend for clarification in the revised manuscript. Figure 1f (in the revised manuscript) is simply a graphic representation of cortical language high gamma power (HGp) activation during speech production with side-by-side comparison of tumor-infiltrated and “normal appearing” non-tumor regions. The goal was to demonstrate for the general non-neuroscience audience that the “normal appearing” non-tumor regions demonstrates similar appearance to classic physiology during speech production as evidenced by language activation in the canonical cortical language regions (top image). Bottom image shows that although the speech initiation neural signature in tumor-infiltrated cortex appears to mirror normal physiology, the spatial representation of cortical activation is greater compared to the non-tumor regions. The intent is to demonstrate the spatial heterogeneity within glioma-infiltrated cortex and reinforce the concept that brain cancer, in this context maintains functional activity and may be viewed as more than a neoplastic process. A revised figure and legend are included below.*

Speech initiation sites

Graphic representation of cortical language high gamma power (HGp) activation during speech production with side-by-side comparison of tumor-infiltrated and “normal appearing” non-tumor regions. Top image shows that the speech initiation neural signature in non-tumor regions closely resembles normal physiology as illustrated by activation in the canonical cortical language regions. Bottom image demonstrates that although the speech initiation in tumor-infiltrated cortex appears to mirror normal physiology, the spatial representation of cortical activation is much greater compared to the “normal appearing” non-tumor regions. Altogether, these data illustrate task-specific, speech initiation circuit remodeling within cortical language regions and also demonstrates spatial heterogeneity within glioma-infiltrated cortex.

Reviewer 2 Comment 5: Line 72/73, LFC vs HFC tumor regions: what is the characteristics of these regions (MRI? All with pathological signal? CE? Glioma subtype? – etc, see above). – Judged from 2d and 2e, the regions appear very tumor cell-low density, so most probably they are MRI negative? (Everything else would be difficult to comprehend). – This is also important for clinical translation of these findings: which parts of this multi-stage disease (with different stages of progression and brain infiltration at different anatomical sites, which is always present in the same patient in this disease) have been studied here?

Reviewer 2 Response 5: *This is an excellent point. Sampling of functionally connected intratumoral regions using MEG was performed exclusively in participants with perisylvian newly diagnosed IDH wild type glioblastoma. Site-directed tissue biopsies from HFC and LFC regions were taken from imaging defined tumor margins as determined by MRI. Given that surgery goals include resection of the T1 post gadolinium enhanced as well as FLAIR region (when deemed safe by the treating surgeon), primary patient samples were obtained within this context. Please note that all glioma subtypes are GBM IDH-wild type. The table and figure below illustrate site-specific sampling location of each annotated specimen as it related to contrast enhancing (CE) region and FLAIR tumor. Site-specific samples were acquired without regard for whether they originated from enhancing or FLAIR regions. A greater number of HFC samples originated from contrast enhanced regions of IDH-wild type glioblastoma however direct comparison did not meet statistical significance. We have incorporated these analyses as Extended Data Fig. 5 in the revised manuscript.*

RAW COUNTS			Fraction of Total (%)		
	CE	FLAIR		CE count	FLAIR count
HFC	26	19	HFC	57.78	42.22
LFC	26	32	LFC	44.83	55.17

All site-specific primary patient samples were acquired from newly diagnosed IDH-wild-type glioblastoma within perisylvian language regions of the frontal-temporal-parietal lobes. While samples were not acquired based on whether they originated from contrast enhancing or FLAIR regions, stereotactic coordinates of each sample were acquired. While 57.78% of HFC samples originated from contrast enhancing (CE) regions, this did not reach statistical significance $P = 0.1923$ chi square, $P = 0.235$ Fisher's exact test.

The manuscript text has been revised to read as follows (updated text is in bold italics):

Synptogenic glioma cells are functionally organized

Having established that gliomas functionally remodel neuronal circuits, we next wanted to understand whether functional integration heterogeneity exists within a specific molecularly defined high-grade glioma subtype. Given the heterogeneity of glioblastoma subpopulations and our previous finding that oligodendrocyte precursor cell-like subpopulations are enriched for synaptic gene expression, functionally connected regions may vary within tumors and differences in functional connectivity between tumor regions may be due at least in part to precursor-like subpopulations of glioma cells with differential synaptic enrichment or astrocyte-like subpopulations with differential synptogenic function. With the goal of sampling functionally connected regions within tumor, we measured neuronal oscillations within glioma-infiltrated brain by magnetoencephalography (MEG) and imaginary coherence functional connectivity for subjects with newly diagnosed IDH-WT glioblastoma. The functional connectivity of an individual voxel was derived by the mean imaginary coherence between the index voxel and the rest of the brain. Functional connectivity brain regions were separated into upper tertile (high connectivity, HFC) and low tertile (low connectivity, LFC) which permitted site-directed tumor sampling. ***Intratumoral functional connectivity correlated with tumor-intrinsic neuronal activity and HFC voxels were identified within both enhancing intratumoral regions as well as FLAIR hyperintense regions (Extended Data Fig. 5a-d).***

Reviewer 2 Comment 6: Line 88: ONLY non-tumor astrocytes express TSP-1: the data tell something different. According to Ext. Data Fig. 5 legend, 2.95% of HFC vs 1.59% of LFC express TSP-1. Moreover, statistics is required to substantiate the claim that there is a difference between HFC vs LFC.

Reviewer 2 Response 6: Thank you for your comment. We agree that (page 6 line 88 in the initial submitted manuscript) is confusing as written and requires clarification. As mentioned in the figure legend, 'percentage' denotes the number of TSP-1-positive tumor cells and non-tumor astrocyte populations in HFC and LFC primary patient samples. It is important to note that most single cell sequencing pipeline does not quantify the expression level of any particular gene. For example, cells with many reads

and cells with few reads are both considered TSP-1-positive above a defined threshold. The above statement was specifically made in reference to TSP-1 expression level (gray vs. red scale that indicates the level of expression among the positive cells) in our non-tumor astrocyte cell population as compared to tumor cells from low connectivity sampled regions (bottom row of the dot plot). We have modified the main text to clarify the above point and have added statistical analyses to the figure to highlight the striking difference in the level of TSP-1 expression among tumor cell population in HFC and LFC samples. We believe that this analysis allows better quantification along the y-axis. The central theme of these experiments is that a subpopulation of astrocyte like glioblastoma cells maintain the capacity to express TSP-1 which contributes to capacity for these regions to maintain network connectivity within the larger neuronal network. The figure below therefore illustrates the increased TSP-1 expression of tumor cell population within HFC derived glioblastoma primary patient samples compared with LFC ($P = 1.4 \times 10^{-7}$). We have added these additional analyses to Extended Data Fig. 8 in the revised manuscript.

Single-cell RNA seq

a, Dot plots showing TSP-1 expression (gray to red scale) and percentage (number of cells expressing the gene) of TSP-1-positive cells in tumor cells and non-tumor astrocyte populations in HFC and LFC samples ($n = 3$ per group). Out of the total HFC tumor cells ($n = 5325$, 3 patients), 157 cells are TSP-1 positive accounting for a percentage of 2.95, while only 1.59% (51 cells out of a total of 3212 LFC tumor cells [$n = 3$ patients]) express TSP-1. However, in the non-tumor astrocyte population, the number of TSP-1-positive cells are higher in LFC ($n = 34$ out of a total of 41 astrocytes, accounting for 82.9%) compared to HFC ($n = 15$ out of a total of 20 astrocytes, accounting for 75%) samples. **b**, Violin plots illustrating significantly increased TSP-1 expression within HFC region glioblastoma cells

relative to LFC regions ($P = 1.4 \times 10^{-7}$). **c**, Compared to HFC, slight trend of increased *TSP-1* expression within non-tumor astrocytes of LFC population. However, this trend did not reach statistical significance, likely due to the small number of non-tumor astrocytes captured ($P = 0.45$).

The manuscript text under the heading **'Synaptogenic glioma cells are functionally organized'** has been revised to read as follows (updated text is in bold italics):

To further determine whether any specific tumor cell population may be contributing to *TSP-1* expression, we performed single-cell sequencing of biopsy samples from HFC and LFC tumor regions (Extended Data Table 4). Malignant tumor cells were inferred based on the expression programs and detection of tumor-specific genetic alterations including copy number variants (Extended Data Fig. 7a-d). We found that 2.44% of all tumor cells (HFC and LFC combined) expressed *TSP-1* and within this *TSP-1*-positive tumor cell population, HFC cells exhibit higher levels of *TSP-1* compared to LFC (Fig. 2a). Within LFC region samples, *TSP-1* expression originated primarily from a non-tumor astrocyte population (as determined by *S100* expression) (Extended Data Fig. 7e-g). ***This data suggests that within low connectivity intratumoral regions, it is the non-tumor astrocyte population that shows higher levels of TSP-1, while within HFC regions, high-grade glioma cells express TSP-1 in addition to non-tumor astrocytes and myeloid cells, which may promote increased connectivity (Extended Data Fig. 8a-c).***

Reviewer 2 Comment 7: Fig. 2b: Why is TSP1 also upregulated in myeloid cells? How can the authors be sure that the upregulation is specific? Could it be normalized to the expression of e.g. myeloid cells and re-analyzed? Were all datasets integrated analyzed?

Reviewer 2 Response 7: Thank you for asking this fascinating question. While *TSP-1* expression in myeloid cells is not the focus of the paper, we can offer a speculative answer supported by the new analysis of our existing dataset. Myeloid cells, which include bone marrow-derived macrophages, microglia, myeloid-derived suppressor cells, dendritic cells and neutrophils are the critical regulators of immune response in the glioblastoma tumor microenvironment. It has previously been established that microglial cell surface molecules, including CD36 and CD47 function as *TSP-1* receptors^{24,25}. Hence the higher expression of *TSP-1* in myeloid cell population of HFC region primary patient samples could be attributed to the elevated overall *TSP-1* expression profile of high connectivity regions which could in turn enhance the binding of *TSP-1* to the myeloid cell surface receptors. We therefore based on prior publications within gliomas, believe that this effect is glioma-specific and may highlight the importance of glioma-neuron-immune crosstalk. We have confirmed these findings with the above violin plot sub analysis) as well as orthogonal validation via protein expression of *TSP-1* within patient derived glioblastoma tissues (Revised manuscript Fig. 2b and Extended Data Fig.9). As for the relative contributions of *TSP-1* mediated connectivity outside of the immune microenvironment i.e., myeloid cell normalization, the single-cell sequencing data is already normalized and scaled using standard workflows (offered in detail in the methods section). A direct and quantitative comparison of *TSP-1* expression in glioma cells can be seen in the Extended Fig. 8 of

the revised manuscript- provided above). All single-cell sequencing datasets were integrated and analyzed together. All the feature plots splitting up cells by tissue type (HFC vs LFC) are done for visualization purposes, however the underlying data were analyzed together.

*The manuscript text under the heading ‘**Synaptogenic glioma cells are functionally organized**’ has been revised to read as follows (updated text is in bold italics):*

To further determine whether any specific tumor cell population may be contributing to *TSP-1* expression, we performed single-cell sequencing of biopsy samples from HFC and LFC tumor regions (Extended Data Table 4). Malignant tumor cells were inferred based on the expression programs and detection of tumor-specific genetic alterations including copy number variants (Extended Data Fig. 7a-d). We found that 2.44% of all tumor cells (HFC and LFC combined) expressed *TSP-1* and within this *TSP-1*-positive tumor cell population, HFC cells exhibit higher levels of *TSP-1* compared to LFC (Fig. 2a). Within LFC region samples, *TSP-1* expression originated primarily from a non-tumor astrocyte population (as determined by *S100* expression) (Extended Data Fig. 7e-g). This data suggests that within low connectivity intratumoral regions, it is the non-tumor astrocyte population that shows higher levels of *TSP-1*, while within HFC regions, high-grade glioma cells express *TSP-1* in addition to non-tumor astrocytes and myeloid cells, which may promote increased connectivity (Extended Data Fig. 8a-c).

Interestingly, myeloid cells, which include bone marrow-derived macrophages, microglia, dendritic cells and neutrophils, are the critical regulators of immune response in the glioblastoma tumor microenvironment (Extended Data Fig. 7e) and microglial cell surface molecules, including CD36 and CD47, can function as TSP-1 receptors. While the role of TSP-1 in the tumor immune microenvironment is not yet clear, myeloid cell expression of TSP-1 suggests that multiple cell types in the tumor microenvironment of HFC regions may contribute to altered synaptic connectivity.

Reviewer 2 Comment 8: Figure 2d: Can the authors explain why TSP1 is staining whole cells in glioma tissue? From published data a more punctate staining would be expected. Specificity experiments are needed (negative and positive control).

Reviewer 2 Response 8: *Thank you for your comment. Recently, Daubon et al. assessed TSP-1 expression in human patient samples from glioma grades II, III and IV by immunohistochemistry (IHC) and found that TSP-1 expression is significantly higher in grade IV compared to other lower grade gliomas²⁴. Importantly, the IHC staining pattern of TSP-1 we observed in our grade IV human glioblastoma tissue samples are identical to the staining demonstrated by Daubon’s group²⁴. To ensure that the IHC signal we observed is specific for the TSP-1 target, we ran a no primary antibody negative control in human GBM samples which showed complete absence of TSP-1 signal. This new negative control has been added to the existing figure (Revised manuscript- Extended Data Fig. 9). For further reference and comparison, we are also providing an image below of the whole cell TSP-1 staining pattern of human breast ductal carcinoma tissue taken from the product data sheet of the TSP-1 antibody we used for the IHC experiment.*

TSP-1 IHC- human breast ductal carcinoma tissue

Reviewer 2 Comment 9: Figure 2e: The nestin staining does not seem to specifically stain cells. Normally, this staining should also stain somata with the nucleus spared. The co-localization does not look convincing and could be attributed to nonspecific staining.

Reviewer 2 Response 9: *Non-specific staining is always a concern, and we agree with reviewer 2's astute comment. We considered a number of tumor-specific markers in this study. The ideal tumor specific marker would be IDH-1 however as stated above, we believe that the circuit dynamic experiments described in figure 1 may differ between IDH-wild type and IDH-mutant gliomas. Out of the remaining glioblastoma antibodies, Nestin is not expressed by mature neurons as demonstrated in manuscript extended figure 12 and is therefore optimal for this particular study (relative to other tumor and neuron-specific antibodies). We have revisited every immunofluorescence section used for these experiment, repeated quantification, confirmed with our UCSF Brain Tumor Center Neuropathology lead (Dr. Joanna Phillips) and feel that the presented results hold true. The representative image presented in the paper was not perfect and we feel*

that the following image better represents the data and is added as Fig. 2 in the revised manuscript. The typical Nestin staining of somata with nucleus spared is shown in HFC samples however our LFC representative image is now improved. We have also added a no primary antibody negative control image for both Nestin and TSP-1 to illustrate the degree of non-specific staining.

Reviewer 2 Comment 10: Line 110: “synaptogenesis and consequent remodeling of connectivity” – how well is this statement generally established in neuroscience? More data in this respect would be helpful.

Reviewer 2 Response 10: We really appreciate this important question. In our work its essential to frame the brain cancer biology discussion into a neuroscience framework. The future of rehabilitation strategies will be engineering efforts which modulate and decode neuronal activity within disease infiltrated cortex; therefore, understanding the mechanisms of circuit remodeling is essential. Beginning with the seminal ideas of Ramon Cajal, synaptically connected neurons are the essential substrate for higher cognitive functions²⁶. In 1894, Schleich et al. suggested that astrocytes may modulate

synaptic transmissions and the fundamental role for astroglia in cognition emerged²⁷. The process of synapse formation or synaptogenesis is tightly regulated in order to ensure that the correct connections exist between neurons. The concept of the tripartite synapse that revolutionized neuroscience by suggesting astrocytes participation in synaptic transmission through vesicular release of neurotransmitters²⁸. In clinical neuroscience, abnormalities in synaptogenesis are thought to be responsible for several common brain disorders including autism, Rett syndrome, mental retardation^{29,30}. While the molecular mechanisms underlying synapse formation are incompletely understood, several proteins including the neuroligin-neurexin complex and TSP-1 stand out as important regulators³¹. Thrombospondins are extracellular-matrix glycoproteins secreted by a number of cells including astrocytes, and thereby promote synaptogenesis³²⁻³⁴. In vivo validation of neural structures, circuit dynamics, and cognitive/behavioral assays are rare. Therefore, outside of the cancer neuroscience value, this study will have great general neuroscience appeal.

Reviewer 2 Comment 11: Figure 2f/g: PSD-95 seem partially to form bigger cluster than expected for synapses. Quantification of cluster size needed. Colocalisation of PSD95 and synapsin is needed to be sure that indeed synapses are detected. Differences of regions could explain the different ratios between NFHM/synapsin and NFHM/PSD95 ratios but this needs to be explained.

Reviewer 2 Response 11: *Thank you for this interesting observation which we had not initially made. To address this comment, we performed additional analyses to quantify cluster size and Syn-1 and PSD95 colocalization in primary patient-derived HFC and LFC tissue samples. Similar to the total number of puncta, we found that the PSD95 cluster size, a measurement of the relative area of PSD95-positive synapses as well as the number of colocalized Syn-1 and PSD95 puncta was significantly higher in highly connected (HFC) glioblastoma tissue samples compared to low connectivity regions. The increased PSD95 cluster size along with increased PSD95 puncta density is indicative of an increased synapse stability and synapse formation in high connectivity regions of glioblastoma³⁵. The new colocalization staining and analyses are added to the revised manuscript as Extended Data Fig. 11.*

Representative confocal images of primary patient-derived HFC and LFC tissues showing regions of synaptic puncta colocalization (white arrows). Orange, synapsin-1 (presynaptic puncta); red, PSD95 (postsynaptic puncta); green, neurofilament (neurons). Scale bar, 15 µm. Quantification of the number of co-localized pre- and postsynaptic puncta (HFC vs. LFC: 371.1 ± 37.13 vs. 212.1 ± 56.42; n = 2-3 per group) (P = 0.05). Quantification of postsynaptic PSD95 puncta size (HFC vs. LFC: 2.27 ± 0.67 µm² vs. 1.04 ± 0.10 µm²) (P = 0.0441). Data presented as mean ± s.e.m. P values determined by two-tailed Student's t-test.

The manuscript text under the heading **'Synaptogenic glioma cells are functionally organized'** has been revised to read as follows (updated text is in bold italics):

We similarly found increased postsynaptic **puncta density and cluster size** on neurons (PSD95-positive/neurofilament-positive) and synapsin-PSD95 puncta colocalization within HFC regions compared with LFC regions (Fig. 2d, Extended Data Fig. 10, 11). **Together these data are indicative of an increased synapse stability and synapse formation in high connectivity regions of glioblastoma suggesting a role for TSP-1 in glioma-associated neural circuit remodeling.**

Reviewer 2 Comment 12: Figure 2i: Colocalisation analyses together with synapsin are needed. In all synaptic analyses the cluster size needs to be determined. Are the cluster sizes different between glioma and normal synapses? – All in all, as it is, the data does not allow to convincingly assess the question whether structural synapse formation is really promoted or not.

Reviewer 2 Response 12: Thank you for your comment. Reviewer #2 raises an important consideration. In this study, we computed total synapses in HFC and LFC samples which includes quantification of both neuron-neuron combined with glioma-neuron synapses within all fields of view. We are building on the hypothesis that a population of TSP-1 positive astrocyte-like glioma cells through paracrine signaling support tumor intrinsic neuron-neuron synapse formation and therefore maintain functional circuit stability. The notion that glioma-neuron interactions promote functional connectivity and therefore by electrochemical synapses support cognition and behavior is entirely provocative and not addressed by this set of experiments. It is important to note, that for this experiment, our goal was to quantify puncta along neurons, not glioma-neuron synapses. However, in line with Reviewer #2's recommendation, Figures 2c and 2d in the revised manuscript illustrates that glioma cells from high connectivity regions have an enriched synaptogenic profile as determined by the expression of pre-synaptic marker, namely synapsin-1 and PSD95 postsynaptic marker. In line with reviewer 2 comments, we have quantified both Homer-1 puncta size and Syn-1 and Homer colocalization points in neuron-HFC/LFC glioma co-cultures. The new analyses

showing increased Homer puncta size and pre-and post-synaptic colocalization in neuron-HFC glioma co-culture compared to LFC cells are added to the revised manuscript as Fig. 2e.

Reviewer 2 Comment 13: Line 124: throughout the manuscript, it is important to understand how the technologies were exactly applied to measure intra-patient heterogeneity and inter-patient heterogeneity. How is intratumoral functional connectivity per individual patient measured and quantified – is it a composite value?

Reviewer 2 Response 13: There are many available options to measure functional connectivity within the human brain. All measures have inherent strengths and weaknesses. Our intent was to assess long-range ipsilateral hemispheric functional connectivity, knowing that IDH wild type glioblastoma often has regions of hypervascularity which could present a difficult to control confounding variable. We therefore centered on magnetoencephalography (MEG) as a measure of neuronal oscillations within a glioma segmentation mask with minimal influence of regional heterogeneity and internal normalization within patient controls. There are several important considerations for the source data in these experiments. Functional connectivity estimates were calculated using imaginary coherence (IC), a technique known to reduce overestimation biases in MEG data generated from common references, cross-talk, and volume conduction^{36,37}. Within the alpha frequency band, an artifact free 1-minute resting state recording was sampled. Each whole-brain oscillatory activity was co-registered to an individual patient's structural MRI which offered spatial normalization. These normalized values generated individual volume-of-interest voxels across the entire brain, each measuring 8 mm in diameter resulting in approximately 3000 voxels per subject. Our previously described NUTMEG software suite was used to generate connectivity maps. The functional connectivity of an individual voxel was derived by the mean IC between the index voxel and the rest of the brain, referenced to its contralesional pair (serving as an internal control). Therefore, as raised by reviewer 2, there are regions within gliomas with varying amounts of functional connectivity.

Additionally, there are individual patients with more or less functional connectivity. We have addressed these differences in our experimental model. Intratumoral

differences in functional connectivity were addressed by the following. In comparison to contralesional voxels, we used a two-tailed t-test to test the null hypothesis that the Z-transformed connectivity IC between the index voxel and non-tumor voxel is equal to the mean of the Z-transformed connectivity between all contralateral voxels and the same set of voxels. The resultant functional connectivity values were separated into tertiles: upper tertile (high functional connectivity [HFC]) and lower tertile (low connectivity [LFC]). Thereafter, functional connectivity maps were projected onto each individual patient's preoperative structural MR images. It was therefore the extremes of intratumoral connectivity (high-and low connectivity, HFC and LFC, respectively) that were analyzed for these experiments. Rather than raw values, each functional connectivity measure represents a z transformed value therefore it remains likely that the HFC distinction for one patient does not perfectly coincide with the HFC distinction in another patient's tumor (intertumoral heterogeneity). Finally, the population based overall survival analysis was based on the presence and absence of any HFC voxels within an individual tumor (Revised manuscript- Extended Data Fig. 21). Of the patients included in this analysis 62% had no functional connectivity voxels identified. We agree that greater clarification within the methods would be useful and have added the above additional details under the magnetoencephalography (MEG) recordings and data analysis methods section in the revised manuscript.

The manuscript text has been revised to read as follows (updated text is in bold italics):

Functional connectivity map

The functional connectivity of an individual voxel was derived by the mean IC between the index voxel and the rest of the brain, referenced to its contralesional pair. ***It is possible that there are regions within gliomas with varying amounts of functional connectivity. Additionally, there are individual patients with more or less functional connectivity. We have addressed these differences in our experimental model. Intratumoral differences in functional connectivity were addressed by the following:*** in comparison to contralesional voxels, we used a two-tailed t-test to test the null hypothesis that the Z-transformed connectivity IC between the index voxel and nontumor voxel is equal to the mean of the Z-transformed connectivity between all contralateral voxels and the same set of voxels. The resultant functional connectivity values were separated into tertiles: upper tertile (high functional connectivity [HFC]) and lower tertile (low connectivity [LFC]). Functional connectivity maps were created by projecting connectivity data onto each individual patient's preoperative structural MR images and imported into the operating room neuro-navigation console. Stereotactic site-directed biopsies from HFC (upper tertile) and LFC (lower tertile) intratumoral regions were taken and X, Y, Z coordinates determined using Brainlab neuro-navigation. ***Thus, only the extremes of intratumoral connectivity (high-and low connectivity, HFC and LFC, respectively) were analyzed for these experiments. Rather than raw values, each functional connectivity measure represents a Z-transformed value and therefore it remains likely that the HFC distinction for one patient does not perfectly coincide with the HFC distinction in another patient's tumor (intertumoral heterogeneity).***

Reviewer 2 Comment 14: Line 134: a 1.4 -fold upregulation of CLU is not impressive in such (proteomics) screening experiments. What is the statistics? What about the other factors here? Why was CLU selected (and many others not which appear much more upregulated?)

Reviewer 2 Response 14: Clusterin was selected as a protein of interest among others because it is structurally similar to TSP-1 and remains understudied. Furthermore, both proteins are of astrocyte origin. Both Clusterin and TSP-1 belong to the thrombospondin type I repeat (TSR) protein superfamily and similar to TSP-1, Clusterin has been demonstrated to promote invasion via regulation of TGFB-1 signaling^{38,39}. In addition to mediating the invasive tumor phenotype, astrocyte-secreted Clusterin has been found to promote excitatory synaptic transmission⁴⁰. However, we completely agree with Reviewer #2's comment that although it is fascinating that this secreted protein was identified within conditioned media from HFC samples, the Clusterin story was not further developed by additional functional experiments. Hence, we have omitted Clusterin from the manuscript in the revised version.

Reviewer 2 Comment 15: Figure 3a: Why was homer intensity quantified? What does this parameter tell us? What about Homer punctae density?

Reviewer 2 Response 15: Thank you for this comment. We quantified fluorescence intensity as a general measure to demonstrate immunoreactivity of Homer in HFC and LFC samples. Determination of Homer-1 expression by quantifying staining intensity has been demonstrated in prior publications^{41,42}. However, to address the reviewer comment, we have also analyzed Homer puncta density of neuron-organoid-HFC/LFC co-cultures. Consistent with immunofluorescence intensity data, the expression of Homer as indicated by homer puncta density (calculated by dividing the total number of puncta colocalized on GFP+ neurons measured with the area of the image field) is significantly higher in neuron organoid-HFC co-culture as compared to LFC cells. This new analysis has been added to the revised manuscript as Extended Data Fig. 13.

Neuron organoids (GFP-labeled) were generated from an iPSC cell line integrated with doxycycline-inducible human NGN2 transgene and co-cultured with RFP- labeled HFC and LFC cells (pseudo-colored white) for two weeks. Quantification of postsynaptic Homer-1 puncta density (calculated by dividing the number of puncta measured with the area of the image field) in 2-week induced neuron (iN) organoid sections (HFC vs. LFC: $0.044 \pm 0.0044 \mu\text{m}^{-2}$ vs.

$0.0181 \pm 0.0042 \mu\text{m}^2$; $n=2/\text{group}$, $n=2 \text{ organoids/group}$, $n= 30-40 \text{ cells/organoid section analyzed}$). $P = 0.0009$. Scale bar, $10 \mu\text{m}$. Data presented as mean \pm s.e.m. P values determined by two-tailed Student's t -test.

*The manuscript text has been revised to read as follows within methods section titled **Induced neuron organoid and glioma co-culture** (updated text is in bold italics):*

Live cell image analyses were performed using ImageJ software. Briefly, a region of interest (ROI) was drawn around each GFP-positive neuron organoid and the fluorescence intensity (integrated density) of the RFP-positive glioblastoma cells was measured in the outlined ROIs for each of the indicated timepoints. At the end of two weeks, organoids from HFC and LFC co-cultures were embedded in OCT and sectioned at $10 \mu\text{m}$ thickness for Homer-1 immunofluorescence staining.

Determination of Homer-1 expression was determined by analyzing Homer puncta density of neuron-organoid-HFC and LFC co-cultures.

Reviewer 2 Comment 16: Fig 3d, EM images: Please make clear where exactly the Immunogold particles are located. Color coding: for LFC-PDX, it appears that a synapse between two non-malignant neuronal structures is shown (pre- and postsynaptic). If yellow means pre-synaptic, only one of the two marks can be correct. Moreover, specificity of RFP is unclear. The clusters of immunogold in LFC-PDX that are clumped together are typically seen when non-specific staining occurs. Single immunogold particles are localized in the presynaptic bouton (HFC-PDX). Specificity controls are needed (negative control - not glioma bearing). What does the quantification mean (total number of synapses? Synapses per field of view? Synaptic density needs to be determined properly with either 3D reconstructions or at least stereological quantifications. Which role do perisynaptic contacts play? How many models have been analyzed? At least three pairs, rather six pairs are needed to make a point about HFC vs LFC. In general, it would be desirable to see more EM (and also patch clamp) experimental data for important parts of the study: to A) substantiate the existence of synapses, and B) to define the synaptic subtypes, and the mode of transmission (fast vs slow waves).

Reviewer 2 Response 16: *Thank you for your comment. We have addressed each of these concerns in detail including better annotation of the location of immunogold particles in revised manuscript figure 3b. The underlying premise of our work is the influence of a population of astrocyte-like glioblastoma cells which through paracrine signaling promote synapse structure assembly including neuro-neuron synapses. The electron microscopy data analysis and synapse quantification in this study was performed as previously described by Venkatesh et al.¹⁶ In this revised submission we now separately quantified neuron-neuron and neuron-glioma synapses in TSP-1 elevated HFC and TSP-1 deficient LFC patient derived xenografts. We found that in addition to the total number of synapses (neuron-neuron combined with neuron-glioma synapses), the number of neuron-glioma synapses (per high power field [hpf]) was significantly different between the HFC and LFC groups. One important consideration for these experiments is the nature of the primary cells utilized for the experiment. In each experiment, primary patient cultures are obtained directly from the operating room, without serial passage and represent passage (P) 0-1. For all immuno-EM staining, as a*

negative control, we performed secondary antibody only (no RFP primary antibody) condition and observed minimal non-specific randomly distributed immunogold particles across the tissue specimen (Extended Data Fig. 15). Furthermore, we did not observe any clustering of immunogold particles in the negative control group, indicating that the clusters of immunogold particles observed in LFC-PDX group reflects the actual presence and distribution of the RFP antigen recognized by the primary antibody. Reviewer 2 proposed an excellent additional control experiments to include immuno-EM of non-tumor bearing mice and we have incorporated this additional negative control in the revised manuscript (Extended Data Fig. 15). Reviewer 2 also requested clarification regarding the definition of total number of synapses. Total synapses number included the quantification of both neuron-neuron combined with glioma-neuron synapses per high power field. In response to the color-coding comment, we would like to make it clear that we used yellow to mark neurons (both pre-and postsynaptic) and red to denote postsynaptic RFP+ glioma cells (Fig. 3b in the revised manuscript). We have modified the figure legend and reworded the sentence to make the above point clear. The quantification represents the number of synapses per field of view, and we have used 2 HFC and 2 LFC patient lines with each line xenografted in 2 mice for a total of 8 mice repeated twice. Lastly, Reviewer #2 made the excellent comment that it would be desirable to perform electrophysiology analysis of our glioma-neuron co-culture conditions focused specifically on our PDX models. We agree completely that these experiments would nicely supplement and validate the human electrophysiology analysis. In order to substantiate the synapse structures identified by electron microscopy, we attempted hippocampal slice two-photon calcium imaging of neurons following xenografts of patient-derived P0-1 primary patient cultures to evaluate changes in neuronal hyperexcitability at baseline and following pharmacological and knockdown of TSP-1. However, after 2 attempts, we found no spontaneous activity which we believe may be due at least in part to the relative slow proliferation of primary patient cultures compared with established cell lines.

Previously published work using established commercial glioma cell lines and patient-derived glioblastoma stem cell lines have elegantly characterized neuron-glioma synapses and neuronal activity-induced calcium signaling and depolarization of glioblastoma cells^{15,16,24}, which in turn has been recently demonstrated to drive tumor microtubule formation and brain invasion^{17,43}. However, unlike the above published papers that used glioma cell cultures subjected to relatively long-term expansion and passaging in vitro, we used low-passage primary patient-derived brain tumor cells for generating our xenografts that were maintained in culture only for short period of time. Hence, the limited replicative capacity and narrow culture time of primary patient-derived cells used in our experiments precludes the extensive proliferative and invasive growth of tumor cells in vivo and also makes it difficult to recapitulate the neuronal population dynamics that has been previously reported with PDX models using calcium imaging and electrophysiology recordings^{16,43}. We therefore transitioned to a micro-electrode array glioma-neuron co-culture system known to perform well with low-passage primary patient cultures.

a, b Quantification of neuron-to-neuron (HFC vs. LFC: 4.352 ± 0.254 vs. 3.860 ± 0.184 , $P = 0.1381$) and neuron-glioma synapses (HFC vs. LFC: 0.704 ± 0.094 vs. 0.256 ± 0.075 , $P = 0.0005$) per high power field (hpf) in HFC and LFC xenografts. **c**, Specificity negative controls for immunogold labeling. (Left) HFC xenograft with secondary antibody only (no primary antibody) control and (right) non-glioma bearing negative control tissue demonstrating few randomly distributed immunogold particles across the tissue specimen. Scale bar, 1000 nm. Data presented as mean \pm s.e.m. P values determined by two-tailed Student's t -test.

Reviewer 2 Comment 17: Fig. 3f: How do the authors explain that LFC glioma cells have a HIGHER proliferation index than HFC glioma cells as baseline, and after co-culture with neurons, both show very similar proliferation indices? Isn't that in contrast to the other findings?+

Reviewer 2 Response 17: *Thank you for this important comment which is a central theme to the paper. TSP-1 high-expressing HFC cells proliferate in response to neuronal signals while LFC condition primary patient cultures proliferate independent of the influence of neurons. This idea is supported by the following evidence. HFC cells in the presence of neurons exhibited increased proliferation compared with LFC cells in co-culture condition however head-to-head comparison of these distinct populations of cells is not the primary point of these analysis. It should be noted that primary patient-derived sampling from HFC and LFC regions comprise distinct populations of cells, including neurons, tumor cells and non-tumor astrocytes (as demonstrated in Extended Figure 7d, e in the revised manuscript). Hence the higher proliferative potential observed at a tissue level in primary patient derived HFC samples (demonstrated by increased Ki67 staining, Extended Fig. 9 in the revised manuscript) could be mediated/driven by the presence of non-malignant cells such as neurons present in the tumor microenvironment. These results suggest that the ability of HFC cells to proliferate is contingent on the presence of neurons/neuronal secreted factors and that in the absence of neuronal signals, they acquire a dormant non-proliferative phenotype. In fact, the presence of glioma-neuron synapse structures within glioblastoma may represent a mechanism of treatment resistance, as identified by Varn et al. recent publication in Cell⁴⁴. We regret that this important point was not explained in great detail in the manuscript, and we have added the above point to provide further clarification in the revised manuscript.*

*The manuscript text has been revised to read as follows within methods section titled **Neurons promote glioma circuit integration** (updated text is in bold italics):*

HFC glioma cells exhibit a 5-fold increase in proliferation when cultured with neurons (from 4% to 21% EdU+ cells). In contrast, LFC glioma *in vitro* cell proliferation index (determined as the fraction of DAPI cells co-expressing EdU) is similar with and without hippocampal neurons *in vitro* (Fig. 3d, Extended Data Fig. 16). ***These results indicate that the ability of HFC cells to proliferate is contingent on the presence of neuronal secreted factors and that in the absence of neuronal signals, they tend to acquire more of a dormant tumor phenotype.***

Reviewer 2 Comment 18: Fig. 3g: Provide high-res images / ideally histological sections to validate TM nature. - Again, when assessing the spheroid invasion area, what sticks out as particularly low (significantly lower than all other groups) is HFC cells without conditioned medium, while HFC+mCM, and both LFC groups are higher. The question is: why is that? Together with 3f, it appears that HFC cells without neuronal interactions are particularly “malignancy-deficient”. Any hypothesis why this is the case? Any data to explain it?

Reviewer 2 Response 18: *We have performed Scanning Electron Microscopy (SEM) in the revised resubmission to examine tumor microtubes in greater detail¹⁷. Below we provide high resolution SEM images of HFC and LFC cells cultured in the presence or absence of neuronal conditioned media (NCM) that shows increased tumor microtube formation of HFC cells in the presence of neuronal conditioned media in comparison to TSP-1 high-expressing HFC cells alone in culture. Glioma cells actively communicate with non-glioma cells, including neurons and glial cells in the tumor microenvironment through paracrine signaling⁴⁵. Prior studies have demonstrated that this dynamic communication between glioma cells and neurons is critical for glioma growth and progression within a subset of malignant glioma cells. We believe that this important population of glioblastoma cells are enriched within HFC intratumoral regions. Hence it is not surprising that HFC glioma cells alone in culture, in the absence of neuronal signal, is exhibiting a “malignancy-deficient” phenotype. Our working hypothesis is that this glioblastoma subpopulation contributes to the invasive glioblastoma phenotype when in contact with secreted factors originating from neurons. It may therefore be entirely possible that the invasive drive of a subpopulation of cells to colocalize with neurons is a targetable phenotype to prevent tumor invasion. We are aware of recent work under review which is specifically focused on activity dependent glioma invasion. Because of this excellent comment, the SEM analysis was added as figure 3e of the resubmitted manuscript and included below.*

Reviewer 2 Comment 19: Figure 4a: The in-vitro monoculture proliferative capacity should be determined. How many cell lines? How many patient pairs? Knockdown/knockout of TSP1? Can this be addressed pharmacologically?

Reviewer 2 Response 19: Thank you for raising this point and we would be delighted to provide additional clarification regarding the above comments. In-vitro monoculture proliferative capacity of high (HFC) and low-connectivity (LFC) cells in culture is provided in manuscript Extended Figure 16a in the revised manuscript. In this experiment, we used 4 distinct primary patient-derived P0-1 cultures each from TSP-1 high-expressing HFC and TSP-1 deficient LFC conditions (without any genetic engineering of TSP-1 gene) to quantify the in-vitro monoculture proliferative capacity. This includes four separate patient pairs. As mentioned in the responses above, this EdU assay demonstrated that HFC glioma cells in vitro show low proliferation on their own; however, addition of neurons markedly increases their proliferative potential. In contrast, LFC glioma cell proliferation index (determined as the fraction of DAPI cells co-expressing EdU) is similar with and without hippocampal neurons in vitro (Extended Figure 16a in the revised manuscript).

In line with the reviewer comments, we believe that there would be great value in determining whether the observed HFC phenotype is causally related to TSP-1, and we have therefore performed additional experiments by both genetic and pharmacological targeting approaches to address the causal relationship of TSP-1 with the invasive and proliferative tumor phenotype of HFC glioblastoma cells. Primary patient-derived HFC tumor cells were either transduced with shRNA targeting TSP-1 to knockdown thrombospondin-1 or treated with the FDA approved drug gabapentin to pharmacologically inhibit TSP-1. We found that compared to control shRNA condition,

HFC cells transduced with TSP-1-shRNA exhibited significantly fewer number of tumor microtubes (Revised manuscript Fig. 3g), consistent with the known role of TSP-1 in tumor microtube formation¹⁷. Interestingly, knockdown of TSP-1 also resulted in significant reduction in the number of Ki67-positive proliferating tumor cells in the neuron-HFC glioma co-culture (Revised manuscript Fig. 4e). Changes in the proliferative potential of HFC cells in the presence of the TSP-1 inhibitor, gabapentin, was further assessed in both *in vitro* neuron-glioma co-culture and *in vivo* patient-derived HFC xenograft models. We found that similar to the gene editing results, pharmacological inhibition of TSP-1 using gabapentin significantly decreased the proliferation of HFC cells both *in vitro* (Revised manuscript Fig. 4f) and *in vivo* (Revised manuscript Fig. 4g, h). We are adding the relevant new figures below for your convenience.

In vitro- TSP-1 shRNA- Tumor microtubes and Ki67 analysis

(Left) Primary patient-derived HFC cells were transduced with shRNA control or shRNA TSP-1. Representative SEM images showing tumor microtubes and quantification of TMTs per cell from HFC shRNA-control and HFC shRNA TSP-1 conditions (HFC-shControl vs. HFC-shTSP-1: 1.44 ± 0.09 vs. 0.44 ± 0.18 , $n = 2/\text{group}$). ($P = 0.0012$). Scale bar, 20 μm . (Right) Primary patient-derived HFC cells were transduced with shRNA control or shRNA TSP-1. Representative confocal images from neuron-HFC glioma co-culture showing marked decrease in proliferation of HFC cells (as measured by the total number of human nuclear antigen (HNA)-positive cells co-labelled with Ki67 divided by the total number of HNA-positive tumor cells counted across all areas quantified) upon TSP-1 silencing using shRNA (HFC-shControl vs. HFC-shTSP-1: $59.63 \pm 4.88\%$ vs. $36.17 \pm 5.92\%$, $n = 2/\text{group}$) ($P = 0.0068$). Red, HNA (human nuclei); white, Ki67. Scale bar, 30 μm .

In vitro- Gabapentin treatment- Ki67 analysis

Representative confocal images from neuron-HFC glioma co-culture showing marked decrease in proliferation of HFC cells (as measured by the total number of human nuclear antigen (HNA)-positive cells co-labelled with Ki67 divided by the total number of HNA-positive tumor cells counted across all areas quantified) upon pharmacological TSP-1 inhibition using (32 μM) gabapentin (HFC vs. HFC + GBP: $66.67 \pm 5.82\%$ vs. $38.77 \pm 4.33\%$, $n = 2/\text{group}$) ($P = 0.0007$). Red, HNA (human nuclei); white, Ki67. Scale bar, 30 μm .

In vivo- Primary patient-derived HFC xenograft- Gabapentin treatment- Ki67 analysis

(Top row) Schematic representation of the in vivo gabapentin treatment paradigm of HFC patient-derived xenografted (PDX) mice. (Bottom row) Representative confocal images, and quantification demonstrating marked decrease in proliferation index (Ki67+HNA+/HNA+) of gabapentin treated mice bearing HFC xenografts (HFC + Vehicle vs. HFC + GBP: 1.00 ± 0.17 vs. 0.76 ± 0.14 , $n = 9$ mice/group) ($P = 0.046$). Red, HNA (human nuclei); white, Ki67. Scale bar, 70 μm . Data presented as mean \pm s.e.m (c-f, h). P values determined by two-tailed Student's t -test. * $P < 0.05$. ** $P < 0.01$. *** $P < 0.001$.

Reviewer 2 Comment 20: Line 215: tumor boundary: Needs better specification (see above). Any other factors (residual tumor mass, which could be higher in this situation and at the same time is a negative prognostic factor?). One would need to know more parameters to gain better confidence that the survival differences are (partly or mainly) due to the different MEG parameters.

Reviewer 2 Response 20: Thank you for raising this important point. Line 215 (now line 258 in the revised manuscript) in the discussion refers to a prior publication by Daniel et al which provides conflicting data demonstrating that intratumoral functional connectivity confers a survival advantage⁴⁶. This study is however not limited to patients with IDH WT glioblastoma and utilizes resting state functional connectivity which is confounded by tumor vascularity. Within this study we have outlined in the methods section about tumor boundary, definitions for extent of tumor burden, as well as volumetric extent of resection. We have now provided additional clarification as well as added a new and enhanced analysis which drives home the critical relationship between intratumoral functional connectivity and patient overall survival including the interactive effects of these variables. The Kaplan-Meier statistics provided in the initial submitted draft of the manuscript controlled for known variables of survival outcome and employed a homogenous sample of patients with chemoradiation treated IDH-wild type glioblastoma. However, in line with reviewer #2 point, it is well known that further confounds may impact overall survival outcomes and furthermore, the interactions

between these variables are essential. To address this, we have included a patient summary table of all molecular clinical variables (Extended Table 2 in the revised manuscript). Additionally, we have employed a second unsupervised multivariable analysis called recursive partitioning. Recursive partitioning creates a decision tree that strives to correctly classify members of a population by splitting into different survival risk groups. Each split in the tree therefore offers a hierarchy of importance with respect to the primary outcome which in this situation is overall survival. Recently, Molinaro et al.¹⁸ (new collaborator in this study) demonstrated the interactive effects of clinical, molecular, and therapeutic variables on survival outcomes in patients with newly diagnosed glioblastoma. The complete list of clinical and molecular variables in this study are listed in Molinaro and Hervey-Jumper et al. table 1 and supplementary table 1. Within this homogenous study population of Stupp protocol chemoradiation treated patients, we applied the same measure of intratumoral neuronal oscillations and classified patients by the presence or absence of intratumoral functional connectivity. This nested dataset included 70 patients, 35 events and a 20-month median follow-up period. Using this approach, three risk groups were determined by risk group stratification and it should be noted that our groupings changed in comparison with previous published results. We were astonished to see that intratumoral functional connectivity emerged as an important overall survival risk variable alongside extent of tumor resection and age at diagnosis. Risk group 1 (black) had the worst outcomes and are the combination of patients older than 72 and patients younger than 72 with less than 97% extent of tumor resection. Risk group 3 (gray) have the best survival, and these are patients younger than 62 with over 97% extent of tumor resection and without functional connectivity in the tumor. Intermediate risk group 2 (red) revealed an interesting interaction between age and HFC. This group had two subsets: patients with over 97% resection of tumor and age younger than 72 with intratumoral connectivity; and those between 62 and 72 years old without functional integration. Therefore, taken together, these data suggest that in humans, neuronal activity within malignant gliomas negatively impacts survival with importance demonstrated by machine learning segmentation of outcomes and quantified to the extent that the presence of neuronal activity may be the equivalent to older patient age regardless of the extent of tumor surgically removed. We have added these new enhanced analyses to the revised manuscript as Fig. 4a, b. New figure 4 and extended table 2 are listed below.

Modeling of survival risk in patients incorporating the effects of glioma intrinsic neuronal activity, therapeutic, and clinical factors on overall survival by recursive partitioning demonstrates 3 risk groups. Risk group 1 (black) have the worst outcomes and are the combination of 1) patients older than 72 and 2) patients younger than 72 with extent of tumor resection under 97.1%. Risk group 3 (gray) have the best survival, and these are patients younger than 62 with extent of tumor resection over 97.1% and no intratumoral neural oscillations. Intermediate risk group 2 (red) is the combination of patients with over 97% extent of tumor resection and either (1) age younger than 72 with neural oscillations identified within the tumor and (2) patients 10 years younger without functional integration.

Extended Data Table 2. Patient summary-clinical and molecular features

Study #	Sex	Age (yr)	Preoperative tumor volume (ml)	Residual tumor (ml)	EOR (%)	MGMT methylation	EGFR amplification	Tumor Location	Tumor type	IDH status
SF#1	M	56.07	23.67	0.00	100.00	no	nonamp	R frontal	GBM	wt
SF#2	M	64.93	9.20	0.00	100.00	yes	amp	L temporal	GBM	wt
SF#3	M	60.96	49.68	3.95	92.06	yes	nonamp	L temporal	GBM	wt
SF#4	M	60.42	20.30	0.00	100.00	yes	amp	L frontal/insula	GBM	wt
SF#5	M	76.72	8.94	0.00	100.00	no	nonamp	L frontal	GBM	wt
SF#6	M	72	23.59	4.79	79.70	yes	nonamp	L temporal	GBM	wt
SF#7	F	64.14	78.37	0.00	100.00	yes	amp	R frontal	GBM	wt
SF#8	M	78.12	77.49	0.00	100.00	yes	amp	L frontal	GBM	wt
SF#9	F	62.2	14.13	0.00	100.00	no	amp	L temporal	GBM	wt
SF#10	F	59.02	6.38	0.00	100.00	yes	amp	R frontal	GBM	wt
SF#11	M	57.26	46.07	1.09	97.64	no	nonamp	L temporal	GBM	wt
SF#12	M	55.34	53.10	7.33	86.19	yes	nonamp	L frontal	GBM	wt
SF#13	F	66.92	3.20	0.00	100.00	yes	amp	L temporal	GBM	wt
SF#14	M	29.3	36.88	0.00	100.00	yes	amp	R insula	GBM	wt
SF#15	M	51.15	29.67	2.63	91.14	no	nonamp	L thalamus	GBM	wt
SF#16	M	48.92	31.05	0.00	100.00	no	nonamp	L frontal	GBM	wt
SF#17	F	49.3	55.55	3.49	93.72	yes	amp	L frontal	GBM	wt
SF#18	M	60.98	67.50	0.00	100.00	yes	amp	L frontal	GBM	wt
SF#19	M	80.1	45.41	0.75	98.35	yes	nonamp	L parietal	GBM	wt
SF#20	F	72.89	6.12	0.00	100.00	yes	nonamp	R frontal	GBM	wt
SF#21	F	56.34	81.00	0.00	100.00	yes	amp	L parietal	GBM	wt
SF#22	M	63.22	7.01	0.53	92.41	yes	amp	L parietal	GBM	wt
SF#23	F	60.04	15.69	0.00	100.00	yes	amp	L parietal	GBM	wt
SF#24	F	69.62	92.46	0.00	100.00	no	nonamp	R frontal	GBM	wt

SF#25	M	60.3 8	101.96	0.21	99.7 9	yes	nonamp	R frontal	GBM	wt
SF#26	F	52.6 1	7.32	0.00	100. 00	yes	nonamp	L temporal	GBM	wt
SF#27	M	52.7 2	41.21	3.58	91.3 1	no	amp	R frontal	GBM	wt
SF#28	F	59.8 9	9.51	0.00	100. 00	no	nonamp	L temporal	GBM	wt
SF#29	M	44.9 2	34.59	0.00	100. 00	yes	amp	R temporal	GBM	wt
SF#30	F	65.5 3	15.77	0.48	96.9 6	no	unknown	Parietal	GBM	wt
SF#31	F	67.8 9	14.14	0.00	100. 00	yes	nonamp	L multifocal	GBM	wt
SF#32	M	71.4 9	21.80	0.00	100. 00	no	amp	L temporal	GBM	wt
SF#33	F	61.7 2	52.27	0.17	99.6 7	no	amp	R occipital	GBM	wt
SF#34	M	69.0 3	57.01	0.00	100. 00	yes	nonamp	L parietal	GBM	wt
SF#35	F	46.9	35.99	0.00	100. 00	yes	yes	R temporal	GBM	wt
SF#36	M	51.7 3	37.45	2.20	94.1 3	no	amp	L frontal	GBM	wt
SF#37	M	69.6 7	59.66	0.33	99.4 5	yes	nonamp	R parieto- occipital	GBM	wt
SF#38	M	69.0 6	74.40	2.49	96.6 5	yes	nonamp	R parietal	GBM	wt
SF#39	F	69.8 3	6.85	0.17	97.5 1	yes	amp		GBM	wt
SF#40	M	54.0 5	61.69	3.00	95.1 3	yes	amp	L frontal	GBM	wt
SF#41	F	57.3 9	9.25	0.00	100. 00	yes	nonamp	L frontal	GBM	wt
SF#42	M	60.2 1	101.06	0.41	99.5 9	no	nonamp	L temporal	GBM	wt
SF#43	M	76.7 5	44.02	1.13	97.4 3	yes	nonamp	R temporal	GBM	wt
SF#44	M	43.8 1	17.72	0.00	100. 00	no	amp	R frontal	GBM	wt

Reviewer 2 Comment 21: No electrophysiology from xenograft is shown. This is needed to understand which role the fast and slow currents (Venkatesh 2019, Venkataramani 2019) play. Can also more synapses be observed functionally with electrophysiology?

Reviewer 2 Response 21: *Thank you for this important comment. Reviewer 2 has proposed an excellent experiment. Previously published work using established commercial glioma cell lines and patient-derived glioblastoma stem cell lines have elegantly characterized neuron-glioma synapses and neuronal activity-induced calcium signaling and depolarization of glioblastoma cells^{15,16,24}, which in turn has been recently demonstrated to drive tumor microtubule formation and brain invasion^{17,43}. However, unlike the above published papers that used glioma cell cultures subjected to relatively long-term expansion and passaging in vitro, we used low-passage primary patient-derived brain tumor cells for generating our xenografts that were maintained in culture only for short period of time. Hence, the limited replicative capacity and narrow culture time of primary patient-*

derived cells used in our experiments precludes the extensive proliferative and invasive growth of tumor cells in vivo and also makes it difficult to recapitulate the neuronal population dynamics that has been previously reported with PDX models using calcium imaging and electrophysiology recordings^{16,43}.

As previously mentioned, this body of work is focused on the influence of malignant glioma cells on neuron-neuron interactions which in turn influence glioma-network circuit dynamics. Recent work published by Venkatesh and Monje et al is focused on both electrochemical synapses between malignant gliomas and neurons¹⁶. Further, we do not believe that functional synapses between malignant gliomas and neurons represents a driving force behind glioma-network network interactions. However, it is entirely possible that within our HFC patient derived xenografts, we may see neuronal hyperexcitability. To test this possibility of whether our observed glioma-associated microenvironment changes impact neuronal excitability, we employed a multielectrode array (MEA) to assess the electrophysiological properties of TSP-1 high-expressing HFC and TSP-1 deficient LFC cells in co-culture with neurons in an in vitro setting. To characterize the neuronal activity profile, we computed mean firing rate (MFR) and network burst frequency of the neuronal networks. MFR is defined as the mean number of spikes (extracellular action potentials) per second computed over the total 30 min MEA recording duration. Network bursts consist of densely packed spikes, called bursts, occurring simultaneously at multiple electrodes/channels and network burst frequency is defined as the total number of network burst computed over the total recording time. In addition to the neuronal activity analysis, we computed network synchrony by calculating the area under the normalized cross-correlogram (AUNCC), as described previously^{47,48}. AUNCC represents area under inter-electrode cross-correlation normalized to the autocorrelations, with higher values indicating greater synchronicity of the network.

Interestingly, we found that HFC cells in co-culture with neurons demonstrated a neuronal signature of hyperexcitability as evidenced by the significant increase in network burst frequency and AUNCC in comparison to the neuron only and neuron-LFC co-culture conditions (see figures below). More importantly, given the premise that HFC glioma cell-derived TSP-1 could serve as a molecular driver of this observed neuronal hyperexcitability, we targeted TSP-1 therapeutically using gabapentin (GBP), as demonstrated previously⁴⁹. We found that in neuron-HFC glioma co-cultures, the neuronal activity measures such as individual spikes (extracellular action potentials) and bursts (cluster of spikes in blue) as well as synchronized network bursts (pink) were significantly reduced after 24-48 h exposure to GBP. These new MEA electrophysiology analyses are incorporated in the revised manuscript (Figs. 2g, 4d, Extended Data Fig. 14). We have provided all relevant figures below as well as updated text in the manuscript below for your convenience.

a. Magnified view of MEA electrodes, showing RFP-labeled glioma cells in co-culture with neurons

b. Representative raster plot- mouse cortical neuron only condition

a. Magnified view of multi-electrode array (MEA), showing RFP-labeled glioma cells in co-culture with neurons. **b.** Representative raster plot showing individual spikes/extracellular action potentials (tick mark), bursts (cluster of spikes in blue) and synchronized network bursts (pink) of mouse cortical neuron only condition (DIV 18 of neuronal culture and 48 h timepoint for neuron-glioma co-culture). The cumulative trace above the raster plots depicts the population spike time histogram indicating the synchronized activity between the different electrodes (network burst).

Electrophysiological properties of glioma cells in co-culture with neurons were analyzed using multi-electrode array (MEA). Representative raster plots showing individual spikes/extracellular action potentials (tick mark), bursts (cluster of spikes in blue) and synchronized network bursts (pink) after 48 h co-culture of neurons with HFC and LFC glioma cells (outlined in red and blue, respectively). The cumulative trace above the raster plots depicts the population spike time histogram indicating the synchronized activity between the different electrodes (network burst). Quantification of network burst frequency (Hz) (defined as the total number of network bursts divided by recording time) and network synchrony (as measured by area under normalized cross-correlation, defined as the area under inter-electrode cross-correlation normalized to the autocorrelations) from HFC and LFC glioma-neuron coculture. Network burst frequency: cortical neuron (CN) only vs. CN + HFC vs. CN + LFC: 0.75 ± 0.21 Hz vs. 1.08 ± 0.10 Hz vs. 0.42 ± 0.20 Hz, $n = 2$ wells/condition, $n = 2$ per HFC/LFC group) ($P = 0.05$); Area under normalized cross-correlation: cortical neuron (CN) only vs. CN + HFC vs. CN + LFC: 0.37 ± 0.004 vs. 1.35 ± 0.03 vs. 0.68 ± 0.18 , $n = 2$ wells/condition, $n = 2$ per HFC/LFC group) ($P = 0.0129$ for CN vs. CN+HFC; $P = 0.0308$ for CN+HFC vs. CN+LFC); Data presented as mean \pm s.e.m (b-e, g). P values determined by two-tailed Student's t -test. * $P < 0.05$. ** $P < 0.01$. **** $P < 0.0001$. N.S., not significant.

The manuscript text has been revised to read as follows (updated text is in bold italics):

Synaptogenic glioma cells are functionally organized

We next sought to further investigate functional distinctions between malignant subpopulations isolated from HFC and LFC regions by testing neuron-glioma interactions in a 3D neuronal organoid model. We co-cultured HFC and LFC glioma cells with GFP-

labelled human neuron organoids, generated from an iPSC cell line integrated with a doxycycline-inducible human *NGN2* transgene to drive neuronal differentiation. Quantification of postsynaptic Homer-1 in induced neuron (iN) organoids revealed a relative increase in postsynaptic puncta density when co-cultured with HFC glioma cells compared to LFC glioma cells ($P = 0.0006$) (Extended Data Fig. 13). Live cell imaging of neuronal organoids co-cultured with HFC and LFC glioma cells revealed that HFC glioma cultures exhibit prominent neuronal tropism and integrate extensively in the organoids, while LFC glioma cells displayed minimal integration with neuron organoids (Fig. 2f, Supplementary Videos 1, 2). Strikingly, exogenous administration of TSP-1 to iN-LFC co-culture reversed this phenotype and promoted robust LFC glioma integration into the neuronal organoid (Fig. 2f, Supplementary Video 3), further implicating TSP-1 in neuron-glioma interactions. ***The electrophysiological properties of TSP-1-high expressing cells in co-culture with neurons were analyzed using multi-electrode array (MEA). After 48 h of coculture, the total number of network bursts (a measure of neuronal activity) from cortical neuron-coculture with TSP-1 over-expressing HFC cells was increased relative to cortical neurons alone and LFC co-culture conditions. Neurons in co-culture with HFC cells also demonstrated increased network synchrony as measured area under normalized cross-correlation (the area under inter-electrode cross-correlation normalized to the autocorrelations) (Fig. 2g, Extended Data Fig. 14).***

Electrophysiological properties of HFC-cortical neuron (CN) co-cultures exposed to a working concentration of 50 μ M gabapentin (GBP) were analyzed using multi-electrode array (MEA). Representative raster plots showing individual spikes/extracellular action potentials (tick mark), bursts (cluster of spikes in blue) and synchronized network bursts (pink) of neuron-HFC co-culture (outlined in red) and 24-48 h exposure of neuron-HFC co-culture to GBP (outlined in orange). The cumulative trace above the raster plots depicts the population spike time histogram indicating the synchronized activity between the different electrodes (network burst). Quantification of weighted mean firing rate (Hz) (defined as the spike rate per well multiplied by the number of active electrodes in the associated well) and network synchrony (as measured by area under normalized cross-correlation, defined as the area under inter-electrode cross-correlation normalized to the autocorrelations) from HFC and HFC+GBP glioma-neuron coculture. Weighted mean firing rate: CN + HFC vs. CN + HFC + GBP: 1.28 ± 0.09 Hz vs. 0.81 ± 0.16 Hz, $n = 2$ wells/condition, $n = 2$ /group); Area under normalized cross-correlation: CN + HFC vs. CN + HFC + GBP: 1.35 ± 0.03 vs. 1.09 ± 0.04 , $n = 2$ wells/condition, $n = 2$ /group).

The manuscript text has been revised to read as follows (updated text is in bold italics):

Glioma functional connectivity shortens survival

Given the premise that TSP-1 serves as a molecular driver of neuronal activity-driven proliferation, we sought to target TSP-1 therapeutically using gabapentin

(GBP). In neuron-glioma co-cultures, individual spikes (extracellular action potentials), bursts (cluster of spikes in blue) and synchronized network bursts (pink) were reduced after 24-48 h exposure to GBP (Fig. 4d). Primary patient-derived HFC cells were transduced with shRNA-control or shRNA-TSP-1 (Fig. 4e) or treated with gabapentin (Fig. 4f); genetic or pharmacological targeting of TSP-1 resulted in a marked decrease in HFC glioma cell proliferation when co-cultured with neurons. Finally, we treated mice bearing HFC patient-derived xenografts (PDX) and found a marked decrease in glioma proliferation index (Ki67+HNA+/HNA+) in gabapentin-treated mice bearing HFC xenografts relative to controls (Fig. 4g, h).

Reviewer 2 Comment 22: Extended Data Fig. 10: Glioma cell marker are needed, and quantification of MET-positive glioma cell density; mean pixel intensity is not really helpful here.

Reviewer 2 Response 22: Thank you for this excellent comment. Based on reviewer 2 comments we have reanalyzed tissue immunofluorescence images. The immunofluorescence image below shows increased MET-positive glioma cell staining as evidenced by the increased expression of Nestin and MET-double positive staining in primary patient-derived HFC tissue compared to LFC. We have included this new analysis to the existing IHC image as Extended Data Fig. 18 in the revised manuscript and provided an updated figure and legend below.

Reviewer 2 Comment 23: Line 236/237: "...and that distinct intratumoral regions maintain functional connectivity through a subpopulation of TSP-1 expressing malignant cells (HFC glioma cells)." - Functional connectivity? To make this claim, optimally

electrophysiological single cell data would be required, and/or ultramicroscopy/EM of TSP1 pos vs neg cells.

Reviewer 2 Response 23: *Thank you for bringing this point to our attention. This phrase was used in reference to (1) the physiological connectivity in our human in vivo electrophysiology experiments, (2) plus the observation of TSP-1 expressing malignant glioma cells within this region, and (3) the increased number of synapse structures identified following orthotopic xenografting. As reviewer #2 suggests, we have strengthened this claim with additional experiments demonstrating TSP-1 mediated neuron-neuron functional connectivity and have incorporated these findings in the revised manuscript (Figs. 2g, 4d, Extended Data Fig. 14). As mentioned in the comment above, we utilized MEA to characterize the electrophysiological properties of HFC and LFC cells and used a cross-correlation approach for functional connectivity analysis. Cross-correlation approach, which is based on the co-occurrence of spikes across different MEA channels is a well validated model for functional connectivity analysis and synaptic connectivity estimation of MEA electrophysiological recordings, as reported by prior studies^{47,50,51}. Cross-correlation measures the probability of a spike on one channel relative to the other, and if the spikes from one electrode tend to occur at a fixed time in relation to the spikes from the other electrode, a peak in the cross-correlogram occurs, which further indicates synchronized activity of the neuronal networks. We found that besides generating a hyperexcitable state, the TSP-1 expressing malignant HFC cells exhibits greater functional connections as observed by the significant increase in the area under normalized cross-correlation network synchrony measure compared to the LFC condition. More importantly, this increase in the network synchrony observed in the neuron-HFC glioma co-culture is eliminated by TSP-1 pharmacological inhibition using gabapentin. These data indicate a causal relationship between glioma cell-derived TSP-1 and network functional connectivity.*

*Electrophysiological properties of glioma cells in co-culture with neurons were analyzed using multi-electrode array (MEA). Representative raster plots showing individual spikes/extracellular action potentials (tick mark), bursts (cluster of spikes in blue) and synchronized network bursts (pink) after 48 h co-culture of neurons with HFC and LFC glioma cells (outlined in red and blue, respectively). The cumulative trace above the raster plots depicts the population spike time histogram indicating the synchronized activity between the different electrodes (network burst). Quantification of network burst frequency (Hz) (defined as the total number of network bursts divided by recording time) and network synchrony (as measured by area under normalized cross-correlation, defined as the area under inter-electrode cross-correlation normalized to the autocorrelations) from HFC and LFC glioma-neuron coculture. Network burst frequency: cortical neuron (CN) only vs. CN + HFC vs. CN + LFC: 0.75 ± 0.21 Hz vs. 1.08 ± 0.10 Hz vs. 0.42 ± 0.20 Hz, $n = 2$ wells/condition, $n = 2$ per HFC/LFC group) ($P = 0.05$); Area under normalized cross-correlation: cortical neuron (CN) only vs. CN + HFC vs. CN + LFC: 0.37 ± 0.004 vs. 1.35 ± 0.03 vs. 0.68 ± 0.18 , $n = 2$ wells/condition, $n = 2$ per HFC/LFC group) ($P = 0.0129$ for CN vs. CN+HFC; $P = 0.0308$ for CN+HFC vs. CN+LFC);. Data presented as mean \pm s.e.m (b-e, g). P values determined by two-tailed Student's t -test. * $P < 0.05$. ** $P < 0.01$. **** $P < 0.0001$. N.S.. not significant.*

The manuscript text has been revised to read as follows (updated text is in bold italics):

Synaptogenic glioma cells are functionally organized

We next sought to further investigate functional distinctions between malignant subpopulations isolated from HFC and LFC regions by testing neuron-glioma interactions in a 3D neuronal organoid model. We co-cultured HFC and LFC glioma cells with GFP-labelled human neuron organoids, generated from an iPSC cell line integrated with a doxycycline-inducible human *NGN2* transgene to drive neuronal differentiation. Quantification of postsynaptic Homer-1 in induced neuron (iN) organoids revealed a relative increase in postsynaptic puncta density when co-cultured with HFC glioma cells compared to LFC glioma cells ($P = 0.0006$) (Extended Data Fig. 13). Live cell imaging of neuronal organoids co-cultured with HFC and LFC glioma cells revealed that HFC glioma cultures exhibit prominent neuronal tropism and integrate extensively in the organoids, while LFC glioma cells displayed minimal integration with neuron organoids (Fig. 2f, Supplementary Videos 1, 2). Strikingly, exogenous administration of TSP-1 to iN-LFC co-culture reversed this phenotype and promoted robust LFC glioma integration into the neuronal organoid (Fig. 2f, Supplementary Video 3), further implicating TSP-1 in neuron-glioma interactions. ***The electrophysical properties of TSP-1 high expressing cells in co-culture with neurons were analyzed using multi-electrode array (MEA). After 48 hours of coculture, the total number of network bursts (a measure of neuronal activity) from cortical neuron-coculture with TSP-1 over-expressing HFC cells was increased relative to cortical neurons alone and LFC co-culture conditions. Neurons in co-culture with HFC cells also demonstrated increased network***

synchrony as measured area under normalized cross-correlation (the area under inter-electrode cross-correlation normalized to the autocorrelations) (Fig. 2g, Extended Data Fig. 14).

Electrophysiological properties of HFC-cortical neuron (CN) co-cultures exposed to a working concentration of 50 μ M gabapentin (GBP) were analyzed using multi-electrode array (MEA). Representative raster plots showing individual spikes/extracellular action potentials (tick mark), bursts (cluster of spikes in blue) and synchronized network bursts (pink) of neuron-HFC co-culture (outlined in red) and 24-48 h exposure of neuron-HFC co-culture to GBP (outlined in orange). The cumulative trace above the raster plots depicts the population spike time histogram indicating the synchronized activity between the different electrodes (network burst). Quantification of weighted mean firing rate (Hz) (defined as the spike rate per well multiplied by the number of active electrodes in the associated well) and network synchrony (as measured by area under normalized cross-correlation, defined as the area under inter-electrode cross-correlation normalized to the autocorrelations) from HFC and HFC+GBP glioma-neuron coculture. Weighted mean firing rate: CN + HFC vs. CN + HFC + GBP: 1.28 ± 0.09 Hz vs. 0.81 ± 0.16 Hz, $n = 2$ wells/condition, $n = 2$ /group); Area under normalized cross-correlation: CN + HFC vs. CN + HFC + GBP: 1.35 ± 0.03 vs. 1.09 ± 0.04 , $n = 2$ wells/condition, $n = 2$ /group).

The manuscript text has been revised to read as follows (updated text is in bold italics):

Glioma functional connectivity shortens survival

Given the premise that TSP-1 serves as a molecular driver of neuronal activity-driven proliferation, we sought to target TSP-1 therapeutically using gabapentin (GBP). In neuron-glioma co-cultures, individual spikes (extracellular action potentials), bursts (cluster of spikes in blue) and synchronized network bursts (pink) were reduced after 24-48 h exposure to GBP (Fig. 4d). Primary patient-derived HFC cells were transduced with shRNA-control or shRNA-TSP-1 (Fig. 4e) or treated with gabapentin (Fig. 4f); genetic or pharmacological targeting of TSP-1 resulted in a marked decrease in HFC glioma cell proliferation when co-cultured with neurons. Finally, we treated mice bearing HFC patient-derived xenografts (PDX) and found a marked decrease in glioma proliferation index (Ki67+HNA+/HNA+) in gabapentin-treated mice bearing HFC xenografts relative to controls (Fig. 4g, h).

Reviewer 2 Comment 24: Minor points:

1. What is "organoid intensity?" (Fig. 3b)
2. Figure 3, headline: "functional" not "funictonal"
3. Line 58: can this really be concluded at this point? I would suggest to tone down the

language here.

4. Line 65: make clearer to the reader: first MEG – then surgery.

5. Line 70: examples of this methodology? How exactly performed? Maps?

6. Line 85: higher levels: quantification is hidden in Fig. Legend ED Fig. 5 – reference better for clarity. Please provide statistics, too.

7. Line 145: “assuming” – since there is so little known about this area, I would make it clearer that there is a big black spot regarding this point.

8. Line 226: negatively influences: appears a too strong statement, at least with the current data provided. Currently, it is more “might/could influence”.

9. Line 150: “activity-dependent potassium-evoked currents in more astrocyte-like glioma cells”. I do not think this is fully established.

Reviewer 2 Response 24: *All minor comments above are addressed. Thank you so very much for your careful review and for bringing these errors to our attention.*

Referee #3 (Remarks to the Author):

Reviewer 3 Comment 1: I commend the authors on a very large body of work that has culminated into this manuscript. However, the work permeates a variety of fields in neuroscience and molecular biology and is likely going to be too complicated for all except a small niche of experts with knowledge of all of the many domains in which data are collected and analyzed.

Reviewer 3 Response 1: *Thank you for making this comment and for your thorough review of our work. While gliomas are considered a rare cancer, we believe that this study addresses principles with broad appeal while using human data to expand on prior concepts. Our collective understanding of nervous system control of glioblastoma initiation, progression, as well as influence of glioblastoma on nervous system function is based largely on preclinical glioma models. Human disease has remained difficult to interrogate largely due to limited access. This paradigm shifting body of work published over the past 7 years includes (but is not limited to) publications by Venkataramani et al Cell 2022⁴³, Anastasaki et al. Nat Commun 2022⁵², Pan et al Nature 2021⁵³, Yu et al Nature 2020⁵⁴, Venkatesh et al Nature 2019¹⁶, Venkataramani et al Nature 2019¹⁵, Venkatesh et al 2017⁵⁵, John Lin et al. Nat Neurosci 2017⁵⁶, Venkatesh Cell et al 2015⁴⁵. Broadly speaking, Venkataramani et al 2019 and 2022 are largely focused on mechanisms of neuronally driven tumor progression through gap channels between glioma cells interactions^{15,43}. Alternatively, Pan and Anastasaki et al convincingly demonstrated the role of neuronal activity on glioma initiation^{52,53}. Venkatesh 2019, 2017, and 2015 focused on the mechanisms of glioma-neuron interactions through both direct electrochemical synapses (2019) and paracrine signaling (2017, 2015)^{16,45,55}. And finally, Yu et al and John Lin et al illustrate mechanisms and pathological significance of glioma-induced neuronal hyperexcitability^{54,56}. Preclinically, it is therefore evident that glioma proliferation induces neuronal activity while neuronal activity drives glioma proliferation. How much of this work translates into human disease and influences cortical processing in the human brain is still unknown. The goal of this manuscript is to bridge this gap in knowledge focused on glioblastoma remodeling of neuron-neuron interactions (i.e., neuronal circuits).*

Quantitative approaches to assess task-specific neuronal activity in the human brain represents an opportunity to advance our understanding of neuroscience in addition to cancer biology. As the reviewer knows, brain cancer is one of several brain lesions. Considering our newfound understanding that gliomas may not function as purely ablative brain lesions means interrogation of cortical processing may present new insight in cortical circuit remodeling by human disease. For example, while stroke and traumatic brain injury lesions are extremely common, the human brain is not accessible for direct interrogation in these settings. Therefore, principles learned in the study of brain cancer may guide our understanding of the human brain similar to the systems in which speech processing has been influenced by cortical signal interrogation under the clinical context of adult epilepsy. Within the clinical context however, we understand that each experimental model has inherent strengths and weaknesses. Therefore, in this body of work we layered on top of the human electrophysiology, RNA transcriptomics, mouse xenografting, in vitro electrophysiology, pharmacological and gene knockdown, in addition to molecular techniques. The goal of this multimodal approach is to offer a balanced, more coherent model of disease built on convergent data between methodologies. We certainly appreciate our Reviewer's concern regarding readability and general interest therefore we have fully edited our revised manuscript with an eye towards ensuring general appeal across disciplines.

Reviewer 3 Comment 2: Given the effort that has gone into this, and some of the interesting findings, I would urge them to parcellate this into more readily digestible bodies of work.

Reviewer 3 Response 2: *Thank for this comment. Readability is critically important. We have addressed the readability issue in the revised manuscript in order to ensure cross discipline appeal.*

Broad commentary asides, the paper suffers from some fundamental flaws that I outline below.

Reviewer 3 Comment 3: ECoG analysis: The point of this work is to say that gliomas remodel functional circuits. Using recordings in the OR during awake craniotomies for the resections of gliomas – they make the argument that there is greater gamma activation in the electrodes overlying tumor.

Reviewer 3 Response 3: *Thank you for offering this concise summary. In general, this statement is true however there are important caveats and control experiments that may have not been fully apparent in our original submission and we have corrected these regretful oversights in the revised manuscript. Nervous-system cancer crosstalk is bidirectional therefore cancer infiltration may induce nervous system remodeling and dysfunction. Entering this project, we hypothesized that glioma integration into neural circuits and glioma-induced neuronal changes would make tumor-infiltrated brain physiologically disorganized and this functional circuit disarray could be the primary contributor to the cognitive impairments experienced by glioma patients. Alternatively, it remains possible that neuron-neuron interactions may be remodeled by glioma infiltration such that cognitive tasks specific neuronal activity would be preserved. Therefore, the preservation of task-specific synchronized neural activity within*

glioblastoma-infiltrated cortex is the take home point of the study which we believe represents a fundamental change in the way that we view brain cancer- neural network integration. Our initial suspicion was that task-relevant neural responses would demonstrate reduced task evoked activity however the alternate was observed. The greater gamma activation from glioma-infiltrated cortex supports preclinical models and we have provided additional evidence further supporting this finding, which is addressed specifically in Reviewer 3, Response 5. Additionally, we have established both in vitro and in vivo patient derived neuron-glioma co-culture models which we believe will add clarity regarding the finding that TSP-1-positive glioblastoma cells which are identified within HFC intratumoral regions promote both neuronal hyperexcitability and neuronal synchrony. While we address control conditions and experimental details of human ECOG experiments in Response 5, we will address new cross-model validation experiments here.

As stated above, work published by Venkatesh and Monje et al is focused on electrochemical synapses between malignant gliomas and neurons. While glioma-neuronal synapses are compelling, these data do not demonstrate that functional synapses between glioblastoma and neurons contribute to the preservation of task-specific neuronal activity (or hyperexcitability). We do however believe that our work supports remodeling of neuron-neuron circuits by a subpopulation of glioblastoma cells through tumor-secreted synaptogenic protein TSP-1. To validate the human ECOG model of glioma-associated microenvironment changes on neuronal excitability, we employed a multielectrode array (MEA) to assess the electrophysiological properties of patient derived TSP-1 high-expressing HFC and TSP-1 deficient LFC cells in co-culture with neurons in an in vitro setting. To characterize the neuronal activity profile, we computed mean firing rate (MFR) and network burst frequency of the neuronal networks. MFR was defined as the mean number of spikes (extracellular action potentials) per second computed over the total 30 min MEA recording duration. Network bursts consist of densely packed spikes, called bursts, occurring simultaneously at multiple electrodes/channels and network burst frequency was defined as the total number of network burst computed over the total recording time. In addition to the neuronal activity analysis, we computed network synchrony by calculating the area under the normalized cross-correlogram (AUNCC), as described previously^{47,48}. AUNCC represents area under inter-electrode cross-correlation normalized to the autocorrelations, with higher values indicating greater synchronicity of the network.

Interestingly, we found that HFC cells in co-culture with neurons demonstrated a neuronal signature of hyperexcitability after only 24 h of co-culture as evidenced by the increase in network burst frequency. Furthermore, this model demonstrated glioblastoma-induced neuronal synchrony after 48 h (as determined by AUNCC) in comparison to the neuron only and neuron-LFC co-culture conditions (see figures below). More importantly, given the premise that HFC glioma cell-derived TSP-1 could serve as a molecular driver of neuronal hyperexcitability, we inhibited TSP-1 therapeutically using gabapentin (GBP)⁴⁹. We found that in neuron-HFC glioma co-cultures, neuronal activity as well as synchronized network bursts (pink) were significantly reduced after 24-48 h exposure to GBP. These new MEA electrophysiology analyses are incorporated in the revised manuscript (Figs. 2g, 4d, Extended Data Fig. 14). We have provided all relevant figures below as well as updated text in the manuscript below for your convenience.

Despite the cross-model validation, the critically important points raised by Reviewer 3 have given us cause to reframe the discussion highlighting a balanced view of both the significance of task-specific hyperexcitability and importance of the discovery that synchronized neuronal activity exists within glioma-infiltrated cortex. The setting in which our study incorporates ECoG analysis is critically important, and we appreciate reviewer 3's caution. The context under which the human brain can directly be interrogated are few and include movement disorders, epilepsy, and brain cancer. It is therefore under this clinical constraint, that passive recordings during awake craniotomies have been obtained for the analysis in manuscript figure 1. Given these concerns, we outlined in the original submission both qualitative and quantitative steps taken to ensure that study participants are appropriate for behavioral/cognition testing and data is of sufficient quality for further study⁵⁷. Detailed validation of these steps has been provided below in Reviewer 3 Responses 4 and 5. Study co-author David Brang validated all human speech initiation ECoG data. However, in-line with Reviewer 3 comments, as an additional validation step for our human models, we have added as a co-author Edward Chang who has now reviewed all behavioral tasks and analysis given his laboratory's focus on human ECoG speech analysis over the past decade.

a. Magnified view of MEA electrodes, showing RFP-labeled glioma cells in co-culture with neurons

b. Representative raster plot- mouse cortical neuron only condition

a. Magnified view of multi-electrode array (MEA), showing RFP-labeled glioma cells in co-culture with neurons. **b.** Representative raster plot showing individual spikes/extracellular action potentials (tick mark), bursts (cluster of spikes in blue) and synchronized network bursts (pink) of mouse cortical neuron only condition (DIV 18 of neuronal culture and 48 h timepoint for neuron-glioma co-culture). The cumulative trace above the raster plots depicts the population spike time histogram indicating the synchronized activity between the different electrodes (network burst).

Electrophysiological properties of glioma cells in co-culture with neurons were analyzed using multi-electrode array (MEA). Representative raster plots showing individual spikes/extracellular action potentials (tick mark), bursts (cluster of spikes in blue) and synchronized network bursts (pink) after 48 h co-culture of neurons with HFC and LFC glioma cells (outlined in red and blue, respectively). The cumulative trace above the raster plots depicts the population spike time histogram indicating the synchronized activity between the different electrodes (network burst). Quantification of network burst frequency (Hz) (defined as the total number of network bursts divided by recording time) and network synchrony (as measured by area under normalized cross-correlation, defined as the area under inter-electrode cross-correlation normalized to the autocorrelations) from HFC and LFC glioma-neuron coculture. Network burst frequency: cortical neuron (CN) only vs. CN + HFC vs. CN + LFC: 0.75 ± 0.21 Hz vs. 1.08 ± 0.10 Hz vs. 0.42 ± 0.20 Hz, $n = 2$ wells/condition, $n = 2$ per HFC/LFC group ($P = 0.05$); Area under normalized cross-correlation: cortical neuron (CN) only vs. CN + HFC vs. CN + LFC: 0.37 ± 0.004 vs. 1.35 ± 0.03 vs. 0.68 ± 0.18 , $n = 2$ wells/condition, $n = 2$ per HFC/LFC group ($P = 0.0129$ for CN vs. CN+HFC; $P = 0.0308$ for CN+HFC vs. CN+LFC); Data presented as mean \pm s.e.m (b-e, g). P values determined by two-tailed Student's t -test. * $P < 0.05$. ** $P < 0.01$. **** $P < 0.0001$. N.S., not significant.

Electrophysiological properties of HFC-cortical neuron (CN) co-cultures exposed to a working concentration of $50 \mu\text{M}$ gabapentin (GBP) were analyzed using multi-electrode array (MEA). Representative raster plots showing individual spikes/extracellular action potentials (tick mark), bursts (cluster of spikes in blue) and synchronized network bursts (pink) of neuron-HFC co-culture (outlined in red) and 24-48 h exposure of neuron-HFC co-culture to GBP (outlined in

orange). The cumulative trace above the raster plots depicts the population spike time histogram indicating the synchronized activity between the different electrodes (network burst). Quantification of weighted mean firing rate (Hz) (defined as the spike rate per well multiplied by the number of active electrodes in the associated well) and network synchrony (as measured by area under normalized cross-correlation, defined as the area under inter-electrode cross-correlation normalized to the autocorrelations) from HFC and HFC+GBP glioma-neuron coculture. Weighted mean firing rate: CN + HFC vs. CN + HFC + GBP: 1.28 ± 0.09 Hz vs. 0.81 ± 0.16 Hz, $n = 2$ wells/condition, $n = 2$ /group); Area under normalized cross-correlation: CN + HFC vs. CN + HFC + GBP: 1.35 ± 0.03 vs. 1.09 ± 0.04 , $n = 2$ wells/condition, $n = 2$ /group).

Reviewer 3 Comment 4: Comparisons are made in amplitude of activation in the same region across individuals and between functional regions in the same individual. Comparison of the amplitude of activations across individuals in the same brain regions (some with a tumor in that region and some without) is flawed, as this assumes that all individuals must activate equally if recordings are performed in homologous regions.

Reviewer 3 Response 4: *Thank you for this insightful comment. Reviewer 3 comments on the across- and within subject experimental design in figure 1. We regret that we did not provide adequate experimental details regarding how human electrophysiology experiments were conducted and we appreciate the opportunity to clarify. Additionally, all methods and analysis in the paper have gone through full biostatistics review with addition of co-author Annette Molinaro and Edward Chang. We agree that gross anatomy (gyral and sulcal patterns) imperfectly delineates functional boundaries and does not guarantee that the same neural populations are sampled across individuals. However, anatomical alignment remains a common and useful approach in systems neuroscience research as functional boundaries are nonetheless well-predicted by anatomical landmarks. We also agree that this likely adds some variability in the data measured across individuals. However, this concern is tempered in our view by several points. First, all data were consistently preprocessed and normalized, putting all electrodes from subjects in the same range. Second, multiple electrodes were included from each subject within anatomical regions of interest, improving the likely overlap between functional and anatomical boundaries. Third, as the main tests involved comparisons of normal-appearing and tumor-infiltrated electrodes, subject-specific variance was accounted for (see mixed model information below).*

Linear mixed effects modeling was used to perform statistical comparisons with repeated measures in Matlab via the fitlme package. The signal's origin (i.e., normal-appearing/glioma-infiltrated cortex) was modeled as a fixed effect and the participants were modeled as random effects. This model is designed to account for between and within subject variance. The normal appearing task-related neural responses were then explored further by pair-matching in order to ensure analysis of an equal number of tumor-infiltrated and normal appearing electrodes for select anatomical regions within the context of the broader perisylvian language network. Linear mixed effects modeling therefore accounts for the across-subject differences in raw magnitude. Human electrophysiology analyses have increasingly applied LME to account for variance between individuals and within electrode differences⁵⁸. To address this comment, we have updated the study methods which are also included below for convenience.

Human Electrocochography (ECoG) and Data Analyses

The hemisphere of language dominance was determined using baseline magnetic source imaging. Briefly, participants sat in a 275-channel whole-head CTF Omega 2000 system (CTF Systems, Inc., Coquitlam, BC, Canada) sampling at 1,200 Hz while they performed an auditory-verb generation task. The resulting time series were then reconstructed in source space with an adaptive spatial filter after registration with high-resolution MRI. Finally, changes in beta-band activity during verb generation were compared across hemispheres to generate an overall laterality index. All participants were left-dominant and underwent electrophysiologic recording of the left hemisphere. We implemented an intraoperative testing paradigm previously established. Noise in the operating room was minimized through rigorous enforcement of the following: 1) all personnel were requested to cease verbal communication, 2) telephones and alarms were muted, and 3) surgical suction and all other non-essential machinery were temporarily shut down. A 15-inch laptop computer (60 Hz refresh rate) running a custom MATLAB script integrated with PsychToolbox 3 (<http://psychtoolbox.org/>) was placed 30 cm away from each participant. The script initiated a picture naming task which consisted of a single block of 48 unique stimuli, each depicting a common object or animal via colored line drawings. Each stimulus was presented at the point of central fixation and occupied 75% of the display. Upon presentation of each stimulus, participants were required to vocalize a single word that best described the item.

Intra-operative photographs with and without subdural electrodes present were used to localize each electrode contact combined with stereotactic techniques. Images were registered using landmarks from gyral anatomy and vascular arrangement to preoperative T1- and T2-weighted MRI scans. Tumor boundaries were localized on MRI scans and electrodes within 10 mm of necrotic tumor core tissue were identified as ‘tumor’ contacts. Electrodes overlying the hypointense core of the tumor extending from the contrast enhancing rim to the edge of FLAIR were considered “tumor electrodes”, and electrodes completely outside of any T1 post gadolinium or FLAIR signal were considered “non-tumor” or “normal appearing” by a trained co-author blinded to the electrophysiologic data. Glioma-infiltrated regions were defined based on two criteria previously established in the literature including mass-like region of T2-weighted FLAIR sequences signal. Imaging was confirmed with gross inspection of the cortex confirming dilation and/or an abnormal vascular pattern. Prior work has shown that regions of “non-enhancing” disease consist of infiltrating tumor cells intermixed with neurons and normal glial cells. These labels were reviewed by study principal investigator and compared to labels derived during intraoperative stereotactic neuro-navigation to reach a consensus (Brainlab; Munich, Germany).

Each participant received a training session two days prior to participation to ensure familiarity with the task. Electrocorticography (ECoG) signals were acquired during a period after stopping the administration of anesthetics (minimum drug wash out period of 20 minutes) and the patient was judged to be alert and awake after an extensive post-emergence wakefulness assessment to ensure adequate arousal⁵⁹. Intraoperative tasks consisted of naming pictorial representations of common objects and animals (Picture Naming, PN) and naming common objects and animals via auditory descriptions (Auditory Naming, AN)⁶⁰. Postoperative videos were re-analyzed to ensure all data was collected and correct responses only included for analysis. Audio was sampled at 44.1 kHz from a dual-channel microphone placed 5 cm from the participant and electrophysiologic signals were amplified (g.tec; Schiedlberg, Austria).

Recordings were acquired at 4800 Hz and down-sampled to 1200 Hz during the initial stages of processing. During offline analyses, audio and electrophysiologic recordings were manually aligned, resampled, and segmented into epochs (speech-locked). These epochs set time = 0 ms as speech onset and included $\pm 2,000$ ms for a total of 4,000 ms of signal per trial. Trials were discarded if a) an incorrect response was given (including fillers and interjections) or b) there was a greater than 2 second delay between stimulus presentation and response so as to maintain consistent trial dynamics and ensure that the neural signal indeed reflected the experimental manipulations. Channels with excessive noise artifacts were visually identified and removed if their kurtosis exceeded 5.0. Following the rejection of artifactual channels, data were referenced to a common average, high-pass filtered at 0.1 Hz to remove slow drift artifacts, and bandpass filtered between 70–110 Hz using a 300-Order FIR filter to focus the analyses on the high-gamma band range, which is strongly related to local mean population spiking rates. To extract the ERSPs, electrophysiologic signals were first down-sampled to 600 Hz, then high-pass filtered at 0.1 Hz to remove DC-offset and low frequency drift, notch filtered at 60 Hz and its harmonics to remove line noise, and bandpass filtered between 70 and 170 Hz (i.e., the high-gamma range) using a Hamming windowed sinc FIR filter. These signals were finally smoothed using a 100ms Gaussian kernel, down-sampled to 100 Hz, and z-scored across each trial. Electrodes were subsequently re-referenced to the common average for each participant to facilitate group comparisons and regions of interest were defined according to the Automated Anatomical Labeling atlas (<https://www.gin.cnrs.fr/en/tools/aal/>). The location of grid implantation was solely directed by clinical indications. The accuracy of the final registration for each participant was independently confirmed using gyral and sulcal anatomy to triangulate the location of each electrode registered to the template surface and was then compared to intraoperative photographs of the actual cortex with the overlying grid(s). High-gamma band power (HGp) was then calculated using the square of the Hilbert transform on the filtered data. HGp was then averaged across the resting-state time-series, yielding a single measure of neural responsivity for each electrode contact. HGp was then averaged across patients during the task response period, yielding a single measure of neuronal responsivity for each channel. HGp levels were then compared between tumor and normal appearing channels. Linear mixed effects modeling was used to perform statistical comparisons with repeated measures via the nlme package in R (<https://cran.r-project.org/web/packages/nlme/citation.html>). The signal's origin (i.e., normal-appearing/glioma-infiltrated cortex) was modeled as a fixed effect and the participants were modeled as random effects. For continuous variables without repeated measures, t-tests were used. A threshold of $P < 0.05$ was used to denote statistical significance and corrections for multiple comparisons were made using the Bonferroni method.

Reviewer 3 Comment 5: When I look at the maps of electrodes over tumor and non-tumor cortex, the area that is distinctly different between these is prefrontal cortex. Estimating distinctions in activation between these two groups of electrodes is meaningless. They are comparisons across regions – and not surprisingly there is greater activation prior to the onset of articulation in prefrontal cortex relative to primary motor cortex. The “pair matching” in Extended data 2 is once again biased by spatial distinctions – these comparisons of the amplitude of activation between cortical sites in the same individual with very large numbers of electrodes and trials are only weakly

significant, with a relatively modest p value, and no measure of the magnitude of the effect is provided. Comparisons within individuals are also confounded by amplitudes of activation intrinsic to these regions (e.g. ventral prefrontal cortex may activate more than dorsal). The highly variable spectro-temporal responses are unaccounted for in the analysis. As such this is not an appropriate use of ECoG data.

Reviewer 3 Response 5: *Thank you for this comment. Again, we regret for not offering a complete description of our ECoG methods and value this opportunity to provide clarification. All experiments have undergone full statistics review in addition to positive and negative control conditions. It should be stated upfront that beyond the observed task-related hyperexcitability, the notion that synchronized neuronal activity is maintained within glioma-infiltrated cortex was an unexpected finding given that prior work led us to expect disorganized patterns of neuronal activity within these regions of cortex⁶¹. However, given this surprising result, we felt compelled to understand any differences in mean HGp between our two cortical conditions (“non-tumor” and “tumor-infiltrated” electrodes). Linear mixed effects (LME) modeling was applied which account for between and within subject variance. We fully acknowledge that there is slight variability in spatial distribution given the geographical area representing language within the human cortex. For this reason, we next applied pair-matching of electrodes by discrete cortical location in order to account for these differences. Below, we will address each of reviewer 3 concerns and have provided effect sizes and full model parameters. This approach is in line with prior publication⁶²⁻⁶⁴ and we have rearranged the resubmitted manuscript to reflect these analyses.*

First, LME was applied given the fact that within the greater perisylvian language network of regional activation for auditory naming and picture naming tasks, there is across-subject variability and a combination of tumor-infiltrated and normal appearing electrodes coverage. Again, these data are obtained under clinical context therefore this variability is common within epilepsy and brain tumor ECoG datasets. Of the 14 study participants, 9 had electrode coverage demonstrating both tumor-infiltrated and normal-appearing regions. We therefore have now performed a restricted analysis of only participants with both normal appearing and tumor-infiltrated conditions ($n = 9$), averaging all electrodes within the paired analysis (thereby yielding 2 values per participant), and ran a paired t test, mean HGp activation demonstrating greater mean HGp within tumor-infiltrated cortex, $t(8) = 4.841$, Cohen's $D = 1.6136$, $P = .0013$. As further confirmation of the differences in mean HGp between tumor-infiltrated and normal appearing electrodes, we next retained the entire study population and applied linear mixed effects modeling. However, rather than presenting each electrode as an individual value (as presented in Extended Data figure 4e), we show below individual electrodes that is pair-matched so that both non-tumor and tumor-infiltrated electrodes came from the same anatomical regions of cortex (panel left). We then averaged these electrodes together within each patient (while averaging the effect across the sampled region of cortex for an individual) to enable a simple group-level comparison (panel right). This analysis demonstrates greater high gamma (HG) power within electrodes overlying tumor-infiltrated cortex ($F(1,21) = 9.2218$, $P = 0.0062715$).

Using this restrictive analysis, the possibility still existed that the timing of neuronal activation could differ between glioma-infiltrated and non-tumor cortex therefore we plotted pair-matched LME HGp time-series estimates across electrodes (see figure below). Our pair-matched group level analysis presented as a time-series demonstrates a similar pattern of neuronal activation between the two conditions. We were reassured by the observation that the peak cortical activity in normal-appearing, and glioma-infiltrated cortex occurred at the same time. Prior work has shown that the relative time course of neural responses during speech may carry greater physiological significance than classic anatomic localizations⁶⁵. Again, we greatly appreciate reviewer 3 comments and believe that these added analyses provide important validity to our results. Because of these insightful comments, we feel that it is best to present the most conservative estimates of the observed effects between our two cortical conditions. Therefore, figure 1c in the revised manuscript now includes averaged electrodes within each patient, while averaging the effect across the sampled region of cortex for an individual and figure 1d presents a time series of HGp between non-tumor and glioma-infiltrated regions.

Time-series of human subjects picks up high gamma band power (HGp) within tumor-infiltrated brain between -1000 milliseconds (ms) and speech onset (0 ms). Electrodes were compared between non-tumor and tumor infiltrated regions, FDR corrected HGp demonstrates normal appearing neuronal activation across both conditions including task-relevant hyperexcitability.

Unfortunately, it is not possible to perfectly match electrodes given that these data are acquired under clinical constraint using human study participants. However, addressing the concerns suggested by reviewer 3 further supports the observed task-related differences in mean HGp between our two cortical conditions. Furthermore, these data support the resting state hyperexcitability within glioma-infiltrated cortex presented in Venkatesh Nature 2019 and in vitro glioma-neuronal co-culture experiments outlined later in the manuscript (outlined in Reviewer 3 response 3). However, these results should not take away from the salient take-home message and main premise of our work which challenges the classic dogma that brain cancer-infiltrated cortex represents purely a destructive disordered process. A reasonable alternative hypothesis would be that tumor-infiltrated cortex would decrease neuronal activity i.e., mean HGp. On the contrary, we discovered that glioma-infiltrated cortex maintained normal appearing patterns of task-relevant neuronal activity. In the resubmitted manuscript, we have carefully managed our word choices as to provide a balanced discussion including alternate hypothesis.

Reviewer 3 Comment 6: It is never made clear which electrodes lie over the tumor and which do not. This is hard to derive from the group figures. In one example, there appears to be a deep-seated tumor with intact cortex over it and in another the tumor is directly below the recording electrode

Reviewer 3 Response 6: *Thank you for raising this comment. We regret that this important point was missed in the figure legend in the original submission. We have modified the figure (with larger black and white circles representing tumor and non-tumor electrodes, respectively for better visualization) and the figure legend (Extended Data Fig. 2a in the revised manuscript; also provided below for your convenience) to state that the electrodes over non-tumor regions are in white and those over tumor-infiltrative regions in black. In addition, we have now included in extended figures all cortical photos demonstrating expansile tumor infiltrated cortex as Extended Data Fig. 3 in the revised manuscript. As reviewer 3 points out, the distinction of “tumor-infiltrated” and “normal-appearing” brain is a complicated question which can be addressed either by (1) histology, and/or (2) imaging with visual confirmation. It is well established that glioma infiltration extends beyond the necrotic tumor core therefore the region of expansile FLAIR signal on MRI was designated as “tumor-infiltrated” in line with prior*

publications from our group as well as others ^{16,19,20}. We hope that these added datapoints provides greater clarity and data transparency.

Individual cognitive task-related neuronal activity from a cohort of 14 adult patients with cortically projecting glioma infiltration in the lateral prefrontal cortex. Electrodes over non-tumor regions are shown in white and those over tumor-infiltrated regions in black.

Cortex of surface projecting perisylvian gliomas

Images redacted

Cortex of surface projecting perisylvian gliomas.

Reviewer 3 Comment 7: No individual spectral data are presented to illustrate the quality of these intra-operative recordings that are often contaminated by movement, RF interference and epileptiform activity, which if not recognized and used to clean the data could easily confound the derivation of the mean gamma power responses.

Reviewer 3 Response 7: *Thank you bringing this oversight to our attention. Inclusion of spectral data is an important control which was omitted from our original submission. The procedures for recording human electrophysiology data obtained passively under clinical context in the intraoperative setting has been established over the past 10 years at our institution. This includes dedicated operating room shielding which prevents RF interference. Furthermore, within the operating room, standard of care includes clinical neurology who review the continuous ECoG monitoring for epileptiform activity. New co-author Edward Chang has published extensively using epilepsy and brain tumor datasets including intraoperative recordings and his expertise was added to both review all aspects of our protocol as well as statistical analysis. In addition to these steps, following surgery, the intraoperative videos were re-analyzed to ensure all data was collected and correct responses only included for analysis. Channels with excessive noise artifacts were visually identified and removed (enhanced and revised study methods have now been included in the manuscript and shared under Reviewer 3 Response 4 comments). Below we have added the spectrogram of time vs frequency from 10-110 Hz both together for all electrodes as well as separated by tumor-infiltrated and normal appearing cortex (see below). Together, we provide evidence that the source data was sufficiently cleaned and usable for analysis. We have added these analyses to the extended data in the revised manuscript (Fig. 1a, Extended Data Fig. 4a).*

Our spectra data show the expected normal pattern of high gamma power (HGp) increasing above 50 Hz and beta band suppression in addition to clear separation of frequencies including all frequencies and across tumor and non-tumor electrodes.

Reviewer 3 Comment 8: In extended figure 2a the amplitude of activation of the two relatively homologous electrodes, both over tumor and both in “premotor” cortex varies enormously – 5 fold in the second patient (SF0059) relative to the first (UM003), illustrating the pitfalls of comparisons across regions in small groups.

Reviewer 3 Response 8: Thank you for raising this excellent point. Linear mixed effects modeling was used which accounts for the across-subject differences in amplitude of activation⁶³. Additionally, there is no reason to expect that all electrodes sampled from glioma-infiltrated brain would be sampled in such a way that they show increased activity. Again, it should be restated that the true significance of these results is that tumor electrodes obtained from glioma-infiltrated cortex show normal synchronized patterns of task-related activity. After statistical review we feel linear mixed effects models is the best approach to account for these differences in variability combined with the across-subject and pair-matched analysis provided in Reviewer 3, Response 5). The variable amplitude of responses described by reviewer 3 are known for any and all cortical measures of neural responses and therefore not unique to this experimental condition.

Reviewer 3 Comment 9: Overall, I feel that these ECoG data and analysis, flawed as they are, are a distraction from the other points made by the paper and I wonder if whether it was in any way critical to make some of the other points in the paper.

Reviewer 3 Response 9: *The mechanisms by which functional cortex coordinates information for transfer within the neuronal environment to maintain multiple goals has only been addressed neurophysiologically in the human brain. While epilepsy itself represents an alteration of network dynamics separate from normal, much of speech physiology originates from this clinical patient population. This collective body of work includes many references which have changed the way we think about cognition and language. All data obtained under clinical constraint have defined speech and language processing using cortical neuronal recordings which on the surface represents a homogenous population of epilepsy patients; however, upon deeper inspection patients included in these highly references landmark studies represent a mix of pathology subtypes. We believe that human electrocorticography obtained under clinical context has inherent limitations however appreciating the uniqueness and richness of being able to interrogate task-related activity within human subjects', themes emerge which make this model of investigation particularly intriguing. Electrophysiology recording of human glioma in the intraoperative setting is not new. In fact, the following clinical disease types have been included in landmark studies defining "normal" human physiology: (1) mesial temporal epilepsy with an identified lesion, (2) mesial temporal epilepsy without a lesion identified yet not described within the publication, (3) adult epilepsy in the context of diffuse low-grade gliomas, (4) epilepsy in the context of malignant gliomas, (5) arachnoid cyst. The impressive body of work includes many but not limited to the following references⁷⁻¹²: Chang et al PNAS 2013 (supplement includes S1 which includes glioma diagnosis), Fonken et al J Neurophysiol 2016 (table 1 different locations and differing pathologies), Haller et al Nat Hum Behav 2018 (pathology not given), Chang et al Nat Neurosci 2010 (table 1). A larger number of human studies simply do not publish the underlying disease or indication for surgery. The human in-vivo electrophysiology data analyzed in these experiments are identical to experiments performed in epilepsy patients using both implanted and intraoperative recordings. Reviewer 3 has significantly improved our analysis for which we are thankful. Applying separate, comparative analysis of "normal-appearing" and tumor infiltrated cortex in the setting of malignant glioma represents a new and dynamic model through which we will gain greater insight into the biology of brain cancer initiation and cancer induced network alterations.*

Reviewer 3 Comment 10: Magnetoencephalography (MEG) was used to categorize cells in the outer tertiles as coming from HFC vs. LFC sites. All connectivity was estimated in the alpha band. MEG suffers from relatively poor spatial localization capacity. The impact of cortical edema, brain shift and the inability to compute inverse models in the absence of accurate individualized cortical models, which is almost always the case in gliomas due to failure of automated parcellation schema, all limit the ability of MEG. Thus, the premise via which these cells are categorized is questionable. Given that ECoG was performed in all these cases, measures of functional connectivity derived from such direct recordings should be feasible and utilizing them to categorize

tumor cells based on connectivity, would have been much more accurate and meaningful. It would be relatively straightforward to make such estimates using ECoG data.

Reviewer 3 Response 10: *Relative to ECoG, MEG does indeed have less spatial resolution. Nevertheless, since the magnetic permeability properties of different tissue such as edema and glioma tissues are no different from grey matter and air, the magnetic field patterns generated by the neural-glia signals are largely undistorted by the tissue type. The large brain tumor functional mapping literature with MEG has shown that brain shift, edema and the use of more simplified forward models with sophisticated inverse modeling algorithms allow us to reconstruct both brain activity and functional connectivity with accuracy and high fidelity approaching that of ECoG⁶⁶⁻⁶⁹. Co-author Sri Nagarajan is a pioneer in this field and led this portion of the study. Therefore, we respectfully disagree with the reviewer about the premise by which voxels are characterized based on MEG data.*

Our goal here was to apply experimental models that best suited the question at hand. Broadly speaking, brain tumor-induced alterations to functional circuits may take multiple forms: (1) local perilesional circuit changes, (2) ipsilateral hemispheric network adaptations, and (3) contralateral hemispheric compensation. Our initial goal was to study task-related neuronal activation within the local perilesional area. Therefore, the exquisite spatial and temporal resolution of ECoG was optimal. Our next goal was to identify mechanistic drivers of glioma-network adaptations. We hypothesized that if a glial precursor or glioma subpopulation supporting glioma-network integration existed, that it might be within intratumoral regions with elevated functional connectivity with the broader network (having already identified that intratumoral regions maintain synchronized neuronal activity in a task specific manner). One would expect that these intratumoral regions would not be plentiful. Therefore, restricting our analysis to cortical glioma, with sampling limited to regions of grid coverage would severely constrain our analysis. Furthermore, our estimate of the required spatial resolution required for such an experiment would necessitate using only high-density electrodes with 1-2 mm of tissue available for biological assays. For this reason, we sacrificed some of the spatial temporal resolution, in order to gain a more complete ipsilesional hemispheric view of functional connectivity. MEG presents advantages over other existing measures because of its robustness despite tumor vascularity⁵⁵. Site-specific sampling of intratumoral regions guided by imaging is a technique common to our research group with experimental workflow to avoid brain shift and decades of NIH funded research (including NCI SPORE, NCI P01, NINDS and NCI R01 studies) in addition to numerous peer reviewed publications using this approach⁷⁰⁻⁷². Furthermore, patient-derived tissue sampling from highly connected intratumoral regions are subjects to extensive control steps outlined in methods including the use of tumor-specific molecular characterization (including copy number variants). Given the goal of advancing our understanding of human disease, our goal was to remain as close as possible to patient-derived cells and tissues and apply multiple overlapping experimental models.

Reviewer 3 Comment 11: Direction of causality: Even assuming that the MEG data are spatially accurate, a possible alternate explanation for the molecular findings may lie in the fact that functionally eloquent regions are more strongly connected – thus they have

a greater number and broader distribution of fiber pathways via which glial cells can disseminate across the brain, encouraging distinct GBM sub-populations that are more capable of migration to be seen at these sites. Thus, it could well be the brain that influences what type of tumor exists in eloquent vs non eloquent sites, and not vice versa.

Reviewer 3 Response 11: *Reviewer 3 raises an absolutely fascinating hypothesis in this comment. Thank you. It should be stated that a causal direction of cancer initiation within functional regions of brain was not given in the paper and is beyond the experiments as currently outlined. In fact, the first papers focused on neuronally driven cancer initiation in a causal direction have recently been published by Pan et al. Nature 2021⁵³ and Chen et al. Nature 2022⁷³ (both of which are referenced in the revised submission). We have proposed this interesting and provocative alternate hypothesis in our resubmitted discussion.*

To address this comment, we have updated the discussion which is also included below for convenience (new text in bold italics).

The neuronal microenvironment has emerged as a crucial regulator of glioma growth. Both paracrine signaling as well as connectivity remodeling may contribute to network level changes in patients impacting both cognition and survival. In patients, the role of network dynamics on survival and cognition remains poorly understood. In fact, using a heterogenous population of patients with both IDH-wild type and mutant WHO grade III and IV gliomas, some have suggested that functional connectivity improves overall survival⁷⁴⁻⁷⁶ and how glioma-network interactions influences cognition remains unanswered; such prior work has been confounded by functional connectivity methods heavily influenced by the presence of tumor vascularity, limited spatial resolution, in addition to a heterogenous patient cohort. ***Nonetheless, the evidence in this study that glioblastomas remodel functional circuits, does not address direction of causality. The concept of brain tumor eloquence has never been a significant risk variable after controlling for prognostic variables such as patient age, extent of resection, and WHO grade. However, it remains possible that functionally eloquent cortical regions are more strongly connected and thus may have greater network distribution thereby encouraging distinct glioblastoma sub-populations which are capable of migration.*** A better understanding of the cross-talk between neurons and gliomas as well as how functional integration impacts clinical outcomes may open the door to a range of pharmacological and neuromodulation therapeutic strategies focused on improving cognitive outcomes as well as survival.

Reviewer 3 Comment 12: Thus, the question becomes: what is the normal variation in the glial expression of TSP-1 in eloquent vs non eloquent regions?

Reviewer 3 Response 12: *TSP-1 was first discovered in platelets and subsequent studies have indicated its function in astrocyte regulation of neuron development. Specifically, in brain development, TSP-1 is a key astrocyte-secreted protein that is involved in the synapse development. In line with known mechanisms of cancer initiation, these preserved mechanisms of human development suggest a causal role of*

TSP-1 in the development of several cancers including glioblastoma²⁴. Recently Daubon et al. assessed the differential expression of TSP-1 across glioma grades and reported a higher level of TSP-1 in patient samples of glioblastoma compared to glioma grades II, III, or normal brains²⁴. To the best of our knowledge (after an extensive literature search), the variability of glial expression of TSP-1 within presumed “eloquent” and “non-eloquent” regions of brain is unknown. Recently the prognostic value of TSP-1 and its expression across glioblastoma molecular subtypes has been explored⁷⁷. Hypomethylation and over expression of TSP-1 in glioblastoma predicted a less favorable prognosis in patients with glioblastoma, particularly patients with the proneuronal transcriptional subgroup in The Cancer Genome Atlas (TCGA)- GBM dataset. Within malignant glioma, the TGFβ canonical pathway transcriptionally regulates THBS1, through SMAD3 binding to the THBS1 gene promoter. THBS1 silencing inhibits tumor cell invasion and growth, alone and in combination with anti-angiogenic therapy. TGFβ1 therefore induces TSP-1 expression through Smad3 which contributes to the invasive behavior during glioblastoma growth²⁴. Reviewer 3 however raises a fascinating point, that it is entirely possible that functional regions of brain through unknown mechanisms may maintain this connectivity at least in part through astrocyte-derived (secreted) TSP-1. Therefore, malignant glioma initiation within these regions may inherently remain enriched for a subpopulation of astrocyte like glioma cells capable of producing TSP-1. While this experiment is challenging to design, we have done the following. Functional and regional variability in TSP-1 expression of the adult human brain remains unknown. However, we were able to obtain a whole brain donation from a patient with glioblastoma within 1 hour of passing. The brain was immediately perfused for tissue sampling. We have sampled (1) visibly abnormal areas of glioblastoma infiltration, (2) the dominant left hemisphere frontal (lateral prefrontal cortex- LPFC) and temporal cortex, and (3) non dominant right frontal and temporal cortex. We demonstrate in the figure below that while glioblastoma infiltrated brain region expresses higher levels of TSP-1 protein, dominant and non-dominant normal appearing frontal-temporal regions show minimal TSP-1 protein expression by immunofluorescence. We thank Reviewer #3 for this recommendation. We have included these data in rebuttal but not the resubmitted manuscript. However, we would be happy to include if requested by Reviewer #3 or Editorial staff.

62 year old,
R handed
Male
GBM
Tumor is in Splenium, CC, L frontal
Tx- Bx, TMZ/XRT, Avastin

Regional and functional specificity in TSP-1 expression

Representative confocal images and quantification showing regional and functional specificity of TSP-1 expression in the postmortem brain tissue of a glioblastoma patient. While TSP-1 protein is not detectable in the dominant left and non-dominant right frontal and temporal brain regions, there is a strong expression of TSP-1 in the tumor-infiltrated region of splenium in the corpus callosum. Green, TSP-1; Red, GFAP and Nestin (astrocytes and tumor cells, respectively); Blue, DAPI. $P < 0.0001$. Data presented as mean \pm s.e.m. P value determined by one-way ANOVA test.

Reviewer 3 Comment 13: The spatial disparity in the locations sampling may also impact the

molecular distinctions [the finding via RNA transcriptomics and IHC that in LFC tumoral regions, only non-tumor astrocytes express TSP-1, while in HFC regions, high-grade glioma cells also express TSP-1] that are proposed as a mechanism of potential increased connectivity. It is entirely possible that TSP-1 may be a normal mechanism of enhanced connectivity in HFC regions, and amplified in their neoplastic manifestation.

Reviewer 3 Response 13: *This is an excellent point given the astrocytic origin of TSP-1 in non-cancer conditions including brain development. This point builds off comment 12 in which the question remains as to whether select regions of brain which maintain greater functional connectivity (relative to less connected regions) are predisposed to glioblastoma infiltration because of the favorable microenvironment and therefore if this is possible then it remains plausible that the dominant LPFC which demonstrates preserved task related responses and HFC intratumoral regions sampled could simply be a byproduct of normal brain physiology into which cancer has invaded. While intriguing, in the adult brain, TSP-1 expression remains low under normal physiological conditions and the postmortem cadaveric experiment in response 12 supports this claim which is also in line with other prior publications^{24,78}. However, as stated above the question of cancer initiation is not addressed in this study therefore this important alternate hypothesis is now stated clearly in the revised manuscript results as well as in discussion.*

The following text has been added to the manuscript Results:

Glioblastomas are known for heterogenous cellular subpopulations which resemble both astrocyte and oligodendrocyte lineage^{79,80}, and previous studies have shown that astrocyte-like malignant cells can secrete synaptogenic factors that promote neuronal excitability^{54,56}. Our findings suggest that regions of tumor-infiltrated brain that exhibit increased functional connectivity include a synaptogenic subpopulation of malignant tumor cells. It is therefore possible that these synaptogenic glioma cells not only promote neuronal hyperexcitability but also potentially contribute to the functional circuit-level remodeling demonstrated above. ***We do not address cancer initiation in this study therefore it remains possible that TSP-1 within HFC intratumoral regions may represent regionally enhanced functional connectivity which becomes amplified in the setting of glioma infiltration.*** However, these data are consistent with the cancer biology principle that cellular subpopulations assume distinct roles within the heterogenous cancer ecosystem which may be defined at least in part by functional connectivity measures.

The following text (in bold italics) has been added to the manuscript Discussion:

The neuronal microenvironment has emerged as a crucial regulator of glioma growth. Both paracrine signaling as well as connectivity remodeling may contribute to network level changes in patients impacting both cognition and survival. In patients, the role of network dynamics on survival and cognition remains poorly understood. In fact, using a heterogenous population of patients with both IDH-wild type and mutant WHO grade III and IV gliomas, some have suggested that functional connectivity improves overall

survival⁵²⁻⁵⁴ and how glioma-network interactions influences cognition remains unanswered; such prior work has been confounded by functional connectivity methods heavily influenced by the presence of tumor vascularity, limited spatial resolution, in addition to a heterogenous patient cohort. ***Nonetheless, the evidence in this study that glioblastomas remodel functional circuits, does not address direction of causality. The concept of brain tumor eloquence has never been a significant risk variable after controlling for prognostic variables such as patient age, extent of resection, and WHO grade. However, it remains possible that functionally eloquent cortical regions are more strongly connected and thus may have greater network distribution thereby encouraging distinct glioblastoma sub-populations which are capable of migration.*** A better understanding of the cross-talk between neurons and gliomas as well as how functional integration impacts clinical outcomes may open the door to a range of pharmacological and neuromodulation therapeutic strategies focused on improving cognitive outcomes as well as survival.

Reviewer 3 Comment 14: The same factors may impact the greater connectivity in HFC xenografts and in organoids.

Reviewer 3 Response 14: *Yes, we completely agree that astrocyte derived TSP-1 (within LFC regions) and glioblastoma derived TSP-1 (within HFC regions) may represent the known affinity of cancer to use normal development to promote tumor growth. We have therefore raised this important alternate hypothesis throughout the results and discussion.*

Reviewer 3 Comment 15: At the very least, this alternate interpretation should permeate the discussion. Optimally, experiments to disambiguate the activity derived impact of neurons in eloquent cortex in rendering HFC glial cells distinct from LFC glial cells should be derived.

Reviewer 3 Response15: *This is an excellent point which we have added to the manuscript discussion. Furthermore, throughout the paper we have expanded on experiments to disambiguate the activity-derived impact of neurons in eloquent cortex in rendering HFC glial cells distinct from LFC glial cells. First, we performed a full molecular characterization of HFC glioblastoma cells relative to LFC glioblastoma cells via bulk and single-cell sequencing followed by protein level validation of TSP-1 as a molecular driver in primary patient tissues and P0-P1 primary patient cultures which were co-cultured with mouse cortical neuron and neuron organoids. Additionally, the role of neuronal activity on glioma proliferation and the role of gliomas on neuronal activity and synchrony have also been studied in the revised manuscript including pharmacological inhibition of TSP-1. We have provided these enhanced analyses in Reviewer 3 response 3 and have updated the manuscript with extensive edits to support these new and exciting data.*

Reviewer 3 Comment 16: In humans, the impressive survival differences in the KM plots fit well with the established literature for much poorer prognosis of patients with tumors in the eloquent cortex (that is essentially a surrogate for the HFC terminology),

who also suffer from a lower functional performance score. As such this is more confirmatory than a discovery

1) It is not made clear whether the two groups received the same and roughly equivalent treatments. It would be helpful to know the PFS as well as the reason for death. Was there a difference in spread locally or more distant between HFC and non HFC groups?

2) Is it possible that these different subpopulations may be more resistant to chemotherapy or even radiotherapy – this may be may be worth adding this to the discussion as potential translational strategy.

3) For a small group of patients such as this, MGMT status is important to know and to account for in the analysis, as it may affect disproportionately affect survival in small sample sizes.

Reviewer 3 Response 16: *Thank you for asking for clarification regarding the human survival experiments. Both cohorts in this study received the same cancer-directed therapies and all patients had glioblastoma WHO grade 4 according to WHO 2021 diagnostic criteria. Furthermore, below we have added a full summary of clinical and molecular features for each patient including MGMT status. All patients died from disease progression. The concept of chemoradiation resistance promoted by the presence of intratumoral connectivity is yet another intriguing concept which we have not addressed in the paper. However, these is new and compelling evidence in support of this concept recently published by Varn et al. Cell 2022⁴⁴. While this study does not analyze human survival data pre- and post-chemoradiation (our study is currently underpowered to address this topic as a single institution), the multinational GLASS consortium recently demonstrated that neuronal signaling is increased in recurrent glioblastoma. While outside of the scope of our work, this important consideration has been added to the manuscript.*

Additionally, inspired by Reviewer #3's comments, we have now reanalyzed the data adding additional analyses to tightly control for potential differences in treatment, clinical and molecular variables. The interplay between glioblastoma molecular classification (IDH, MGMT status, etc.), patient characteristics (age, functional status), and treatment (such as extent of tumor resection and chemoradiation) has been a topic of intense interest¹⁸. Recent cancer research studies have attempted to move beyond preselected multivariate statistical models. In the original submitted draft we demonstrated using mouse (figure 3h) and human (Extended Data figure 21) Kaplan-Meier survival analysis illustrating 71-week overall survival for patients with HFC voxels as determined by contrast-enhanced T1-weighted images as compared to 123-weeks for participants without HFC voxels. We have now performed an added multivariate statistical analysis called Recursive partitioning analysis (RPA) for Post-Stupp era (2005) IDH wild-type WHO grade 4 glioblastoma patients. Variables analyzed for this experiment included those published in Molinaro and Hervey-Jumper et al Jama Oncology¹⁸. A nested dataset within this cohort of 70 patients had MEG measures of intratumoral neuronal oscillations (35 events 20-month median follow-up) all of whom received identical Stupp protocol chemoradiation. Patients were stratified in a binary manner as having any neuronal oscillations within the tumor or none. Using this approach, three risk groups were determined by RPA (see below). Risk group 1 (black in figure below) have the worst outcomes and are the combination of 1) patients older than 72 and 2) patients

younger than 72 with extent of tumor resection under 97.1%. Risk group 3 (gray in figure below) have the best survival, and these are patients younger than 62 with extent of tumor resection over 97.1% and no intratumoral neural oscillations. Intermediate risk group 2 (red in figure below) is the combination of patients with over 97% extent of tumor resection and either (1) age younger than 72 with neural oscillations identified within the tumor and (2) patients age between 62- 72 and no functional integration. These data demonstrate that neuronal activity within malignant gliomas negatively impacts survival with importance demonstrated by machine learning segmentation of outcomes and quantified to the extent that the presence of neuronal activity may be the equivalent to older patient age regardless of the extent of tumor surgically removed. We have added this enriched analysis to the revised manuscript (Fig. 4a, b) as it controls for the variables mentioned above by Reviewer 3.

Extended Data Table 2. Patient summary-clinical and molecular features

Study #	Sex	Age (yr)	Preoperative tumor volume (ml)	Residual tumor (ml)	EOR (%)	MGMT methylation	EGFR amplification	Tumor Location	Tumor type	IDH status
SF#1	M	56.07	23.67	0.00	100.00	no	nonamp	R frontal	GBM	wt
SF#2	M	64.93	9.20	0.00	100.00	yes	amp	L temporal	GBM	wt
SF#3	M	60.96	49.68	3.95	92.06	yes	nonamp	L temporal	GBM	wt
SF#4	M	60.42	20.30	0.00	100.00	yes	amp	L frontal/insula	GBM	wt
SF#5	M	76.72	8.94	0.00	100.00	no	nonamp	L frontal	GBM	wt
SF#6	M	72	23.59	4.79	79.70	yes	nonamp	L temporal	GBM	wt
SF#7	F	64.14	78.37	0.00	100.00	yes	amp	R frontal	GBM	wt

SF#8	M	78.1 2	77.49	0.00	100. 00	yes	amp	L frontal	GBM	wt
SF#9	F	62.2	14.13	0.00	100. 00	no	amp	L temporal	GBM	wt
SF#10	F	59.0 2	6.38	0.00	100. 00	yes	amp	R frontal	GBM	wt
SF#11	M	57.2 6	46.07	1.09	97.6 4	no	nonamp	L temporal	GBM	wt
SF#12	M	55.3 4	53.10	7.33	86.1 9	yes	nonamp	L frontal	GBM	wt
SF#13	F	66.9 2	3.20	0.00	100. 00	yes	amp	L temporal	GBM	wt
SF#14	M	29.3	36.88	0.00	100. 00	yes	amp	R insula	GBM	wt
SF#15	M	51.1 5	29.67	2.63	91.1 4	no	nonamp	L thalamus	GBM	wt
SF#16	M	48.9 2	31.05	0.00	100. 00	no	nonamp	L frontal	GBM	wt
SF#17	F	49.3	55.55	3.49	93.7 2	yes	amp	L frontal	GBM	wt
SF#18	M	60.9 8	67.50	0.00	100. 00	yes	amp	L frontal	GBM	wt
SF#19	M	80.1	45.41	0.75	98.3 5	yes	nonamp	L parietal	GBM	wt
SF#20	F	72.8 9	6.12	0.00	100. 00	yes	nonamp	R frontal	GBM	wt
SF#21	F	56.3 4	81.00	0.00	100. 00	yes	amp	L parietal	GBM	wt
SF#22	M	63.2 2	7.01	0.53	92.4 1	yes	amp	L parietal	GBM	wt
SF#23	F	60.0 4	15.69	0.00	100. 00	yes	amp	L parietal	GBM	wt
SF#24	F	69.6 2	92.46	0.00	100. 00	no	nonamp	R frontal	GBM	wt
SF#25	M	60.3 8	101.96	0.21	99.7 9	yes	nonamp	R frontal	GBM	wt
SF#26	F	52.6 1	7.32	0.00	100. 00	yes	nonamp	L temporal	GBM	wt
SF#27	M	52.7 2	41.21	3.58	91.3 1	no	amp	R frontal	GBM	wt
SF#28	F	59.8 9	9.51	0.00	100. 00	no	nonamp	L temporal	GBM	wt
SF#29	M	44.9 2	34.59	0.00	100. 00	yes	amp	R temporal	GBM	wt
SF#30	F	65.5 3	15.77	0.48	96.9 6	no	unknown	Parietal	GBM	wt
SF#31	F	67.8 9	14.14	0.00	100. 00	yes	nonamp	L multifocal	GBM	wt
SF#32	M	71.4 9	21.80	0.00	100. 00	no	amp	L temporal	GBM	wt
SF#33	F	61.7 2	52.27	0.17	99.6 7	no	amp	R occipital	GBM	wt
SF#34	M	69.0 3	57.01	0.00	100. 00	yes	nonamp	L parietal	GBM	wt
SF#35	F	46.9	35.99	0.00	100. 00	yes	yes	R temporal	GBM	wt
SF#36	M	51.7 3	37.45	2.20	94.1 3	no	amp	L frontal	GBM	wt
SF#37	M	69.6 7	59.66	0.33	99.4 5	yes	nonamp	R parieto-occipital	GBM	wt
SF#38	M	69.0 6	74.40	2.49	96.6 5	yes	nonamp	R parietal	GBM	wt

SF#39	F	69.8 3	6.85	0.17	97.5 1	yes	amp		GBM	wt
SF#40	M	54.0 5	61.69	3.00	95.1 3	yes	amp	L frontal	GBM	wt
SF#41	F	57.3 9	9.25	0.00	100. 00	yes	nonamp	L frontal	GBM	wt
SF#42	M	60.2 1	101.06	0.41	99.5 9	no	nonamp	L temporal	GBM	wt
SF#43	M	76.7 5	44.02	1.13	97.4 3	yes	nonamp	R temporal	GBM	wt
SF#44	M	43.8 1	17.72	0.00	100. 00	no	amp	R frontal	GBM	wt

The manuscript text has been revised to read as follows (updated text is in bold italics):

Glioma functional connectivity shortens survival

We next explored the effects of high functional connectivity within gliomas on survival and cognition. First, we tested the hypothesis that gliomas exhibiting increased functional connectivity may be more aggressive, given the robust influence of neuronal activity on tumor progression. To investigate patient outcomes, we performed a human survival analysis of patients with molecularly uniform newly diagnosed IDH-WT glioblastoma. After controlling for known correlates of survival (age, tumor volume, completion of chemotherapy and radiation, and extent of tumor resection), neural oscillations and functional connectivity were measured within tumor-infiltrated brain using MEG (Extended Tables 2 and 5). Subjects were classified by the presence or absence of HFC voxels within the tumor boundary. Kaplan-Meier survival analysis illustrates 71-week overall survival for patients with HFC voxels as compared to 123-week overall survival for participants without HFC voxels, illustrating a striking inverse relationship between survival and functional connectivity of the tumor (mean follow-up months 50.5 months) ($P = 0.04$) (Extended Data Fig. 20). ***To identify clinically relevant survival risk groups created by the presence or absence of functional integration for patients with chemoradiation treated newly diagnosed glioblastoma in a multivariate setting, we employed recursive partitioning via the partDSA algorithm. Survival trees use recursive partitioning to divide patients into risk groups based on the interactive effects of all known prognostic variables including age at diagnosis, sex, tumor location, chemotherapy, radiotherapy, the presence of functional connectivity within the tumor, pre- and post-operative tumor volume, and extent of resection. Risk group 1 (black) had the worst outcomes and are the combination of patients older than 72 and patients younger than 72 with extent of tumor resection under 97.1%. Risk group 3 (gray) have the best survival, and these are patients younger than 62 with extent of tumor resection over 97.1% and functional connectivity within the tumor. Intermediate risk group 2 (red) revealed an interesting interaction between patients with over 97% extent of tumor removed and either age younger than 72 with intratumoral connectivity or patients 10 years younger without functional integration (Fig. 4a, b). These results demonstrate the striking prognostic value of HFC on survival.*** We next examined whether TSP-1, a secreted synaptogenic protein, can be identified in patient serum and whether circulating TSP-1 is correlated with functional connectivity as measured by magnetoencephalography imaginary coherence. Circulating TSP-1 levels in patient serum exhibited a striking positive correlation with intratumoral functional connectivity ($P = 0.01$) (Fig. 4c). Identifying a possible clinical correlate for functional connectivity in glioma patients.

Minor points:

Reviewer 3 Comment 17: It does not appear that measurements of tumor volume in the mice to demonstrate differences between the xenografted HFC or LFC cells was performed – this must have been performed and it would be good to look at to explain such a different survival.

Reviewer 3 Response 17: Thank you for this comment. Tumor cell burden of HFC and LFC hippocampal xenografts were assessed using blinded rank order analysis as previously reported⁸¹. Interestingly, we noticed that instead of forming a compact mass, HFC cells xenografted in the CA1 region of the hippocampus showed a diffuse infiltrative pattern while they recapitulated the laminar hippocampal layers. In contrast, xenografted LFC cells showed significantly lower tumor engraftment with minimal infiltration and invasion of tumor cells. We demonstrate below a significantly higher tumor burden in mice bearing HFC cells compared to the LFC counterpart and have included this analysis in the revised manuscript (Extended Data Fig. 20).

Representative confocal images showing the diffuse infiltrative pattern of HFC cells in the hippocampus in comparison to the LFC cells. Quantification of tumor burden of HFC and LFC hippocampal xenografts using rank order analysis (HFC vs. LFC: 16.462 ± 1.70 vs. 7.818 ± 1.52 , $P = 0.002$). Data presented as mean \pm s.e.m. P value determined by two-tailed Mann-Whitney test.

Reviewer 3 Comment 18: The claim that “gliomas remodel functional neural circuitry such that task-relevant neural responses activate tumor-infiltrated cortex, beyond cortical excitation normally recruited in the healthy brain” is overblown. The finding of non-traditional language sites are activated during lexical access is hardly surprising as functional reorganization secondary to gliomas is well known and is entirely expected in such cases.

Reviewer 3 Response 18: *While we realize that this is a striking finding, it is indeed what the data show. We have modified wording in the manuscript, but the core finding expressed is supported by the data. First, the preservation of task-relevant responses within glioma-infiltrated brain (or any lesioned brain in humans) has not been demonstrated to the best of our knowledge. Known glioma-induced remodeling of functional circuits in the human brain has previously been suggested almost entirely with respect to contralateral functional network compensation, including language lateralization and whole brain network dynamics studies using rs-fMRI²⁻⁵. The role of perilesional glioma-infiltrated cortex on functional circuit reorganization remains unknown and is a critically important consideration. In fact, Brandt et al. previously demonstrated diminished patterns of resting state cortical neuronal activity in the setting of malignant glioma⁶¹.*

The notion that within the periglioma microenvironment, glioma-infiltrated cortex engages in network level activity is an entirely new way of viewing brain cancer. In fact, in line with reviewer 3's comments, we have been fascinated by the notion that despite task-relevant neuronal activity, patients with gliomas often present with neurological impairments which raises the question of why. Inspired by the control experiments recommended by reviewer 3, we have now applied neuronal decoding of glioma-infiltrated and normal appearing substrate²¹. Engineering efforts now apply neurobiological findings, together with advances in machine learning to demonstrate that speech can be decoded from neuronal activity in patients with and without speech impairments⁶. Learning algorithms to create computational models for the detection and classification of words from patterns in recorded cortical activity represents the future of rehabilitative medicine. We believe that these results will set the stage for this entire field of speech and cognitive rehabilitation for cancer patients. Recently Leonard and others have studied how speech sequence are neurally encoded, by using high-resolution direct cortical recordings from human lateral superior temporal cortex as subjects listened to words and non-words with varying transition probabilities between sound segments. They found that neural responses encoded language-level probability of upcoming speech sounds suggesting acoustic representations with linguistic information is encoded within neural substrate. We therefore invited co-author Edward Chang and applied a similar information theory across tumor-infiltrated and normal appearing cortex. In this experiment, audiovisual stimuli are separated and evenly presented to the study participant based on level of complexity. Beyond maintained temporal representations of neuronal activity within glioma infiltrated cortex, glioma-infiltrated cortex was unable to predict nuanced aspects of word retrieval such as word frequency (see below). Vocalization of low frequency words, for instance, requires a more intricate coordination of articulatory elements than that of high frequency words. To identify differences in computational properties of normal-appearing and glioblastoma-infiltrated cortex, we determined the decodability of their signals using a regularized logistic regression classifier to distinguish between low and high frequency word trial conditions using event-related responses (Fig. 1e). We implemented identical training and leave-one-subject-out cross-validation paradigms for both conditions. We have incorporated this new analysis in the revised manuscript as Fig. 1e. Despite normal appearing task-specific neuronal activity, glioma-infiltrated cortex maintains the ability to perform basic computational properties, yet loses the ability to perform complex aspects of speech and cognition. Therefore, beyond maintained temporal representations of neuronal activity within glioma infiltrated cortex, tumor-infiltrated

cortex was unable to decode or predict nuanced aspects of speech production such as word frequency (see figure below). Therefore, despite normal appearing task-specific neuronal activity, glioma-infiltrated cortex maintains non-specific and what appears to be spatially less discrete linguistic information. This analysis may provide balance to our work therefore we are delighted to include within the main figures in the revised manuscript as Fig. 1e.

The manuscript text under the heading '**Glioblastomas remodel functional neural circuits**' has been revised to read as follows (updated text is in bold italics):

Task-evoked neural responses from tumor-infiltrated regions may be less than in non-tumor tissues. Thus, we wanted to understand whether the magnitude of task-related neural activity within tumor-infiltrated regions of brain oscillates similar to non-tumor regions. We therefore pair-matched each cortical electrode array (Extended Data Fig. 4d, e) which confirmed increased HGp within this expanded region of cortex infiltrated by tumor ($P = 0.016$) (Fig. 1c, d). **Given the presence of coordinated neural responses, we set out to determine whether there are alterations in the high gamma activity elicited by computationally demanding tasks varied across cortical conditions. Vocalization of low frequency words, for instance, requires a more intricate coordination of articulatory elements than that of high frequency words. Therefore, in order to identify differences in computational properties of normal-appearing and glioblastoma-infiltrated cortex we determined the decodability of their signals using a regularized logistic regression classifier to distinguish between low and high frequency word trial conditions using event-related responses in the anterior**

temporal lobe (Fig. 1e). We implemented identical training and leave-one-subject-out cross-validation paradigms for both conditions. Normal-appearing cortex produced above-chance decoding between low and high frequency word trials (mean classifier accuracy = 0.56, $P < 0.001$). By contrast, glioblastoma-infiltrated cortex was not able to decode word trials above chance (mean classifier accuracy = 0.49, $P = 0.72$). These data further demonstrate that glioblastoma integration into cortical regions results in functional neural circuit remodeling and task-specific hyperexcitability, yet there is loss of computational properties within these cortical regions (Fig. 1f).

Reviewer 3 Comment 19: The number of patients in the HFC and no HFC groups in figure 4b need to be explicit, as it's a bit confusing and difficult to visualize each death on the plot.

Reviewer 3 Response 19: Thank you for raising this point. We have explicitly stated the number of patients in HFC ($n = 25$) and no HFC ($n = 41$) groups in the figure legend and have changed each censored hash mark from gray to black to provide clarification in the revised manuscript (Extended Data Fig. 20). Thank you for raising this important point and for the opportunity to clarify our data.

Kaplan-Meier human survival analysis illustrates 71-week overall survival for patients with HFC voxels ($n = 25$) as determined by contrast-enhanced T1-weighted images as compared to 123-weeks for participants without HFC voxels ($n = 41$) (mean follow-up months 50.5, range 4.9-155.9 months).

Reviewer 3 Comment 20: This work appears to miss the opportunity to build upon prior publications (Venkatesh et al - Electrical and synaptic integration of glioma into neural circuits - Nature 2019), by not seeking to modulate the influence of glioma activity on neuronal excitability via potassium fluxes in vivo, or to directly modulate activity

regulated glioma growth – natural directions given the rich datasets and the skill they have brought to bear in performance of this work.

Reviewer 3 Response 20: *This is an excellent point which was also raised by Reviewer 2. We provided context for this body of work in Reviewer 3 comment however it seems appropriate to restate here as well. Broadly speaking, over the past 7-10 years the following publications have used largely preclinical experimental models to study the influence of the neurons on glioma initiation, progression, and invasion. Venkataramani et al 2019 and 2022 are largely focused on mechanisms of neuronally driven tumor progression through gap channels between glioma cells (i.e., glioma-glioma interactions)^{15,43}. Alternatively, Pan and Anastasaki et al demonstrated the role of neuronal activity on glioma initiation (optic and olfactory neurons)^{52,53}. Venkatesh 2019, 2017, and 2015 focused on the mechanisms of glioma-neuron interactions through both direct electrochemical synapses (2019) and paracrine signaling (2017, 2015)^{16,45,55}. And Yu et al and John Lin et al illustrate mechanisms and pathological significance of glioma to induce neuronal hyperexcitability (including astrocyte type C cells which closely mirror glioblastoma cells and induce neuronal excitability)^{54,56}. Preclinically, it is therefore evident that glioma proliferation induces neuronal activity while neuronal activity drives glioma proliferation. It remains unknown how much of this work translates into human disease and cortical processing of information in the human brain. Our goal was to bridge this gap in knowledge focused on the mechanisms by which glioblastoma remodel the brain using human datasets and low-passaged (P0-1) primary patient cultures. Therefore, we focus on neuron-neuron interactions (i.e., neuronal circuits). While building on the groundbreaking work by Venkatesh and Monje Nature 2019, we do not suspect that glioma-neuron synapses would maintain functional circuits within highly connected intratumoral regions. However, while not the point of this work we have started experiments focused on this new and intriguing concept.*

REFERENCES

- 1 Chang, Y. N. & Lambon Ralph, M. A. A unified neurocomputational bilateral model of spoken language production in healthy participants and recovery in poststroke aphasia. *Proc Natl Acad Sci U S A* **117**, 32779-32790, doi:10.1073/pnas.2010193117 (2020).
- 2 Amoruso, L. *et al.* Oscillatory and structural signatures of language plasticity in brain tumor patients: A longitudinal study. *Hum Brain Mapp* **42**, 1777-1793, doi:10.1002/hbm.25328 (2021).
- 3 Cargnelutti, E., Ius, T., Skrap, M. & Tomasino, B. What do we know about pre- and postoperative plasticity in patients with glioma? A review of neuroimaging and intraoperative mapping studies. *Neuroimage Clin* **28**, 102435, doi:10.1016/j.nicl.2020.102435 (2020).
- 4 Sharma, V. V. *et al.* Beta synchrony for expressive language lateralizes to right hemisphere in development. *Sci Rep* **11**, 3949, doi:10.1038/s41598-021-83373-z (2021).

- 5 Traut, T. *et al.* MEG imaging of recurrent gliomas reveals functional plasticity of hemispheric language specialization. *Hum Brain Mapp* **40**, 1082-1092, doi:10.1002/hbm.24430 (2019).
- 6 Moses, D. A. *et al.* Neuroprosthesis for Decoding Speech in a Paralyzed Person with Anarthria. *N Engl J Med* **385**, 217-227, doi:10.1056/NEJMoa2027540 (2021).
- 7 Chang, E. F., Niziolek, C. A., Knight, R. T., Nagarajan, S. S. & Houde, J. F. Human cortical sensorimotor network underlying feedback control of vocal pitch. *Proc Natl Acad Sci U S A* **110**, 2653-2658, doi:10.1073/pnas.1216827110 (2013).
- 8 Durschmid, S. *et al.* Oscillatory dynamics track motor performance improvement in human cortex. *PLoS One* **9**, e89576, doi:10.1371/journal.pone.0089576 (2014).
- 9 Voytek, B. *et al.* Oscillatory dynamics coordinating human frontal networks in support of goal maintenance. *Nat Neurosci* **18**, 1318-1324, doi:10.1038/nn.4071 (2015).
- 10 Fonken, Y. M. *et al.* Frontal and motor cortex contributions to response inhibition: evidence from electrocorticography. *J Neurophysiol* **115**, 2224-2236, doi:10.1152/jn.00708.2015 (2016).
- 11 Haller, M. *et al.* Persistent neuronal activity in human prefrontal cortex links perception and action. *Nat Hum Behav* **2**, 80-91, doi:10.1038/s41562-017-0267-2 (2018).
- 12 Chang, E. F. *et al.* Categorical speech representation in human superior temporal gyrus. *Nat Neurosci* **13**, 1428-1432, doi:10.1038/nn.2641 (2010).
- 13 Bakken, T. E. *et al.* Single-nucleus and single-cell transcriptomes compared in matched cortical cell types. *PLoS One* **13**, e0209648, doi:10.1371/journal.pone.0209648 (2018).
- 14 Couturier, C. P. *et al.* Single-cell RNA-seq reveals that glioblastoma recapitulates a normal neurodevelopmental hierarchy. *Nat Commun* **11**, 3406, doi:10.1038/s41467-020-17186-5 (2020).
- 15 Venkataramani, V. *et al.* Glutamatergic synaptic input to glioma cells drives brain tumour progression. *Nature* **573**, 532-538, doi:10.1038/s41586-019-1564-x (2019).
- 16 Venkatesh, H. S. *et al.* Electrical and synaptic integration of glioma into neural circuits. *Nature* **573**, 539-545, doi:10.1038/s41586-019-1563-y (2019).
- 17 Joseph, J. V. *et al.* TGF-beta promotes microtubule formation in glioblastoma through thrombospondin 1. *Neuro Oncol* **24**, 541-553, doi:10.1093/neuonc/noab212 (2022).
- 18 Molinaro, A. M. *et al.* Association of Maximal Extent of Resection of Contrast-Enhanced and Non-Contrast-Enhanced Tumor With Survival Within Molecular Subgroups of Patients With Newly Diagnosed Glioblastoma. *JAMA Oncol* **6**, 495-503, doi:10.1001/jamaoncol.2019.6143 (2020).
- 19 Ji, M. *et al.* Detection of human brain tumor infiltration with quantitative stimulated Raman scattering microscopy. *Sci Transl Med* **7**, 309ra163, doi:10.1126/scitranslmed.aab0195 (2015).
- 20 Pekmezci, M. *et al.* Detection of glioma infiltration at the tumor margin using quantitative stimulated Raman scattering histology. *Sci Rep* **11**, 12162, doi:10.1038/s41598-021-91648-8 (2021).

- 21 Leonard, M. K., Bouchard, K. E., Tang, C. & Chang, E. F. Dynamic encoding of speech sequence probability in human temporal cortex. *J Neurosci* **35**, 7203-7214, doi:10.1523/JNEUROSCI.4100-14.2015 (2015).
- 22 Chartier, J., Anumanchipalli, G. K., Johnson, K. & Chang, E. F. Encoding of Articulatory Kinematic Trajectories in Human Speech Sensorimotor Cortex. *Neuron* **98**, 1042-1054 e1044, doi:10.1016/j.neuron.2018.04.031 (2018).
- 23 Criss, A. H., Aue, W. R. & Smith, L. The effects of word frequency and context variability in cued recall. *J Mem Lang* **64**, 119-132, doi:10.1016/j.jml.2010.10.001 (2011).
- 24 Daubon, T. *et al.* Deciphering the complex role of thrombospondin-1 in glioblastoma development. *Nat Commun* **10**, 1146, doi:10.1038/s41467-019-08480-y (2019).
- 25 Jeanne, A., Schneider, C., Martiny, L. & Dedieu, S. Original insights on thrombospondin-1-related antireceptor strategies in cancer. *Front Pharmacol* **6**, 252, doi:10.3389/fphar.2015.00252 (2015).
- 26 Augusto-Oliveira, M. *et al.* Astroglia-specific contributions to the regulation of synapses, cognition and behaviour. *Neurosci Biobehav Rev* **118**, 331-357, doi:10.1016/j.neubiorev.2020.07.039 (2020).
- 27 Galambos, R. A glia-neural theory of brain function. *Proc Natl Acad Sci U S A* **47**, 129-136, doi:10.1073/pnas.47.1.129 (1961).
- 28 Araque, A., Parpura, V., Sanzgiri, R. P. & Haydon, P. G. Tripartite synapses: glia, the unacknowledged partner. *Trends Neurosci* **22**, 208-215, doi:10.1016/s0166-2236(98)01349-6 (1999).
- 29 Sudhof, T. C. Neuroligins and neurexins link synaptic function to cognitive disease. *Nature* **455**, 903-911, doi:10.1038/nature07456 (2008).
- 30 Zoghbi, H. Y. Postnatal neurodevelopmental disorders: meeting at the synapse? *Science* **302**, 826-830, doi:10.1126/science.1089071 (2003).
- 31 Tan, C. X., Burrus Lane, C. J. & Eroglu, C. Role of astrocytes in synapse formation and maturation. *Curr Top Dev Biol* **142**, 371-407, doi:10.1016/bs.ctdb.2020.12.010 (2021).
- 32 Adams, J. C. Thrombospondins: multifunctional regulators of cell interactions. *Annu Rev Cell Dev Biol* **17**, 25-51, doi:10.1146/annurev.cellbio.17.1.25 (2001).
- 33 Christopherson, K. S. *et al.* Thrombospondins are astrocyte-secreted proteins that promote CNS synaptogenesis. *Cell* **120**, 421-433, doi:10.1016/j.cell.2004.12.020 (2005).
- 34 Xu, J., Xiao, N. & Xia, J. Thrombospondin 1 accelerates synaptogenesis in hippocampal neurons through neuroligin 1. *Nat Neurosci* **13**, 22-24, doi:10.1038/nn.2459 (2010).
- 35 Cane, M., Maco, B., Knott, G. & Holtmaat, A. The relationship between PSD-95 clustering and spine stability in vivo. *J Neurosci* **34**, 2075-2086, doi:10.1523/JNEUROSCI.3353-13.2014 (2014).
- 36 Brookes, M. J. *et al.* Measuring functional connectivity using MEG: methodology and comparison with fcMRI. *Neuroimage* **56**, 1082-1104, doi:10.1016/j.neuroimage.2011.02.054 (2011).
- 37 Tarapore, P. E. *et al.* Magnetoencephalographic imaging of resting-state functional connectivity predicts postsurgical neurological outcome in brain gliomas. *Neurosurgery* **71**, 1012-1022, doi:10.1227/NEU.0b013e31826d2b78 (2012).

- 38 Fox, J. L., Dews, M., Minn, A. J. & Thomas-Tikhonenko, A. Targeting of TGFbeta signature and its essential component CTGF by miR-18 correlates with improved survival in glioblastoma. *RNA* **19**, 177-190, doi:10.1261/rna.036467.112 (2013).
- 39 Peng, M. *et al.* The role of Clusterin in cancer metastasis. *Cancer Manag Res* **11**, 2405-2414, doi:10.2147/CMAR.S196273 (2019).
- 40 Chen, F. *et al.* Clusterin secreted from astrocyte promotes excitatory synaptic transmission and ameliorates Alzheimer's disease neuropathology. *Mol Neurodegener* **16**, 5, doi:10.1186/s13024-021-00426-7 (2021).
- 41 Jia, S., Rodriguez, M., Williams, A. G. & Yuan, J. P. Homer binds to Orai1 and TRPC channels in the neointima and regulates vascular smooth muscle cell migration and proliferation. *Sci Rep* **7**, 5075, doi:10.1038/s41598-017-04747-w (2017).
- 42 Luo, P. *et al.* Postsynaptic scaffold protein Homer 1a protects against traumatic brain injury via regulating group I metabotropic glutamate receptors. *Cell Death Dis* **5**, e1174, doi:10.1038/cddis.2014.116 (2014).
- 43 Venkataramani, V. *et al.* Glioblastoma hijacks neuronal mechanisms for brain invasion. *Cell*, doi:10.1016/j.cell.2022.06.054 (2022).
- 44 Varn, F. S. *et al.* Glioma progression is shaped by genetic evolution and microenvironment interactions. *Cell* **185**, 2184-2199 e2116, doi:10.1016/j.cell.2022.04.038 (2022).
- 45 Venkatesh, H. S. *et al.* Neuronal Activity Promotes Glioma Growth through Neuroligin-3 Secretion. *Cell* **161**, 803-816, doi:10.1016/j.cell.2015.04.012 (2015).
- 46 Daniel, A. G. S. *et al.* Functional connectivity within glioblastoma impacts overall survival. *Neuro Oncol* **23**, 412-421, doi:10.1093/neuonc/noaa189 (2021).
- 47 Passaro, A. P., Aydin, O., Saif, M. T. A. & Stice, S. L. Development of an objective index, neural activity score (NAS), reveals neural network ontogeny and treatment effects on microelectrode arrays. *Sci Rep* **11**, 9110, doi:10.1038/s41598-021-88675-w (2021).
- 48 Pastore, V. P., Massobrio, P., Godjoski, A. & Martinoia, S. Identification of excitatory-inhibitory links and network topology in large-scale neuronal assemblies from multi-electrode recordings. *PLoS Comput Biol* **14**, e1006381, doi:10.1371/journal.pcbi.1006381 (2018).
- 49 Eroglu, C. *et al.* Gabapentin receptor alpha2delta-1 is a neuronal thrombospondin receptor responsible for excitatory CNS synaptogenesis. *Cell* **139**, 380-392, doi:10.1016/j.cell.2009.09.025 (2009).
- 50 Narayanan, N. S. & Laubach, M. Methods for studying functional interactions among neuronal populations. *Methods Mol Biol* **489**, 135-165, doi:10.1007/978-1-59745-543-5_7 (2009).
- 51 Ullo, S. *et al.* Functional connectivity estimation over large networks at cellular resolution based on electrophysiological recordings and structural prior. *Front Neuroanat* **8**, 137, doi:10.3389/fnana.2014.00137 (2014).
- 52 Anastasaki, C. *et al.* Neuronal hyperexcitability drives central and peripheral nervous system tumor progression in models of neurofibromatosis-1. *Nat Commun* **13**, 2785, doi:10.1038/s41467-022-30466-6 (2022).
- 53 Pan, Y. *et al.* NF1 mutation drives neuronal activity-dependent initiation of optic glioma. *Nature* **594**, 277-282, doi:10.1038/s41586-021-03580-6 (2021).
- 54 Yu, K. *et al.* PIK3CA variants selectively initiate brain hyperactivity during gliomagenesis. *Nature* **578**, 166-171, doi:10.1038/s41586-020-1952-2 (2020).

- 55 Venkatesh, H. S. *et al.* Targeting neuronal activity-regulated neuroligin-3 dependency in high-grade glioma. *Nature* **549**, 533-537, doi:10.1038/nature24014 (2017).
- 56 John Lin, C. C. *et al.* Identification of diverse astrocyte populations and their malignant analogs. *Nat Neurosci* **20**, 396-405, doi:10.1038/nn.4493 (2017).
- 57 Aabedi, A. A. *et al.* Assessment of wakefulness during awake craniotomy to predict intraoperative language performance. *J Neurosurg* **132**, 1930-1937, doi:10.3171/2019.2.JNS183486 (2019).
- 58 Kadipasaoglu, C. M. *et al.* Network dynamics of human face perception. *PLoS One* **12**, e0188834, doi:10.1371/journal.pone.0188834 (2017).
- 59 Aabedi, A. A. *et al.* Assessment of wakefulness during awake craniotomy to predict intraoperative language performance. *J Neurosurg*, 1-8, doi:10.3171/2019.2.JNS183486 (2019).
- 60 Chang, E. F., Raygor, K. P. & Berger, M. S. Contemporary model of language organization: an overview for neurosurgeons. *J Neurosurg* **122**, 250-261, doi:10.3171/2014.10.JNS132647 (2015).
- 61 Bandt, S. K. *et al.* The impact of high grade glial neoplasms on human cortical electrophysiology. *PLoS One* **12**, e0173448, doi:10.1371/journal.pone.0173448 (2017).
- 62 Karthik, G. *et al.* Visual speech differentially modulates beta, theta, and high gamma bands in auditory cortex. *Eur J Neurosci* **54**, 7301-7317, doi:10.1111/ejn.15482 (2021).
- 63 Aabedi, A. A. *et al.* Functional alterations in cortical processing of speech in glioma-infiltrated cortex. *Proc Natl Acad Sci U S A* **118**, doi:10.1073/pnas.2108959118 (2021).
- 64 Kadipasaoglu, C. M. *et al.* Surface-based mixed effects multilevel analysis of grouped human electrocorticography. *Neuroimage* **101**, 215-224, doi:10.1016/j.neuroimage.2014.07.006 (2014).
- 65 Flinker, A. *et al.* Redefining the role of Broca's area in speech. *Proc Natl Acad Sci U S A* **112**, 2871-2875, doi:10.1073/pnas.1414491112 (2015).
- 66 Dalal, S. S. *et al.* Spatial localization of cortical time-frequency dynamics. *Annu Int Conf IEEE Eng Med Biol Soc* **2007**, 4941-4944, doi:10.1109/IEMBS.2007.4353449 (2007).
- 67 Guggisberg, A. G., Dalal, S. S., Findlay, A. M. & Nagarajan, S. S. High-frequency oscillations in distributed neural networks reveal the dynamics of human decision making. *Front Hum Neurosci* **1**, 14, doi:10.3389/neuro.09.014.2007 (2007).
- 68 Tarapore, P. E. *et al.* Resting state magnetoencephalography functional connectivity in traumatic brain injury. *J Neurosurg* **118**, 1306-1316, doi:10.3171/2013.3.JNS12398 (2013).
- 69 Tarapore, P. E. *et al.* Preoperative multimodal motor mapping: a comparison of magnetoencephalography imaging, navigated transcranial magnetic stimulation, and direct cortical stimulation. *J Neurosurg* **117**, 354-362, doi:10.3171/2012.5.JNS112124 (2012).
- 70 Bulubas, L. *et al.* Motor Cortical Network Plasticity in Patients With Recurrent Brain Tumors. *Front Hum Neurosci* **14**, 118, doi:10.3389/fnhum.2020.00118 (2020).

- 71 Mancini, A. *et al.* Disruption of the beta1L Isoform of GABP Reverses Glioblastoma Replicative Immortality in a TERT Promoter Mutation-Dependent Manner. *Cancer Cell* **34**, 513-528 e518, doi:10.1016/j.ccell.2018.08.003 (2018).
- 72 Verburg, N. *et al.* Spatial concordance of DNA methylation classification in diffuse glioma. *Neuro Oncol* **23**, 2054-2065, doi:10.1093/neuonc/noab134 (2021).
- 73 Chen, P. *et al.* Olfactory sensory experience regulates gliomagenesis via neuronal IGF1. *Nature* **606**, 550-556, doi:10.1038/s41586-022-04719-9 (2022).
- 74 Daniel, A. G. S. *et al.* Functional connectivity within glioblastoma impacts overall survival. *Neuro Oncol*, doi:10.1093/neuonc/noaa189 (2020).
- 75 Belgers, V. *et al.* Postoperative oscillatory brain activity as an add-on prognostic marker in diffuse glioma. *J Neurooncol* **147**, 49-58, doi:10.1007/s11060-019-03386-7 (2020).
- 76 Derks, J. *et al.* Oscillatory brain activity associates with neuroligin-3 expression and predicts progression free survival in patients with diffuse glioma. *J Neurooncol* **140**, 403-412, doi:10.1007/s11060-018-2967-5 (2018).
- 77 Qi, C. *et al.* Thrombospondin-1 is a prognostic biomarker and is correlated with tumor immune microenvironment in glioblastoma. *Oncol Lett* **21**, 22, doi:10.3892/ol.2020.12283 (2021).
- 78 Lawler, J. *et al.* Identification and characterization of thrombospondin-4, a new member of the thrombospondin gene family. *J Cell Biol* **120**, 1059-1067, doi:10.1083/jcb.120.4.1059 (1993).
- 79 Filbin, M. G. *et al.* Developmental and oncogenic programs in H3K27M gliomas dissected by single-cell RNA-seq. *Science* **360**, 331-335, doi:10.1126/science.aao4750 (2018).
- 80 Venteicher, A. S. *et al.* Decoupling genetics, lineages, and microenvironment in IDH-mutant gliomas by single-cell RNA-seq. *Science* **355**, doi:10.1126/science.aai8478 (2017).
- 81 Keough, M. B. *et al.* An inhibitor of chondroitin sulfate proteoglycan synthesis promotes central nervous system remyelination. *Nat Commun* **7**, 11312, doi:10.1038/ncomms11312 (2016).

Reviewer Reports on the First Revision:

Referees' comments:

Referee #1 (Remarks to the Author):

The authors have addressed all of the concerns I raised through considerable additional experiments and revisions of the text. The manuscript in its current form is simply superb, a fascinating addition to the literature. I suspect that many papers in the future will spring from the topics and concepts that are first described within. Well done.

Referee #2 (Remarks to the Author):

The author did a fantastic job in revising the manuscript, performing additional analyses and experiments, providing important additional data, and editing the text. Collectively, these additions and changes made the story much stronger and even more compelling.

I have three remaining points that should be addressed before publication can be recommended:

1. to my Comment 1: the authors did not answer my question with respect to the high MGMT promotor hypermethylation numbers found here, and more importantly, whether survival differences found can be explained by MGMT status imbalances between groups. Furthermore, why did they did not include MGMT status as one crucial and really strong, established prognostic (since predictive for response to alkylating chemotherapy) factor into the new survival calculations? If MGMT was not showing up as an independent prognostic factor (was this tested?), then the entire analysis needs to be questioned.

2. to my Comment 4: the answer does not appear sufficient, at least does not help me to comprehend much better what I am looking at. What about the quantifications here to support/substantiate the claims? Are these representative images (if so, this should be clarified)? Or are these "composite" images of n=? patients?

3. the new Gabapentin data is interesting, and provides important translational aspects, but also strengthens the link between TSP-1, potentially tumor microtubules, and neuronal hyperexcitability / neuron-glioma interactions. However, one important control experiment is missing: considering the many direct and indirect actions of Gabapentin one can imagine on neurons and on tumor cells, it would be important to see that Gabapentin exposure has no relevant effects (or at least differential effects) on glioblastoma cells in vitro when growing in monoculture (without neurons).

Referee #4 (Remarks to the Author):

This is a an outstanding manuscript that examines the interaction between high grade gliomas and neural circuitry. The authors use several converging and complementary approaches and techniques to present compelling evidence that these brain tumors are associated with increased excitability and connectivity in surrounding human cortex, yet these surrounding circuits maintain functionality. At a molecular level, this increased connectivity appears related to increased TSP-1 expression, a known synaptogenic factor, at least compared to those tumor cells expressing TSP-1 in regions of low connectivity. This is a nice demonstration of a molecular basis to the observed

changes in connectivity. Increased expression of TSP-1 correlates with worse survival in animal models and also in subpopulations of their human subjects, and inhibition of TSP-1 appears to decrease proliferation in vitro. Together, this is an impressive amount of work and the conclusions are well supported in my view.

I was specifically asked to comment on the concerns raised by Reviewer 3. There are several concerns raised, and my opinion is that the authors have sufficiently addressed these concerns. Overall, I felt the manuscript is well written and clear. There were a number of initial concerns about how to interpret the ECoG recording data, and comparisons that were drawn and therefore conclusions that were inferred. In my view, the authors have sufficiently addressed these concerns, and have provided greater clarification regarding the experimental methods and have presented more detailed data regarding the recorded neural responses. The reviewer raises a good point about the method in which regions of high connectivity are identified using MEG since one would potentially ask why not simply identify those regions using the direct ECoG recordings. The authors have addressed the concern about MEG localization, and have provided a reasonable justification as to why they chose to use MEG (since it provides broader coverage). Certainly it would be helpful if they could supplement this analysis with an analysis of connectivity based on the ECoG data, as this would provide stronger evidence. But this is not required for the overall conclusions of the manuscript. The reviewer also raises a very good point about causality and the normal expression of TSP-1. The authors have clarified this point by amending their discussion which is likely sufficient. An additional possibility would be to examine TSP-1 expression in other brain samples (epilepsy patients) for control, but again, this would be supporting evidence but not absolutely required. The reviewer also raised a question about the survival curves and the distinctions made between different subgroups, but this appears well addressed. The minor points raised by the reviewer have also been sufficiently addressed.

Author Rebuttals to First Revision:

Referee #1 (Remarks to the Author)

Referee 1 Comment 1: *The authors have addressed all of the concerns I raised through considerable additional experiments and revisions of the text. The manuscript in its current form is simply superb, a fascinating addition to the literature. I suspect that many papers in the future will spring from the topics and concepts that are first described within. Well done.*

Referee 1 Response 1: Thank you for offering valuable critique which has significantly improved the quality of our work.

Referee #2 (Remarks to the Author)

The author did a fantastic job in revising the manuscript, performing additional analyses and experiments, providing important additional data, and editing the text. Collectively, these additions and changes made the story much stronger and even more compelling. I have three remaining points that should be addressed before publication can be recommended:

Referee 2 Comment 1: *The authors did not answer my question with respect to the high MGMT promotor hypermethylation numbers found here, and more importantly, whether survival differences found can be explained by MGMT status imbalances between groups. Furthermore, why did they did not include MGMT status as one crucial and really strong, established prognostic (since predictive for response to alkylating chemotherapy) factor into the new survival calculations? If MGMT was not showing up as an independent prognostic factor (was this tested?), then the entire analysis needs to be questioned.*

Referee 2 Response 1: Thank you for this important comment and asking for clarification. MGMT promoter methylation as a control was included in all human overall survival analysis for this paper. We agree with reviewer #2 that MGMT promoter methylation status is a well-known predictive and prognostic biomarker for survival outcomes in patients with glioblastoma. Importantly, MGMT methylation is associated with a longer survival, as demonstrated by previous studies (PMID: 26885283, PMID: 36009577, PMID: 27904447). Our overarching experimental goal throughout all experiments in this study was to use human data and primary patient-derived tissues whenever possible. Therefore, Extended Data Table 2 summarizes the MGMT methylation status for each individual patient-derived sample, culture, and tissues used in this study (not purely the patients included in our survival analysis). As the referee is aware, there are multiple methods used to quantify MGMT promoter methylation and our institution uses methylation-specific PCR which yields a binary variable (methylation yes/no). We do not have a semiquantitative methylation data available for the patients and samples used in this study. We appreciated the idea that the survival analysis in Figure 4, in which glioma-intrinsic neuronal oscillations influenced survival outcomes, could be a provocative finding. Therefore, we applied a rigorously studied and previously validated data set of patients with newly diagnosed glioblastoma (summarized in extended data table 5). We recently published a retrospective multicenter cohort study of patients with newly diagnosed glioblastoma, in which age, IDH, extent of resection of enhancing and non-contrast enhancing disease, in addition to the use of chemoradiation were significant predictors of overall survival (Molinaro and Hervey-Jumper, *Jama Oncology* 2020- PMID: 32027343). The strength of this approach is that the risk model generated by the discovery cohort of glioblastoma patients used in the Molinaro *Jama Oncology* 2020 study has been validated externally, therefore suggesting generalizability of results. Using these data, we identified 64 patients, all of whom received standard-of-care Stupp protocol temozolomide plus 60 Gy brain irradiation. Of these individuals n = 40 (62.5%) were MGMT promoter methylated, and n = 24 (37.5%) were not methylated. We then quantified

glioma-intrinsic neuronal oscillations in the analysis. By univariate Cox proportional-hazard modeling, MGMT methylation status was not associated with overall survival outcomes in this subset of patients. However, given the prognostic significance of MGMT for overall survival outcomes, we still included MGMT status in our recursive partitioning survival tree (using the partDSA algorithm). MGMT, however was not chosen as a significant variable for overall survival while tumor-intrinsic functional connectivity was chosen. This result does not by any means suggest that MGMT status is not a significant correlate of overall survival (that fact was already well established in the literature including in our Molinaro, Hervey-Jumper, et al., JAMA Oncology 2020 publication using this patient cohort). It simply means that using a three-partition RPA, age, extent of tumor resection, and tumor-intrinsic connectivity are primary drivers of overall survival in this cohort of patients. It would be a fascinating experiment to explore the interactive effects of MGMT promoter methylation and tumor-intrinsic connectivity. However, our existing dataset appears underpowered to address this important question. This important translational question is now a topic of future investigation for our group based on reviewer 2's excellent comment. The manuscript has been updated to confirm that each analysis included proper controls for MGMT promoter methylation status.

Referee 2 Comment 2: *To my Comment 4: the answer does not appear sufficient, at least does not help me to comprehend much better what I am looking at. What about the quantifications here to support/substantiate the claims? Are these representative images (if so, this should be clarified)? Or are these "composite" images of n=? patients?*

Referee 2 Response 2: We regret that the explanation we provided for Fig. 1f was not sufficient. We would like to make it clear that this figure and panel was not based on any quantification. Rather, we used this schematic diagram as a visual representation to compare and contrast the spatial differences in the cortical neuronal activity across electrodes during speech initiation between tumor-infiltrated regions of the cortex as opposed to the "normal appearing" regions. We are aware that this illustration only serves to summarize the findings from other experiments listed in Figure 1 and do not add a new experimental data or observation to the manuscript. The intent was to improve readability for general audiences. We have removed it from the manuscript given that it appears to confuse rather than clarify concepts for the general reader.

Referee 2 Comment 3: *The new Gabapentin data is interesting, and provides important translational aspects, but also strengthens the link between TSP-1, potentially tumor microtubules, and neuronal hyperexcitability / neuron-glioma interactions. However, one important control experiment is missing: considering the many direct and indirect actions of Gabapentin one can imagine on neurons and on tumor cells, it would be important to see that Gabapentin exposure has no relevant effects (or at least differential effects) on glioblastoma cells in vitro when growing in monoculture (without neurons).*

Referee 2 Response 3: Thank you for this important point. We have addressed your comment by performing additional experiments to enumerate the effect of gabapentin treatment on glioblastoma proliferation in monoculture. Consistent with prior experiments, primary patient-derived glioblastoma cultures were treated with gabapentin (32 μ M) for 72h. We found that in contrast to the marked decrease in proliferation of TSP-1 overexpressing HFC cells in neuron-HFC glioma co-culture, gabapentin (GBP) in monoculture failed to produce any differential effect on glioblastoma proliferation. This data suggests that the antiproliferative effects of gabapentin are restricted to inhibition of neuronal activity-dependent glioblastoma proliferation. The manuscript has been revised to reflect this additional control experiment on page 14, lines 295-296. We have also incorporated these additional analyses to the revised manuscript as Extended Data Fig. 22 (Page 87- lines 735-740).

Representative confocal images from HFC glioma monoclulture showing no significant change in glioblastoma proliferation (as measured by the number of human nuclear antigen (HNA)-positive cells co-labelled with Ki67 divided by the total number of HNA-positive tumor cells counted across all areas quantified) upon pharmacological TSP-1 inhibition using (32 μ M) gabapentin (HFC vs. HFC + GBP: 1.84 ± 1.48 % vs. 0.84 ± 0.56 %, $n = 2$ /group) ($P = 0.50$). Red, HNA (human nuclei); white, Ki67. Scale bar, 30 μ m.

Referee #4 (Remarks to the Author)

Referee 1 Comment 1: *This is an outstanding manuscript that examines the interaction between high grade gliomas and neural circuitry. The authors use several converging and complementary approaches and techniques to present compelling evidence that these brain tumors are associated with increased excitability and connectivity in surrounding human cortex, yet these surrounding circuits maintain functionality. At a molecular level, this increased connectivity appears related to increased TSP-1 expression, a known synaptogenic factor, at least compared to those tumor cells expressing TSP-1 in regions of low connectivity. This is a nice demonstration of a molecular basis to the observed changes in connectivity. Increased expression of TSP-1 correlates with worse survival in animal models and also in subpopulations of their human subjects, and inhibition of TSP-1 appears to decrease proliferation in vitro. Together, this is an impressive amount of work and the conclusions are well supported in my view.*

I was specifically asked to comment on the concerns raised by Reviewer 3. There are several concerns raised, and my opinion is that the authors have sufficiently addressed these concerns. Overall, I felt the manuscript is well written and clear. There were a number of initial concerns about how to interpret the ECoG recording data, and comparisons that were drawn and therefore conclusions that were inferred. In my view, the authors have sufficiently addressed these concerns, and have provided greater clarification regarding the experimental methods and have presented more detailed data regarding the recorded neural responses. The reviewer raises a good point about the method in which regions of high connectivity are identified using MEG since one would potentially ask why not simply identify those regions using the direct ECoG recordings. The authors have addressed the concern about MEG localization, and have provided a reasonable justification as to why they chose to use MEG (since it provides broader

coverage). Certainly, it would be helpful if they could supplement this analysis with an analysis of connectivity based on the ECoG data, as this would provide stronger evidence. But this is not required for the overall conclusions of the manuscript. The reviewer also raises a very good point about causality and the normal expression of TSP-1. The authors have clarified this point by amending their discussion which is likely sufficient. An additional possibility would be to examine TSP-1 expression in other brain samples (epilepsy patients) for control, but again, this would be supporting evidence but not absolutely required. The reviewer also raised a question about the survival curves and the distinctions made between different subgroups, but this appears well addressed. The minor points raised by the reviewer have also been sufficiently addressed.

Referee 4 Response 1: We greatly appreciate the careful review and comments made by Reviewer #3 and #4 for this manuscript. We are thrilled to know that we have satisfactorily addressed the questions and concerns raised by the Reviewer #3. While not requested for this submission we agree that the analysis of spectral features of diffuse gliomas across subtype with cross validation between MEG and ECoG could be of great value. The present study is under sized for this experiment however we look forward to report these results in future studies. We strongly believe that addition of the new experiments and analyses requested by Reviewer comments have strengthened our manuscript. The findings in this manuscript illustrates a fundamental change in our understanding of molecular drivers of glioblastoma proliferation as well as radically change the way human brain cancers are studied.

Reviewer Reports on the Second Revision:

Referees' comments:

Referee #2 (Remarks to the Author):

The authors have responded very well to all of my comments, and I have no remaining points.

Regarding my 3rd comment: The authors might want to consider to shortly mention (maybe with a few words in the result sentence) the fact that under non-treatment conditions, co-culture with neurons is massively increasing tumor cell proliferation when compared to the monoculture conditions (Ext Data Fig. 22 vs Fig. 4f) - IF they believe this comparison is scientifically meaningful because of otherwise comparable growth/medium/....conditions. - This would strengthen the point that neuronal proximity and/or activity per se is massively pushing tumor cell proliferation in their models.